

SciPost Phys. Lect. Notes 73 (2023)

# Introduction to Hamiltonian formulation of general relativity and homogeneous cosmologies

**Rishabh Jha⋆**

Institute for Theoretical Physics, Georg-August-Universität Göttingen, Germany

⋆ rishabh.jha@uni-goettingen.de

## Abstract

We give a pedagogical introduction to the Hamiltonian formalism of general relativity at an advanced undergraduate and graduate levels. After covering the mathematical prerequisites as well as the $3 + 1$-decomposition of spacetime, we proceed to discuss the Arnowitt-Deser-Misner (ADM) formalism (a Hamiltonian approach) of general relativity. Then we proceed to give a brief but self-contained introduction to homogeneous (but not necessarily isotropic) universes and discuss the associated Bianchi classification. We first study their dynamics in the Lagrangian formulation, followed by the Hamiltonian formulation to show the equivalence of both approaches. We present a variety of examples to illustrate the ADM formalism: (i) free & massless scalar field coupled to homogeneous (in particular, Bianchi IX) universe, (ii) scalar field with a potential term coupled to Bianchi IX universe, (iii) electromagnetic field coupled to gravity in general, and (iv) electromagnetic field coupled to Bianchi IX universe.

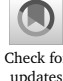
# 1   Introduction

General relativity is generally introduced in the Lagrangian formalism (the so-called *standard formalism*) to the students. This illustrates the importance of the principle of general covariance. Similar to what we have in classical mechanics where the Hamiltonian formalism (the

so-called *canonical formalism*) is an equivalent description as the Lagrangian formalism, this is true in the context of general relativity as well. But this canonical approach to general relativity is not as completely obvious. For example, in the covariant formalism, space and time are treated on equal footing but in the Hamiltonian approach, we need to parametrize time and create a slicing of "time+space" of the spacetime manifold. This is non-trivial because general relativity does not admit a natural parametrization for time and thus choosing a particular time coordinate remains arbitrary in the canonical approach. The purpose of this work is to go through the details of this procedure and allow for a Hamiltonian description of general relativity.

Once we have the canonical formalism ready, we see that the formulation of the initial value problem (the so-called *Cauchy problem*) is vastly simplified. A vast amount of progress has been made in the context of the initial-value problem in general relativity [1] and the associated works of York, Choquet-Bruhat and O'Murchadha [2,3]. One of the central pillars of the Cauchy problem formulation in general relativity is the Arnowitt-Deser-Misner (ADM) formalism [4,5] and its applications to various Lorentz invariant classical field theoretical formulations. Simultaneously, the Hypersurface Deformation Algebra (HDA) [6] have played a significant role in the development of general relativity. This is sometimes taken as an independent starting point to develop general relativity. Physics emerging from further deforming the HDA [7] are the topical areas of interest. We briefly discuss this in Chapter 3.3 and present a detailed derivation of the HDA in Appendix H.

We cover two major topics in this work: (i) the ADM (Hamiltonian) formulation of general relativity, & (ii) homogeneous cosmological solutions of the Einstein field equations (the so-called "Bianchi" class of universes). Mathematically speaking, the isotropic and homogeneous universe, namely the FRWL cosmological model is a subset of the Bianchi universe in which the anisotropy parameters vanish.[1] FRWL cosmological model is highly relevant for our universe as it lies within the experimental limits placed through CMB (Cosmic Microwave Background) observations and the paradigm of inflation [8]. However the equations of motion of general relativity predict that a deviation from isotropy might have happened at very early epochs (before the inflation), so studying the anisotropic homogeneous models makes sense in these regards.

We have tried to be detailed and self-contained in this work while addressing both of these pre-requisites. The readers are expected to have familiarity with the basic concepts in general relativity and know how to derive the FRWL cosmological solution from the Einstein field equations. The structure is as follows.

Chapter 2 deals with the mathematical preliminaries required for this work. It focuses on developing the mathematics required for the two approaches towards general relativity, namely the Lagrangian formulation as well as the Hamiltonian formulation. In particular, Section 2.2 provides a brief but rigorous derivation of Einstein field equations using the Einstein-Hilbert action in the Lagrangian formulation. Concepts of hypersurfaces, embeddings and other related foundational topics are discussed which form the basis of $3+1-$description of general relativity.

Chapter 3 deals with the Arnowitt-Deser-Misner formalism (a Hamiltonian approach) of general relativity. Section 3.1 delves into decomposing the spacetime into $3+1-$foliation of space and time. After describing this procedure, the ADM formalism is discussed in Section 3.2. The chapter concludes with the discussion on Hypersurface Deformation Algebra (sometimes known as the Dirac algebra) in Section 3.3 which can be viewed as an independent starting

---

[1]Different Bianchi universes have different topologies, just like FRWL universes. Thus to make this sentence more precise, the closed FRWL universe is the isotropic case of Bianchi IX, the flat FRWL universe is a special case of Bianchi I & Bianchi $VII_0$ and the open FRWL is of Bianchi V & Bianchi $VII_a$. See the chart 129 for the Bianchi classification.

point of general relativity [9,10]. A detailed derivation of the HDA is provided in Appendix H.

Chapter 4 provides a brief but self-contained introduction to homogeneous but anisotropic universes (the "Bianchi" class of universes) where we start with the classification of topologically different homogeneous cosmologies in Section 4.1. We introduce a form of basis, known as the *invariant basis*, in Section 4.2 which we show to be particularly well-suited to study homogeneous cosmologies. We express the Einstein field equations in invariant basis in this section as well. Then we discuss the dynamics, as examples, of Bianchi I and Bianchi IX universes in the Lagrangian formulation in Section 4.3 where all the results are re-derived in the Hamiltonian formulation in Chapters 5.1 & 5.2, thereby showing their equivalence.

Chapter 5 is completely devoted to do the canonical analyses of the homogeneous cosmologies that we encountered in Chapter 4. Again as examples, we present the ADM analysis of Bianchi I and Bianchi IX universes in Sections 5.1 & 5.2 where we re-establish the results obtained in the Lagrangian formulation in Chapter 4.3. We then proceed to give two more examples to further practice the ADM formulation: (i) (Section 5.3) a free & massless classical scalar field coupled to Bianchi IX universe, and (ii) (Section 5.4) we extend the previous system to the case of a classical scalar field with a potential term. Through these two examples, we study their dynamics and phenomena such as *Mixmaster dynamics* (first encountered in Chapter 4.3.2 and re-established in Section 5.2.4) & *quiescence* (introduced in Section 5.3.1).

Chapter 6 extends the ADM analysis done in Chapter 5 to the case of Einstein/Bianchi IX-Maxwell-Scalar Field system. Section 6.1 contains a $3+1-$decomposition of Maxwell's equations and the continuity equation which we use in Subsection 6.1.1 to present the full Einstein-Maxwell equations of motion for the general case of electromagnetic field coupled to gravity. Then in Section 6.2, we take a step back and derive the ADM action whose variations lead to the equations of motion presented in Subsection 6.1.1. Finally in Subsection 6.2.1, we specialize to the case of homogeneous cosmology, in particular Bianchi IX universe and do the ADM analysis of Bianchi IX-Maxwell system. In Section 6.3, we study a free & massless classical scalar field coupled to the Bianchi IX-Maxwell system in the Hamiltonian formalism and calculate explicitly its equations of motion. Although the procedure has been known in the literature, the explicit calculations and the results obtained in Sections 6.2 & 6.3 have not been reported to the best knowledge of the author. Thus these two sections can be considered a new component of this work, albeit not original.

Chapter 7 summarises this work and discusses future prospects such as extending these results to Yang-Mills field as well as to other more general inhomogeneous universes. Appendices contain involved & detailed calculations that have been taken out from the corresponding chapters and relegated therein to maintain the flow of reading.

For the purposes of this work, we will always be interested in the bulk and will always (unless stated otherwise) ignore the boundary terms arising, say, due to integration-by-parts. Therefore, two of the crucial concepts missing in this work are: ADM mass & ADM momentum. The sign of the metric $g_{\mu\nu}$ will be taken as $(-,+,+,+)$ and cosmological constant $\Lambda$ will be set to zero (unless stated otherwise). The units we will be working with are the natural units where we set the Newton's gravitational constant $G$ and the speed of light $c$, both equal to 1. This work is completely based on classical Physics and every entity encountered should be taken as classical objects. Greek indices, such as $\mu, \nu, \alpha, \ldots$, denote the full spacetime components (which in $3+1-$D means running over $\{0,1,2,3\}$) while Latin indices, such as $i,j,a,\ldots$, denote the spatial components only (which in $3-$D means running over $\{1,2,3\}$). The only exception will be when we introduce invariant basis in Chapter 4.2 where both Greek and Latin indices will denote spatial components with Greek denoting invariant basis while Latin denoting coordinate basis. There should be no confusion for the readers as what Greek indices mean (spacetime components versus spatial components in invariant basis) will always be clear from the context. See footnote 6 in Chapter 4 for further comments.

## 2 Mathematical preliminaries

In this chapter, we set up the mathematical machinery behind general relativity. In Section 2.1, we start with defining crucial mathematical operations which are inevitable in the study of general relativity. The basic definitions and useful formulae of general relativity are already summarized in Appendix A. After briefly discussing the definitions, we proceed to directly deal with general relativity. There are two major approaches: the Lagrangian (or the so-called *standard*) as well as the Hamiltonian formulations of general relativity. Section 2.2 completely derives from the basic the Einstein field equations using the Lagrangian approach starting from the Einstein-Hilbert action. Section 2.3 prepares the readers for the Hamiltonian formulation which is discussed at length in Chapter 3.

### 2.1 Definitions

**Covariant derivative or connection**

We consider a differentiable manifold $\mathcal{M}$ over which we define a *covariant derivative* (or *connection*) $\nabla$ as a map:

$$\nabla : T(r,s) \longmapsto S(r,s+1), \tag{1}$$

where $T$ and $S$ are tensor fields of rank $(r,s)$ and $(r,s+1)$, satisfying the following properties:

(a) $\nabla$ is linear: $\nabla(aT + bS) = a\nabla T + b\nabla S$ where $T$ and $S$ are tensor fields of same rank and $\{a,b\}$ are scalar constants.

(b) For a given tensor field $T$ and a scalar field $f$, we have $df$ as a tensor of rank $(0,1)$ with tensor components $\partial_\mu f$, and the connection satisfies: $\nabla(fT) = df \otimes T + f\nabla T$.

(c) Given the bases sets $\{e_\mu\}$ and $\{\theta^\mu\}$ of the tangent and the cotangent spaces $T_p(\mathcal{M})$ and $T_p^\star(\mathcal{M})$ respectively, we have: $\nabla e_\mu = \Lambda^\alpha_{\beta\mu}\theta^\beta \otimes e_\alpha$, where $\Lambda^\alpha_{\beta\mu}$ are the connection coefficients defined in Appendix A.

The connection becomes a *metric connection* if we have a well-defined metric $g_{\mu\nu}$ on the differentiable manifold $\mathcal{M}$ and the connection satisfies $\nabla g_{\mu\nu} = 0$. In this particular case, the connections are known as *Christoffel symbols* whose formula is provided in Appendix A.

In terms of components, we have:

$$\begin{aligned}
\nabla_\mu A^\nu = A^\nu_{;\mu} &= \frac{\partial A^\nu}{\partial x^\mu} + \Lambda^\nu_{\mu\lambda}A^\lambda = A^\nu_{,\mu} + \Lambda^\nu_{\mu\lambda}A^\lambda, \\
\nabla_\mu A_\nu = A_{\nu;\mu} &= \frac{\partial A_\nu}{\partial x^\mu} - \Lambda^\lambda_{\nu\mu}A_\lambda = A_{\nu,\mu} - \Lambda^\lambda_{\nu\mu}A_\lambda.
\end{aligned} \tag{2}$$

**Tensor density**

For a given tensor $T$ of rank $(r,s)$, the corresponding tensor density is defined as:

$$\mathcal{T}^{\alpha_1\alpha_2\cdots\alpha_r}{}_{\beta_1\beta_2\ldots\beta_s} \equiv \sqrt{|g|}^W T^{\alpha_1\alpha_2\ldots\alpha_r}{}_{\beta_1\beta_2\ldots\beta_s}, \tag{3}$$

where $g = \text{determinant}(g_{\mu\nu})$ and $W \in \mathbb{R}$ is the *weight of the tensor density*.

With this defined, the covariant derivative of a tensor density is a direct generalization (using $\nabla_\mu g_{\alpha\beta} = 0$, so $\nabla_\mu g = 0$):

$$\begin{aligned}
\nabla_\mu \mathcal{T}^{\alpha_1\alpha_2\cdots\alpha_r}{}_{\beta_1\beta_2\ldots\beta_s} &= \sqrt{|g|}^W \nabla_\mu \left[ \frac{\mathcal{T}^{\alpha_1\alpha_2\cdots\alpha_r}{}_{\beta_1\beta_2\ldots\beta_s}}{\sqrt{|g|}^W} \right] \\
&= \sqrt{|g|}^W \nabla_\mu T^{\alpha_1\alpha_2\ldots\alpha_r}{}_{\beta_1\beta_2\ldots\beta_s}.
\end{aligned} \tag{4}$$

**Integral curve and flow map**

For any given vector field $X = X^\mu \partial_\mu$ on a differentiable manifold $\mathcal{M}$ and an open subset $I \subset \mathbb{R}$, we define the *integral curve of $X$* at point $p$ as follows:

$$\alpha_p : I \longmapsto \mathcal{M},$$
$$s \longmapsto \alpha_p(s),$$

(5)

such that:

$$\alpha_p(0) = 0,$$
$$\frac{d\alpha_p}{ds}\bigg|_{s_0} = \dot{\alpha}_p(s_0) = X_{s_0}(\alpha_p) \ \forall \ s_0 \in I.$$

(6)

Then the integral curve defines the *flow map $\phi_s^X$* as follows (where $U \subset \mathcal{M}$ is an open subset):

$$\phi_s^X : U \longmapsto \mathcal{M},$$
$$p \longmapsto \alpha_p(s),$$

(7)

such that:

$$\frac{d\alpha_p}{ds}\bigg|_{s_0} = \dot{\alpha}_p(s_0) = X_{s_0}(\alpha_p).$$

(8)

This flow map $\phi_s^X$ has the following properties:

(a) $\phi_0^X(p) = \alpha_p(0) = p \implies \phi_0^X = \mathbb{I}$,

(b) $\phi_s^X \circ \phi_t^X = \phi_{s+t}^X \quad \forall s, t \in \mathbb{R}$,

(c) $\phi_s^X$ is a diffeomorphism, and

(d) $\left[\phi_s^X\right]^{-1} = \phi_{-s}^X$.

**Lie derivative**

The flow map allows us to define something known as Lie derivative of any differentiable tensor field of rank $(r, s)$ along the vector $X$, evaluated at point $p$, as follows:

$$[\mathcal{L}_X(T)]_p \equiv \frac{d}{ds}\bigg|_{s=0} \left[\left(\phi_{-s}^X\right)_* T_{\phi_s^X(p)}\right].$$

(9)

In terms of components, which is most commonly used by physicists, we have:

$$\begin{aligned}
[\mathcal{L}_X(T)]^{\mu_1 \dots \mu_r}{}_{v_1 \dots v_s} &= X^\lambda \partial_\lambda T^{\mu_1 \dots \mu_r}{}_{v_1 \dots v_s} \\
&\quad - T^{\lambda \dots \mu_r}{}_{v_1 \dots \mu_s} \partial_\lambda X^{\mu_1} - \dots - T^{\mu_1 \dots \lambda}{}_{v_1 \dots v_s} \partial_\lambda X^{\mu_r} \\
&\quad + T^{\mu_1 \dots \mu_r}{}_{\lambda \dots v_s} \partial_{v_1} X^\lambda + \dots + T^{\mu_1 \dots \mu_r}{}_{v_1 \dots \lambda} \partial_{v_s} X^\lambda.
\end{aligned}$$

(10)

For our purposes, we are interested in the special case of $\nabla$ being torsion-free, namely $\Gamma^\lambda_{\mu v} = +\Gamma^\lambda_{v\mu}$, where the Lie derivative takes the form:

$$\begin{aligned}
[\mathcal{L}_X(T)]^{\mu_1 \dots \mu_r}{}_{v_1 \dots v_s} &= X^\lambda \nabla_\lambda T^{\mu_1 \dots \mu_r}{}_{v_1 \dots v_s} \\
&\quad - T^{\lambda \dots \mu_r}{}_{v_1 \dots \mu_s} \nabla_\lambda X^{\mu_1} - \dots - T^{\mu_1 \dots \lambda}{}_{v_1 \dots v_s} \nabla_\lambda X^{\mu_r} \\
&\quad + T^{\mu_1 \dots \mu_r}{}_{\lambda \dots v_s} \nabla_{v_1} X^\lambda + \dots + T^{\mu_1 \dots \mu_r}{}_{v_1 \dots \lambda} \nabla_{v_s} X^\lambda.
\end{aligned}$$

(11)

Lie derivative satisfies the following properties as can be checked by direct computation:

(a) $\mathcal{L}_X(T)$ is linear in both $X$ and $T$,

(b) $\mathcal{L}_X : T(r,s) \longmapsto S(r,s)$: Lie derivative of a tensor field of rank $(r,s)$ is another tensor field of rank $(r,s)$,

(c) for a given scalar field $f$, we have $\mathcal{L}_X(f) = X(f) = X^\mu \partial_\mu f$, and

(d) for a given vector field $V^\mu$, we have $\mathcal{L}_X V^\mu = [\vec{X}, \vec{V}]$.

## 2.2 Lagrangian formulation of general relativity

As a reminder, we are using the natural units ($G = c = 1$), setting the cosmological constant $\Lambda = 0$, and always ignoring all boundary terms (unless stated otherwise) throughout this work. After discussing all the variations with respect to the metric in the following paragraphs, we will derive the Einstein field equations using the Lagrangian formulation.

**Variation of metric**

Variations $\delta g^{\mu\nu}$ and $\delta g_{\mu\nu}$ are related as:

$$g^{\alpha\lambda} g_{\lambda\beta} = \delta^\alpha_\beta \quad \Rightarrow \boxed{\delta g_{\mu\nu} = -g_{\mu\alpha} g_{\nu\beta} \delta g^{\alpha\beta}\,,} \tag{12}$$

where minus sign is noted.

We also have the *Jacobi's formula*:

$$\boxed{\delta g = g g^{\mu\nu} \delta g_{\mu\nu} = -g g_{\mu\nu} \delta g^{\mu\nu}\,,} \tag{13}$$

**Variation of Christoffel symbols**

The variation is:

$$
\begin{aligned}
\delta \Gamma_{\lambda\mu\nu} &= \frac{1}{2} \left( \delta g_{\lambda\nu,\mu} + \delta g_{\mu\lambda,\nu} - \delta g_{\mu\nu,\lambda} \right) \\
&= \frac{1}{2} \left( \nabla_\mu \delta g_{\lambda\nu} + \nabla_\nu \delta g_{\mu\lambda} - \nabla_\lambda \delta g_{\mu\nu} \right) \\
&\quad + \frac{1}{2} \left[ \Gamma^\sigma_{\mu\lambda} \delta g_{\sigma\nu} + \Gamma^\sigma_{\mu\nu} \delta g_{\lambda\sigma} + \Gamma^\sigma_{\nu\mu} \delta g_{\sigma\lambda} + \Gamma^\sigma_{\nu\lambda} \delta g_{\mu\sigma} - \Gamma^\sigma_{\lambda\mu} \delta g_{\sigma\nu} - \Gamma^\sigma_{\lambda\nu} \delta g_{\mu\sigma} \right]
\end{aligned}
\tag{14}
$$

$$\Rightarrow \boxed{\delta \Gamma_{\lambda\mu\nu} = \frac{1}{2} \left( \nabla_\mu \delta g_{\lambda\nu} + \nabla_\nu \delta g_{\mu\lambda} - \nabla_\lambda \delta g_{\mu\nu} \right) + \Gamma^\sigma_{\mu\nu} \delta g_{\sigma\lambda}\,.} \tag{15}$$

We will also be needing:

$$
\begin{aligned}
\delta \Gamma^\rho_{\mu\nu} &= \delta g^{\rho\lambda} \Gamma_{\lambda\mu\nu} + g^{\rho\lambda} \delta \Gamma_{\lambda\mu\nu} \\
&= -g^{\rho\lambda} \Gamma^\sigma_{\mu\nu} \delta g_{\sigma\lambda} + \frac{1}{2} g^{\rho\lambda} \left( \nabla_\mu \delta g_{\lambda\nu} + \nabla_\nu \delta g_{\mu\lambda} - \nabla_\lambda \delta g_{\mu\nu} \right) + g^{\rho\lambda} \Gamma^\sigma_{\mu\nu} \delta g_{\sigma\lambda}
\end{aligned}
\tag{16}
$$

$$\Rightarrow \boxed{\delta \Gamma^\rho_{\mu\nu} = \frac{1}{2} g^{\rho\lambda} \left( \nabla_\mu \delta g_{\lambda\nu} + \nabla_\nu \delta g_{\mu\lambda} - \nabla_\lambda \delta g_{\mu\nu} \right)\,,} \tag{17}$$

which can be shown to be a tensor of rank $(1,2)$,

and

$$\delta\Gamma^{\mu}_{\mu\nu} = \delta g^{\mu\lambda}\Gamma_{\lambda\mu\nu} + g^{\mu\lambda}\delta\Gamma_{\lambda\mu\nu}$$

$$= -g^{\alpha\mu}g^{\beta\lambda}\delta g_{\alpha\beta}\Gamma_{\lambda\mu\nu} + g^{\mu\lambda}\left[\frac{1}{2}\left(\nabla_{\mu}\delta g_{\lambda\nu} + \nabla_{\nu}\delta g_{\mu\lambda} - \nabla_{\lambda}\delta g_{\mu\nu}\right) + \Gamma^{\sigma}_{\mu\nu}\delta g_{\sigma\lambda}\right] \quad (18)$$

$$= -\Gamma^{\beta}_{\mu\nu}g^{\alpha\mu}\delta g_{\alpha\beta} + \frac{1}{2}g^{\lambda\mu}\nabla_{\nu}\delta g_{\lambda\mu} + \Gamma^{\beta}_{\mu\nu}g^{\alpha\mu}\delta g_{\alpha\beta}$$

$$\Rightarrow \boxed{\delta\Gamma^{\mu}_{\mu\nu} = \frac{1}{2}g^{\lambda\mu}\nabla_{\nu}\delta g_{\lambda\mu}.} \quad (19)$$

**Variation of curvature**

We can start from the complete definition of the Riemann curvature tensor as provided in Appendix A and vary it with respect to the metric, or we can make our lives easier by choosing a local inertial frame where we can always make $\Gamma^{\lambda}_{\mu\nu} = 0$ which is valid in any Lorentz frame (tangential to the spacetime manifold). Accordingly this greatly simplifies the variation of the Riemann curvature tensor as follows:

$$\delta R^{\rho}_{\sigma\mu\nu} = \delta\left[\Gamma^{\rho}_{\sigma\nu,\mu} - \Gamma^{\rho}_{\sigma\mu,\nu}\right] \qquad \text{(Lorentz frame)} , \quad (20)$$

where we now replace the partial derivative $\partial_{\mu}$ with covariant derivative $\nabla_{\mu}$ and realize that this is a tensor identity, therefore it should be valid in all frames of reference. This leads to the *Palatini identity*:

$$\Rightarrow \boxed{\delta R^{\rho}_{\sigma\mu\nu} = \nabla_{\mu}\delta\Gamma^{\rho}_{\sigma\nu} - \nabla_{\nu}\delta\Gamma^{\rho}_{\sigma\mu}.} \quad (21)$$

Accordingly we get:

$$\Rightarrow \boxed{\delta R_{\mu\nu} = \nabla_{\lambda}\delta\Gamma^{\lambda}_{\mu\nu} - \nabla_{\mu}\delta\Gamma^{\lambda}_{\lambda\nu}.} \quad (22)$$

Proof: By definition, we have:

$$R_{\mu\nu} = R^{\lambda}_{\mu\lambda\nu} = \partial_{\lambda}\Gamma^{\lambda}_{\mu\nu} - \partial_{\nu}\Gamma^{\lambda}_{\mu\lambda} + \Gamma^{\lambda}_{\lambda\rho}\Gamma^{\rho}_{\nu\mu} - \Gamma^{\lambda}_{\nu\rho}\Gamma^{\rho}_{\lambda\mu}$$

$$\Rightarrow \delta R_{\mu\nu} = \partial_{\lambda}\delta\Gamma^{\lambda}_{\mu\nu} - \partial_{\nu}\delta\Gamma^{\lambda}_{\mu\lambda} + \delta\Gamma^{\lambda}_{\lambda\rho}\Gamma^{\rho}_{\nu\mu} + \Gamma^{\lambda}_{\lambda\rho}\delta\Gamma^{\rho}_{\nu\mu} - \delta\Gamma^{\lambda}_{\nu\rho}\Gamma^{\rho}_{\lambda\mu} - \Gamma^{\lambda}_{\nu\rho}\delta\Gamma^{\rho}_{\lambda\mu}. \quad (23)$$

Then we use eqs. (17, 19) to get the desired result: $\delta R_{\mu\nu} = \nabla_{\lambda}\delta\Gamma^{\lambda}_{\mu\nu} - \nabla_{\mu}\delta\Gamma^{\lambda}_{\lambda\nu}$.

Next we evaluate another important result which will also be used in the context of 3−dimensions in Appendix F (recall $\nabla_{\mu}g_{\alpha\beta} = 0$):

$$g^{\mu\nu}\delta R_{\mu\nu} = \nabla_{\lambda}\left(g^{\mu\nu}\delta\Gamma^{\lambda}_{\mu\nu}\right) - \nabla_{\mu}\left(g^{\mu\nu}\delta\Gamma^{\lambda}_{\lambda\nu}\right)$$

$$= \nabla_{\lambda}\left(g^{\mu\nu}\delta\Gamma^{\lambda}_{\mu\nu} - g^{\mu\lambda}\delta\Gamma^{\nu}_{\nu\mu}\right). \quad (24)$$

We again use eqs. (17, 19) to get:

$$\Rightarrow g^{\mu\nu}\delta R_{\mu\nu} = \left(\nabla^{\mu}\nabla^{\nu} - g^{\mu\nu}\nabla^{\alpha}\nabla_{\alpha}\right)\delta g_{\mu\nu}$$

$$= \left(g^{\mu\alpha}g^{\nu\beta} - g^{\mu\nu}g^{\alpha\beta}\right)\nabla_{\mu}\nabla_{\nu}\delta g_{\alpha\beta} \quad (25)$$

$$\Rightarrow \boxed{g^{\mu\nu}\delta R_{\mu\nu} = \nabla_{\lambda}\left[g^{\lambda\alpha}g^{\nu\beta} - g^{\lambda\nu}g^{\alpha\beta}\right]\nabla_{\nu}\delta g_{\alpha\beta}.}$$

Now using eq. (22), we can also calculate the variation of Ricci scalar with respect to the metric as follows:

$$
\begin{aligned}
\delta R &= -R^{\mu\nu}\delta g_{\mu\nu} + g^{\mu\nu}\delta R_{\mu\nu} \\
&= -R^{\mu\nu}\delta g_{\mu\nu} + g^{\mu\nu}\left[\nabla_\lambda \delta\Gamma^\lambda_{\mu\nu} - \nabla_\mu \delta\Gamma^\lambda_{\lambda\nu}\right] \\
&= -R^{\mu\nu}\delta g_{\mu\nu} + \underbrace{\nabla_\lambda \delta V^\lambda} \\
&= g^{\mu\nu}g^{\rho\lambda}\nabla_\rho\left(\nabla_\mu \delta g_{\lambda\nu} - \nabla_\lambda \delta g_{\mu\nu}\right) \\
&= \nabla^\mu\nabla^\nu \delta g_{\mu\nu} - \nabla^\lambda\nabla_\lambda \delta\ln|g|
\end{aligned}
\tag{26}
$$

Thus we have finally:

$$
\Rightarrow \boxed{\delta R = -R^{\mu\nu}\delta g_{\mu\nu} + \nabla^\mu\nabla^\nu \delta g_{\mu\nu} - \nabla^\lambda\nabla_\lambda \delta\ln|g|\,.}
\tag{27}
$$

**Action of general relativity**

The total action functional is:

$$
S = S_H + S_m\,,
\tag{28}
$$

where $S_H$ is the Hilbert term for pure gravity and $S_m$ is the matter action, both given by:

$$
\begin{aligned}
S_H &= \int_V d^4x\,\sqrt{-g}\,\mathcal{L}_H = \frac{1}{16\pi}\int_V d^4x\,\sqrt{-g}R\,, \\
S_m &= \int_V d^4x\,\sqrt{-g}\,\mathcal{L}_m\,,
\end{aligned}
\tag{29}
$$

where $V$ is the volume over which the integration is done.

**Variation of the Hilbert term**

We apply the chain rule to get:

$$
\begin{aligned}
\delta\mathcal{L}_H &= \frac{1}{16\pi}\left[\delta\left(g^{\mu\nu}R_{\mu\nu}\sqrt{-g}\right)\right] \\
&= -\frac{1}{16\pi}\left[\frac{\delta g}{2\sqrt{-g}}g^{\mu\nu}R_{\mu\nu} + \left(\delta g^{\mu\nu}R_{\mu\nu} + g^{\mu\nu}\delta R_{\mu\nu}\right)\sqrt{-g}\right]\,.
\end{aligned}
\tag{30}
$$

Then we use the Jacobi's formula (eq. (13)) to get:

$$
\delta\mathcal{L}_H = \frac{1}{16\pi}\left[\left(R_{\mu\nu} - \frac{1}{2}g_{\mu\nu}R\right)\delta g^{\mu\nu} + g^{\mu\nu}\delta R_{\mu\nu}\right]\sqrt{-g}\,.
\tag{31}
$$

Then we use Palatini identity (eq. (21)) to get:

$$
\begin{aligned}
\sqrt{-g}g^{\mu\nu}\delta R_{\mu\nu} &= \sqrt{-g}g^{\mu\nu}\left[\nabla_\rho \delta\Gamma^\rho_{\mu\nu} - \nabla_\mu \delta\Gamma^\rho_{\rho\nu}\right] \\
&= \sqrt{-g}\nabla_\rho\left[g^{\mu\nu}\delta\Gamma^\rho_{\mu\nu} - g^{\rho\nu}\delta\Gamma^\mu_{\mu\nu}\right] \doteq \partial_\rho\left(\sqrt{-g}\delta V^\rho\right)\,,
\end{aligned}
\tag{32}
$$

where we already introduced the variation $\delta V^\mu$ above. This becomes a full derivative, which we choose to ignore as we are never considering boundary contributions. We are, therefore, finally left with:

$$
\boxed{\delta\mathcal{S}_H = \frac{1}{16\pi}\int_V \left(R_{\mu\nu} - \frac{1}{2}g_{\mu\nu}R\right)\sqrt{-g}\delta g^{\mu\nu}\mathrm{d}^4x\,.}
\tag{33}
$$

**Variation of the Matter Term**

We get by applying the chain rule again:

$$
\begin{aligned}
\delta S_m &= \int_V \left[ \frac{\delta \mathcal{L}_m}{\delta g^{\mu\nu}} \delta g^{\mu\nu} \sqrt{-g} + \mathcal{L}_m \delta \sqrt{-g} \right] \mathrm{d}^4 x \\
&= \int_V \left[ \frac{\delta \mathcal{L}_m}{\delta g^{\mu\nu}} - \frac{1}{2} \mathcal{L}_m g_{\mu\nu} \right] \sqrt{-g} \delta g^{\mu\nu} \mathrm{d}^4 x \, .
\end{aligned}
\tag{34}
$$

We define the *stress-energy tensor* $T_{\mu\nu}$ as $T_{\mu\nu} \equiv -2 \frac{\delta \mathcal{L}_M}{\delta g^{\mu\nu}} + \mathcal{L}_M g_{\mu\nu}$ to get:

$$
\delta S_m = -\frac{1}{2} \int_V T_{\mu\nu} \sqrt{-g} \delta g^{\mu\nu} \mathrm{d}^4 x \, .
\tag{35}
$$

**Einstein Field Equations**

Combining the Hilbert term and the matter term, we get the variation of the full metric as:

$$
\delta S = \delta S_H + \delta S_m = \frac{1}{16\pi} \int_{\mathcal{V}} \left[ R_{\mu\nu} - \frac{1}{2} g_{\mu\nu} R - 8\pi T_{\mu\nu} \right] \sqrt{-g} \delta g^{\mu\nu} \mathrm{d}^4 x \, .
\tag{36}
$$

Then we enforce the action principle and equate $\delta S = 0$ to get the Einstein field equations:

$$
\underbrace{R_{\mu\nu} - \frac{1}{2} g_{\mu\nu} R}_{G_{\mu\nu} = G_{\nu\mu}} = 8\pi T_{\mu\nu} \, .
\tag{37}
$$

from which we get the desired conservation of the stress-energy tensor (see the text below eq. (A.12)):

$$
\nabla_\mu T^{\mu\nu} = 0 \, .
\tag{38}
$$

Note that we have ignored the cosmological constant $\Lambda$ throughout but the entire analysis goes through if we replace $\boxed{R \to (R - 2\Lambda)}$ to get the full Einstein field equations (this prescription of replacing $R$ with $(R - 2\Lambda)$ is in general a powerful heuristic of restoring the cosmological constant):

$$
R_{\mu\nu} - \frac{1}{2} g_{\mu\nu} R + g_{\mu\nu} \Lambda = 8\pi T_{\mu\nu} \, .
\tag{39}
$$

## 2.3 Prerequisites of Hamiltonian formulation of general relativity

We now set up the space where we shall be working. We consider a submanifold $\mathcal{N} \subset \mathcal{M}$ through the *embedding* $\tilde{\Phi} : \mathcal{N} \to \mathcal{M}$ (injective and structure preserving). In particular, $\tilde{\Phi} : \mathcal{N} \to \tilde{\Phi}(\mathcal{N})$ is a diffeomorphism where $\tilde{\Phi}(\mathcal{N}) \subset \mathcal{M}$ is a $k-$dimensional submanifold ($k < n$). We will identify $\mathcal{N}$ and $\tilde{\Phi}(\mathcal{N})$. This is shown in Fig.(1) [11].

We now assume that the spacetime $(\mathcal{M}, g_{\mu\nu})$ is *globally hyperbolic*[2] namely that its topology is $\mathbb{R} \times \Sigma$ where $\Sigma$ is an orientable $3-$dimensional manifold (see Fig.(2) [11]). Accordingly we

---

[2]A spacetime $\mathcal{M}$ is said to be globally hyperbolic if it admits a spacelike hypersurface $\Sigma$ (called the Cauchy surface) such that every timelike or null curve without end points intersects $\Sigma$ only once.

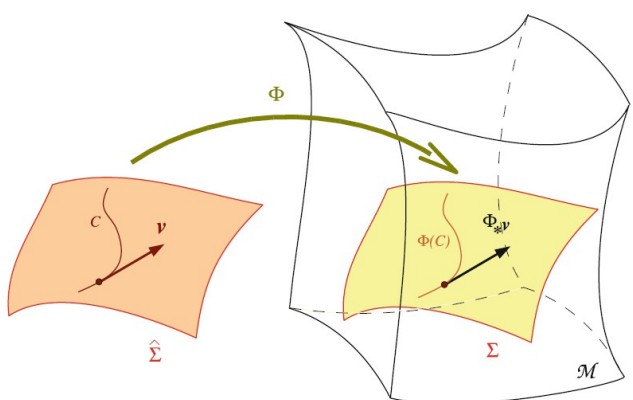

Figure 1: Embedding 3−D manifold in 4−D manifold [11].

can foliate the spacetime by 3−manifolds (hypersurfaces) $\Sigma_t$ ($t \in \mathbb{R}$) such that (we identify $\Sigma_t$ with $\{t\} \times \Sigma$):

$$\mathcal{M} = \bigcup_{t \in \mathbb{R}} \Sigma_t \,. \tag{40}$$

Then we assume the following about $\Sigma_t$:

(a) No two $\Sigma_t$ will intersect with each other.

(b) The initial hypersurface $\Sigma_{t=0}$ will encode the initial information giving rise to the spacetime as prescribed by the equations of motion.

(c) Hypersurfaces $\Sigma_t$ arise as level surfaces of a scalar function $t$ which will be interpreted as a *global function time*.

(d) All $\Sigma_t$ are *spacelike*.

As an aside, we are imposing the assumption (d) for our purposes but in general the foliation allows to have hypersurfaces $\Sigma$ of three types (recall our convention of the signature of the metric: $(-, +, +, +)$):

 (i) spacelike hypersurface if the induced 3−metric (defined below) is positive definite, i.e. signature is $(+, +, +)$ having a timelike normal vector,

 (ii) timelike hypersurface if the induced 3−metric is Lorentzian, i.e. signature is $(-, +, +)$ having a spacelike normal vector, and

 (iii) null hypersurface is the induced 3−metric is degenerate, i.e. signature is $(0, +, +)$.

We will always stick to the first type, namely a spacelike hypersurface with a timelike normal vector.

This construction allows us to define a *normal vector* $n^\mu$ on each of the spatial hypersurface $\Sigma_t$. This is shown in Fig. (2). We can interpret $n^\mu$ as the 4−velocity of a normal observer whose worldline is always orthogonal to $\Sigma_t$. Clearly $n^\mu$ is a *timelike vector* which we shall always take to be normalized. In our metric signature convention, this means $n^\mu n_\mu = -1$.

We have defined our hypersurface $\Sigma_t$ as that of a surface with constant $t$ where $t$ is a scalar field on $\mathcal{M}$. So the 1-form $\mathrm{d}t$ is normal to $\Sigma_t$ in the sense that every vector on $\Sigma_t$ has a vanishing inner product with $\mathrm{d}t$. Accordingly the metric dual of $\mathrm{d}t$, namely $\partial_\mu t$, is also normal to the hypersurface $\Sigma_t$ where $\partial_\mu t$ is timelike if $\Sigma_t$ is spacelike. Thus we see a resemblance between the structure of $\partial_\mu t$ and $n_\mu$. Indeed upto a normalization constant, we

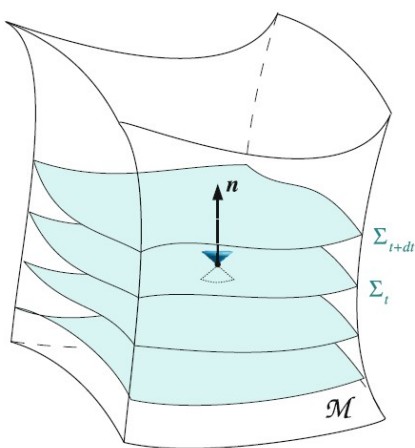

Figure 2: Foliation of globally hyperbolic spacetime $\mathcal{M}$ [11].

can write $n_\mu = \Omega \partial_\mu t$ where $\Omega = \Omega(x^\mu)$ is a normalization constant which is fixed by the condition $n^\mu n_\mu = -1$. Also $n^\mu = n_\mu = g^{\mu\nu} n_\mu n_\nu = g^{tt}\Omega^2$. Thus we have $\Omega = \pm\frac{1}{\sqrt{g^{00}}}$. We choose a negative $\Omega$ to allow $n^\mu$ to be a timelike vector and we thus get:

$$
\begin{aligned}
n_\mu &= -\frac{\delta^0_\mu}{\sqrt{-g^{00}}}\,, \\
n^\mu &= -\frac{g^{0\mu}}{\sqrt{-g^{00}}}\,.
\end{aligned}
\tag{41}
$$

Then the spacetime metric $g_{\mu\nu}$ (the 4−metric) *induces* a 3−dimensional Riemannian metric $\gamma_{ij}$ on $\Sigma_t$ such that $\gamma_{\mu\nu} = g_{\mu\nu} + n_\mu n_\nu \Leftrightarrow \gamma^{\mu\nu} = g^{\mu\nu} + n^\mu n^\nu$, where despite being a 3−D object, we have still used Greek indices for $\gamma_{ij}$ because we can regard it as an object living on spacetime. Any time Greek indices can be converted to Latin indices to get back the 3−dimensional results on spacelike hypersurfaces $\Sigma_t$. Then we get explicitly for the induced 3−metric:

$$
\gamma^\mu_\nu = \delta^\mu_\nu + n^\mu n_\nu = \left(\begin{array}{c|c} 0 & -\frac{g^{0j}}{g^{00}} \\ \hline 0^i & +\delta^i_j \end{array}\right).
\tag{42}
$$

This induced metric is also used as a *projector*. We have two types of projection:

(a) Spatial projection (spacelike):  given a tensor $T_{\mu\nu}$, its spatial part is given by $T^S_{\mu\nu} = \gamma^\alpha_\mu \gamma^\beta_\nu T_{\alpha\beta}$.

(b) Normal projection (timelike): *Normal projector $N^\mu_\nu$ is defined as* $N^\mu_\nu \equiv -n_\nu n^\mu = \delta^\mu_\nu - \gamma^\mu_\nu$.

Accordingly any vector $V^\mu$ can be decomposed into spatial and temporal parts as follows:

$$
V^\mu = \delta^\mu_\nu V^\nu = (\gamma^\mu_\nu + N^\mu_\nu)V^\nu = V^S + V^T\,.
\tag{43}
$$

Just like $g_{\mu\nu}$ on $\mathcal{M}$ defines a unique covariant derivative $\nabla_\mu$, the 3−metric $\gamma_{ij}$ defines in a unique way a covariant derivative $D_i$ (the Levi-Civita connection) on $\Sigma_t$. This can be taken to be torsion free and compatible with the metric in 3−D on each hypersurface $\Sigma_t$, just like the full 3+1−D case. Accordingly, in 3−D we have $\boxed{D_\mu \gamma_{\alpha\beta} = 0}$, just like in 3+1−D we have $\nabla_\mu g_{\alpha\beta} = 0$. The relation between the 3− and 4−covariant derivatives is given in eq. (B.4) in Appendix B.

The 3−metric then defines the 3−Christoffel symbols as:

$$^{(3)}\Gamma^{\mu}_{\alpha\nu} = \frac{1}{2}\gamma^{\mu\sigma}\left(\partial_{\alpha}\gamma_{\nu\sigma} + \partial_{\nu}\gamma_{\sigma\alpha} - \partial_{\sigma}\gamma_{\alpha\nu}\right).$$

(44)

Like 3+1−D, the covariant derivative in 3−D defines the *intrinsic curvature* of each spacelike hypersurface $\Sigma_t$ as follows:

$$\left[D_{\mu}, D_{\nu}\right]V^{\alpha} = {}^{(3)}R^{\alpha}_{\beta\mu\nu}V^{\beta},$$

(45)

where ${}^{(3)}R^{\alpha}_{\beta\mu\nu}n^{\beta} = 0$, Ricci tensor ${}^{(3)}R_{\alpha\beta} = {}^{(3)}R^{\mu}_{\alpha\mu\beta}$ and Ricci scalar ${}^{(3)}R = {}^{(3)}R_{\alpha\beta}\gamma^{\alpha\beta}$

But this only provides the information about the curvature intrinsic to the hypersurface and provides no information at all that how $\Sigma_t$ fits in $(\mathcal{M}, g_{\mu\nu})$. This is what is captured by *extrinsic curvature tensor $K_{\mu\nu}$* defined as:

$$K_{\mu\nu} \equiv -\gamma^{\alpha}_{\mu}\gamma^{\beta}_{\nu}\nabla_{\alpha}n_{\beta}.$$

(46)

The properties of the extrinsic curvature tensor $K_{\mu\nu}$ are:

(a) symmetric in $\mu$ and $\nu$ by construction,

(b) purely spatial by construction: $n^{\mu}K_{\mu\nu} = -\gamma^{\alpha}_{\mu}\gamma^{\beta}_{\nu}\frac{1}{2}\nabla_{\alpha}\left(n_{\beta}n^{\mu}\right) = 0$ where we made use of eq. (B.4), $D_{\mu}\gamma_{\alpha\beta} = 0$ and $n^{\mu}n_{\mu} = -1$ is just a constant,

(c) measures how the normal to the hypersurface changes from point to point, &

(d) also measures the rate at which the hypersurface deforms as it is carried along the normal, thereby capturing intuitive notion of how the curvature varies from one hypersurface to the next.

There is an associated concept known as the *acceleration of a foliation $a_{\mu}$* that, as the name suggests, captures how rapidly the curvature changes from one hypersurface to the next. It is defined as:

$$a_{\mu} \equiv n^{\nu}\nabla_{\nu}n_{\mu}.$$

(47)

This allows us to express the extrinsic curvature tensor in two other equivalent ways than eq. (46). They are:

(a) $\boxed{K_{\mu\nu} = -\nabla_{\mu}n_{\nu} - n_{\mu}a_{\nu}.}$

Proof: We realize $n^{\mu}\nabla_{\nu}n_{\mu} = \frac{1}{2}\nabla_{\nu}\underbrace{\left(n^{\mu}n_{\mu}\right)}_{=-1} = 0$. Thus we have from the definition in eq. (46) that:

$$\begin{aligned}
K_{\mu\nu} &= -\gamma^{\alpha}_{\mu}\gamma^{\beta}_{\nu}\nabla_{\alpha}n_{\beta} = -\left(\delta^{\alpha}_{\mu} + n_{\mu}n^{\alpha}\right)\left(\delta^{\beta}_{\nu} + n_{\nu}n^{\beta}\right)\nabla_{\alpha}n_{\beta}\\
&= -\left(\delta^{\alpha}_{\mu} + n_{\mu}n^{\alpha}\right)\left(\delta^{\beta}_{\nu}\right)\nabla_{\alpha}n_{\beta}\\
&= -\nabla_{\mu}n_{\nu} - n_{\mu}a_{\nu}.
\end{aligned}$$

(b) $\boxed{K_{\mu\nu} = -\frac{1}{2}\mathcal{L}_{n}\gamma_{\mu\nu}.}$

Proof: We start from the RHS and use $\mathcal{L}_n g_{\mu\nu} = 2\nabla_{(\mu} n_{\nu)}$ to get:

$$\mathcal{L}_n \gamma_{\mu\nu} = \mathcal{L}_n \left( g_{\mu\nu} + n_\mu n_\nu \right) = 2\nabla_{(\mu} n_{\nu)} + n_\mu \mathcal{L}_n n_\nu + n_\nu \mathcal{L}_n n_\mu$$
$$= 2 \left[ \nabla_{(\mu} n_{\nu)} + n_{(\mu} n_{\nu)} \right] = -2K_{\mu\nu}$$
$$\Rightarrow K_{\mu\nu} = -\frac{1}{2} \mathcal{L}_n \gamma_{\mu\nu}.$$

Clearly either of these two definitions also satisfy the aforementioned properties of $K_{\mu\nu}$ and indeed in the literature, sometimes the definition of $K_{\mu\nu}$ is taken to be either of these two instead of eq. (46).

Just like the Ricci scalar in $3+1-$D, we have something known as the *mean curvature* or *extrinsic curvature scalar*, defined as (keeping in mind that $K_{\mu\nu}$ and thus $K$ are $3-$objects living on $\Sigma_t$):

$$K \equiv g^{\mu\nu} K_{\mu\nu} = \gamma^{\mu\nu} K_{\mu\nu}. \tag{48}$$

It can be shown to be equivalent to $K = -\nabla_\mu n^\mu = -\mathcal{L}_n \left( \ln \left( \det(\gamma) \right) \right)$. The physical meaning captured by $K$ is that it measures the fractional change of $3-$dimensional volume along the normal $n^\mu$ from one spacelike hypersurface to the next.

There is a note to be made. Even though the indices used are Greek for the $3-$metric, it is understood that only the spatial components are non-trivial. This is a rule in general that if Greek indices are used for any mathematical object which are $3-$objects living on a spacelike hypersurface $\Sigma_t$, only the spatial components matter and we can safely replace all Greek indices with Latin ones. Accordingly, for example, the covariant derivative induced by $\gamma_{\mu\nu}$ is denoted by $D_\mu$ that satisfies $D_\alpha \gamma_{\mu\nu} = 0$ simply means $D_i \gamma_{ab} = 0$. Thus, $\{\gamma_{\mu\nu}, D_\mu, {}^{(3)}\Gamma^\lambda_{\mu\nu}, {}^{(3)}R^\mu_{\nu\alpha\beta}, K_{\mu\nu}, K\}$ are $3-$objects (as their respective contractions with the normal vector $n^\mu$ are zero), living on $\Sigma_t$ and accordingly the Greek indices can be replaced with Latin ones as only the spatial components are relevant.

The final ingredient that is required as a mathematical pre-requisite are the famous Gauss, Codazzi, Mainardi relations. Without them, the $3+1-$decomposition cannot be done and this is the foundation of the Arnowitt-Deser-Misner (ADM) formalism of general relativity. They have been proven in complete detail in Appendix B. Here we list the final results.

- Gauss Identities:

    - Gauss relation:

    $$\gamma^\mu_\alpha \gamma^\nu_\beta \gamma^\gamma_\rho \gamma^\sigma_\delta {}^{(4)}R^\rho_{\sigma\mu\nu} = {}^{(3)}R^\gamma_{\delta\alpha\beta} + K^\gamma_\alpha K_{\delta\beta} - K^\gamma_\beta K_{\alpha\delta}. \tag{49}$$

    - Contracted Gauss relation

    $$\gamma^\mu_\alpha \gamma^\nu_\beta {}^{(4)}R_{\mu\nu} + \gamma_{\alpha\mu} n^\nu \gamma^\rho_\beta n^\sigma {}^{(4)}R^\mu_{\nu\rho\sigma} = {}^{(3)}R_{\alpha\beta} + KK_{\alpha\beta} - K_{\alpha\mu} K^\mu_\beta. \tag{50}$$

    - Scalar Gauss relation (or generalized *Theorema Egregium*):

    $${}^{(4)}R + 2{}^{(4)}R_{\mu\nu} n^\mu n^\nu = {}^{(3)}R + K^2 - K_{ij} K^{ij}. \tag{51}$$

The original Theorem Egregium proposed by Gauss is a special case of this result and is derived using this result in Appendix B.

- Codazzi-Mainardi Identities:

  – Codazzi-Mainardi relation:

  $$\gamma^\gamma_\rho n^\sigma \gamma^\mu_\alpha \gamma^\nu_\beta {}^{(4)}R^\rho_{\sigma\mu\nu} = D_\beta K^\gamma_\alpha - D_\alpha K^\gamma_\beta \, . \tag{52}$$

  – Contracted Codazzi relation:

  $$\gamma^\mu_\alpha n^\nu {}^{(4)}R_{\mu\nu} = D_\alpha K - D_\mu K^\mu_\alpha \, . \tag{53}$$

This completes our requirement of all the required mathematical machinery and we are now in a position to decompose spacetime into spatial and temporal parts.

# 3 Hamiltonian formulation of general relativity

In this chapter, we develop the methodology of decomposing general relativity, which is a Lorentz invariant theory, into temporal and spatial components. In doing so we realize that general relativity *apriori* does not admit a natural parametrization for time and there always remains an arbitrary choice for the time coordinate. But having such a split of "time+space" enables us to deal with time-varying tensor fields on spatial hypersurfaces. This allows for the formulation of the so-called *Cauchy problem* (the initial value problem) in general relativity [1–3]. In Section 3.1, we discuss in detail the setup required to do so. One of the crucial elements of this section is to introduce 4 new functions, namely the lapse function $N$ and the shift functions $\vec{N}$ which are functions of spacetime. Then the entire $3+1-$decomposition of the globally hyperbolic spacetime manifold $\mathcal{M}$, based on the mathematical machinery developed in Chapter 2.3 as well as these four new functions, are detailed. In Section 3.2, we finally develop the canonical formulation and derive the Hamiltonian of general relativity based on the works of Arnowitt-Deser-Misner [4,5]. As a reminder, we will only be considering bulk terms and will throughout ignore the boundary terms. Accordingly the discussions on the ADM mass & momentum are excluded from this treatment. In Section 3.3, we discuss the Hypersurface Deformation Algebra (HDA), or the Dirac algebra, which will provide us the insight into the Hamiltonian and diffeomorphism constraints that we derive in Section 3.2. The HDA is sometimes taken as an independent starting point to develop general relativity [6,7,9,10]. In the Minkowski limit, the HDA boils down to the well-known Poincaré algebra.

## 3.1 3+1-decomposition of spacetime

As seen in Chapter 2.3, we do the dimensional splitting between time and space by assuming the spacetime manifold $\mathcal{M}$ to be globally hyperbolic and endowed with a metric $g_{\mu\nu}$. As shown in eq. (40), the spacetime is foliated into spacelike hypersurfaces $\Sigma_t$ on which a $3-$metric $\gamma_{\mu\nu}$ (or $\gamma_{ij}$) is induced by the $4-$metric $g_{\mu\nu}$. But despite the machinery developed in Chapter 2.3, this is not sufficient for the $3+1-$formalism to be completely equivalent to the $4-$geometry of the full spacetime. We still need to specify the geometry between the hypersurfaces. This is done by introducing four new variables: the lapse function $N$ and the shift functions $\vec{N}$ which provide the additional information required for a complete description of the spacetime manifold $\mathcal{M}$.

## Lapse & shift functions

The definition of the lapse function $N$ is:

$$N \equiv \frac{1}{\sqrt{-g^{00}}}, \tag{54}$$

and that of shift function $\vec{N}$ is:

$$N^i \equiv N^2 g^{0i}. \tag{55}$$

There is another object known as the *normal evolution vector*, that will be useful later and is defined as:

$$m^\mu \equiv N n^\mu, \tag{56}$$

where $n^\mu$ is the normal vector defined in eq. (41). Physically it shows the evolution from one hypersurface to another as shown in Fig. (3) [12].

The claim is: $\{N, N^i, \gamma_{ij}\}$ *completely determine the spacetime geometry* which we will prove it below in the rest of this Section 3.1. Before proceeding to show this, we first need to develop an intuition about the lapse & shift functions and what they mean physically. The geometrical interpretation of these functions is shown in Fig. (3) [12]. $n^\mu$ is the normal vector to the hypersurface and thus $n^\mu N dt \equiv m^\mu dt$ leads "$a$" to the next adjacent hypersurface $\Sigma_{t+dt}$ at point "$b$" and then shift functions measure the difference of coordinates on the hypersurface $\Sigma_{t+dt}$ between "$b$" and the time evolution point of "$a$", namely "$c$". There is another interpretation of the lapse function: if we consider an observer moving with the 4−velocity $n^\mu$, then the elapsed proper time $\delta\tau$ between two events as measured by this *normal observer* is given by $\delta\tau = N\delta t$ ($t$ is the coordinate time) which simply means that the lapse function $N$ associates an infinitesimal interval of coordinate time $t$ to the proper time $\tau$ as measured by a normal observer whose world lines are orthogonal to $\Sigma_t$. Thus the lapse and the shift functions tell how to relate coordinates between two hypersurfaces where the lapse function measures the proper time to go from one hypersurface to the next one and the shift functions measure changes in the spatial coordinates on the same hypersurface. In this way, these 4 functions capture the geometry in between the hypersurfaces and coupled with the 3−metric $\gamma_{ij}$ (i.e. the set $\{N, \vec{N}, \gamma_{ij}\}$) completely determine the spacetime geometry of $\mathcal{M}$ (which is completely captured by the 4−metric $g_{\mu\nu}$). We will now make this statement more precise.

## 4−metric & its inverse

Using the definitions of $N$ and $\vec{N}$ from eqs. (54, 55), we already some of the components of the inverse metric $g^{\mu\nu}$. Then we can write the whole matrix as:

$$g^{\mu\nu} = \left( \begin{array}{c|c} -\frac{1}{N^2} & \frac{N^j}{N^2} \\ \hline \frac{N^i}{N^2} & \frac{\Omega^{ij}}{N^2} \end{array} \right), \tag{57}$$

where $\Omega^{ij}$ are the unknown functions which will determine the inverse 3−metric. For the metric $g_{\mu\nu}$, we take an ansatz by keeping in mind that the only knowledge we already have in advance in that the 3−metric $g_{ij}$ must be the $\gamma_{ij}$:

$$g_{\mu\nu} = \left( \begin{array}{c|c} A & B_j \\ \hline B_i & \gamma_{ij} \end{array} \right), \tag{58}$$

where $A, B_i$ ($B_j$ is the same as $B_i$ with a different index as $g_{\mu\nu}$ is symmetric in its indices) are unknown functions.

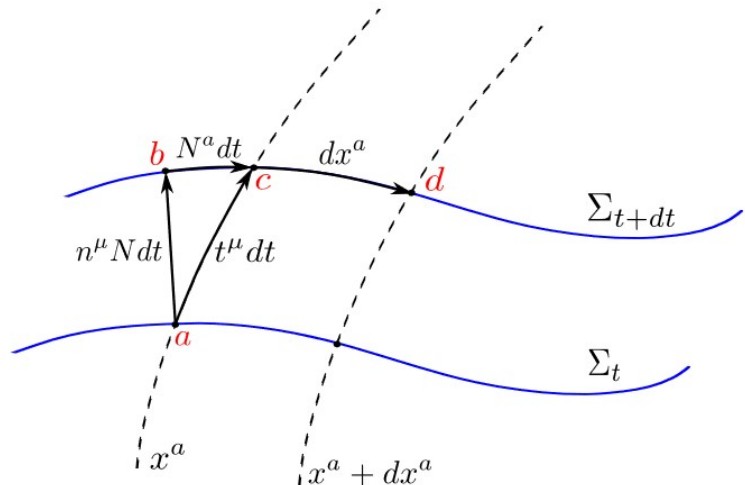

Figure 3: Geometric interpretation of lapse and shift functions [12].

Thus we solve for $A, B_i, \Omega$ using the identity $g_{\mu\rho} g^{\rho\nu} = \delta^{\nu}{}_{\mu}$ as follows:

$$
\begin{aligned}
g_{i\rho} g^{\rho 0} = \frac{1}{N^2} \left( -B_i + \gamma_{ij} N^j \right) = 0 &\implies B_i = \gamma_{ik} N^k, \\
g_{0\rho} g^{\rho 0} = \frac{1}{N^2} \left( -A + \gamma_{ik} N^k N^i \right) = 1 &\implies A = \gamma_{ik} N^k N^i - N^2, \\
g_{i\rho} g^{\rho j} = \frac{1}{N^2} \gamma_{ik} \left( N^k N^j + \Omega^{kj} \right) = \delta_i{}^j &\implies \Omega^{ij} = N^2 \gamma^{ij} - N^i N^j.
\end{aligned}
\tag{59}
$$

Thus we have finally:

$$
g_{\mu\nu} = \left( \begin{array}{c|c} N_i N^i - N^2 & N_j \\ \hline N_i & \gamma_{ij} \end{array} \right), \qquad
g^{\mu\nu} = \left( \begin{array}{c|c} -\frac{1}{N^2} & \frac{N^j}{N^2} \\ \hline \frac{N^2}{N^2} & \gamma^{ij} - \frac{N^i N^j}{N^2} \end{array} \right).
\tag{60}
$$

Then if we take $\det(g_{\mu\nu}) = g$ (where $g < 0$ due to signature of 4$-$metric being $(-, +, +, +)$) and $\det(\gamma_{ij}) = \gamma$ (where $\gamma > 0$ on the spacelike hypersurface $\Sigma_t$ due to signature being $(+, +, +)$), then they are related as (upon direct computation):

$$
\sqrt{-g} = N \sqrt{\gamma}.
\tag{61}
$$

Finally, using eq. (41) and definitions in eqs. (54, 55), we can express the normal vector as:

$$
\begin{aligned}
n_\mu &= (-N, 0, 0, 0), \\
n^\mu &= \left( \frac{1}{N}, -\frac{N^i}{N} \right).
\end{aligned}
\tag{62}
$$

Before we start to decompose the 4$-$Riemannian curvature in $3 + 1-$form, we need to express the remaining 3$-$objects introduced in Chapter 2.3 and in this chapter in terms of lapse and shift functions. We state the final results here whose detailed proofs can be found in Appendix C.

$$
\begin{aligned}
a_\mu &= D_\mu \ln(N), & \mathcal{L}_m \gamma^\mu{}_\nu &= 0, \\
\nabla_\mu n_\nu &= -K_{\mu\nu} - n_\mu D_\nu \ln(N), & \nabla_\mu m^\nu &= -N K_\mu^\nu - n_\mu D^\nu N + n^\nu \nabla_\mu N, \\
\mathcal{L}_m \gamma^{\mu\nu} &= 2 N K^{\mu\nu}, & \mathcal{L}_m \gamma_{\mu\nu} &= -2 N K_{\mu\nu}, \\
\mathcal{L}_m K_{\mu\nu} = N \gamma_\mu^\alpha \gamma_\nu^\beta \nabla_n K_{\alpha\beta} - 2 N K_{\mu\rho} K_\nu^\rho, & K_{\mu\nu} &= \frac{1}{2N} \left[ D_\mu N_\nu + D_\nu N_\mu - \dot{\gamma}_{\mu\nu} \right].
\end{aligned}
\tag{63}
$$

The last equation on the right, upon rearranging, gives the equation of motion for 3−metric $\gamma_{\mu\nu}$ in terms of 3−objects and govern the evolution of 3−metric on a spacelike hypersurface $\Sigma_t$.

There is an additional important result which allows us to deduce a crucial corollary. The result is obtained from the Lie derivative of $K_{\mu\nu}$ to get the Lie derivative of $K$ along $m^\mu$, given as follows:

$$\boxed{\mathcal{L}_m K = N \mathcal{L}_n K = N\gamma^{\mu\nu}\nabla_n K_{\mu\nu}\,.} \tag{64}$$

Proof: Consider the LHS and make use of eq. (63):

$$
\begin{aligned}
\mathcal{L}_m K &= \mathcal{L}_m\left(\gamma^{ij}K_{ij}\right)\\
&= \left(\mathcal{L}_m\gamma^{ij}\right)K_{ij} + \left(\mathcal{L}_m K_{ij}\right)\gamma^{ij}\\
&= 2NK^{ij}K_{ij} + \left(N\gamma_i^a\gamma_j^b\nabla_n K_{ab} - 2NK_{ic}K_j^c\right)\gamma^{ij}\\
&= N\gamma_i^a\gamma_j^b\nabla_n K_{ab} - 2NK_{ic}K_j^c\gamma^{ij}\\
&= N\gamma^{ij}\nabla_n K_{ij} = \gamma^{ij}\nabla_m K_{ij}\,.
\end{aligned}
$$

Corollary: We know in general that $\gamma_{ij}$ and $\nabla_n$ (or $\nabla_m$) do not commute (as 3−derivatives $D_i$ are compatible with 3−metric, not 4−derivatives). But this identity suggests that we can replace $\gamma^{ij}\nabla_n K_{ij}$ with $\mathcal{L}_n K$. Thus the corollary we have is that for a scalar field $f$ and a vector $\vec{X}$, $\mathcal{L}_X f = \nabla_X f$ and thus we are able to write $\gamma^{ij}\nabla_n K_{ij} = \mathcal{L}_n K$. Thus contraction with $\gamma^{ij}$ commutes with $\nabla$ even though $\nabla_\lambda\gamma^{ij} \neq 0$ due to the presence of a 3−object $K$. This also means that $\gamma^{ij}\nabla_m K_{ij} = \mathcal{L}_m K$.

Now we are in a position to $3+1$−decompose the Riemann 4−curvature (LHS of Einstein field equations) and 4−stress-energy-momentum tensor (RHS of Einstein field equations) following which we will decompose the Einstein field equations in $3+1$−variables. We will only present the results here and the derivations of all the results presented can be found in Appendix D.

**Projection of 4−curvature**

With the aforementioned complete set of results obtained, we can proceed to decompose the Riemann curvature tensor. We start with the definition of the 4−Riemann tensor when applied to normal vector $n^\mu$, namely:

$$\left[\nabla_\mu, \nabla_\nu\right]n^\rho = {}^{(4)}R^\rho_{\sigma\mu\nu}n^\sigma\,. \tag{65}$$

We now project this twice onto the hypersurface $\Sigma_t$ using the induced 3−metric and once along the normal $n^\mu$ to get:

$$\boxed{\gamma_{\rho\alpha}\gamma_\beta^\mu {}^{(4)}R^\rho_{\sigma\mu\nu}n^\sigma n^\nu = -K_{\alpha\lambda}K_\beta^\lambda + \gamma_\alpha^\mu\gamma_\beta^\nu\nabla_n K_{\mu\nu} + \frac{1}{N}D_\alpha D_\beta N\,.} \tag{66}$$

Similarly we do the same procedure of projecting the Ricci tensor and the Ricci scalar to get them in terms of the $3+1$−variables:

$$\boxed{\gamma_\mu^\alpha\gamma_\nu^\beta {}^{(4)}R_{\alpha\beta} = {}^{(3)}R_{\mu\nu} + KK_{\mu\nu} - \gamma_\mu^\alpha\gamma^\beta{}_\nu\nabla_n K_{\alpha\beta} - \frac{1}{N}D_\mu D_\nu N\,,} \tag{67}$$

and

$$\boxed{{}^{(4)}R = {}^{(3)}R + K^2 + K^{ij}K_{ij} - 2\nabla_n K - \frac{2}{N}D^i D_i N\,.} \tag{68}$$

Thus we have successfully $(3+1)$—decomposed spacetime curvature in terms of the 3—dimensional objects, namely $K_{\mu\nu}$ (and the associated $K$), $\gamma_{\mu\nu}$, the lapse function $N$, the shift functions $\vec{N}$ and 3—Riemann tensor of $\Sigma_t$. See Appendix D for the detailed proofs.

**Projection of 4-stress-energy-momentum tensor**

Once the curvature tensor has been decomposed, this finally helps us to project Einstein field equations into $3+1$—formalism. Without the cosmological constant, Einstein field equations in natural units read as $^{(4)}R_{\mu\nu} - \frac{1}{2}g_{\mu\nu}{}^{(4)}R = 8\pi T_{\mu\nu}$, where $T_{\mu\nu}$ is the *stress-energy-momentum tensor* which is symmetric in its indices. We have already projected the LHS of this equation and now we need to project the RHS, namely $T_{\mu\nu}$, into *energy density* (projected twice along the normal, as measured by a normal observer moving with 4—velocity $n^\mu$), *momentum density* (projected once along the normal and once along the hypersurface, making it tangent to $\Sigma_t$) and *stress tensor* (projected twice along the hypersurface) as follows:

$$
\begin{aligned}
E &\equiv T_{\mu\nu} n^\mu n^\nu && \text{(Energy Density)}, \\
p_\alpha &\equiv -T_{\mu\nu} n^\mu \gamma^\nu_\alpha && \text{(Momentum Density)}, \\
S_{\alpha\beta} &\equiv T_{\mu\nu} \gamma^\mu_\alpha \gamma^\nu_\beta && \text{(Stress Tensor)}.
\end{aligned}
\tag{69}
$$

Here we can define *stress scalar* as $\boxed{S \equiv \gamma^{\alpha\beta} S_{\alpha\beta}}$ and *stress-energy-momentum scalar* as $T \equiv g^{\mu\nu} T_{\mu\nu}$. Then we see that $S, T$ and $E$ are not all independent but related by:

$$
\begin{aligned}
T &= T_{\mu\nu} g^{\mu\nu} = T_{\mu\nu} (\gamma^{\mu\nu} - n^\mu n^\nu) \\
&\Rightarrow \boxed{T = S - E}.
\end{aligned}
\tag{70}
$$

With these projections of $T_{\mu\nu}$ taken, we have projected the LHS as well as the RHS of the Einstein field equations separately and it's time to combine them.

**Projection of Einstein field equations**

We finally combine the results obtained in the last two subsections to finally project the Einstein field equations in terms of $3+1$—variables. Since the equations involve rank 2 tensors, we can take two projections corresponding to each indices and we have three possibilities for doing so: (i) projecting twice along the spatial hypersurface $\Sigma_t$, (ii) projecting twice along the normal vector $n^\mu$, & (iii) mixed projections involving (once) along $\Sigma_t$ as well as along $n^\mu$. We perform the calculations for all three cases and we get the final results as (derivations can be found in Appendix D):

- Both Projections along $\Sigma_t$: This is purely spatial projection:

$$
\boxed{{}^{(3)}R_{\alpha\beta} - 2K_{\alpha\lambda}K^\lambda_\beta + KK_{\alpha\beta} - \frac{1}{N}\left[\mathcal{L}_m K_{\alpha\beta} + D_\alpha D_\beta N\right] = 8\pi\left[S_{\alpha\beta} - \frac{1}{2}\gamma_{\alpha\beta}(S-E)\right]}.
\tag{71}
$$

- Both Projections along $n^\mu$: This is purely temporal projection (also called the **Hamiltonian constraint**):

$$
\boxed{{}^{(3)}R - K_{ij}K^{ij} + K^2 = 16\pi E}.
\tag{72}
$$

- Mixed Projections along $\Sigma_t$ and $n^\mu$ (also called the **momentum constraint**):

$$
\boxed{D_\beta K^\beta_\alpha - D_\alpha K = 8\pi p_\alpha}.
\tag{73}
$$

These three equations collectively contain the same amount of information as the covariant form of the Einstein field equations: $^{(4)}R_{\mu\nu} - \frac{1}{2}g_{\mu\nu}{}^{(4)}R = 8\pi T_{\mu\nu}$. We know that a symmetric matrix $A$ of size $n \times n$ has $\frac{n(n+1)}{2}$ independent elements, therefore the Einstein field equations in covariant form is a set of 16 equations out of which 6 are dependent leaving us with 10 independent equations (since it is symmetric in $\mu$ & $\nu$ where they run over spacetime components $\{0, 1, 2, 3, 4\}$, so $n = 4$). These 10 independent equations solve for the exactly 10 independent components of the metric $g_{\mu\nu}$ (as it is symmetric in its spacetime indices as well). This is true in terms of $3 + 1-$variables too: eq. (71) is symmetric in the indices $\alpha$ & $\beta$ where the indices are spatial (thus $n = 3$ & it contains 6 independent equations), eq. (72) is a scalar equation (so 1 independent equation), and eq. (73) is a vector equation with one free spatial index $\alpha$ (therefore 3 independent equations), putting the total count in $3 + 1-$variables to 10 independent equations just like the covariant form.

Now we are in a position to finally introduce the formalism on which this entire work is based upon and that is the *Arnowitt-Deser-Misner (ADM)* formulation of general relativity.

## 3.2  ADM formalism of general relativity

In the canonical Hamiltonian formalism, just like the case of classical mechanics, time holds a privileged position among the coordinates $x^\mu$ and the time evolution of tensor fields on spacelike hypersurfaces are governed by Hamilton's equations of motion which are first-order differential equations in time derivatives. The advantage of this approach is that it allows for a clear formulation of the initial value problem (also called the Cauchy problem). But it is significantly difficult to obtain a Hamiltonian picture because the metric $g_{\mu\nu}$ contain some redundancies in the covariant approach to general relativity. Capturing those redundancies can be tricky in the Hamiltonian approach. Moreover we know that Hamilton's equations of motion are closely connected to Poisson brackets which in turn is closely related to commutation relations in quantum mechanics. Therefore, the Hamiltonian formalism becomes a pre-cursor in the grand attempt to canonically quantize gravity. But we need to first identify the unconstrained canonical variables in order to be able to write down their Poisson brackets and this identification of unconstrained canonical variables from the total set of variables can require significant effort. The Hamiltonian formalism we are interested in is known as the *Arnowitt-Deser-Misner (ADM)* formulation of general relativity and we will develop it in this section based on the entire mathematical machinery built in Chapters 2.3 & 3.1. The structure will be as follows: we will first derive the Einstein-Hilbert action in $3 + 1-$variables, then proceed to find conjugate momenta to dynamical variables in order to write down the Hamiltonian of general relativity, and finally evaluate the equations of motion from this Hamiltonian. In principle, they should contain the same amount of information as the Einstein field equations in covariant form. *We only focus on the vacuum case (pure gravity) in the bulk.*

**Einstein-Hilbert action in $3 + 1-$variables**

We derive this using the projection of $4-$curvature (eq. (68)) but an alternate derivation using the scalar Gauss relation (eq. (B.16)) is possible and is done in Appendix E.

The Einstein-Hilbert action for pure gravity without the cosmological constant is given by:

$$S_H = \frac{1}{16\pi} \int {}^{(4)}R \sqrt{-g}\, d^4x\,. \tag{74}$$

Now we already know the decomposition of $^{(4)}R$ from eq. (68) as well as for $\sqrt{-g}$ from

eq. (61). Also $d^4x = dt\,d^3x$. Thus we have:

$$\Rightarrow S_H = \frac{1}{16\pi} \int_{t_1}^{t_2} dt \int_{\Sigma_t} d^3x N\sqrt{\gamma} \left[ {}^{(3)}R + K^2 + K^{ij}K_{ij} - 2\nabla_n K - \frac{2}{N} D^i D_i N \right]. \tag{75}$$

But the last two terms contain pure divergences which can be ignored (since we are only interested in the bulk), as we will show now. Just like 4−divergence in terms of 4−covariant derivative is given by eq. (A.4), we have a similar result in $3+1$−variables for a scalar function $f$:

$$D_i D^i f = \frac{1}{\sqrt{\gamma}} \frac{\partial}{\partial x^i} \left( \sqrt{\gamma} \frac{\partial f}{\partial x_i} \right). \tag{76}$$

We use this relation to simplify:

$$\sqrt{\gamma} D_i D^i N = \partial_i \left( \sqrt{\gamma} \partial^i N \right). \tag{77}$$

Similarly, we make use of the eq. (A.3), reproduced here for convenience:

$$\nabla_\mu V^\mu = \frac{1}{\sqrt{-g}} \partial_\alpha \left( \sqrt{-g} V^\alpha \right). \tag{78}$$

Then we use $\sqrt{-g} = N\sqrt{\gamma}$ (eq. (61)), substitute $V^\alpha = Kn^\alpha$ and recall $K = -\nabla_\alpha n^\alpha$ (below eq. (48)) to get:

$$\begin{aligned}
\partial_\alpha \left( \sqrt{-g} V^\alpha \right) &= N\sqrt{\gamma} \nabla_\alpha (Kn^\alpha) \\
&= N\sqrt{\gamma} \underbrace{(\nabla_\alpha n^\alpha)}_{=-K} K + N\sqrt{\gamma} n^\alpha (\nabla_\alpha K) \\
\Rightarrow N\sqrt{\gamma} n^\alpha (\nabla_\alpha K) &= \partial_\alpha \left( \sqrt{-g} V^\alpha \right) + N\sqrt{\gamma} K^2 \\
\Rightarrow N\sqrt{\gamma} \nabla_n K &= \partial_\alpha \left( \sqrt{-g} V^\alpha \right) + N\sqrt{\gamma} K^2.
\end{aligned} \tag{79}$$

Thus we plug eqs. (77, 79) into eq. (75) and read off the Lagrangian density $\mathcal{L}_H$ (since $S_H = \int_{t_1}^{t_2} dt \int_{\Sigma_t} d^3x \mathcal{L}_H$) to get:

$$\mathcal{L}_H = \frac{1}{16\pi} \left[ \left( {}^{(3)}R - K^2 + K^{ij}K_{ij} \right) N\sqrt{\gamma} - 2 \left( \partial_i \left( \sqrt{\gamma} \partial^i N \right) + \partial_\alpha \left( \sqrt{\gamma} N K n^\alpha \right) \right) \right]. \tag{80}$$

Ignoring the total diverges (boundary terms), we finally get the Einstein-Hilbert action for pure gravity in $3+1$−variables (also known as the *ADM action*) where the pre-factor $\frac{1}{16\pi}$ is ignored without loss of generality:

$$\boxed{S_H = \int_{t_1}^{t_2} dt \int_{\Sigma_t} d^3x N\sqrt{\gamma} \left( {}^{(3)}R - K^2 + K^{ij}K_{ij} \right).} \tag{81}$$

An alternate derivation is provided in Appendix E.

**Hamiltonian Formalism**

From eq. (81), we can read off the Lagrangian density:

$$\mathcal{L}_H = N\sqrt{\gamma} \left( {}^{(3)}R - K^2 + K^{ij}K_{ij} \right), \tag{82}$$

and the Lagrangian is given by $L = \int d^3x \mathcal{L}_H$ which in turn gives the action $S_H = \int dt L$.

The first observation to make is that $S_H$ (or consequently $\mathcal{L}_H$) depends on the set $\{\gamma, \dot{\gamma}_{ij}, N, \vec{N}, \partial_i N, \partial_i \vec{N}\}$. But it does not depend on $\{\dot{N}, \dot{\vec{N}}\}$ which, as shown in Appendix F,

gets translated into the fact that $N$ & $\vec{N}$ serve as Lagrange multipliers (& thus are not dynamical variables). However, on a first glance, it is logical to assume then that $\{\gamma, N, \vec{N}\}$ and their conjugate momenta (defined below) are the dynamical variables of the system but as we will find out, only $\gamma_{ij}$ and its conjugate momenta (denoted by $\pi^{ij}$) are dynamical variables leading to equations of motion while $\{N, \vec{N}\}$ are Lagrange multipliers (& thus are not dynamical variables) leading to constraints relations. Readers are directed to Appendix F for a mathematically detailed discussion of this.

As a quick refresher, in classical mechanics having the Lagrangian $L = \int d^3x \mathcal{L}$ where $\mathcal{L} = \mathcal{L}(q, \dot{q})$ ($q = \{q_i\}$ for $i = 1, 2, 3, ..., n$ is the generalized coordinate) has canonical momenta corresponding to $q_i$ defined as $\pi^i = \frac{\partial \mathcal{L}}{\partial q_i}$. Thus the Legendre transformation of $\mathcal{L}$ gives the Hamiltonian $H$ as: $H(q, \pi) = \sum_i \pi^i \dot{q}_i - L$. We are going to have the same approach here where we will find the conjugate momenta corresponding to $\{\gamma, \dot{\gamma}_{ij}, N, \vec{N}\}$ and then take the Legendre transform of $\mathcal{L}_H$ (eq. 82).

The simplest are the conjugate momenta corresponding to $N$ and $\vec{N}$:

$$\boxed{\begin{aligned} \pi_N &= \frac{\partial \mathcal{L}_H}{\partial \dot{N}} = 0, \\ \pi_{N^i} &= \frac{\partial \mathcal{L}_H}{\partial \dot{N}^i} = 0, \end{aligned}} \tag{83}$$

because, as discussed above, $\mathcal{L}_H$ is independent of $\dot{N}$ and $\dot{\vec{N}}$.

Next we need to evaluate the conjugate momenta $\pi^{ij}$ conjugate to the components of $\gamma_{ij}$ as $\pi^{ij} = \frac{\partial \mathcal{L}_H}{\partial \gamma_{ij}}$. $\pi^{ij}$ contains *six independent components* and is *symmetric* in its indices. To evaluate this, we first note that $\gamma$ and $\dot{\gamma}$ are taken as independent variables, much like what we do in classical mechanics, and the set $\{\gamma_{ij}, \pi^{ij}, \dot{\gamma}_{ij}, \dot{\pi}^{ij}\}$ is taken as an independent set of variables. Next we realize that in the definition of 3−Ricci scalar $^{(3)}R$, we just have 3−metric appearing and not its derivatives, thus $\frac{\partial^{(3)}R}{\partial \dot{\gamma}_{ij}} = 0$. Finally we make use of the relation in eq. (63), namely $K_{\mu\nu} = \frac{1}{2N}\left[D_\mu N_\nu + D_\nu N_\mu - \dot{\gamma}_{\mu\nu}\right]$, to get:

$$\frac{\partial K_{ab}}{\partial \dot{\gamma}_{ij}} = -\frac{1}{2N}\delta_a^i \delta_b^j. \tag{84}$$

Then we are in a position to finally explicitly evaluate $\pi^{ij}$ as follows:

$$\begin{aligned} \pi^{ij} &= \frac{\partial \mathcal{L}_H}{\partial \gamma_{ij}} = \frac{\partial}{\partial \dot{\gamma}_{ij}}\left[N\sqrt{\gamma}\left(^{(3)}R - K^2 + K^{ab}K_{ab}\right)\right] \\ &= N\sqrt{\gamma}\frac{\partial}{\partial \dot{\gamma}_{ij}}\left[-K^2 + K^{ij}K_{ij}\right] = N\sqrt{\gamma}\frac{\partial}{\partial \dot{\gamma}_{ij}}\left[-2K\frac{\partial K}{\partial \dot{\gamma}_{ij}} + 2K^{ab}\frac{\partial K_{ab}}{\partial \dot{\gamma}_{ij}}\right] \\ &= N\sqrt{\gamma}\frac{\partial}{\partial \dot{\gamma}_{ij}}\left[-2K\frac{\partial\left(\gamma^{ab}K_{ab}\right)}{\partial \dot{\gamma}_{ij}} + 2K^{ab}\frac{\partial K_{ab}}{\partial \dot{\gamma}_{ij}}\right] \\ &= N\sqrt{\gamma}\frac{\partial}{\partial \dot{\gamma}_{ij}}\left[-2K\gamma^{ab}\frac{\partial\left(K_{ab}\right)}{\partial \dot{\gamma}_{ij}} + 2K^{ab}\frac{\partial K_{ab}}{\partial \dot{\gamma}_{ij}}\right] \\ &= N\sqrt{\gamma}\left[-2K\gamma^{ab}\left(-\frac{1}{2N}\delta_a^i\delta_b^j\right) + 2K^{ab}\left(-\frac{1}{2N}\delta_a^i\delta_b^j\right)\right] \\ \Rightarrow &\boxed{\pi^{ij} = \sqrt{\gamma}\left(K\gamma^{ij} - K^{ij}\right).} \end{aligned} \tag{85}$$

Clearly $\pi^{ij}$ is a contravariant tensor density of weight one as $\sqrt{\gamma}^W$ with $W = 1$ enters the expression. Also the 3−metric is responsible for shuffling the indices up or down in $\pi^{ij}$.

But *we want our final result to be completely written in terms of* $\{^{(3)}R, \gamma, N, \vec{N}\}$, so we need to simplify it further. We have the expression from eq. (63) about $K_{ij} = \frac{1}{2N}\left[D_i N_j + D_j N_i - \dot{\gamma}_{ij}\right]$ which we use now to eliminate the reference of the extrinsic curvature completely from $\pi^{ij}$ as follows:

$$\pi^{ij} = \sqrt{\gamma}\left(K\gamma^{ij} - K^{ij}\right) = \sqrt{\gamma}\left[K_{ab}\gamma^{ab}\gamma^{ij} - K_{ab}\gamma^{ia}\gamma^{jb}\right], \tag{86}$$

where we plug the expression for $K_{ab}$ and finally get:

$$\Rightarrow \boxed{\pi^{ij} = \frac{\sqrt{\gamma}}{2N}\left[2\gamma^{ij}D_k N^k - D^i N^j - D^j N^i + \left(\gamma^{ik}\gamma^{jl} - \gamma^{ij}\gamma^{kl}\right)\dot{\gamma}_{kl}\right].} \tag{87}$$

Now we have the ingredients to calculate the ADM Hamiltonian density but before we proceed, we realize that the extrinsic curvature appears in $\mathcal{L}_H$ in eq. (82) and we need to get rid of these terms in favour of the lapse and shift functions. Using eq. (85), we can re-arrange it to get:

$$K^{ij} = \frac{1}{2\sqrt{\gamma}}\left(\pi\gamma^{ij} - 2\pi^{ij}\right). \tag{88}$$

Contracting with the 3−metric gives for $\gamma^{ij}K_{ij} = K$ as follows (using $\gamma^{ij}\gamma_{ij} = 3$ and $\gamma^{ij}\pi_{ij} = \pi$):

$$\boxed{K = \gamma^{ij}K_{ij} = \frac{1}{2\sqrt{\gamma}}\left(3\pi - 2\pi\right) = \frac{\pi}{2\sqrt{\gamma}}.} \tag{89}$$

Finally we can write the evolution equation of the 3−metric using the expression from eq. (63) about $K_{ij} = \frac{1}{2N}\left[D_i N_j + D_j N_i - \dot{\gamma}_{ij}\right]$ and replacing $K_{ij}$ with eq. (88) to get:

$$\boxed{\dot{\gamma}_{ij} = D_i N_j + D_j N_i - \frac{N}{\sqrt{\gamma}}\left(\pi\gamma_{ij} - 2\pi_{ij}\right).} \tag{90}$$

Plugging eqs. (88, 89) into eq. (82), we get for $\mathcal{L}_H$:

$$\begin{aligned}
\mathcal{L}_H &= N\sqrt{\gamma}\left(^{(3)}R - K^2 + K^{ij}K_{ij}\right) \\
&= N\sqrt{\gamma}^{(3)}R - N\sqrt{\gamma}\frac{\pi^2}{4\gamma} + N\sqrt{\gamma}\left(\frac{1}{2\sqrt{\gamma}}\right)^2\left(\pi\gamma^{ij} - 2\pi^{ij}\right)\left(\pi\gamma_{ij} - 2\pi_{ij}\right) \\
&= N\sqrt{\gamma}^{(3)}R - N\sqrt{\gamma}\frac{\pi^2}{4\gamma} + \frac{N\sqrt{\gamma}}{4\gamma}\left(3\pi^2 - 2\pi^2 - 2\pi^2 + 4\pi^{ij}\pi_{ij}\right).
\end{aligned} \tag{91}$$

Thus we have for the Lagrangian density in terms of $\{^{(3)}R, \gamma, N, \vec{N}\}$ as follows:

$$\Rightarrow \boxed{\mathcal{L}_H = N\sqrt{\gamma}^{(3)}R + \frac{N}{\sqrt{\gamma}}\left(\pi^{ij}\pi_{ij} - \frac{1}{2}\pi^2\right).} \tag{92}$$

We can finally calculate the ADM Hamiltonian density using this form of Lagrangian density (eq. 92) and eqs. (83, 90) as follows:

$$\begin{aligned}
\mathcal{H}_H &= \pi_N \dot{N} + \pi_{N^i}\dot{N}^i + \pi^{ij}\dot{\gamma}_{ij} - \mathcal{L}_H \\
&= \pi^{ij}\left[D_i N_j + D_j N_i - \frac{N}{\sqrt{\gamma}}\left(\pi\gamma_{ij} - 2\pi_{ij}\right)\right] - \left[N\sqrt{\gamma}^{(3)}R + \frac{N}{\sqrt{\gamma}}\left(\pi^{ij}\pi_{ij} - \frac{1}{2}\pi^2\right)\right],
\end{aligned} \tag{93}$$

to finally get for the *ADM Hamiltonian density*:

$$\Rightarrow \boxed{\mathcal{H}_H = 2\pi^{ij}D_i N_j - N\sqrt{\gamma}^{(3)}R + \frac{N}{\sqrt{\gamma}}\left(\pi_{ij}\pi^{ij} - \frac{\pi^2}{2}\right).} \tag{94}$$

An alternate expression is:

$$\Rightarrow \mathcal{H}_H = 2D_i\left(\pi^{ij}N_j\right) - 2N_j D_i\pi^{ij} - N\sqrt{\gamma}^{(3)}R + \frac{N}{\sqrt{\gamma}}\left(\pi_{ij}\pi^{ij} - \frac{\pi^2}{2}\right). \tag{95}$$

Note that $D_i\pi^{ij}$ is the 3−covariant density of a tensor density with weight $W = 1$ and just like eq. (4), we have $D_i\pi^{ij} = \sqrt{\gamma}^W D_i\left[\frac{\pi^{ij}}{\sqrt{\gamma}^W}\right]$ with $W = 1$.

The *ADM Hamiltonian* $H_{\text{ADM}} = \int_{\Sigma_t} d^3x \mathcal{H}_H$ can be written in a more meaningful way as follows:

$$H_{\text{ADM}} = \mathbb{H}[N] + \mathbb{D}\left[N^j\right] = N\mathbb{H} + N^j\mathbb{D}_j, \tag{96}$$

where $\mathbb{H}[N]$ is the *Hamiltonian constraint* given by:

$$\mathbb{H}[N] \equiv \int_{\Sigma_t} d^3x N\left[-\sqrt{\gamma}^{(3)}R - \frac{1}{\sqrt{\gamma}}\left(\frac{\pi^2}{2} - \pi^{ij}\pi_{ij}\right)\right], \tag{97}$$

and $\mathbb{D}\left[N^i\right]$ are the *diffeomorphism constraints* given by:

$$\mathbb{D}[\vec{N}] \equiv \int_{\Sigma_t} d^3x N^i\left[-2D^j\pi_{ij}\right]. \tag{98}$$

We will see the physical interpretation of the Hamiltonian and diffeomorphism constraints in the Section 3.3. Also by varying the ADM action (eq. (81)) with respect to $N$ and $\vec{N}$, as done in Appendix F, we get the following constraint relations that needs to be satisfied on any spacelike hypersurface $\Sigma_t$:

$$\begin{aligned} \mathbb{H} &\approx 0, \\ \mathbb{D}_j &\approx 0, \end{aligned} \tag{99}$$

where $\approx$ symbol is called *weakly equal to* which simply means that this equality needs to be satisfied only on the hypersurfaces and not in between them. Note that the Hamiltonian constraint is 1 constraint equation while the diffeomorphism constraints contain 3 constraint equations.

**Hamilton's equations of motion**

We are now in a position to determine the 12 Hamilton's equations of motion corresponding to:

$$\dot{\gamma}_{ij} = \frac{\delta\mathcal{H}}{\delta\pi^{ij}}, \qquad \dot{\pi}^{ij} = -\frac{\delta\mathcal{H}}{\delta\gamma_{ij}}. \tag{100}$$

We start with the ADM action eq. (81) and use eq. (94) to get:

$$\begin{aligned} S_{\text{ADM}} &= \int_{t_1}^{t_2} dt \int_{\Sigma_t} d^3x\left(\pi^{ij}\dot{\gamma}_{ij} - \mathcal{H}\right) \\ &= \int_{t_1}^{t_2} dt \int_{\Sigma_t} d^3x\left[\pi^{ij}\dot{\gamma}_{ij} - \left(2\pi^{ij}D_iN_j - N\sqrt{\gamma}R + \frac{N}{\sqrt{\gamma}}\left(\pi_{ij}\pi^{ij} - \frac{\pi^2}{2}\right)\right)\right]. \end{aligned} \tag{101}$$

In order to calculate the equations of motion, we need to impose the *boundary conditions* (where $\partial\Sigma_t$ denotes the boundary of the hypersurface $\Sigma_t$):

$$\delta N|_{\partial\Sigma_t} = \delta N^i|_{\partial\Sigma_t} = \delta\gamma_{ij}|_{\partial\Sigma_t} = 0. \tag{102}$$

However there are no restrictions on the conjugate momenta $\pi^{ij}$ which are treated as independent variables.

So we have to vary this ADM action with respect to $\{N, \vec{N}, \pi^{ij}, \gamma_{ij}\}$ which we have taken to be a set of independent variables from the start. This is done in complete detail in Appendix F. We will present the results here. Variations with respect to $N$ & $\vec{N}$ lead to something known as constraint relations that need to be satisfied on a hypersurface, and with respect to $\pi^{ij}$ & $\gamma_{ij}$ lead to equations of motion telling about actual evolution of tensor fields in time on a spacelike hypersurface $\Sigma_t$. They are given by:

- Constraint equations:

  - $\dfrac{\delta S_{ADM}}{\delta N} \overset{!}{=} 0 \quad \Rightarrow \boxed{\mathbb{H} = 0} \quad$ ($N$ is a Lagrange multiplier),

  - $\dfrac{\delta S_{ADM}}{\delta N_i} \overset{!}{=} 0 \quad \Rightarrow \boxed{\mathbb{D}^i = 0} \quad$ ($N_i$ are Lagrange multipliers).

- Hamilton's equations of motion:

  - $\dfrac{\delta S_{ADM}}{\delta \pi^{ij}} \overset{!}{=} 0 \quad \Rightarrow \boxed{\dot{\gamma}_{ij} = \dfrac{\delta \mathcal{H}}{\delta \pi^{ij}} = D_i N_j + D_j N_i - 2N K_{ij}}$,

  - $\dfrac{\delta S_{ADM}}{\delta \gamma_{ij}} \overset{!}{=} 0 \quad \Rightarrow \dot{\pi}^{ij} = -\dfrac{\delta \mathcal{H}}{\delta \gamma_{ij}}$,

    which for the sake of completeness is given by the full expression as follows:

$$
\boxed{
\begin{aligned}
\Rightarrow \dot{\pi}^{ij} = {}& -N\sqrt{\gamma}\left(R^{ij} - \frac{1}{2}\gamma^{ij}R\right) + \frac{N}{2\sqrt{\gamma}}\left(\pi_{cd}\pi^{cd} - \frac{\pi^2}{2}\right)\gamma^{ij} \\
& -\frac{2N}{\sqrt{\gamma}}\left(\pi^{ic}\pi_c^{\ j} - \frac{1}{2}\pi\pi^{ij}\right) + \sqrt{\gamma}\left(D^i D^j N - \gamma^{ij}D_c D^c N\right) \\
& + D_c\left(\pi^{ij}N^c\right) - \pi^{ic}D_c N^j - \pi^{jc}D_c N^i.
\end{aligned}
}
$$

As a redundancy check, we realize that the equation of motion for the 3−metric is the same as what we had obtained in eq. (90) where we replaced $K_{ij}$ with eq. (88). The evolution of the conjugate momenta is a new result while the 4 constraint relations simply help us conclude what we stated above that the lapse and the shift functions are Lagrange multipliers (& thus are not dynamical variables). These dynamical equations also lead to *fundamental Poisson brackets* among the *canonical variables* of the system, namely:

$$
\begin{aligned}
\{\gamma_{ij}, \gamma_{kl}\}|_{(\gamma,\pi)} &= 0, \\
\{\pi^{ij}, \pi^{kl}\}|_{(\gamma,\pi)} &= 0, \\
\{\gamma_{ij}, \pi^{kl}\}|_{(\gamma,\pi)} &= \delta_i^k \delta_j^l,
\end{aligned}
\tag{103}
$$

where the Poisson brackets are calculated with respect to variables $\gamma_{ij}$ and $\pi^{kl}$. The definition of the Poisson bracket is taken as:

$$
\{f(x), h(y)\}|_{(\gamma,\pi)} = \int d^3 z \left[\frac{\delta f(x)}{\delta \gamma_{ij}(z)}\frac{\delta h(y)}{\delta \pi^{ij}(z)} - \frac{\delta f(x)}{\delta \pi^{ij}(z)}\frac{\delta h(y)}{\delta \gamma_{ij}(z)}\right].
\tag{104}
$$

As an aside, without proof, we note that reintroducing the cosmological constant $\Lambda$ has no effect on the definition of $\pi^{ij}$ and a simple relabelling $R \to (R - 2\Lambda)$ leads to correct result throughout without any exception. Thus we conclude that the Hamiltonian formalism of general relativity has no change with the reintroduction of $\Lambda$ through this simple relabelling. Although we are never considering the boundary terms, it can be stated here without proof that even with the inclusion of boundary terms in the Lagrangian formulation as well as the Hamiltonian formulation, nothing changes after the substitution $R \to (R - 2\Lambda)$ everywhere.

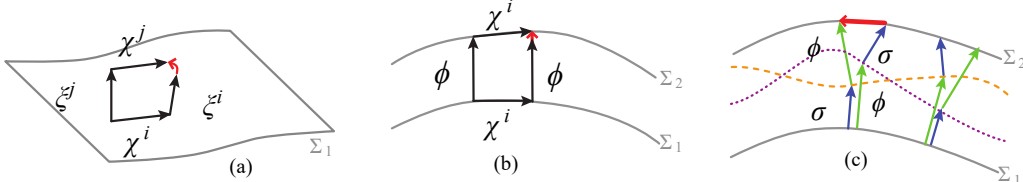

Figure 4: Geometric interpretation of constraints: (a) diffeomorphism constraint, (b) diffeomorphism + Hamiltonian constraints, (c) Hamiltonian constraint [1].

Now we move to the *Hypersurface Deformation Algebra (HDA)*, also known as the *Dirac algebra*, where we will have a physical interpretation of the Hamiltonian constraint and the diffeomorphism constraints.

### 3.3 Hypersurface deformation algebra

We realize that $\mathbb{H}[N]$ and $\mathbb{D}[\vec{N}]$ are constraints as both vanish on any hypersurface $\Sigma_t$. Hence these constraints must be preserved as the system evolves. Accordingly we expect that $\mathbb{H}[N]$ and $\mathbb{D}[\vec{N}]$ form a *first class set of constraints*,[3] i.e. they will form a closed system of constraints under the action of the Poisson brackets. Indeed they do and the *constraint algebra* or more commonly known as the *Hypersurface Deformation Algebra* (HDA) or the *Dirac algebra* is [9,10] (which can be taken as an independent starting point for general relativity):

$$
\begin{aligned}
&\text{(a)} \ \left\{\mathbb{D}[\vec{\xi}], \mathbb{D}[\vec{\chi}]\right\}\big|_{(\gamma,\pi)} = \mathbb{D}\left[\mathcal{L}_{\vec{\xi}}\vec{\chi}\right] = \mathbb{D}\left[[\vec{\xi}, \vec{\chi}]\right], \\
&\text{(b)} \ \left\{\mathbb{D}[\vec{\xi}], \mathbb{H}[\phi]\right\}\big|_{(\gamma,\pi)} = \mathbb{H}\left[\mathcal{L}_{\vec{\xi}}\phi\right], \\
&\text{(c)} \ \left\{\mathbb{H}[\phi], \mathbb{H}[\sigma]\right\}\big|_{(\gamma,\pi)} = \mathbb{D}\left[\gamma^{jk}(x)(\phi\partial_j\sigma - \sigma\partial_j\phi)\right].
\end{aligned}
\tag{105}
$$

A detailed derivation of the HDA is provided in Appendix H. We focus here on the graphical interpretation of constraints as shown in Fig. (4) [1]. The vector fields generating the tangential and normal deformations of the spatial slice form an algebra[4] under the commutator, which finds a representation in the deformation algebra formed by the phase space quantities $\mathbb{H}[N]$ and $\mathbb{D}[\vec{N}]$ under the Poisson bracket. Thus we have a clear geometrical interpretation for the Hamiltonian constraint $\mathbb{H}[N]$ and the three diffeomorphism constraints $\mathbb{D}[\vec{N}]$: $\mathbb{H}[N]$ takes us from one hypersurface $\Sigma_1$ to another hypersurface $\Sigma_2$ (whose travel length is proportional to $N$) while $\mathbb{D}[\vec{N}]$ allows for movement on one hypersurface itself (whose travel length is proportional to $\vec{N}$). And irrespective of which operation is done first followed by the next, the algebra remains closed.

In other words, if we consider (a) in eq. (105), the Poisson bracket for two diffeomorphism constraints is yet another diffeomorphism constraint, implying that if we compute two diffeomorphism constraints in different orders, then this would lead us to two different points on the same hypersurface and thus we need another diffeomorphism to connect the two points on the hypersurface. Similarly (b) in eq. (105) implies that if we compute Hamiltonian constraint followed by the diffeomorphism constraint, then we reach a point on the next hypersurface but the reverse order of computing would take us somewhere in between the two hypersurfaces and that's why we need a Hamiltonian constraint at the end to get us to the same point

---

[3]A function $f$ defined on the full phase-space is called a *first class constraint* if its Poisson brackets with all the constraints of the system vanish weakly, namely $\{f, \Phi_i\} \approx 0$ (where $\Phi_i$ are the constraints of the system). Any function that is not a first class constraint is a *second class constraint*, namely it has one or more non-weakly vanishing Poisson brackets with all the constraints of the system.

[4]$\gamma^{jk}(x)$ appearing in (c) are spacetime functions, not just constants, that emerge from the geometrical deformations of hypersurface.

as reached first. Finally, (c) in eq. (105) means that computing two different Hamiltonian constraints in different orders both lead to the next hypersurface but at different points and hence we need a diffeomorphism constraint on the next hypersurface to connect those two points. This is exactly what is shown in Fig. (4).

As a special case, if we restrict to linear deformations, namely linear coordinate changes of flat slices $\gamma_{ij} = \delta_{ij}$ (*the Minkowski limit*), it can be shown [13] that the *HDA reduces to the Poincaré algebra* under linear diffeomorphisms (where $P_\mu$ is the generator of translations, $M_{\mu\nu}$ is the generator of Lorentz transformations and $\eta_{\mu\nu}$ is the Minkowski metric):

$$
\begin{aligned}
\{P_\mu, P_\nu\}|_{(\gamma,\pi)} &= 0\,, \\
\{M_{\mu,\nu}, P_\rho\}|_{(\gamma,\pi)} &= \eta_{\mu\rho} P_\nu - \eta_{\nu\rho} P_\mu\,, \\
\{M_{\mu\nu}, M_{\rho,\sigma}\}|_{(\gamma,\pi)} &= \eta_{\mu\rho} M_{\nu\sigma} - \eta_{\mu\sigma} M_{\nu\rho} - \eta_{\nu\rho} M_{\mu\rho} + \eta_{\nu\sigma} M_{\mu\rho}\,.
\end{aligned}
\tag{106}
$$

## 4 Homogeneous cosmologies

After giving a detailed introduction to the ADM formalism of general relativity, we now shift towards giving a brief but self-contained introduction to homogeneous cosmologies. We stick to the assumption of homogeneity but would relax the criterion of isotropy. As we are aware in case of homogeneous and isotropic universe, we have one independent variable and that is the acceleration parameter $a(t)$ that appears in the Friedmann–Lemaître–Robertson–Walker (FRWL) metric [14]. In the case of homogeneous but anisotropic universe (FRWL being a special case[5]), we will see that there are more than one type of cosmological solutions that are inequivalent to each other. In Section 4.1, we discuss this classification of homogeneous but anisotropic universes, something known as the *Bianchi classification*. Then in Section 4.2, we discuss a set of basis vectors well-suited for homogeneous cosmologies, namely the invariant basis and go on to show how the Einstein field equations look in this basis. In Section 4.3, we discuss the dynamics of the Bianchi models in the Lagrangian formulation and then, as examples, specialize to the case of Bianchi I & Bianchi IX universes. We recover these results for Bianchi I & Bianchi IX universes in the Hamiltonian formulation in Chapters 5.1 & 5.2 respectively, thereby showing the equivalence of these two formulations. *We always consider the vacuum case in the bulk*.

### 4.1 Bianchi classification

We rely heavily on the resources [15, 16] for this section. By homogeneous, we are referring to those spacetimes which have *spatial homogeneity* [15, 16]. We do not deal with the case of entire spacetime manifold being homogeneous where the metric is the same at all points in time and space because such a universe cannot expand at all. A *spatially homogeneous spacetime* (or simply the homogeneous spacetime from now onward) is defined as a manifold possessing a group of transformations that leave the metric invariant, or in other work a group of isometries. Accordingly there is a set of vectors, known as the Killing vectors $\xi$, that generate such invariant transformations ($\mathcal{L}_\xi g = 0$) whose orbits are spacelike hypersurfaces foliating the spacetime manifold which we encountered in Chapter 2.3. So we start with a brief overview of Killing vector fields following which we make the definition of homogeneity mathematically more precise and what it means in the context of cosmology. Then finally we discuss the Bianchi classification.

---

[5]Different Bianchi universes have different topologies, just like FRWL universes. Thus to make this sentence more precise, the closed FRWL universe is the isotropic case of Bianchi IX, the flat FRWL universe is a special case of Bianchi I & Bianchi VII$_0$ and the open FRWL is of Bianchi V & Bianchi VII$_a$. See the chart 129 for the Bianchi classification.

## Killing vector fields

The Lie algebras of Killing vector fields are responsible for generating infinitesimal displacements that can lead to conserved quantities and allows for a classification of homogeneous cosmologies. Before we delve into that, let's focus on Killing vector fields themselves.

Consider a group of transformations

$$x^\mu \longmapsto x'^\mu = f^\mu(x, a), \tag{107}$$

on a manifold $\mathcal{M}$ where $\{a^b\}|_{b=1,2,\dots,r}$ are $r-$independent variables which parametrize the group and let $a_0$ be the identity such that $f^\mu(x, a_0) = x^\mu$. Then taking an infinitesimal transformation $a_0 + \delta a$ about identity leads to:

$$x^\mu \to x'^\mu = f^\mu(x, a_0 + \delta a) \approx \underbrace{f^\mu(x, a_0)}_{=x^\mu} + \underbrace{\left(\frac{\partial f^\mu}{\partial a^a}\right)(x, a_0)}_{\equiv \xi_a^\mu(x)} \delta a^a$$

$$\Rightarrow x^\mu \to x'^\mu \approx x^\mu + \xi_a^\mu \delta a^a \tag{108}$$

$$= (1 + \delta a^b \xi_b) x^\mu.$$

The first-order differentiable operators $\{\xi_a\}$ (total $r$ of them) are the *generating vector fields,* also called the *Killing vector fields*, defined as:

$$\xi_b \equiv \xi_b^\mu \frac{\partial}{\partial x^\mu}, \tag{109}$$

where the components are given by $\{\xi_b^\mu\}$ and satisfy $\mathcal{L}_\xi g_{\mu\nu} = 0$. Thus we have $x'^\mu \approx (1 + \delta a^b \xi_b) x^\mu \approx e^{\delta a^b \xi_b} x^\mu$. This is for infinitesimal transformations of the group.

Finite transformations of the group are represented by:

$$x^\mu \to x'^\mu = e^{\theta^a \xi_a} x^\mu, \tag{110}$$

where $\{\theta^a\}_{a=1,2,\dots,r}$ are $r$ new parameters of the group.

The Killing vector fields form a Lie algebra where the basis $\{\xi_a\}$ is closed under commutation:

$$[\xi_a, \xi_b] = \pm C_{ab}^c \xi_c, \tag{111}$$

where $C_{ab}^c$ are the structure constants of the Lie algebra and $\pm$ refer to the left-/right-invariant groups.

This algebra allows us to define a natural inner product as follows. Suppose $\{e_a\}$ is a basis of the Lie algebra **g** of a group $G$:

$$[e_a, e_b] = C_{ab}^c e_c. \tag{112}$$

Then we can define $\gamma_{ab} \equiv C_{ad}^c C_{bc}^d = \gamma_{ba}$ (symmetric by definition) that allows for a natural inner product on the Lie algebra (when $\det(\gamma_{ab}) \neq 0$) as follows: $\gamma_{ab} = e_a.e_b = \gamma(e_a, e_b)$. This is known as the semi-simple group which is our primary interest.

## Mathematical definition of homogeneity

With this introduction about Killing vector fields, we now proceed to define homogeneity. Suppose that the group acts of a manifold $\mathcal{M}$ as a group of transformations $x^\mu \to f^\mu(x, a) \equiv f_a^\mu(x)$ and let us define the orbit of $x$: $f_G(x) = \{f_a(x) | a \in G\}$. This constitutes a set of all points that can be reached from $x$ under the group of transformations. Thus we define the group of isometry at $x$ is $G_x = \{a \in G | f_a(x) = x\}$ (it is the subgroup of $G$ which leaves $x$ fixed). Suppose $G = \{a_0\}$ and $f_G(x) = \mathcal{M}$ and every point in $\mathcal{M}$ can be reached from $x$ by a unique

transformation. Then $G|G_x = \{aa_0|a \in G\} = G$ where $G_x$ is the group isotropy at $x$. Thus $G$ is diffeomorphic to $\mathcal{M}$. Then for a given basis $\{e_a\}$ of the Lie algebra of a three dimensional Lie group $G$, the spatial metric at each point of time is specified by the *spatially constant inner products*: $e_a.e_b = g_{ab}(t)$ (6 functions of time). *This is the definition of spatially homogeneous universes that we are interested in*. We will see later that Einstein equations become ordinary differential equations for these six functions for pure gravity case. In three dimensions, the classification of inequivalent 3D Lie groups is called the Bianchi classification and determines various symmetry types for homogeneous 3-spaces which is analogous to how $k = -1, 0, +1$ determines the possible symmetry types for homogeneous and isotropic (FRWL) 3−spaces.

A homogeneous spacetime is defined by spacelike hypersurfaces $\Sigma_t$ such that for any point $p, q \in \Sigma_t$, there exists a unique element $\tau \in G$ such that $\tau(p) = q$ (here the Lie group acts transitively on each $\Sigma_t$). Such uniqueness implies that $\dim(G) = \dim(\Sigma_t) = 3$ and thus $G \cong \Sigma_t$. As an example, the simplest case of translation group has $\Sigma_t = \mathbb{R}^3$. Thus the action of isometries on $\Sigma_t$ is just the left-multiplication on $G$ and tensor fields invariant under isometries are the left-invariant ones on $G$.

From now on, we specializing to $3 + 1$−D where we foliate the spacetime manifold as $\mathcal{M} = \mathbb{R} \times G$. We demand the invariance of the line element on each of the hypersurfaces:

$$dl^2 = \gamma_{ab}\left(x^1, x^2, x^3\right) dx^a dx^b = \gamma_{ab}\left(x'^1, x'^2, x'^3\right) dx'^a dx'^b. \tag{113}$$

In general for any non-Euclidean homogeneous 3−D space, we have three independent differential forms $\omega^\alpha$ ($\alpha = 1, 2, 3$) which are invariant under the transformations generated by the three independent Killing vector fields. We write them as $\omega^\alpha = e_a^\alpha dx^a$. The components $e_a^\alpha$ in the dual basis satisfy the orthogonal relations: (i) $e_a^\alpha e_\alpha^b = \delta_a^b$, & (ii) $e_a^\alpha e_\beta^a = \delta_\beta^\alpha$. Thus the 3−line element becomes:

$$dl^2 = \eta_{\alpha\beta}\left(e_a^\alpha dx^a\right)\left(e_b^\beta dx^b\right), \tag{114}$$

from which we read off the metric as $\gamma_{ab} = \eta_{\alpha\beta}(t) e_a^\alpha\left(x^i\right) e_b^\beta\left(x^i\right)$ and the inverse metric as $\gamma^{ab} = \eta^{\alpha\beta}(t) e_\alpha^a\left(x^i\right) e_\beta^b\left(x^i\right)$. Defining volume $V = \left|e_a^\alpha\right| = e^1 \cdot \left[e^2 \wedge e^3\right]$ leads to $\det(\gamma_{ab}) = \eta V^2$ where $\eta = \det(\eta_{\alpha\beta})$.

With a given basis $\{e_a\}$, we have the following relation for homogeneous spacetimes (spatial homogeneity):

$$\boxed{e_a^\alpha \frac{\partial e_b^\gamma}{\partial x^\alpha} - e_b^\beta \frac{\partial e_a^\gamma}{\partial x^\beta} = C_{ab}^c e_c^\gamma.} \tag{115}$$

Proof: The invariance of the line element in eq. (114) means that $\omega^a(x) = \omega^a(x') \Rightarrow$ $e_\alpha^a(x) dx^\alpha = e_\alpha^a\left(x'\right) dx'^\alpha$ where $e_\alpha^a$ on both sides is the same function expressed in old and new coordinates respectively. From this equality, we can deduce:

$$\frac{\partial x'^\beta}{\partial x^\alpha} = e_a^\beta\left(x'\right) e_\alpha^a(x). \tag{116}$$

This is the fundamental differential equation defining the change of coordinates $x \leftrightarrow x'$ in terms of given basis vectors $\vec{e}^a$ and its dual $\vec{e}_a$. The integrability condition of eq. (116) is known as the *Schwartz condition* provided by:

$$\frac{\partial^2 x'^\beta}{\partial x^\alpha \partial x^\gamma} = \frac{\partial^2 x'^\beta}{\partial x^\gamma \partial x^\alpha}$$
$$\Rightarrow \frac{\partial}{\partial x^\alpha}\left(e_a^\beta(x') e_\gamma^a(x)\right) = \frac{\partial}{\partial x^\gamma}\left(e_a^\beta(x') e_\alpha^a(x)\right). \tag{117}$$

Taking derivatives of $\vec{e}_a(x)$ and $\vec{e}_a(x')$ on both sides and using the orthogonality conditions of $e_\alpha^a$:

$$\left[ \frac{\partial e_a^\beta(x')}{\partial x'^\delta} e_b^\delta(x') - \frac{\partial e_b^\beta(x')}{\partial x'^\delta} e_a^\delta(x') \right] e_\gamma^b(x) e_\alpha^a(x) = e_a^\beta(x') \left[ \frac{\partial e_\gamma^a(x)}{\partial x^\alpha} - \frac{\partial e_\alpha^a(x)}{\partial x^\gamma} \right]. \quad (118)$$

Multiplying and contracting on both sides by $e_d^\alpha(x) e_c^\gamma(x) e_\beta^f(x')$, we get:

$$\Rightarrow e_c^\beta(x') e_d^\delta(x') \left[ \frac{\partial e_\beta^f(x')}{\partial x'^\delta} - \frac{\partial e_\delta^f(x')}{\partial x'^\gamma} \right] = e_c^\beta(x) e_d^\delta(x) \left[ \frac{\partial e_\beta^f(x)}{\partial x^\delta} - \frac{\partial e_\delta^f(x)}{\partial x^\gamma} \right]. \quad (119)$$

Basically, the LHS and the RHS are the same functions denoted in the new and the old coordinates respectively. Since the coordinate system is arbitrary, we have LHS = RHS = constant, where we choose the group structure constants $C_{ab}^c$ as the constant to get:

$$\Rightarrow \left( \frac{\partial e_\alpha^c}{\partial x^\beta} - \frac{\partial e_\beta^c}{\partial x^\alpha} \right) e_a^\alpha e_b^\beta = C_{ab}^c, \quad (120)$$

which upon contracting with $e_c^\gamma$ gives us eq. (115)

Here $C_{ab}^c$ are the structure constants of the Lie algebra for the Lie group $G$ which is by construction $C_{ab}^c = -C_{ba}^c$. Defining a vector as $X_a \equiv e_a^\alpha \frac{\partial}{\partial x^\alpha}$, we have:

$$\boxed{[X_a, X_b] = C_{ab}^c X_c.} \quad (121)$$

Thus the condition of homogeneity is then expressed by the *Jacobi identity*:

$$\boxed{[[X_a, X_b], X_c] + [[X_b, X_c], X_a] + [[X_c, X_a], X_b] = 0,} \quad (122)$$

or explicitly:

$$\boxed{C_{ab}^f C_{cf}^d + C_{bc}^f C_{af}^d + C_{ca}^f C_{bf}^d = 0.} \quad (123)$$

Introducing $C_{ab}^c \equiv \epsilon_{abd} C^{dc}$ where $\epsilon_{abc}$ is the 3D Levi-Civita tensor with $\epsilon_{123} = +1$, we get for the Jacobi identity:

$$\boxed{\epsilon_{bcd} C^{cd} C^{ba} = 0.} \quad (124)$$

*Thus Bianchi classification of categorizing inequivalent homogeneous spaces reduces to finding all inequivalent sets of structure constants. Each algebra uniquely determines the local properties of a 3D group.*

**Bianchi classification**

In order to have Bianchi classification, we start by realizing that any structure constant can be written as:

$$C_{bc}^a = \epsilon_{bcd} m^{da} + \delta_c^a a_b - \delta_b^a a_c, \quad (125)$$

where $m^{ab} = m^{ba}$.

*Class A* and *Class B Bianchi models* refer to the cases $a_b = 0$ and $a_b \neq 0$ respectively. Accordingly *Jacobi identity* becomes

$$\boxed{m^{ab} a_b = 0.} \quad (126)$$

Without loss of generality, we can take $a_b = (a, 0, 0)$ and matrix $m^{ab}$ can be described by its principal eigenvalues, say, $n_1$, $n_2$ and $n_3$ (in 3D). Then Jacobi identity further simplifies to

$$\Rightarrow n_1 a = 0, \tag{127}$$

which means either $a$ or $n_1$ has to vanish. Explicitly we have *Jacobi identity* as:

$$\begin{aligned}
[X_1, X_2] &= -aX_2 + n_3 X_3, \\
[X_2, X_3] &= n_1 X_1, \\
[X_3, X_1] &= n_2 X_2 + aX_3,
\end{aligned} \tag{128}$$

where $a > 0$ and $(a, n_1, n_2, n_3)$ are all rescaled to unity without loss of generality. Thus Bianchi classification is given as [17]:

| Type | $a$ | $n_1$ | $n_2$ | $n_3$ |
|---|---|---|---|---|
| Bianchi I | 0 | 0 | 0 | 0 |
| Bianchi II | 0 | 1 | 0 | 0 |
| Bianchi III | 1 | 0 | 1 | −1 |
| Bianchi IV | 1 | 0 | 0 | 1 |
| Bianchi V | 1 | 0 | 0 | 0 |
| Bianchi VI $_a$ | $a$ | 0 | 1 | −1 |
| Bianchi VI $_0$ | 0 | 1 | −1 | 0 |
| Bianchi VII $_a$ | $a$ | 0 | 1 | 1 |
| Bianchi VII $_0$ | 0 | 1 | 1 | 0 |
| Bianchi VIII | 0 | 1 | 1 | −1 |
| Bianchi IX | 0 | 1 | 1 | 1 |

$$\tag{129}$$

Clearly FRWL universe (homogeneous as well as isotropic) is a special case of Bianchi universes (see footnote 5 at the start of this Chapter 4). Note that Bianchi I is isomorphic to the $\mathbb{R}^3$ (3D translation group) for which the flat FRWL model is a particular case (once isotropy is restored). Thus Bianchi I universe has flat spatially homogeneous hypersurfaces. Analogously, Bianchi V contains open FRWL as a special case. Another crucial point to be noted is that not all anisotropic dynamics are compatible with a satisfactory Standard Cosmological Model [8] but some can be represented as "FRWL model + a gravitational waves packet" if certain conditions are satisfied [18, 19]. As we will see later, for example in the case of Bianchi IX universe, there is a type of dynamics as one approaches the big-bang singularity where there is an infinite number of transitions from one free motion to another due to bounces off the potential wall (just like the case of a billiards ball bouncing off the walls of the billiards table and travelling freely in between those collisions). Such a behaviour is what Misner called *Mixmaster* behaviour [20–22].

The line element of a homogeneous universe can be decomposed as follows:

$$ds^2 = ds_0^2 - \delta_{(a)(b)} G_{ik}^{(a)(b)} dx^i dx^k, \tag{130}$$

where $ds^0$ denotes the line element of an isotropic universe having a positive curvature constant $k = 1$ (closed), $G_{ik}^{(a)(b)}$ is a set of spatial tensors and $\delta_{(a)(b)}$ are the amplitude functions which are sufficiently small when far from singularity. These satisfy:

$$G_{ik;l}^{(a)(b);l} = -(n^2 - 3) G_{ik}^{(a)(b)}, \qquad G_{i;k}^{(a)(b)k} = 0, \qquad G_i^{(a)(b)i} = 0. \tag{131}$$

Here Laplacian is referred to the geometry of a unit sphere.

Choosing a basis of dual vector fields $\omega^\alpha$ preserved under isometries and recalling $\gamma^{ij} = \eta^{\alpha\beta} e^i_\alpha e^j_\beta$, we can decompose the 4−D line element as:

$$ds^2 = N^2 dt^2 - \eta_{\alpha\beta} \omega^\alpha \otimes \omega^\beta, \tag{132}$$

which is parametrized by the lapse function $N$ and $\omega^\alpha$ that satisfies the *Maurer-Cartan equations*:

$$d\omega^a = \frac{1}{2}C^a_{bc}\omega^b \wedge \omega^c.$$
(133)

Then we have explicitly the expressions for Bianchi I and IX universes as:

- Bianchi I:

$$
\begin{aligned}
C^a_{bc} &= 0, \\
\omega^1 &= dx^1, \\
\omega^2 &= dx^2, \\
\omega^3 &= dx^3.
\end{aligned}
$$
(134)

- Bianchi IX:

$$
\begin{aligned}
C_{abc} &= \epsilon_{abc}, \\
\omega^1 &= \sin(\psi)\sin(\theta)d\phi + \cos(\psi)d\theta, \\
\omega^2 &= -\cos(\psi)\sin(\theta)d\phi + \sin(\psi)d\theta, \\
\omega^3 &= \cos(\theta)d\phi + d\psi,
\end{aligned}
$$
(135)

which are the coordinates for a unit 3-sphere. Here $C^a_{bc} = \epsilon_{bcd}C^{da}$ where $C^{da} = \mathrm{diag}(1,1,1)$.

## 4.2 Invariant basis & Einstein field equations

We recall that the set of transformations generated by Killing vector fields $\{\xi_i\}$ ($i = 1, 2, 3$) on a manifold $\mathcal{M}$ form a Lie group, also known as the *isometry of the manifold*. They are given by:

$$\left[\xi_i, \xi_j\right] = C^c_{ij}\xi_c,$$
(136)

where $C^\gamma_{\alpha\beta}$ are the structure constants of the Lie algebra which satisfies $C^\gamma_{\alpha\beta} = -C^\gamma_{\alpha\beta}$. Thus in 3−D, we have 9 independent components. Eqs. (123, 124) are also its general properties. Here we are going to discuss about two types of basis states: the *invariant basis* as well as *triad formalism* (a non-coordinate basis). Then we recast all 3−geometrical objects capturing the curvature of spacelike hypersurfaces in terms of non-coordinate basis. Finally we reduce the Einstein field equations for the vacuum case in a homogeneous ansatz (provided in eq. (145)), both in terms of coordinate as well as non-coordinate bases (which coincide with invariant basis).

**Invariant basis**

We start with considering a general coordinate system $\{x^i\}$ whose coordinate basis is $\{\partial_i\}$ and its dual basis $\{dx^i\}$. Then the 3−metric is given in terms of this *coordinate basis* is:

$$\gamma = \gamma_{ij}dx^i dx^j.$$
(137)

But from the definition of Killing vector fields, we have $\mathcal{L}_{\xi^i}\gamma = 0$ which in coordinate basis becomes $\mathcal{L}_{\xi_c}dx^i = 0 \ \forall \ i$. But we also know that the inner product between basis state and its dual is a delta function: $\langle dx^i, \partial_j \rangle = \delta^i_j$. Applying the Lie derivative on this equation and using the chain rule, we get $\langle \mathcal{L}_{\xi_c}dx^i, \partial_j \rangle = -\langle dx^i, \mathcal{L}_{\xi_c}\partial_b \rangle$. Thus we have established:

$$\boxed{\mathcal{L}_{\xi_c}dx^i = 0 \quad \Longleftrightarrow \quad \mathcal{L}_{\xi_c}\partial_b = 0.}$$
(138)

This is the defining relation for *invariant basis*: a set of basis states that are invariant under transformations generated by $\xi_c$. Dual to the invariant basis is called the *dual invariant basis*.

Before we proceed further of how to construct an invariant basis, we list down the advantages of using this basis in the context of Bianchi (homogeneous) universes:

(a) Components of the 3−metric $\gamma$ are spatially constants on each of the hypersurfaces $\Sigma_t$ while only depending on time,

(b) If $\{e_a\}$ are the vector fields associated to the invariant basis, then they form a Lie algebra $[e_i, e_j] = D^c_{ab} e_c$. Generally the *structure functions* $D^c_{ab}$ are dependent on spatial coordinates but in case of invariant basis, these are constants.

(c) $D^c_{ab} = C^c_{ab}$ in an invariant basis for Bianchi universes where $C^c_{ab}$ is introduced in eq. (136). This holds at all points on any spatially homogeneous hypersurfaces.

We realize that in general a coordinate basis need not coincide with an invariant basis. Here we discuss how to construct an invariant basis using (i) a coordinate non-invariant basis, & (ii) three independent Killing vector fields on Bianchi spatial hypersurfaces. We start with taking three independent vectors $V_i$ at any arbitrary point $P$ on a hypersurface. It is often convenient to identify them with Killing vector fields $\xi_i$ so that we have $V_i = \delta^j_i \xi_j(P)$. Then any 3−vector fields $\{A_a\}$ by construction form an invariant basis if it satisfies the following two conditions:

$$(\text{i})\ A_a(P) = V_a\,, \qquad (\text{ii})\ \mathcal{L}_{\xi_a} A_j = [\xi_a, A_j] = 0\,. \tag{139}$$

It is conventional to denote invariant basis by $\{\hat{e}_\alpha\}$ and use Greek indices to label them where $\alpha = 1, 2, 3$. We will be using invariant basis to discuss Bianchi cosmologies in this work. Note that in general invariant basis $\{\hat{e}_\alpha\}$ may not coincide with coordinate basis $\{\partial_i\}$ but if it does then $D^c_{ab} = C^c_{ab} = 0$.

**Triad formalism**

Any vector in a non-coordinate basis can be written as a linear combination of the coordinate basis vectors as follows:

$$\hat{e}_\alpha = e^a_\alpha \partial_a\,, \tag{140}$$

where indices $a$ and $\alpha$ run over $\{1, 2, 3\}$ on spacelike hypersurfaces, $e^a_\alpha$ are called *triads* which can depend on spatial coordinates and we always take $\det\left(e^a_\alpha\right) > 0$ to preserve the orientation of the manifold.[6]

Inverse of the triads $e^\alpha_a$ are defined as the components of the vectors of the dual non-coordinate basis in the dual coordinate basis:

$$\hat{\theta}^\alpha = e^\alpha_a dx^a\,. \tag{141}$$

The triads and its inverse satisfy the orthogonality conditions:

$$e^\alpha_a e^b_\alpha = \delta^b_a\,, \qquad e^\alpha_a e^a_\beta = \delta^\alpha_\beta\,. \tag{142}$$

---

[6]A note on notation: The notation used here can be a source of confusion because Greek indices have been used until now to denote spacetime components while Latin indices are used to denote spatial components. But here Greek indices are used to denote the non-coordinate basis components (which we will later take to coincide with the invariant basis in the context of Bianchi cosmologies) while Latin indices are used to denote coordinate basis components and both runs over spatial parts only. The distinction between these two usages should be clear from the context.

The dual non-coordinate basis satisfies the *Maurer-Cartan structure equation*:

$$
\begin{aligned}
d\hat{\theta}^\alpha &= -\frac{1}{2} D^\alpha_{\beta\gamma} \hat{\theta}^\alpha \wedge \hat{\theta}^\gamma \\
&= -\frac{1}{2} C^\alpha_{\beta\gamma} \hat{\theta}^\alpha \wedge \hat{\theta}^\gamma \,,
\end{aligned}
\tag{143}
$$

where the second line holds only *iff* the non-coordinate basis coincides with the invariant basis which is of our interest. Note that $C^\gamma_{\alpha\beta} = 0$ when an invariant basis coincides with a coordinate basis while $D^\gamma_{\alpha\beta} = 0$ when a non-coordinate basis coincides with a coordinate basis. An example for the triads are $e^a_\alpha = \delta^a_\alpha$ for Bianchi I universe (where $C^\gamma_{\alpha\beta} = 0$, see eq. (134)).

**Geometry of hypersurfaces**

Now we are in a position to start expressing the geometry of hypersurfaces in terms of triads. We defined 3−metric in terms of coordinate basis in eq. (137). Then its components in non-coordinate basis is given by:

$$
h_{\alpha\beta} = \gamma_{ij} e^i_\alpha e^j_\beta \,.
\tag{144}
$$

The 4−line element of a homogeneous universe becomes:

$$
ds^2 = -dt^2 + \gamma_{ij} dx^i dx^j = -dt^2 + h_{\alpha\beta}(t) e^\alpha_i e^\beta_j dx^i dx^j \,.
\tag{145}
$$

Accordingly the connection coefficients in non-coordinate basis are defined as:

$$
D_{\hat{e}_\alpha} \hat{e}_\beta = {}^{(3)}\Gamma^\gamma_{\alpha\beta} \hat{e}_\gamma \,,
\tag{146}
$$

which in terms of triads become:

$$
D_\alpha e^i_\beta = {}^{(3)}\Gamma^\sigma_{\alpha\beta} e^i_\sigma \,, \qquad D_i e^j_\alpha = {}^{(3)}\Gamma^j_{ik} e^k_\alpha \,.
\tag{147}
$$

Then the compatibility with the metric $D_{\hat{e}_\alpha} h_{\alpha\beta} = 0$ implies ${}^{(3)}\Gamma_{\gamma\alpha\beta} = -{}^{(3)}\Gamma_{\beta\alpha\gamma}$. Moreover *torsion* is defined in a non-coordinate basis as:

$$
T^\gamma_{\alpha\beta} \equiv {}^{(3)}\Gamma^\gamma_{\alpha\beta} - {}^{(3)}\Gamma^\gamma_{\beta\alpha} - D^\gamma_{\alpha\beta} \,.
\tag{148}
$$

Thus *torsion-free* implies $T = 0 \Rightarrow {}^{(3)}\Gamma^\gamma_{\alpha\beta} - {}^{(3)}\Gamma^\gamma_{\beta\alpha} = D^\gamma_{\alpha\beta} = C^\gamma_{\alpha\beta}$, where last equality is true only when non-coordinate basis coincides with invariant basis. The crucial point is that in a non-coordinate basis, connection coefficients ${}^{(3)}\Gamma^\gamma_{\alpha\beta}$ are not symmetric in its indices $\alpha$ and $\beta$. Only when a non-coordinate basis coincides with a coordinate basis, then $D^\gamma_{\alpha\beta} = 0$ and we have symmetry restored in case of torsion-free.

Just like the 3−connection coefficients, we can evaluate other 3−geometric objects in non-coordinate basis whose results are listed herewith:

$$
\begin{aligned}
{}^{(3)}\Gamma_{\alpha\beta\gamma} &= -\frac{1}{2} \left( D_{\alpha\gamma\beta} + D_{\gamma\beta\alpha} - D_{\beta\alpha\gamma} \right) , \\
{}^{(3)}R^\alpha_{\beta\gamma\delta} &= \partial_\gamma {}^{(3)}\Gamma^\alpha_{\delta\beta} - \partial_\delta {}^{(3)}\Gamma^\alpha_{\gamma\beta} + {}^{(3)}\Gamma^\alpha_{\gamma\epsilon} {}^{(3)}\Gamma^\epsilon_{\delta\beta} - {}^{(3)}\Gamma^\alpha_{\delta\epsilon} {}^{(3)}\Gamma^\epsilon_{\gamma\beta} - D^\epsilon_{\gamma\delta} {}^{(3)}\Gamma^\alpha_{\epsilon\beta} \,.
\end{aligned}
\tag{149}
$$

Remember that the indices are raised or lowered using the 3−metric in non-coordinate basis, i.e. using $h_{\alpha\beta}$. For example, $D_{\alpha\beta\gamma} = h_{\alpha\delta} D^\delta_{\beta\gamma}$, $h_{\mu\tau} {}^{(3)}\Gamma^\tau_{\sigma\rho} = {}^{(3)}\Gamma_{\mu\sigma\rho}$ and similarly 3−Ricci tensor is defined as ${}^{(3)}R_{\beta\delta} = {}^{(3)}R^\alpha_{\beta\alpha\delta}$ giving us the following result:

$$
\begin{aligned}
{}^{(3)}R_{\alpha\beta} = -\frac{1}{2} \Big( &\partial_\gamma D^\gamma_{\alpha\beta} + \partial_\gamma D^\gamma_{\beta\alpha} + \partial_\beta D^\gamma_{\gamma\alpha} + \partial_\alpha D^\gamma_{\gamma\beta} + D^{\gamma\delta}_\beta D_{\gamma\delta\alpha} \\
&+ D^{\gamma\delta}{}_\beta D_{\gamma\alpha\delta} - \frac{1}{2} D^{\gamma\delta}_\beta D_{\alpha\gamma\delta} + D^\gamma_{\gamma\delta} D^\delta_{\alpha\beta} + D^\gamma_{\gamma\delta} D^\delta_{\beta\alpha} \Big) \,.
\end{aligned}
\tag{150}
$$

Special Case: The case we are interested in is when non-coordinate basis coincides with invariant basis. Then $D_{\alpha\beta}^{\gamma} = C_{\alpha\beta}^{\gamma}$ are constants and their derivatives vanish. This leaves us with:

$$^{(3)}R_{\alpha}^{\beta} = \frac{1}{2h}\left[2C^{\beta\delta}C_{\alpha\delta} + C^{\delta\beta}C_{\alpha\delta} + C^{\beta\delta}C_{\delta\alpha} - C_{\delta}^{\delta}\left(C_{\alpha}^{\beta} + C_{\alpha}^{\beta}\right) + \delta_{\alpha}^{\beta}\left(\left(C_{\delta}^{\delta}\right)^2 - 2C^{\delta\gamma}C_{\delta\gamma}\right)\right], \quad (151)$$

where we define $C^{\alpha\beta}$ as $C_{\alpha\beta}^{\gamma} = \epsilon_{\alpha\beta\delta}C^{\delta\gamma}$ and $h = \det(h_{\alpha\beta})$.

Now we proceed to give explicit expressions for invariant basis and its dual for Bianchi IX universe as this is of the most importance to us in this work. Let the invariant basis be denoted by $\{\chi_{\mu}\}$ ($\mu = 1, 2, 3$) and its dual be $\{\sigma^{\mu}\}$. Then:

- Invariant Basis: We have $\chi_{\mu} = e_{\mu}^{a}(x)\partial_{a}$. Thus:

$$\begin{aligned}
\chi_1 &= -\cos r \cot\theta\, \partial_r - \sin r\, \partial_\theta + \cos r \csc\theta\, \partial_\phi\,, \\
\chi_2 &= \sin r \cot\theta\, \partial_r - \cos r\, \partial_\theta - \sin r \csc\theta\, \partial_\phi\,, \\
\chi_3 &= \partial_r\,,
\end{aligned} \quad (152)$$

from which we can read off the triads:

$$\begin{aligned}
e_1^r, e_1^\theta, e_1^\phi &= -\cos r \cot\theta\,, -\sin r\,, \cos r \csc\theta\,, \\
e_2^r, e_2^\theta, e_2^\phi &= \sin r \cot\theta\,, -\cos r\,, -\sin r \csc\theta\,, \\
e_3^r, e_3^\theta, e_3^\phi &= 1, 0, 0\,.
\end{aligned} \quad (153)$$

- Dual Invariant Basis: Recall $\sigma^{\mu} = e_{a}^{\mu}(x)dx^a$. Then:

$$\begin{aligned}
\sigma^1 &= -\sin r\, d\theta + \cos r \sin\theta\, d\phi\,, \\
\sigma^2 &= -\cos r\, d\theta - \sin r \sin\theta\, d\phi\,, \\
\sigma^3 &= dr + \cos\theta\, d\phi\,,
\end{aligned} \quad (154)$$

from which we can read off the inverse triads:

$$\begin{aligned}
e_r^1, e_\theta^1, e_\phi^1 &= 0, -\sin r\,, \cos r \sin\theta\,, \\
e_r^2, e_\theta^2, e_\phi^2 &= 0, -\cos r\,, -\sin r \sin\theta\,, \\
e_r^3, e_\theta^3, e_\phi^3 &= 1, 0, \cos\theta\,.
\end{aligned} \quad (155)$$

Using these results, a wide variety of useful results can be obtained which will be used later while imposing the homogeneous ansatz on Bianchi IX ADM Hamiltonian (Section 5.2). For example, from eq. (144), the homogeneous metric determinant becomes $h = \gamma \sin^2(\theta)$. Similarly $D_j\left(\sin(\theta)e_{\alpha}^{j}\right) = 0\ \forall\ \alpha$.[7]

---

[7]We can explicitly check this. For example, consider $\alpha = 1$, then we have

$$D_j\left(\sin(\theta)e_1^j\right) = D_r\left(\sin(\theta)e_1^r\right) + D_\theta\left(\sin(\theta)e_1^\theta\right) + D_\phi\left(\sin(\theta)e_1^\phi\right)\,.$$

Now we plug in the explicit forms of triads from eq. (153) to see that $D_j\left(\sin(\theta)e_1^j\right) = 0$. Similarly we can check for $\alpha = \{2, 3\}$.

**Einstein field equations in homogeneous universes**

We are interested in the vacuum case (pure gravity) as usual. This means that the 4−stress-energy-momentum tensor $T_{\mu\nu} = 0$. Using eq. (D.19), we see that the Einstein field equations become ${}^{(4)}R_{\mu\nu} = 0$. We impose on this the homogeneous ansatz for the metric, i.e. eq. (145) reproduced here for convenience:

$$ds^2 = -dt^2 + \gamma_{ij}dx^i dx^j = -dt^2 + h_{\alpha\beta}(t)e_i^\alpha e_j^\beta dx^i dx^j \,. \tag{156}$$

We first present the result in coordinate basis where we start with the metric $ds^2 = g_{\mu\nu}dx^\mu dx^\nu = -dt^2 + \gamma_{ij}dx^i dx^j$ to directly calculate the Christoffel symbols as follows:

$$ {}^{(4)}\Gamma_{ab}^0 = \frac{1}{2}\dot\gamma_{ab}\,, \quad {}^{(4)}\Gamma_{0b}^a = \frac{1}{2}\gamma^{ac}\dot\gamma_{cb}\,, \quad {}^{(4)}\Gamma_{bc}^a = {}^{(3)}\Gamma_{bc}^a\,, \tag{157}$$

while the remaining ones are identically zero. Here ${}^{(3)}\Gamma_{bc}^a$ is defined in an analogous manner (suited to 3−D) to how ${}^{(4)}\Gamma_{\alpha\beta}^\lambda$ is defined in Appendix A. Accordingly the 4−Riemann curvature is evaluated. Thus ${}^{(4)}R_{\mu\nu} = 0$ reduce to the following in a coordinate basis for a homogeneous ansatz:

$$
\begin{aligned}
{}^{(4)}R_{00} &= -\frac{1}{2}\partial_0\dot\gamma_a^a - \frac{1}{4}\dot\gamma_a^b\dot\gamma_b^a = 0\,, \\
{}^{(4)}R_{a0} &= \frac{1}{2}\left(D_b\dot\gamma_a^b - D_a\dot\gamma_b^b\right) = 0\,, \\
{}^{(4)}R_{ab} &= \frac{1}{2}\partial_0\dot\gamma_{ab} + \frac{1}{4}\left(\dot\gamma_{ab}\dot\gamma_c^c - 2\dot\gamma_{bc}\dot\gamma_a^c\right) + {}^{(3)}R_{ab} = 0\,.
\end{aligned} \tag{158}
$$

In non-coordinate basis that coincide with invariant basis, we use ${}^{(3)}R_{ab} = e_a^\alpha e_b^\beta {}^{(3)}R_{\alpha\beta}$. For the homogeneous metric, we now read off the metric from $ds^2 = -dt^2 + h_{\alpha\beta}e_i^\alpha e_j^\beta dx^i dx^j$ and use $\gamma_{ij} = h_{\alpha\beta}e_i^\alpha e_j^\beta$. When non-coordinate basis coincides with invariant basis, recall that $D_{\alpha\beta}^\gamma = C_{\alpha\beta}^\gamma$ are constants and their derivatives vanish. Thus ${}^{(4)}R_{\mu\nu} = 0$ reduce to the following in a non-coordinate basis (which coincide with invariant basis) for a homogeneous ansatz:

$$
\begin{aligned}
{}^{(4)}R_0^0 &= \frac{1}{2}\partial_0\dot h_\alpha^\alpha + \frac{1}{4}\dot h_\alpha^\beta\dot h_\beta^\alpha = 0\,, \\
{}^{(4)}R_\alpha^0 &= e_\alpha^{a\,(4)}R_a^0 = \frac{1}{2}\dot h_\beta^\gamma\left(C_{\gamma\alpha}^\beta - \delta_\alpha^\beta C_{\delta\gamma}^\delta\right) = 0\,, \\
{}^{(4)}R_\alpha^\beta &= e_\alpha^a e_b^\beta {}^{(4)}R_a^b = {}^{(3)}R_\alpha^\beta + \frac{1}{2\sqrt{h}}\partial_0\left(\sqrt{h}\dot h_\alpha^\beta\right) = 0\,.
\end{aligned} \tag{159}
$$

## 4.3   Dynamics of the Bianchi models in Lagrangian formalism

We now proceed towards finding a general solution. A general solution, by definition, means that it has to be completely stable and must satisfy arbitrary initial conditions. A perturbation should not change the form of the solution. First we discuss the methodology towards finding a general solution and then specialize to the case of Bianchi I & Bianchi IX universes as these two cosmological solutions are what we are interested in for the purposes of the work. In Subsection 4.3.1, we present the solutions of the vacuum case for Bianchi I universe (also known as the *Kasner solutions*) while in Subsection 4.3.2, we show Mixmaster dynamics in Bianchi IX universes.

We take the most general ansatz for a diagonal metric and calculate the Einstein field equations corresponding to this ansatz. In order to be able to do so, we introduce three spatial

vectors as $e^a = \ell(x^\gamma), m(x^\gamma), n(x^\gamma)$ and take the most general diagonal ansatz for $h_{\alpha\beta}$ (defined above in eq. (144)) as:

$$h_{\alpha\beta} = a^2(t)\ell_\alpha\ell_\beta + b^2(t)m_\alpha m_\beta + c^2(t)n_\alpha n_\beta . \tag{160}$$

Then just like we did above, using this metric, the *Einstein field equations for a generic homogeneous cosmological model in an empty space for a generic diagonal* $3-$*metric* become:

$$
\begin{aligned}
-R^l_l &= \frac{(\dot{a}bc)^{\cdot}}{abc} + \frac{1}{2a^2b^2c^2}\left[n_1^2 a^4 - \left(n_2 b^2 - n_3 c^2\right)^2\right] = 0\,, \\
-R^m_m &= \frac{(a\dot{b}c)^{\cdot}}{abc} + \frac{1}{2a^2b^2c^2}\left[n_2^2 b^4 - \left(n_1 a^2 - n_3 c^2\right)^2\right] = 0\,, \\
-R^n_n &= \frac{(ab\dot{c})^{\cdot}}{abc} + \frac{1}{2a^2b^2c^2}\left[n_3^2 c^4 - \left(n_1 a^2 - n_2 b^2\right)^2\right] = 0\,, \\
-R^0_0 &= \frac{\ddot{a}}{a} + \frac{\ddot{b}}{b} + \frac{\ddot{c}}{c} = 0\,,
\end{aligned}
\tag{161}
$$

where $(n_1, n_2, n_3) = (C_{11}, C_{22}, C_{33})$ (recall $C^c_{ab} = \epsilon_{abd}C^{dc}$ and eq. (129)) and off-diagonal terms of Ricci tensor vanish identically due to the choice of the diagonal form of $h_{\alpha\beta}$.

We now introduce a new temporal variable $\tau$ as $dt = abc\, d\tau$ (where $t$ is the coordinate time) as well as $\alpha = \ln(a)$, $\beta = \ln(b)$ and $\gamma = \ln(c)$. Then the Einstein field equations further simplify to:

$$
\boxed{
\begin{aligned}
2\alpha_{\tau\tau} &= \left(n_2 b^2 - n_3 c^2\right)^2 - n_1^2 a^4\,, \\
2\beta_{\tau\tau} &= \left(n_1 a^2 - n_3 c^2\right)^2 - n_2^2 b^4\,, \\
2\gamma_{\tau\tau} &= \left(n_1 a^2 - n_2 b^2\right)^2 - n_3^2 c^4\,, \\
\frac{1}{2}(\alpha + \beta + \gamma)_{\tau\tau} &= \alpha_\tau\beta_\tau + \alpha_\tau\gamma_\tau + \beta_\tau\gamma_\tau\,,
\end{aligned}
}
\tag{162}
$$

where $A_\tau = \frac{dA}{d\tau}$ and $A_{\tau\tau} = \frac{d^2 A}{d\tau^2}$.

These are the homogeneous Einstein field equations in vacuum corresponding to the most generic diagonal ansatz for the metric in eq. (160) that needs to be solved.

### 4.3.1 Kasner solution

The vacuum solutions for the case of Bianchi I universe is known as *Kasner solution*. For the case of Bianchi I universe, we realize that from eq. (129) that $a = n_1 = n_2 = n_3 = 0$. They all vanish. Accordingly, the RHS of eq. (162) vanish.

We now proceed to solve for Bianchi I explicitly using eq. (159). We know for Bianchi I universe from eq. (134) that $C^\alpha_{\beta\gamma}$ and the triads $e^a_\alpha = \delta^a_\alpha$. We plug this back in eq. (159) to get (the second equation becomes trivial):

$$
\begin{aligned}
\partial_0\dot{h}^\alpha_\alpha + \frac{1}{2}\dot{h}^\beta_\alpha\dot{h}^\alpha_\beta &= 0\,, \\
\frac{1}{\sqrt{h}}\partial_0\left(\sqrt{h}\dot{h}^\beta_\alpha\right) &= 0\,.
\end{aligned}
\tag{163}
$$

The second equation implies:

$$\sqrt{h}\dot{h}^\beta_\alpha = \text{ constant} = 2\lambda^\beta_\alpha \quad \text{(say)}, \tag{164}$$

where $\lambda^\beta_\alpha$ is a matrix of coefficients that can be reduced to its diagonal form. This makes the first equation as:

$$\partial_0\dot{h}^\alpha_\alpha = -\frac{2}{h}\lambda^\beta_\alpha\lambda^\alpha_\beta . \tag{165}$$

Substituting $\dot{h}^\alpha_\alpha$ from eq. (164) into eq. (165), we get $\frac{\dot{h}}{2\sqrt{h}} = $ constant, solving which gives:

$$h = Ct^2, \tag{166}$$

for some constant $C$. Accordingly eqs. (164, 165) becomes:

$$\dot{h}^\beta_\alpha = \frac{2}{Ct}\lambda^\beta_\alpha, \\ \partial_0 \dot{h}^\alpha_\alpha = -\frac{2}{Ct^2}\lambda^\beta_\alpha \lambda^\alpha_\beta. \tag{167}$$

Without loss of generality, we can always rescale the spacetime coordinates $x^\mu$ such that the constant becomes one, or in other words $\lambda^\alpha_\alpha = 1$. Then we substitute $\dot{h}^\alpha_\alpha$ from the first equation into the second in eq. (167) where we use $\lambda^\alpha_\alpha = 1$ to get from the second equation:

$$\lambda^\alpha_\beta \lambda^\beta_\alpha = 1. \tag{168}$$

Next, putting the constant to unity, if we lower the index $\beta$ in the first equation of eq. (167) using $h_{\beta\gamma}$, we get:

$$\dot{h}_{\alpha\beta} = \frac{2}{t}\lambda^\gamma_\alpha h_{\beta\gamma}. \tag{169}$$

This is the system of ordinary differential equations that needs to be solved to get the 3−metric. In order to do so, we diagonalize $\lambda^\gamma_\alpha$ by its eigenvalues $p_1, p_2, p_3$ (all real & different) having eigenvectors $\vec{n}^{(1)}, \vec{n}^{(2)}, \vec{n}^{(3)}$. Then the solution to the system of ODEs in eq. (168) is given by:

$$h_{\alpha\beta} = t^{2p_1} n^{(1)}_\alpha n^{(1)}_\beta + t^{2p_2} n^{(2)}_\alpha n^{(2)}_\beta + t^{2p_3} n^{(3)}_\alpha n^{(3)}_\beta. \tag{170}$$

Since the triads for Bianchi I is given by $e^a_\alpha = \delta^a_\alpha$, then we can choose the frame of eigenvectors that resemble the spatial coordinates denoted by $x^1, x^2, x^3$. Thus the spatial line element of Bianchi I for the vacuum becomes:

$$\boxed{dl^2 = t^{2p_1}\left(dx^1\right)^2 + t^{2p_2}\left(dx^2\right)^2 + t^{2p_3}\left(dx^3\right)^2,} \tag{171}$$

where $p_1, p_2, p_3$ are called Kasner exponents that satisfy:

- $\lambda^\alpha_\alpha = 1 \qquad \Rightarrow p_1 + p_2 + p_3 = 1,$

- $\lambda^\alpha_\beta \lambda^\beta_\alpha = 1 \qquad \Rightarrow (p_1)^2 + (p_2)^2 + (p_3)^2 = 1.$

Except for the cases $(0,0,1)$ and $(-\frac{1}{3}, \frac{2}{3}, \frac{2}{3})$, the Kasner exponents are never equal and one is always negative while the other two being positive. In fact if we choose a particular ordering for Kasner exponents, say, $p_1 < p_2 < p_3$, then the range of Kasner exponents are:

$$-\frac{1}{3} \le p_1 \le 0, \qquad 0 \le p_2 \le \frac{2}{3}, \qquad \frac{2}{3} \le p_3 \le 1. \tag{172}$$

Thus we have solved the vacuum case of Bianchi I universe and realize that (i) volumes grow linearly in time, (ii) the linear distances grow along two directions while decrease along the third (unlike the FRWL solution), and (iii) the metric obtained has only *one true singularity at* $t = 0$ with the only exception of $\{p_1, p_2, p_3\} = (0,0,1)$ case where using the transformations $t\sinh(x^3) = \zeta$ and $t\cosh(x^3) = \tau$, the metric reduces to a Galilean form having a fictitious singularity in a flat spacetime. We have used the Lagrangian formulation here to get these results and will establish the equivalence with the Hamiltonian formulation of Bianchi I universe in Chapter 5.1.

### 4.3.2 Mixmaster dynamics in Bianchi IX universe

Finally we consider the behaviour of the solutions for the Bianchi IX model which is of interest to us. Referring back to eq. (162), for the case of Bianchi IX universe, we have $(n_1, n_2, n_3) = (C_{11}, C_{22}, C_{33}) = (1, 1, 1)$. Thus the Einstein field equations become:

$$2\alpha_{\tau\tau} = \left(b^2 - c^2\right)^2 - a^4,$$
$$2\beta_{\tau\tau} = \left(a^2 - c^2\right)^2 - b^4,$$
$$2\gamma_{\tau\tau} = \left(a^2 - b^2\right)^2 - c^4, \tag{173}$$
$$\frac{1}{2}(\alpha + \beta + \gamma)_{\tau\tau} = \alpha_\tau \beta_\tau + \alpha_\tau \gamma_\tau + \beta_\tau \gamma_\tau,$$

where we recall the definitions of $\{\alpha, \beta, \gamma\} = \{\ln(a), \ln(b), \ln(c)\}$ while $\{a, b, c\}$ are defined in the ansatz for the metric in eq. (160).

We realize that if we neglect the RHS of the first three equations in eq. (173), we recover the Bianchi I universe (plug $(n_1, n_2, n_3) = (C_{11}, C_{22}, C_{33}) = (0, 0, 0)$ for Bianchi I in eq. (162) to see this) whose solution is the Kasner solution (derived in Subsection 4.3.1). In this case the fourth equation no longer remains an independent equation. So the way we proceed now is to consider the Kasner approximation of the eq. (173) within Bianchi IX universe and study the dynamics and stability of such solutions within the context of Bianchi IX universe.

In the Kasner approximation, we begin with considering the ordering $p_1 < p_2 < p_3$ with $p_1$ as being negative for the Kasner exponents that appeared in the Kasner solution (eq. (171)). We identify $p_1 = p_\ell$, $p_2 = p_m$ and $p_3 = p_n$ where the spatial vectors $\ell, m$ & $n$ are defined in eq. (160). Since flat FRWL universe is an isotropic case of Bianchi I, here $p_\ell$ corresponds to the scale factor of FRWL solution $a(t)$. Moreover, we derived in Subsection 4.3.1 that $t = 0$ is the true singularity of Bianchi I universe, we have the following behaviour of one of the directions in the vicinity of the singularity:

$$p_1 < 0 \rightarrow p_1 = -|p_1|, \quad \begin{cases} \alpha \sim -|p_1| \ln t \\ a \sim \frac{1}{t^{|p_1|}} \end{cases} \quad \text{increases for } t \rightarrow 0, \tag{174}$$

while for other two directions:

$$\begin{aligned} p_2 > 0 \rightarrow p_2 = |p_2|, \beta \sim |p_2| \ln t, \\ p_3 > 0 \rightarrow p_3 = |p_3|, \gamma \sim |p_3| \ln t, \end{aligned} \quad \text{decreases for } t \rightarrow 0. \tag{175}$$

Then we have $a \sim t^{p_1}$, $b \sim t^{p_2}$, $c \sim t^{p_3} \Rightarrow abc = \Lambda t$ ($\Lambda = $ constant) (since $p_1 + p_2 + p_3 = 1$). Since $dt = abc\, d\tau = \Lambda t\, d\tau$, we have $d\tau = \frac{d\ln(t)}{\Lambda}$. The initial time is when $t \rightarrow +\infty$ and the singularity is at $t = 0$. Then in terms of $\tau$, we have the initial time at $\tau \rightarrow +\infty$ and the singularity is approached when $\tau \rightarrow -\infty$. Thus:

$$\alpha_\tau = \Lambda p_1, \qquad \beta_\tau = \Lambda p_2, \qquad \gamma_\tau = \Lambda p_3, \tag{176}$$

which clearly satisfies the Kasner approximation of the four Bianchi IX equations (173) (i.e., RHS $\approx 0$ in the first three equations). These can be taken as the initial conditions ($\tau \rightarrow +\infty$).

Thus the Kasner approximation in eq. (173) cannot persist forever because the RHS always contains one increasing quantity (eqs. (174)) near the singularity. In order to study its effects, we focus on eq. (173) where we only consider the increasing terms in the RHS to further simplify the Einstein field equations as (recall from the definition of $\alpha$ that $a = e^\alpha$):

$$\alpha_{\tau\tau} = -\frac{1}{2}a^4 = -\frac{1}{2}e^{4\alpha},$$
$$\beta_{\tau\tau} = +\frac{1}{2}a^4 = \frac{1}{2}e^{4\alpha},$$
$$\gamma_{\tau\tau} = +\frac{1}{2}a^4 = \frac{1}{2}e^{4\alpha},$$

which can be integrated using initial conditions in eq. (176) as follows:

$$
\begin{aligned}
a^2 &= \frac{2|p_1|\Lambda}{\cosh(2|p_1|\Lambda\tau)}, \\
b^2 &= b_0^2 \exp[2\Lambda(p_2 - |p_1|)\tau]\cosh(2|p_1|\Lambda\tau), \\
c^2 &= c_0^2 \exp[2\Lambda(p_3 - |p_1|)\tau]\cosh(2|p_1|\Lambda\tau),
\end{aligned}
\tag{177}
$$

where $b_0$ and $c_0$ are integration constants.

Towards the singularity $t \to 0$ ($\tau \to -\infty$), we get [23, 24]:

$$
\begin{aligned}
a &\sim \exp[-\Lambda p_1 \tau], \\
b &\sim \exp[\Lambda(p_2 + 2p_1)\tau], \\
c &\sim \exp[\Lambda(p_3 + 2p_1)\tau], \\
t &\sim \exp[\Lambda(1 + 2p_1)\tau].
\end{aligned}
\tag{178}
$$

Thus we have a new Kasner epoch where we express $\{a, b, c\}$ in terms of the new Kasner exponents $\{p_1', p_2', p_3'\}$ as: $a \sim t^{p_1'}$, $b \sim t^{p_2'}$, $c \sim t^{p_3'}$, such that $abc = \Lambda' t$ (compare with the old Kasner epoch below eq. (175)) where:

$$
\begin{aligned}
p_1' &= \frac{|p_1|}{1 - 2|p_1|}, & p_2' &= -\frac{2|p_1| - p_2}{1 - 2|p_1|}, \\
p_3' &= \frac{p_3 - 2|p_1|}{1 - 2|p_1|}, & \Lambda' &= (1 - 2|p_1|)\Lambda.
\end{aligned}
\tag{179}
$$

Thus we see the effect of perturbation over the Kasner regime that a Kasner epoch is replaced by another one so that the negative power of $t$ is transferred from the $\vec{\ell}$ to the $\vec{m}$ direction. So if the original solution had $p_1 < 0$ then in the new solution, $p_2' < 0$ while $p_1' > 0$. The exponents of the new Kasner epoch in terms of the old ones are expressed in eq. (179). Accordingly the previously increasing perturbation in one direction dampens and eventually vanishes while other perturbations in other directions increase. This leads to replacement of one Kasner epoch by another in the Bianchi IX universe. These changes in the Kasner epochs from one straight line motion to the next straight line motion can be visualized as a representative point moving on a straight line (one Kasner epoch) and then bouncing off the potential walls in the Bianchi IX universe and then setting off onto another straight line motion (another Kasner epoch) until the next collision with the potential walls happens. *Thus we conclude that the Kasner solutions in Bianchi IX universe are unstable.* This is the socalled Mixmaster behaviour. We will encounter this again from a different perspective of ADM (Hamiltonian) analysis of Bianchi IX universe in Chapter 5.2 where the graphical picture of a particle bouncing off the potential walls will be made more precise.

# 5 ADM formalism of homogeneous cosmologies

We study in this chapter the homogeneous cosmologies, like we did in Chapter 4, but in the ADM formalism. In Section 5.1, we start with the vacuum case of Bianchi I universe, much like what we derived in Chapter 4.3.1, and recover the results obtained therein. In Section 5.2, we do the ADM analysis of the vacuum case of Bianchi IX universe where we study its dynamics in details using the 3+1−variables. We illustrate the Mixmaster dynamics that we already showed in Chapter 4.3.2 and provide a graphical picture of the mechanism. Mixmaster dynamics lead to infinite number of shifts from one Kasner epoch to another before the particle reaches the big-bang singularity. Then to further practice the ADM formalism, we provide two more examples in this chapter: (i) in Section 5.3, we introduce a free & massless classical scalar field that is coupled to the Bianchi IX universe, and (ii) in Section 5.4, we extend the previous

system to the case of a classical scalar field with a potential term. Through these two examples, we further show that how the Mixmaster dynamics of Bianchi IX universe can be averted in the presence of a scalar field. This phenomenon of having only a finite number of bounces (finite number of shifts from one Kasner epoch to another) before the particle reaches the big-bang singularity is known as *quiescence*. We wish to remind the readers that we are only considering classical objects throughout this work.

## 5.1 Bianchi I universe

We consider the ADM Hamiltonian in eq. (96) and the corresponding Hamiltonian constraint (eq. (97)) & the diffeomorphism constraints (eq. (98)), then apply them in the context of Bianchi I universe. We recall from eqs. (129, 134) that Bianchi I has flat spatially homogeneous hypersurfaces. Moreover, in Bianchi I universe, coordinate basis is the invariant basis and triads are given by $e_\alpha^a = \delta_\alpha^a$. Accordingly the 3−curvature is zero. Therefore for Bianchi I, the ADM Hamiltonian (eqs. (96, 97, 98)) reduces to:

$$H_{\text{ADM}} = \mathbb{H}[N] + \mathbb{D}[N^j] = N\mathbb{H} + N^j\mathbb{D}_j, \tag{180}$$

where $\mathbb{H}[N]$ is the *Hamiltonian constraint* given by:

$$\mathbb{H}[N] \equiv \int_{\Sigma_t} d^3x N\left[-\frac{1}{\sqrt{\gamma}}\left(\frac{\pi^2}{2} - \pi^{ij}\pi_{ij}\right)\right], \tag{181}$$

and $\mathbb{D}[N^i]$ is the *diffeomorphism constraint* given by:

$$\mathbb{D}[\vec{N}] \equiv \int_{\Sigma_t} d^3x N^i\left[-2D^j\pi_{ij}\right]. \tag{182}$$

We impose the *homogeneous ansatz* by making use of the explicit forms of triads $e_\alpha^a = \delta_\alpha^a$ and realizing that in Bianchi I universe, we have the non-coordinate basis coincide with the coordinate basis, to get (see eq. (145)):

$$\begin{aligned} \gamma_{ij} &\to h_{\alpha\beta}(t), \\ \pi^{ij} &\to \pi^{\alpha\beta}(t), \end{aligned} \tag{183}$$

where the spatial homogeneity is reflected in either of the bases.

Then the spatial homogeneity of 3−conjugate momenta in Bianchi I leads to:

$$\boxed{\mathbb{D}_{\text{BI}}[\vec{N}] = 0,} \tag{184}$$

as $\pi^{\alpha\beta}(t)$ depends only on time in the invariant basis.

The Hamiltonian constraint becomes:

$$\mathbb{H}_{\text{BI}}[N] = \left(\int_{\Sigma_t} d^3x N\right)\left(\frac{1}{\sqrt{h}}\left(\pi^{\alpha\beta}\pi_{\alpha\beta} - \frac{\pi^2}{2}\right)\right), \tag{185}$$

where the spatial components are put together in the integrand of $\left(\int_{\Sigma_t} d^3x N\right)$ which we can simply call $n$. Thus we have the total ADM Hamiltonian for Bianchi I universe as the Hamiltonian constraint itself ($n$ being the Lagrange multiplier):

$$\boxed{H_{\text{Bianchi I}} = \mathbb{H}_{\text{BI}}[N] = \frac{n}{\sqrt{h}}\left(\pi^{\alpha\beta}\pi_{\alpha\beta} - \frac{\pi^2}{2}\right),} \tag{186}$$

where the constraint relations become $H_{\text{Bianchi I}} = \mathbb{H}_{\text{BI}}[n] \approx 0$ as the diffeomorphism constraints are trivially satisfied because of being identically zero everywhere.

Of course, the Hamiltonian formalism should lead to the same result as obtained in Section 4.3.1. We will show that it indeed is true. We start with calculating the equations of motion:

$$\dot{h}_{\alpha\beta} = \left\{ h_{\alpha\beta}, H_{\text{Bianchi I}} \right\},$$
$$\dot{\pi}^{\alpha\beta} = \left\{ \pi^{\alpha\beta}, H_{\text{Bianchi I}} \right\}, \tag{187}$$

where we use the fact that $h_{\alpha\beta}$ and $\pi^{\alpha\beta}$ are independent variables but $\pi_{\alpha\beta} = h_{\alpha\delta} h_{\beta\gamma} \pi^{\delta\gamma}$ & $\pi = \pi^{\alpha\beta} h_{\alpha\beta}$ are not independent from $h_{\alpha\beta}$. We recall the definition of Poisson brackets from eq. (104) which we reproduce here suited to our variables:

$$\{f(x), h(y)\}|_{(h,\pi)} = \int d^3z \left[ \frac{\delta f(x)}{\delta h_{\alpha\beta}(z)} \frac{\delta h(y)}{\delta \pi^{\alpha\beta}(z)} - \frac{\delta f(x)}{\delta \pi^{\alpha\beta}(z)} \frac{\delta h(y)}{\delta h_{\alpha\beta}(z)} \right]. \tag{188}$$

Then calculating the brackets and using $\frac{\partial h_{\delta\gamma}}{\partial h_{\alpha\beta}} = \delta_\delta^\alpha \delta_\gamma^\beta$ and $\frac{\partial \pi^{\delta\gamma}}{\partial \pi^{\alpha\beta}} = \delta_\alpha^\delta \delta_\beta^\gamma$, we get:

$$\dot{h}_{\alpha\beta} = \frac{2n}{\sqrt{h}} \left( \pi_{\alpha\beta} - \frac{1}{2} h_{\alpha\beta} \pi \right),$$
$$\dot{\pi}^{\alpha\beta} = -\frac{2n}{\sqrt{h}} \left( \pi_\gamma^\alpha \pi^{\gamma\beta} - \frac{1}{2} \pi^{\alpha\beta} \pi \right), \tag{189}$$

where there is an additional term in $\dot{\pi}^{\alpha\beta}$ corresponding to the variation of $\frac{1}{\sqrt{h}}$ present in $\mathbb{H}_{\text{Bianchi I}}$ in eq. (186). But that variation term has the coefficient $\left( \pi^{\alpha\beta} \pi_{\alpha\beta} - \frac{\pi^2}{2} \right)$, which we take as zero due to the constraint relation $\mathbb{H}_{\text{Bianchi I}} \approx 0$. Thus this equation of motion is weakly equal which is not a problem because we are always doing our analyses on some hypersurface & not in between them (see eq. (99) & the text below it). Here $n \equiv \left( \int d^3x N \right)$ contains the entities dependent on spatial coordinates in its integrand which gets integrated out. Thus imposing the homogeneous ansatz for the metric removes all spatial dependencies thereby leaving us with only the time dependencies. Accordingly the constraint equations when expressed in terms of the homogeneous metric ansatz become global constraints.

We can combine these two equations to get:

$$\partial_0 \left( \pi^{\alpha\delta} h_{\delta\beta} \right) = \dot{\pi}^{\alpha\delta} h_{\delta\beta} + \pi^{\alpha\delta} \dot{h}_{\delta\beta} = 0. \tag{190}$$

Hence $\pi^{\alpha\delta} h_{\delta\beta} = \pi_\beta^\alpha = $ constant. Then the first equation in eq. (189) becomes of the form:

$$\sqrt{h} \dot{h}_\beta^\alpha = \text{ constant}, \tag{191}$$

which is of the same form as eq. (164). The rest of the argument is the same as followed after eq. (164) in Section 4.3.1. Thus we have shown for this case the equivalence between the Lagrangian & the Hamiltonian formulations.

## 5.2 Bianchi IX Universe

We refer the readers to Section 4.1 where in eq. (135) we summarize the expressions that hold for Bianchi IX universe. The important observation to be made is that the non-coordinate basis for Bianchi IX is the same as the coordinate basis used for a 3−sphere embedded in $\mathbb{R}^4$ with a constraint on the radius (which we can set to unity without loss of generality), namely the hyperspherical coordinates $x^\mu$ ($\mu = 1, 2, 3, 4$) given by [1]:

$$\begin{cases} x^1 = \cos r, \\ x^2 = \sin r \cos \theta, \\ x^3 = \sin r \sin \theta \cos \phi, \\ x^4 = \sin r \sin \theta \sin \phi, \end{cases} \tag{192}$$

satisfying $\left(x^1\right)^2 + \left(x^2\right)^2 + \left(x^3\right)^2 + \left(x^4\right)^2 = 1$ where $r, \theta \in [0, \pi]$ and $\phi \in [0, 2\pi]$.

The algebra is non-trivial (see eq. (135)), unlike Bianchi I where $C^{\gamma}_{\alpha\beta} = 0$, and is given by (for Killing vector fields):

$$\left[\xi_\alpha, \xi_\beta\right] = \epsilon_{\alpha\beta\gamma}\xi^{\gamma}. \tag{193}$$

We again start with the full ADM Hamiltonian (eqs. (96, 97, 98)) which we reproduce here for convenience:

$$H_{\text{ADM}} = \mathbb{H}[N] + \mathbb{D}\left[N^j\right] = N\mathbb{H} + N^j\mathbb{D}_j, \tag{194}$$

where $\mathbb{H}[N]$ is the Hamiltonian constraint given by:

$$\mathbb{H}[N] \equiv \int_{\Sigma_t} d^3x N \left[\underbrace{-\sqrt{\gamma}^{(3)}R}_{\text{Term I}} \underbrace{-\frac{1}{\sqrt{\gamma}}\left(\frac{\pi^2}{2} - \pi^{ij}\pi_{ij}\right)}_{\text{Term II}}\right], \tag{195}$$

and $\mathbb{D}\left[N^i\right]$ is the diffeomorphism constraint given by:

$$\mathbb{D}[\vec{N}] \equiv \int_{\Sigma_t} d^3x N^i \left[-2D^j\pi_{ij}\right]. \tag{196}$$

Then we impose the homogeneous ansatz for the metric (eq. (145)) in invariant basis where we make use of the explicit forms of triads given for Bianchi IX universe (eqs. (152)-(155)) and recall that $\pi^{ij}$ is a tensor density to get:

$$\begin{aligned}
\gamma_{ij} &= h_{\alpha\beta}(t)e_i^\alpha e_j^\beta \quad \Rightarrow \gamma = h\sin^2(\theta), \\
\pi^{ij} &= \sin(\theta)\pi^{\alpha\beta}(t)e_\alpha^i e_\beta^j.
\end{aligned} \tag{197}$$

We first focus on the Hamiltonian constraint. Recall the orthogonality conditions of the triads $e_a^\alpha e_\alpha^b = \delta_a^b$ and $e_a^\alpha e_\beta^a = \delta_\beta^\alpha$. Term II is relatively straightforward where we use:

$$\begin{aligned}
-\frac{1}{\sqrt{\gamma}}\frac{\pi^2}{2} &= -\frac{1}{\sqrt{\gamma}}\frac{\left(\pi^{ij}\gamma_{ij}\right)^2}{2} = -\frac{1}{2\sqrt{h}\sin(\theta)}\left(\pi^{\alpha\beta}h_{\alpha\beta}\right)^2\sin^2(\theta) = -\frac{\sin(\theta)}{2\sqrt{h}}\left(\pi^{\alpha\beta}h_{\alpha\beta}\right)^2, \\
\frac{1}{\sqrt{\gamma}}\pi^{ij}\pi_{ij} &= \frac{\sin(\theta)}{\sqrt{h}}\pi^{\alpha\beta}\pi_{\alpha\beta},
\end{aligned} \tag{198}$$

to get for term II:

$$\begin{aligned}
\text{Term II} &= \int_{\Sigma_t} d^3x N \left[-\frac{1}{\sqrt{\gamma}}\left(\frac{\pi^2}{2} - \pi^{ij}\pi_{ij}\right)\right] \\
&= \left(\int_{\Sigma_t} d^3x N \sin(\theta)\right)\left[-\frac{1}{2\sqrt{h}}\left(\pi^{\alpha\beta}h_{\alpha\beta}\right)^2 + \frac{1}{\sqrt{h}}\pi^{\alpha\beta}\pi_{\alpha\beta}\right] \\
&= n\left(-\frac{1}{2\sqrt{h}}\left(\pi^{\alpha\beta}h_{\alpha\beta}\right)^2 + \frac{1}{\sqrt{h}}\pi^{\alpha\beta}\pi_{\alpha\beta}\right),
\end{aligned} \tag{199}$$

where $n \equiv \left(\int_{\Sigma_t} d^3x N \sin(\theta)\right)$ contains all spatial dependence in the integrand which gets integrated out, leaving us with time dependencies only.

Next we simplify term I for which we need to use the expression for 3—Ricci tensor for homogeneous universes in invariant basis provided in eq. (151). We contract the indices $\alpha$ and $\beta$ in eq. (151) to get:

$$^{(3)}R^\alpha_\alpha = \frac{1}{2h}\Big[2C^{\alpha\delta}C_{\alpha\delta} + C^{\delta\alpha}C_{\alpha\delta} + C^{\alpha\delta}C_{\delta\alpha} - C^\delta_\delta\left(C^\alpha_\alpha + C^\alpha_\alpha\right) + \delta^\alpha_\alpha\left(\left(C^\delta_\delta\right)^2 - 2C^{\delta\gamma}C_{\delta\gamma}\right)\Big]. \quad (200)$$

Then we use $\delta^\alpha_\alpha = 3$ and simplify this expression to get:

$$\begin{aligned}
\Rightarrow {}^{(3)}R^\alpha_\alpha &= \frac{1}{2h}\Big[4C^{\alpha\delta}C_{\alpha\delta} - \left(C^\delta_\delta\right)^2 - \left(C^\delta_\delta\right)^2 + 3\left(C^\delta_\delta\right)^2 - 6C^{\alpha\delta}C_{\alpha\delta}\Big] \\
&= \Big[\left(C^\alpha_\alpha\right)^2 - 2C^{\alpha\beta}C_{\alpha\beta}\Big].
\end{aligned} \quad (201)$$

Then we rewrite $C^\alpha_\alpha = C^{\alpha\beta}h_{\alpha\beta}$ and $C^{\alpha\beta}C_{\alpha\beta} = C^{\alpha\beta}C^{\delta\gamma}h_{\alpha\delta}h_{\beta\gamma}$ as well as recall that for Bianchi IX universe, we have $C^{\alpha\beta} = \delta^{\alpha\beta}$ (see eq. (135)). Thus we have:

$$\left(C^\alpha_\alpha\right)^2 = (\mathrm{Tr}(h))^2, \qquad C^{\alpha\beta}C_{\alpha\beta} = \mathrm{Tr}\left(h^2\right). \quad (202)$$

Thus term I becomes:

$$\begin{aligned}
\text{Term I} &= \int_{\Sigma_t} d^3x N\left(-\sqrt{\gamma}^{(3)}R\right) \\
&= \left(\int_{\Sigma_t} d^3x N\sin(\theta)\right)\left[\frac{1}{\sqrt{h}}\mathrm{Tr}\left(h^2\right) - \frac{1}{2\sqrt{h}}(\mathrm{Tr}(h))^2\right] \\
&= \frac{n}{\sqrt{h}}\left(\mathrm{Tr}\left(h^2\right) - \frac{1}{2}(\mathrm{Tr}(h))^2\right),
\end{aligned} \quad (203)$$

where $n \equiv \left(\int_{\Sigma_t} d^3x N\sin(\theta)\right)$ contains all spatial dependence in the integrand which gets integrated out, leaving us with time dependencies only.

Thus we have for the Hamiltonian constraint in Bianchi IX universe the following (with $n$ being the Lagrange multiplier):

$$\boxed{\mathbb{H}_{\text{BIX}}[n] = \frac{n}{\sqrt{h}}\left(\mathrm{Tr}\left(h^2\right) - \frac{1}{2}(\mathrm{Tr}(h))^2 - \frac{1}{2}\left(\pi^{\alpha\beta}h_{\alpha\beta}\right)^2 + \pi^{\alpha\beta}\pi_{\alpha\beta}\right) \approx 0,} \quad (204)$$

where $n \equiv \left(\int_{\Sigma_t} d^3x N\sin(\theta)\right)$ contains all the spatial dependence in the integrand which gets integrated out, leaving us with time dependencies only.

For the diffeomorphism constraint, we similarly use the explicit form for the triads (eqs. (152)-(155)) and define for the spatially dependent integrand in $n^i \equiv \left(\int_{\Sigma_t} d^3x N^i\sin(\theta)\right) = n^\alpha e^i_\alpha$ to get:

$$\boxed{\mathbb{D}_{\text{BIX}}[n^\alpha] = n^\alpha\mathbb{D}_\alpha = n^\alpha\left(2\epsilon^\gamma_{\alpha\beta}\pi^{\beta\delta}h_{\delta\gamma}\right) \approx 0.} \quad (205)$$

Thus we have the total ADM Hamiltonian for Bianchi IX universe as follows:

$$\boxed{\begin{aligned}
H_{\text{Bianchi IX}} &= \mathbb{H}_{\text{BIX}}[n] + \mathbb{D}_{\text{BIX}}[n^\alpha] \\
&= \frac{n}{\sqrt{h}}\left[\mathrm{Tr}\left(h^2\right) - \frac{1}{2}(\mathrm{Tr}(h))^2 - \frac{1}{2}\left(\pi^{\alpha\beta}h_{\alpha\beta}\right)^2 + \pi^{\alpha\beta}\pi_{\alpha\beta}\right] + n^\alpha\left(2\epsilon^\gamma_{\alpha\beta}\pi^{\beta\delta}h_{\delta\gamma}\right),
\end{aligned}} \quad (206)$$

where the constraint relations become $\mathbb{H}_{\text{BIX}}[n] \approx 0$ and $\mathbb{D}_{\text{BIX}}[n^\alpha] \approx 0$ (non-trivial here unlike the Bianchi I universe).

As a reminder, we are throughout using the invariant basis, whose importance can be appreciated by now hopefully in the context of homogeneous cosmologies. Now we proceed to solve the diffeomorphism constraints for the metric and its conjugate momenta. We use the form in eq. (205) :

$$
\begin{aligned}
2n^\delta \epsilon_{\tau\delta\beta} \delta^{\tau\gamma} \pi^{\beta\alpha} h_{\alpha\gamma} &= 2n^\delta h_{\alpha 1} \delta^{11} \epsilon_{1\delta\beta} \pi^{\beta\alpha} + 2n^\delta h_{\alpha 2} \delta^{22} \epsilon_{2\delta\beta} \pi^{\beta\alpha} + 2n^\delta h_{\alpha 3} \delta^{33} \epsilon_{3\delta\beta} \pi^{\beta\alpha} \\
&= 2n^3 h_{1\alpha} \pi^{\alpha 2} \epsilon_{132} + 2n^2 h_{1\alpha} \pi^{\alpha 3} \epsilon_{123} + 2n^1 h_{2\alpha} \pi^{\alpha 3} \epsilon_{213} \\
&\quad + 2n^3 h_{2\alpha} \pi^{\alpha 1} \epsilon_{231} + 2n^2 h_{3\alpha} \pi^{\alpha 1} \epsilon_{321} + 2n^1 h_{3\alpha} \pi^{\alpha 2} \epsilon_{312} \\
&= -2n^1 \left( h_{2\alpha} \pi^{\alpha 3} - h_{3\alpha} \pi^{\alpha 2} \right) - 2n^2 \left( h_{3\alpha} \pi^{\alpha 1} - h_{1\alpha} \pi^{\alpha 3} \right) \\
&\quad - 2n^3 \left( h_{1\alpha} \pi^{\alpha 2} - h_{2\alpha} \pi^{\alpha 1} \right).
\end{aligned}
$$

Thus the diffeomorphism constraints in eq. (205) give us the following:

$$
\Rightarrow -2 \left[ \left( \pi^{\alpha\tau} h_{\tau\beta} - \pi^{\beta\tau} h_{\tau\alpha} \right) \right] \approx 0. \tag{207}
$$

Then we observe that the LHS contains the matrix commutator as follows:

$$
\boxed{[\pi, h]_\beta^\alpha = \left( \pi^{\alpha\tau} h_{\tau\beta} - \pi^{\beta\tau} h_{\tau\alpha} \right) = -[h, \pi]_\beta^\alpha,} \tag{208}
$$

and the diffeomorphism constraints give us for the Bianchi IX universe:

$$
\boxed{[\pi, h]_\beta^\alpha \approx 0.} \tag{209}
$$

For diffeomorphism constraints to be *second class constraints*,[8] we impose a particular form of gauge fixing. To motivate the choice of gauge fixing to be applied, we calculate the Poisson brackets using the definition provided in eq. (188). To be clear, for example, $[\pi, h]_2^1 = \left( \pi^{1\tau} h_{\tau 2} - \pi^{2\tau} h_{\tau 1} \right)$ and we recall that $h_{\alpha\beta} = h_{\beta\alpha}$ as well as $\pi^{\alpha\beta} = \pi^{\beta\alpha}$ in general. Then we have:

$$
\begin{aligned}
&\left\{ [\pi, h]_3^2, h_{12} \right\} = -h_{13} \approx 0, &\quad &\left\{ [\pi, h]_3^1, h_{12} \right\} = -h_{23} \approx 0, \\
&\left\{ [\pi, h]_2^1, h_{23} \right\} = h_{13} \approx 0, &\quad &\left\{ [\pi, h]_3^1, h_{23} \right\} = -h_{12} \approx 0, \\
&\left\{ [\pi, h]_2^1, h_{13} \right\} = -h_{23} \approx 0, &\quad &\left\{ [\pi, h]_3^2, h_{13} \right\} = h_{12} \approx 0,
\end{aligned} \tag{210}
$$

while the remaining three are:

$$
\begin{aligned}
&\left\{ [\pi, h]_2^1, h_{12} \right\} = h_{11} - h_{22}, \\
&\left\{ [\pi, h]_3^2, h_{23} \right\} = h_{22} - h_{33}, \\
&\left\{ [\pi, h]_3^1, h_{13} \right\} = h_{11} - h_{33}.
\end{aligned} \tag{211}
$$

This strongly suggests that the natural choice to impose for gauge fixing is a diagonal 3−metric for $h_{\alpha\beta}$. Since the 3−metric commutes with the conjugate momenta (eq. (209)), accordingly the conjugate momenta $\pi^{\alpha\beta}$ is diagonal as well. The variables used to denote the diagonal representation of the metric and its conjugate momenta are known as *Ashtekar-*

---

[8]See footnote 3 in Chapter 3.3 for the definitions of *first class* and *second class constraints*.

*Henderson-Sloan (AHS) variables* [25, 26]:

$$h_{\alpha\beta} = \text{diag}\,(Q_1, Q_2, Q_3)\,, \qquad \pi^{\alpha\beta} = \text{diag}\left(\frac{P_1}{Q_1}, \frac{P_2}{Q_2}, \frac{P_3}{Q_3}\right), \tag{212}$$

where eq. (211) suggests that $Q_1 \neq Q_2$, $Q_2 \neq Q_3$ and $Q_3 \neq Q_1$ for the diffeomorphism constraints to be a true set of second class constraints. Furthermore the choice of $\pi^{\alpha\beta}$ in terms of AHS variables also suggests that the set $\{Q_1, Q_2, Q_3\} \neq 0$. Positive definitiveness of 3−metric implies $\{Q_1, Q_2, Q_3\} > 0$. Thus we have finally solved the diffeomorphism constraints by imposing an appropriate choice of gauge fixing. We have defined symplectic potential $\mathbb{S}$ below eq. (F.21) which in terms of AHS variables becomes $\boxed{\mathbb{S} = \sum_{i=1}^{3} \frac{P_i \dot{Q}_i}{Q_i}.}$

Once we have solved the diffeomorphism constraints and obtained the form of 3−metric and its conjugate momenta, we proceed to express the Hamiltonian constraint (eq. (204)) in terms of these 3−metric and its conjugate momenta. We reproduce eq. (204) here for our convenience where we now take $n' \equiv \frac{n}{\sqrt{h}}$ as the new Lagrange multiplier (since $n$ is arbitrary and can always be chosen to scale like $h$):

$$\mathbb{H}_{\text{BIX}}[n'] = n'\left(\underbrace{\text{Tr}\left(h^2\right)}_{\text{Term A}} \underbrace{-\frac{1}{2}\left(\text{Tr}(h)\right)^2}_{\text{Term B}} \underbrace{-\frac{1}{2}\left(\pi^{\alpha\beta} h_{\alpha\beta}\right)^2}_{\text{Term C}} + \underbrace{\pi^{\alpha\beta} \pi_{\alpha\beta}}_{\text{Term D}}\right). \tag{213}$$

For terms A and B, we use:

$$\begin{aligned}
\text{Tr}(h) &= \text{Tr}\left(\begin{pmatrix} Q_1 & 0 & 0 \\ 0 & Q_2 & 0 \\ 0 & 0 & Q_3 \end{pmatrix}\right) = Q_1 + Q_2 + Q_3\,, \\
\text{Tr}\left(h^2\right) &= \text{Tr}\left(\begin{pmatrix} Q_1 & 0 & 0 \\ 0 & Q_2 & 0 \\ 0 & 0 & Q_3 \end{pmatrix} \cdot \begin{pmatrix} Q_1 & 0 & 0 \\ 0 & Q_2 & 0 \\ 0 & 0 & Q_3 \end{pmatrix}\right) = (Q_1)^2 + (Q_2)^2 + (Q_3)^2\,.
\end{aligned} \tag{214}$$

For term C, we simply have:

$$\left(\pi^{\alpha\beta} h_{\alpha\beta}\right)^2 = \left(\pi^{11} h_{11} + \pi^{22} h_{22} + \pi^{33} h_{33}\right)^2 = (P_1 + P_2 + P_3)^2\,. \tag{215}$$

For term D, we have:

$$\pi^{\alpha\beta} \pi_{\alpha\beta} = \pi^{\alpha\beta} \pi^{2\sigma} h_{\alpha\tau} h_{\beta\sigma} = (P_1)^2 + (P_2)^2 + (P_3)^2\,, \tag{216}$$

where only the diagonal components of $\pi^{\alpha\beta}$ and $h_{\alpha\beta}$ contribute (see eq. (212)). Plugging eqs. (214, 215, 216) into eq. (213), we finally get for Bianchi IX, the following Hamiltonian constraint subjected to the choice of gauge (eq. (212)) imposed:

$$\begin{aligned}
\mathbb{H}_{\text{BIX}}[n'] = n'\Big[ &\left\{(Q_1)^2 + (Q_2)^2 + (Q_3)^2\right\} + \left\{(P_1)^2 + (P_2)^2 + (P_3)^2\right\} \\
&-\frac{1}{2}(Q_1 + Q_2 + Q_3)^2 - \frac{1}{2}(P_1 + P_2 + P_3)^2 \Big].
\end{aligned} \tag{217}$$

This forms the basis of further progress in the rest of this section.

### 5.2.1 Jacobi variables

We now switch to *Jacobi variables* $\{P_1, P_2, P_3, Q_1, Q_2, Q_3\} \to \{x, y, k_x, k_y, D, \alpha\}$ where $\{k_x, k_y\}$ are conjugate variables of $\{x, y\}$. Here $x$ and $y$ are called *Misner variables* and they are measures of anisotropies (accordingly $x = y = 0$ gives a homogeneous as well as isotropic cosmology). This change of variables to Jacobi coordinates are:

$$P_1 = -\frac{k_x}{\sqrt{2}} + \frac{k_y}{\sqrt{6}} + D, \quad P_2 = \frac{k_x}{\sqrt{2}} + \frac{k_y}{\sqrt{6}} + D, \quad P_3 = -\sqrt{\frac{2}{3}}k_y + D,$$
$$Q_1 = \alpha e^{-\frac{x}{\sqrt{2}} + \frac{y}{\sqrt{6}}}, \qquad Q_2 = \alpha e^{\frac{x}{\sqrt{2}} + \frac{y}{\sqrt{6}}}, \qquad Q_3 = \alpha e^{-\sqrt{\frac{2}{3}}y}. \tag{218}$$

We further impose:

$$\alpha = v^{\frac{2}{3}}, \qquad D = \frac{v\tau}{2}, \tag{219}$$

where $v$ is the 3−volume on the hypersurface and $\tau$ is known as the *York time*. $\tau$ is the conjugate variable to $v$. The Misner variables $\{x, y\}$ are also called *shape degrees of freedom* while the global factor of 3−volume $v$ is called the *scale degree of freedom*.

With these change of variables, we have for the 3−metric as well as the symplectic potential $\mathbb{S}$ (defined in the paragraph below eq. (212)) the following results:

$$\mathbb{S} = \sum_{i=1}^{3} \frac{P_i \dot{Q}_i}{Q_i} = k_x \dot{x} + k_y \dot{y} + \frac{3D\dot{\alpha}}{\alpha} = k_x \dot{x} + k_y \dot{y} + \tau \dot{v},$$
$$h_{\alpha\beta} = \text{diag}(Q_1, Q_2, Q_3) = v^{\frac{2}{3}} \text{diag}\left(e^{-\frac{x}{\sqrt{2}} + \frac{y}{\sqrt{6}}}, e^{\frac{x}{\sqrt{2}} + \frac{y}{\sqrt{6}}}, e^{-\sqrt{\frac{2}{3}}y}\right), \tag{220}$$

where we used $v = \alpha^{\frac{3}{2}}$ and $\tau = \frac{2D}{v}$.

Next we shift to the Hamiltonian constraint and express in terms of these new variables. We refer to eq. (217), which we reproduce here for convenience:

$$\mathbb{H}_{\text{BIX}}[n'] = n' \left[ \left\{ \sum_{i=1}^{3}(Q_i)^2 \right\} + \left\{ \sum_{i=1}^{3}(P_i)^2 \right\} - \frac{1}{2}\left(\sum_{i=1}^{3}Q_i\right)^2 - \frac{1}{2}\left(\sum_{i=1}^{3}P_i\right)^2 \right]. \tag{221}$$

We get the following on substitution for separate terms:

$$\left(\sum_{i=1}^{3} P_i\right)^2 = 9D^2 = \frac{9}{4}v^2\tau^2,$$
$$\sum_{i=1}^{3}(P_i)^2 = k_x^2 + k_y^2 + 3D^2 = k_x^2 + k_y^2 + \frac{3}{4}v^2\tau^2,$$
$$\sum_{i=1}^{3}(Q_i)^2 = \alpha^2\left[e^{-\frac{2x}{\sqrt{2}} + \frac{2y}{\sqrt{6}}} + e^{\frac{2x}{\sqrt{2}} + \frac{2y}{\sqrt{6}}} + e^{-\frac{4}{\sqrt{6}}y}\right] \qquad \left(\alpha = v^{\frac{2}{3}}\right), \tag{222}$$
$$\left(\sum_{i=1}^{3}Q_i\right)^2 = v^{\frac{4}{3}}\left[e^{\frac{-2x}{\sqrt{2}} + \frac{2y}{\sqrt{6}}} + e^{\frac{2x}{\sqrt{2}} + \frac{2y}{\sqrt{6}}} + e^{-\frac{4}{\sqrt{6}}y} + 2\left(e^{2y/\sqrt{6}} + e^{\frac{x}{\sqrt{2}} - \frac{y}{\sqrt{6}}} + e^{-\frac{x}{\sqrt{2}}\frac{-y}{\sqrt{6}}}\right)\right].$$

We can always choose the Lagrange multiplier $n$ in eq. (204) to scale like $\sqrt{h}$ without loss of generality, and therefore we can set $n' = \frac{n}{\sqrt{h}} = 1$. Thus we get for the Hamiltonian constraint as:

$$\boxed{\mathbb{H}_{\text{BIX}} = k_x^2 + k_y^2 - \frac{3}{8}v^2\tau^2 + v^{\frac{4}{3}}U(x, y) \approx 0,} \tag{223}$$

where $U(x,y) \equiv \left[ \frac{1}{2}e^{-\frac{2\sqrt{3}}{\sqrt{6}}x + \frac{2y}{\sqrt{6}}} + \frac{1}{2}e^{\frac{2\sqrt{3}x}{\sqrt{6}} + \frac{2y}{\sqrt{6}}} + \frac{1}{2}e^{\frac{-4}{\sqrt{6}}y} - e^{\frac{2y}{\sqrt{6}}} - e^{\frac{\sqrt{3}x}{\sqrt{6}} - \frac{y}{\sqrt{6}}} - e^{-\frac{\sqrt{3}x}{\sqrt{6}} - \frac{y}{\sqrt{6}}} \right]$ is called the

*shape potential* whose coefficient is the 3−volume $v^{\frac{4}{3}}$.

We can rewrite the shape potential in a succinct way as:

$$U(x,y) = f(-\sqrt{3}x + y) + f(\sqrt{3}x + y) + f(-2y), \tag{224}$$

where:

$$f(z) \equiv \frac{1}{2}e^{2z/\sqrt{6}} - e^{-z/\sqrt{6}}. \tag{225}$$

It is convenient to do another change of variables and express eq. (223) in terms of the new variables to discuss the dynamics. The change of variables are:

$$\begin{aligned}
v &= v_{(0)}e^{-\frac{\sqrt{3}}{2}x^0}, & \tau &= -\frac{2}{\sqrt{3}}v^{-1}p_0, \\
x &= \sqrt{2}x^1, & k_x &= \frac{1}{\sqrt{2}}p_1, \\
y &= \sqrt{2}x^2, & k_y &= \frac{1}{\sqrt{2}}p_2,
\end{aligned} \tag{226}$$

where the conjugate variables are:

$$\{x^0, p_0\} = \{x^1, p_1\} = \{x^2, p_2\} = 1. \tag{227}$$

Thus the Hamiltonian constraint becomes:

$$\mathbb{H}_{\text{BIX}-\Phi} = \frac{1}{2}\left(-p_0^2 + p_1^2 + p_2^2\right) + W(x^0, x^1, x^2) \approx 0, \tag{228}$$

where $W(x^0, x^1, x^2) = v_{(0)}^{\frac{4}{3}}e^{-\frac{2}{\sqrt{3}}x^0}U(x^1, x^2)$ is the Bianchi IX potential. This justifies switching to the new variables in eq. (226) that the Hamiltonian constraint simplifies to eq. (228) which can be interpreted as the motion in a Minkowski space perturbed by a potential $W$. In the context of Bianchi cosmologies, like we discussed in Chapter 4.3.2, this is same as having Bianchi I solution as a subset of Bianchi IX universe for the regimes where the RHS of eq. (173) can be neglected.

### 5.2.2 Shape potential

We start with the expression of shape potential in terms of Misner variables $\{x, y\}$ from eq. (224) where $f(z)$ is defined in eq. (225). Then we express the Misner variables in terms of $x^1 = x/\sqrt{2}$ and $x^2 = y/\sqrt{2}$ as in eq. (226) to get:

$$\Rightarrow U\left(x^1, x^2\right) = f\left(-\sqrt{6}x^1 + \sqrt{2}x^2\right) + f\left(\sqrt{6}x^1 + \sqrt{2}x^2\right) + f\left(-2\sqrt{2}x^2\right). \tag{229}$$

Now we plug the explicit form of the function $f(z)$ at these three places on the RHS to get:

$$\Rightarrow U\left(x^1, x^2\right) = \frac{1}{2}\left[e^{-2x^1 + \frac{2}{\sqrt{3}}x^2} + e^{2x^1 + \frac{2}{\sqrt{3}}x^2} + e^{-\frac{4}{\sqrt{3}}x^2}\right] - \left(e^{x^1 - \frac{x^2}{\sqrt{3}}} + e^{-x^1 - \frac{x^2}{\sqrt{3}}} + e^{\frac{2}{\sqrt{3}}x^2}\right). \tag{230}$$

Then we make use of polar representation for the variables $\{x^1, x^2\}$ as represented in Fig. (5). Thus we have:

$$x^1 = r\cos(\phi), \qquad x^2 = r\sin(\phi), \tag{231}$$

where $r \equiv \sqrt{(x^1)^2 + (x^2)^2}$.

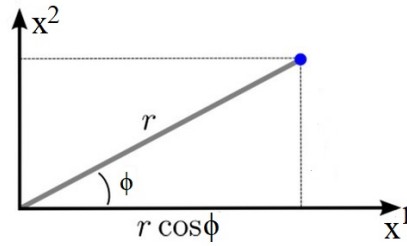

Figure 5: Polar coordinates to simplify Bianchi IX shape potential.

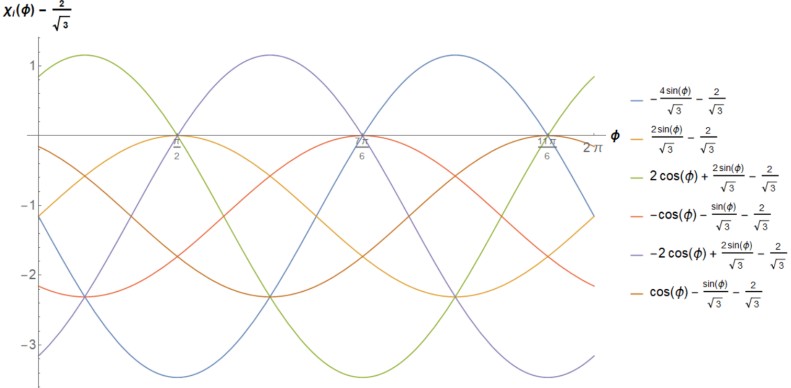

Figure 6: Six functions constituing the shape function $U(x^1, x^2)$.

Finally with the substitution in eq. (231) in eq. (230), we get:

$$\Rightarrow \boxed{U(r, \phi) = \sum_{i=1}^{6} (-1)^i c_i e^{\sqrt{(x^1)^2 + (x^2)^2}} \chi_i(\phi_0),} \tag{232}$$

where the set $c_i = \{\frac{1}{2}, 1, \frac{1}{2}, 1, \frac{1}{2}, 1\}$ and the six functions $\chi_i(\phi)$ are given by:

$$\boxed{\begin{aligned} \chi_1(\phi) &= \frac{-4\sin\phi}{\sqrt{3}}, & \chi_2(\phi) &= \frac{2\sin\phi}{\sqrt{3}}, & \chi_3(\phi) &= 2\cos\phi + \frac{2\sin\phi}{\sqrt{3}}, \\ \chi_4(\phi) &= -\cos\phi - \frac{\sin\phi}{\sqrt{3}}, & \chi_5(\phi) &= -2\cos\phi + \frac{2\sin\phi}{\sqrt{3}}, & \chi_6(\phi) &= \cos\phi - \frac{\sin\phi}{\sqrt{3}}. \end{aligned}} \tag{233}$$

These functions are plotted in Fig. (6). There are few observations to be made from their graphs:

(a) All six functions are bounded.

(b) For any given value of $\phi$, at least one out of six functions $\chi_i(\phi)$ is positive.

These facts are used in this Chapter 5 and the next Chapter 6 to prove or disprove (depending on the system in consideration) the phenomenon of *quiescence*.

### 5.2.3 Singularities in Bianchi IX universe

We realize that due to the presence of $v^{\frac{4}{3}}$ in eq. (223), the overall potential is dynamically changing. In order to study this dynamics, we now proceed to calculating equations of motion for $v$ and $\tau$. This will tell us about the locations of big-bang singularities where we expect $v \to 0$ because at least one of the three spatial dimensions collapses at the singularity, leading

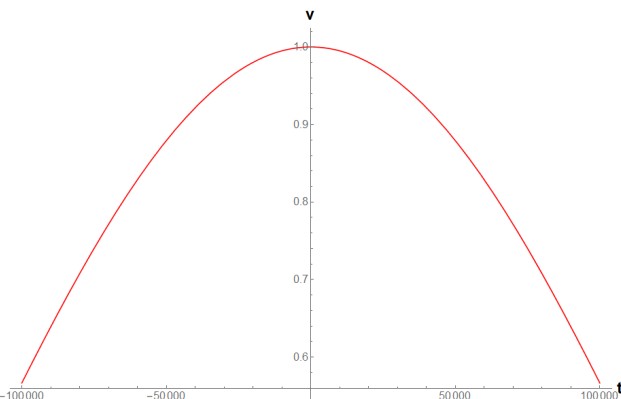

Figure 7: Behaviour of volume with time of a Bianchi IX universe showing two big-bang singularities (one in the past and another in the future).

to vanishing of 3−volume on the hypersurface. Note that since we are considering the vacuum case of Bianchi IX universe, the vanishing of the 3−volume on each hypersurface of the ADM formalism is the only signature we have for the big-bang singularity. The equations of motion for 3−volume and York time are (recall they are conjugate variables):

$$\dot{v} = \frac{\partial \mathbb{H}_{\text{BIX}}}{\partial \tau} = -\frac{3}{4}\tau v^2,$$
$$\dot{\tau} = -\frac{\partial \mathbb{H}_{\text{BIX}}}{\partial v} = \frac{3}{4}\tau^2 v + \frac{4}{3}v^{\frac{1}{3}}U(x,y) \approx \frac{5}{4}\tau^2 v + \frac{4}{3}\left(k_x^2 + k_y^2\right)v^{-1},$$

(234)

where in the second equality of the second line, we used the weak equality $\mathbb{H}_{\text{BIX}} \approx 0$ in eq. (223) to replace the shape potential in terms of other variables.

As expected from its name, therefore, the York time $\tau$ is monotonically increasing as $v > 0$ always. Similarly we can obtain the behaviour of $v$ by noting that $\frac{d}{dt}(v^{-1}) = -\frac{\dot{v}}{v^2} = \frac{3}{4}\tau$ and $\frac{d^2}{dt^2}(v^{-1}) = \frac{3}{4}\dot{\tau}$ which allows us to plot $v$ as a function of the coordinate time $t$ in Fig. (7). Thus we have two singularities that are reached in infinite coordinate time $t$. It has been shown that even though an infinite coordinate time $t \to \pm\infty$ is taken to reach the singularities in the past and in the future, an observer travelling towards the singularity requires only a finite proper time [1, 27]. Thus these two big-bang singularities (in the past and in the future) of Bianchi IX universe are essential and genuine singularities.

**A brief digression: Singularity in Bianchi I universe**

We digress for the moment to discuss the presence of big-bang singularity in Bianchi I universe. We already showed the presence of a true big-bang singularity in the Lagrangian formulation in Section 4.3.1 and now discuss the ADM formulation of it. We know that the shape potential is zero in Bianchi I and we get the following Hamiltonian constraint for Bianchi I universe (using eq. (223)):

$$\boxed{\mathbb{H}_{\text{BI}} = k_x^2 + k_y^2 - \frac{3}{8}v^2\tau^2 \approx 0.}$$

(235)

Accordingly we are again looking for big-bang singularity whose signature is $v \to 0$. So we calculate the equations of motion for 3−volume and York time $\tau$ (recall they are conjugate variables) as follows:

$$\dot{v} = \frac{\partial \mathbb{H}_{\text{BIX}}}{\partial \tau} = -\frac{3}{4}\tau v^2,$$
$$\dot{\tau} = -\frac{\partial \mathbb{H}_{\text{BIX}}}{\partial v} = \frac{3}{4}\tau^2 v,$$

(236)

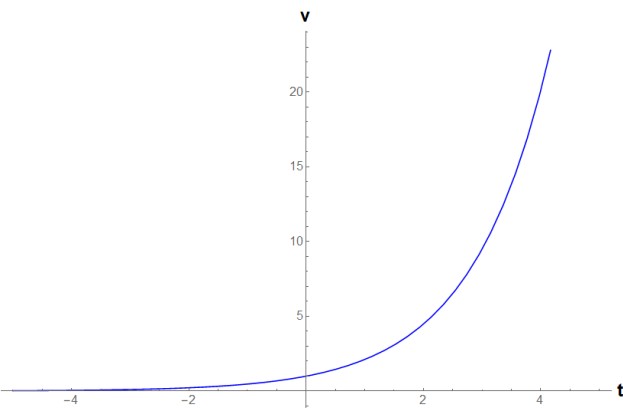

Figure 8: Behaviour of volume with time of a Bianchi I universe showing a big-bang singularity in the past (for $D_{(0)} < 0$).

which can be integrated to get $v(t) = v_{(0)} e^{-\frac{3}{2}D_{(0)}t}$ and $\tau(t) = \tau_{(0)} e^{\frac{3}{2}D_{(0)}t}$. Here $v_{(0)}$ and $\tau_{(0)}$ are two integration constants and $D_{(0)} = 2v_{(0)}\tau_{(0)}$. We realize that $v \to 0$ happens only once, either in the past or in the future, depending on the sign chosen for $D_{(0)}$. For example, if we choose the negative sign for $D_{(0)}$, then the universe contracts till 3−volume goes to 0 as $t \to -\infty$. But we realize that this time $t$ is the coordinate time and again it can be shown [1, 27] that the proper time of an observer travelling towards the big-bang singularity is finite and thus singularity can be reached in a finite amount of proper time, making it an essential and genuine singularity. For $t \to \infty$, the universe will continue to expand forever. This is shown in Fig. (8).

### 5.2.4 Infinite bounces: Mixmaster dynamics

We can now start analyzing the dynamics of Bianchi IX universe. The first observation to make is the presence of the shape potential. We plot the 3D version of shape potential that we obtained in eq. (224) in Fig. (9) while the 2D plot is provided in Fig. (10) [28]. The colour coding is that blue represents low values while red represents large values of the shape potential. These plots tell a story that Bianchi IX universe can be imagined as a Bianchi I universe but with the presence of a triangular billiard table shaped potential. In other words, we can basically imagine "Bianchi IX = Bianchi I + Shape Potential $U(x, y)$". This also makes the study of Bianchi I universe crucial if we wish to study Bianchi IX. Thus from the viewpoint of a representative particle traversing the Bianchi IX universe, it keeps moving freely along a straight line (the so-called *Kasner epochs* as the equations of motion resemble that of a Bianchi I universe because $U(x, y) \approx 0$ when we are far away from the potential walls) till it hits one of the three walls of the potential as shown in Fig. (9). Due to the presence of $v^{4/3}$ in the potential term in eq. (223), the Bianchi IX potential term is dynamic in nature and the bounces off the potential walls are inelastic, causing the momentum of the particle $\sqrt{k_x^2 + k_y^2}$ to decrease with time after each bounce (see [1, 27] for more details). When expressed in terms of new variables in eq. (226), the momentum $\sqrt{p_1^2 + p_2^2}$ decreases after each collision. Each bounce of the particle off the potential walls then changes the direction of the particle and it sets on a new straight line motion till it again hits the potential wall. The transition that happens from bouncing off the potential wall is known as *Taub transition*.

Now what does this tell us about the dynamics of the potential wall? Suppose we go towards the big-bang singularity in the future where $v \to 0$. We know from eq. (223) that $\mathbb{H}_{\text{BIX}} \approx 0$. During a Kasner epoch, away from the potential walls, the kinetic energy $K = k_x^2 + k_y^2$ needs to be conserved. But upon collisions with the potential walls where the bounces are inelastic, the momentum $\left(k_x^2 + k_y^2\right)$ (accordingly $\left(p_1^2 + p_2^2\right)$) decreases with time after each

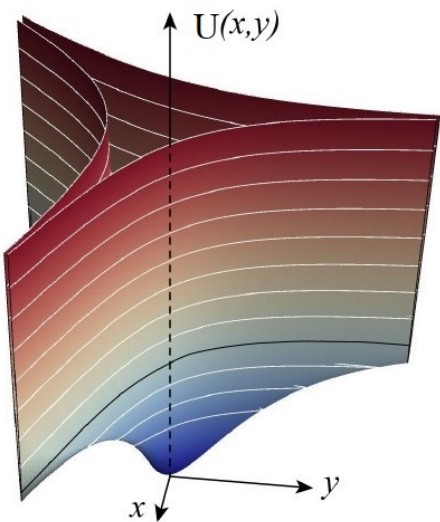

Figure 9: 3D plot of shape potential $U(x, y)$ [28].

bounce. But since $\mathbb{H}_{\text{BIX}} \approx 0$, we can deduce that $K$ must hit the potential wall $U(x, y)$ at points farther and farther away from the origin as the coordinate time $t$ passes. Thus the physical picture of Bianchi IX universe is that it corresponds to a triangular shaped billiards table where motion in between are the Kasner epochs (straight line motions) followed by Taub transitions (inelastically bouncing off the potential walls) with the potential walls moving apart with time (thereby increasing the size of the triangular shaped billiards table). Hence with the passage of time, the particle can explore larger regions as can be seen from the numerical simulation done in Fig. (11) [28].

The next natural question to ask is that since the potential walls are receding, can it happen that they recede fast enough[9] to be never caught up by the particle? If this happens, then there will be one last bounce off the potential wall after which it will set on a straight line (Kasner epoch) motion for eternity before hitting the singularity. As we show now here in this subsection that for the case of pure Bianchi IX universe, this can *never* happen.

The idea is as follows. We stick to the motion of a particle far away from the potential walls (Kasner epoch). Then we show that every such Kasner epoch inevitably ends up with a collision of the potential walls of the shape potential, thereby bouncing off it and setting off on to another Kasner epoch. Therefore, no matter how far in the future or back in the past we consider a Kasner epoch, every Kasner epoch ends with a collision and hence, the potential term in eq. (223) necessarily catches up with the kinetic terms in there. Accordingly there is no one permanent Kasner epoch that lasts forever till the particle hits the big-bang singularity but in fact the *particle bounces off the potential walls for an infinite number of times before the singularity is reached*. We now implement this idea and make it precise.

We start with calculating the equations of motion during a Kasner epoch where the particle is far away from the potential walls ($W \approx 0$). We use the Hamiltonian constraint in eq. (235) to get for $\dot{x}^\mu = \frac{\partial}{\partial p_\mu} \mathbb{H}_{\text{BI}-\Phi}$ as well as $\dot{p}_\mu = -\frac{\partial}{\partial x^\mu} \mathbb{H}_{\text{BI}-\Phi}$:

$$\dot{x}^\mu = \eta^{\mu\nu} p_\nu, \qquad \dot{p}_\mu = 0, \tag{237}$$

where $\mu = \{0, 1, 2\}$ and $\eta^{\mu\nu} = \text{diag}(-1, +1, +1)$. Integrating them gives the *Kasner solutions*

---

[9]That is to say, the potential will decay with coordinate time $t$ to the point of vanishing when $t$ becomes infinitely large.

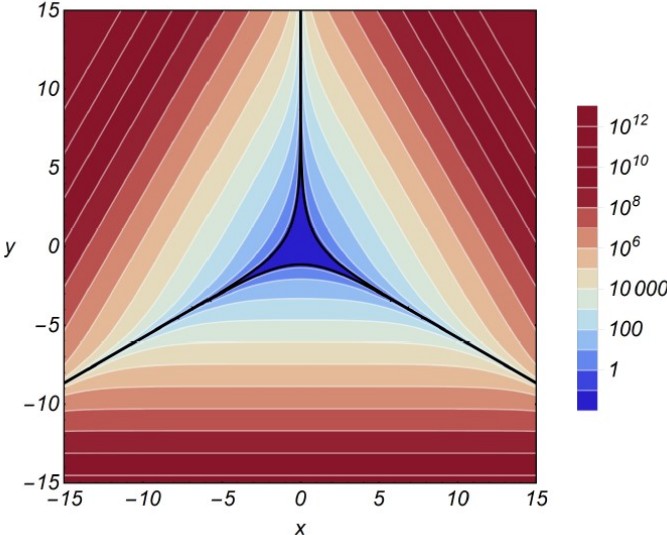

Figure 10: 2D plot of shape potential $U(x, y)$ (minimum is at the origin $U(x = 0, y = 0) = -\frac{3}{2}$) [28].

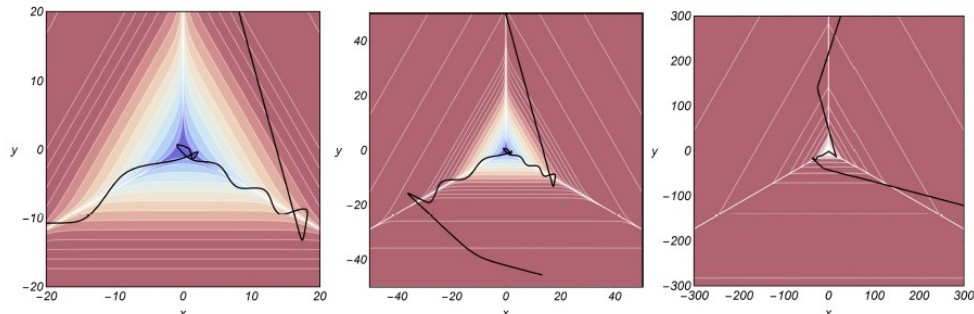

Figure 11: Numerical simulations of a pure Bianchi IX model showing the Mixmaster behaviour as explained in the text [28].

as:

$$x^\mu(t) = \eta^{\mu\nu} p_\nu t + x_0^\mu, \qquad p_\mu(t) = p_\mu^0 \quad \forall t, \tag{238}$$

where $x_0^\mu$ & $p_\mu^0$ are integration constants or the initial conditions.

Using the final result for the shape potential $U(x^1, x^2)$ from Subsection 5.2.2 (eqs. (232, 233)), we have for $W$ in eq. (248):

$$
\begin{aligned}
W(x^0, x^1, x^2) &= v_0^{4/3} e^{-\frac{2}{\sqrt{3}} x^0} U(x^1, x^2) \\
&= v_0^{4/3} e^{-\frac{2}{\sqrt{3}} x^0} \sum_{i=1}^{6} (-1)^i c_i e^{\sqrt{(x^1)^2 + (x^2)^2} \chi_i(\phi_0)} \\
&= v_0^{4/3} \sum_{i=1}^{6} (-1)^i c_i e^{-\frac{2}{\sqrt{3}} x^0 + \sqrt{(x^1)^2 + (x^2)^2} \chi_i(\phi_0)},
\end{aligned}
\tag{239}
$$

where we choose $\phi_0 = \arctan\left(\frac{p_2^0}{p_1^0}\right)$ as the direction of the Kasner solution obtained in eq. (238) and $c_i = \left\{\frac{1}{2}, 1, \frac{1}{2}, 1, \frac{1}{2}, 1\right\}$. The explicit expressions for $\chi_i$ are provided in Subsection 5.2.2 in eq. (233).

Then we plug in the expressions for $\{x^0, x^1, x^2\}$ from the Kasner solutions in eq. (238) into the last line of eq. (239) and use the Kasner Hamiltonian constraint (eq. (235)) being weakly equal to zero, namely $p_0 \approx \pm\sqrt{(p_1)^2 + (p_2)^2}$, to get[10] $W \to$ constant $\sum_{i=1}^{6} \zeta_i$ where $\zeta_i$ are the exponential functions. Focussing on one of the $\zeta_i$, we have:

$$
\begin{aligned}
\zeta_i &= \exp\left[\frac{-2}{\sqrt{3}}\sqrt{p_1^2 + p_2^2}\,t + \sqrt{(p_1)^2 + (p_2)^2}\,\chi_i(\phi_0)\,t\right] \\
&= \exp\left[\sqrt{p_1^2 + p_2^2}\left(\chi_i(\phi_0) - \frac{2}{\sqrt{3}}\right)t\right].
\end{aligned}
\tag{240}
$$

But we know from Subsection 5.2.2 and in particular Fig. (6) that for any value of $\phi$, we have at least one out of the six functions $\chi_i(\phi)$ plotted there such that $\left(\chi_i(\phi_0) - \frac{2}{\sqrt{3}}\right)$ is positive. Accordingly the potential term in eq. (223) which is of the form $v^{4/3}U(x^1, x^2)$ is always exponentially increasing with time $t$ during a Kasner epoch and hence necessarily catches up with the particle (kinetic terms in eq. (223)) causing that Kasner epoch to end by setting off the particle to another Kasner epoch till next collision with the potential walls happen.

Thus we have proved for the vacuum case of Bianchi IX universe that the particle will always be able to catch up with the potential walls and there will be infinite number of bounces before the particle hits the singularity in a finite proper time. Therefore in a pure Bianchi IX model, every Kasner epoch (the time the particle travels freely) will end with a collision with the potential walls and thus the singularity is reached after infinite bounces. This is what is known as the *Mixmaster behaviour* ( [20]) and this is a perfect scrambler of information as all information of the initial conditions get erased due to infinite inelastic bounces as the particle approaches the singularity. Also the precise point where the singularity is reached by the particle remains undetermined and we do not have a defined limit at the singularity. This is similar to the case of taking the limit of $x \to 0$ in the function $\sin\left(\frac{1}{x}\right)$. Hence neither any information of the initial condition is preserved nor the definite limit at the singularity is known where the particle reaches, making this a perfect scrambler of information. *In the vacuum case of Bianchi IX universe, quiescence can never be attained*.

Now we proceed to two more examples in Sections 5.3 & 5.4 where we do the ADM analysis of the Bianchi IX universe coupled to a free & massless scalar field as well as a scalar field with a potential term, respectively. To give the spoiler, the situation is drastically different there. As proved in Section 5.3.1, the potential walls can be shown to recede much faster for the particle to be able to catch up to it after a point of time, thereby causing only a *finite number of bounces* before the particle reaches the singularity. Thus after a certain finite number of bounces off the potential walls, eventually there will be a Kasner epoch that will last forever, until the particle reaches the singularity. This mechanism allows for the scrambling of information to cease after the last bounce and the information of the initial conditions are actually preserved at the singularity. Moreover, there exists a direction on the Misner plane that admits a well-defined limit at the singularity where the particle reaches. Thus the Mixmaster behaviour can be avoided by coupling to a classical scalar field and this has allowed to continue the classical solutions through the singularity as reported in the literature [28, 29].

---

[10]Here a negative $p_0 = -\sqrt{p_1^2 + p_2^2}$ implies a shrinking solution as we realize from the transformations in eq. (226) that we need $x^0 \to +\infty$ for $v \to 0$. The corresponding Hamiltonian constraint in eq. (228) for the final Kasner epoch is $\mathbb{H} = \frac{1}{2}\left(-p_0^2 + p_1^2 + p_2^2\right)$. Thus the equations of motion for $x^0$ & $p_0$ are: $\dot{x}^0 = -p_0$ & $\dot{p}_0 = 0$, thereby implying the solutions: $p_0 = c$ (constant) & $x^0 = -ct + \xi^0$. Thus for $x^0$ to be positive at large time $t \to +\infty$, $c < 0$ and therefore $p_0 = c < 0$.

### 5.3 Free & massless scalar field coupled to Bianchi IX universe

We now consider Bianchi IX universe in the presence of a classical scalar field that is free and massless. Since we have already done a detailed calculation for the ADM action in Chapter 3.2 & Bianchi IX universe in Section 5.2, we redo the same calculations in the presence of a free & massless scalar field to get for the ADM action ($\pi_\Phi$ is the conjugate variable corresponding to the scalar field $\Phi$):

$$S_{\text{ADM}+\Phi} = \int_{t_1}^{t_2} dt \int_{\Sigma_t} d^3x \left( \pi^{ij}\dot{\gamma}_{ij} + \pi_\Phi\dot{\Phi} - \mathcal{H}_{\text{ADM}+\Phi} \right), \tag{241}$$

where the symplectic potential is given by:

$$\boxed{\mathbb{S}_{\text{ADM}+\Phi} \equiv \int_{\Sigma_t} d^3x \left( \pi^{ij}\dot{\gamma}_{ij} + \pi_\Phi\dot{\Phi} \right),} \tag{242}$$

and the total Hamiltonian $H_{\text{ADM}+\Phi} = \int d^3x \mathcal{H}_{\text{ADM}+\Phi} = \mathbb{H}_{\text{ADM}+\Phi}[N] + \mathbb{D}_{\text{ADM}+\Phi}[N^i]$ is given by:

$$\boxed{\begin{aligned} \mathbb{H}_{\text{ADM}+\Phi}[N] &\equiv \int_{\Sigma_t} d^3x N \left[ -\sqrt{\gamma}^{(3)}R - \frac{1}{\sqrt{\gamma}}\left( \frac{\pi^2}{2} - \pi^{ij}\pi_{ij} - \frac{1}{2}\pi_\Phi^2 \right) + \frac{1}{2}\sqrt{\gamma}D^i\Phi D_i\Phi \right], \\ \mathbb{D}_{\text{ADM}+\Phi}[\vec{N}] &\equiv \int_{\Sigma_t} d^3x N^i \left[ -2D^j\pi_{ij} + \pi_\Phi D_i\Phi \right], \end{aligned}} \tag{243}$$

where terms marked in green are the new terms appearing due to the presence of a scalar field. Note that these equations hold true in general when a free & massless scalar field is coupled with gravity. We will specialize to homogeneous cosmologies, in particular Bianchi IX universe, below.

Assumption: We impose a homogeneous scalar field which means that the scalar field has only temporal dependence but no spatial dependence. Then they simplify to (we will denote the entities as $S_\Phi, \mathbb{S}_\Phi, \mathbb{H}_\Phi[N] = N\mathbb{H}_\Phi$ and $\mathbb{D}_\Phi[N_i] = N_i\mathbb{D}_\Phi^i$ for this assumption):

$$\mathbb{S}_\Phi = \int_{\Sigma_t} d^3x \left( \pi^{ij}\dot{\gamma}_{ij} + \pi_\Phi\dot{\Phi} \right),$$

$$\mathbb{H}_\Phi[N] = \int_{\Sigma_t} d^3x N \left[ -\sqrt{\gamma}^{(3)}R - \frac{1}{\sqrt{\gamma}}\left( \frac{\pi^2}{2} - \pi^{ij}\pi_{ij} - \frac{1}{2}\pi_\Phi^2 \right) \right], \tag{244}$$

$$\mathbb{D}_\Phi[\vec{N}] = \int_{\Sigma_t} d^3x N^i \left[ -2D^j\pi_{ij} \right].$$

Now we specialize to homogeneous cosmologies to impose the Bianchi IX homogeneous ansatz for the metric just like we did in Section 5.2. We realize that since the diffeomorphism constraints in eq. (244) are unaffected even in presence of a free & massless homogeneous scalar field, the ansatz that we imposed after solving the diffeomorphism constraint (eq. (209)) remains the same and we still resort to the same AHS variables we did in eq. (212). Then keeping the variables for $\Phi$ and its conjugate $\pi_\Phi$ as they are and imposing the change of variables to Jacobi variables (eq. (218)), we get for Bianchi IX universe coupled to a free & massless homogeneous scalar field the following Hamiltonian constraint (where we have again rescaled the Lagrange multiplier $n$ to scale with $\sqrt{h}$ and chose $n' = \frac{n}{\sqrt{h}}$ to be unity without loss of generality):

$$\boxed{\mathbb{H}_{\text{BIX}-\Phi} = k_x^2 + k_y^2 + \frac{\pi_\Phi^2}{2} - \frac{3}{8}v^2\tau^2 + v^{4/3}U(x,y).} \tag{245}$$

Thus we have "$\mathbb{H}_{\text{BIX}-\Phi} = \mathbb{H}_{\text{BIX}}$+Free & Massless Homogeneous Scalar Field".

Now we enquire about the Mixmaster dynamics & the phenomenon of quiescence corresponding to this Hamiltonian constraint, just like we did for the vacuum case of Bianchi IX universe in Subsection 5.2.4.

### 5.3.1 Quiescence in Bianchi IX-scalar field system

Just like the change of variables in eq. (226), it is convenient to do another change of variables here and express eq. (245) in terms of the new variables. The change of variables & its justification are the same as mentioned in eq. (226) with the addition of two conjugate variables $\{x^3, p_3\}$ corresponding to the introduction of scalar fields:

$$
\begin{aligned}
v &= v_{(0)} e^{-\frac{\sqrt{3}}{2}x^0}, & \tau &= -\frac{2}{\sqrt{3}}v^{-1}p_0, \\
x &= \sqrt{2}x^1, & k_x &= \frac{1}{\sqrt{2}}p_1, \\
y &= \sqrt{2}x^2, & k_y &= \frac{1}{\sqrt{2}}p_2, \\
\Phi &= x^3, & \pi_\Phi &= p_3,
\end{aligned}
\tag{246}
$$

where the conjugate variables are:

$$
\{x^0, p_0\} = \{x^1, p_1\} = \{x^2, p_2\} = \{x^3, p_3\} = 1.
\tag{247}
$$

Thus the Hamiltonian constraint becomes:

$$
\boxed{\mathbb{H}_{\text{BIX}-\Phi} = \frac{1}{2}\left(-p_0^2 + p_1^2 + p_2^2 + p_3^2\right) + W(x^0, x^1, x^2) \approx 0,}
\tag{248}
$$

where $W(x^0, x^1, x^2) = v_{(0)}^{\frac{4}{3}} e^{-\frac{2}{\sqrt{3}}x^0} U(x^1, x^2)$ is the Bianchi IX potential (see Subsection 5.2.2 for a simplified expression). During a Kasner epoch, this potential term can be neglected and we are left with a Hamiltonian constraint that resembles the motion of a free particle in Minkowski spacetime. Using eq. (220), the metric of the Bianchi IX model becomes in this set of variables as follows:

$$
\begin{aligned}
h_{\alpha\beta} &= v_{(0)} e^{-\frac{1}{\sqrt{3}}x^0} \operatorname{diag}\left(e^{-x^1 + \frac{1}{\sqrt{3}}x^2}, e^{x^1 + \frac{1}{\sqrt{3}}x^2}, e^{-\frac{2}{\sqrt{3}}x^2}\right) \\
&\propto \operatorname{diag}\left(e^{\left(\frac{1}{\sqrt{3}}p_0 - p_1 + \frac{1}{\sqrt{3}}p_2\right)t}, e^{\left(\frac{1}{\sqrt{3}}p_0 + p_1 + \frac{1}{\sqrt{3}}p_2\right)t}, e^{\left(\frac{1}{\sqrt{3}}p_0 - \frac{2}{\sqrt{3}}p_2\right)t}\right).
\end{aligned}
\tag{249}
$$

We follow the same idea and the strategy here that we employed while exploring quiescence in pure Bianchi IX universe in Subsection 5.2.4. We proceed to calculate the equations of motion during Kasner epochs where $W \approx 0$. Thus the Hamiltonian constraint in eq. (248) becomes $\mathbb{H}_{\text{BI}-\Phi}$ when $W \approx 0$. Calculating $\dot{x}^\mu = \frac{\partial}{\partial p_\mu}\mathbb{H}_{\text{B}-\Phi}$ as well as $\dot{p}_\mu = -\frac{\partial}{\partial x^\mu}\mathbb{H}_{\text{BI}-\Phi}$ component wise, we get:

$$
\dot{x}^\mu = \eta^{\mu\nu}p_\nu, \qquad \dot{p}_\mu = 0,
\tag{250}
$$

where $\mu = \{0, 1, 2, 3\}$ and $\eta^{\mu\nu} = \operatorname{diag}(-1, +1, +1, +1)$. Integrating them gives the *Kasner solutions* as:

$$
x^\mu(t) = \eta^{\mu\nu}p_\nu t + x_0^\mu, \qquad p_\mu(t) = p_\mu^0 \quad \forall t,
\tag{251}
$$

where $x_0^\mu$ & $p_\mu^0$ are integration constants or the initial conditions.

Now we finally proceed towards proving *quiescence*. Using the final simplified result of the shape potential $U(x^1, x^2)$ from Subsection 5.2.2 (eqs. (232, 233)), we have for $W$ in eq.

(248):

$$W(x^0, x^1, x^2) = v_0^{4/3} e^{-\frac{2}{\sqrt{3}}x^0} U(x^1, x^2)$$

$$= v_0^{4/3} e^{-\frac{2}{\sqrt{3}}x^0} \sum_{i=1}^{6} (-1)^i c_i e^{\sqrt{(x^1)^2+(x^2)^2}} \chi_i(\phi_0) \qquad (252)$$

$$= v_0^{4/3} \sum_{i=1}^{6} (-1)^i c_i e^{-\frac{2}{\sqrt{3}}x^0 + \sqrt{(x^1)^2+(x^2)^2}} \chi_i(\phi_0),$$

where we choose $\phi_0 = \arctan\left(\frac{p_2^0}{p_1^0}\right)$ as the direction of the Kasner solution obtained in eq. (251) and $c_i = \left\{\frac{1}{2}, 1, \frac{1}{2}, 1, \frac{1}{2}, 1\right\}$. The explicit expressions for $\chi_i$ are provided in Subsection 5.2.2 in eq. (233).

Then we plug in the expressions for $\left\{x^0, x^1, x^2\right\}$ from the Kasner solutions in eq. (251) into the last line of eq. (252) and use the Kasner Hamiltonian constraint $p_0 \approx \pm\sqrt{(p_1)^2 + (p_2)^2 + (p_3)^2}$ to get[11] $W \to$ constant $\sum_{i=1}^{6} \zeta_i$ where $\zeta_i$ are the exponential functions. Focussing on one of the $\zeta_i$, we have:

$$\zeta_i = \exp\left[\frac{-2}{\sqrt{3}}\sqrt{p_1^2 + p_2^2 + p_3^2}\, t + \sqrt{(p_1)^2 + (p_2)^2}\, \chi_i(\phi_0)\, t\right],$$

$$= \exp\left[\sqrt{p_1^2 + p_2^2}\, t\left(\chi_i(\phi_0) - \frac{2}{\sqrt{3}}\sqrt{1 + \frac{p_3^2}{p_1^2 + p_2^2}}\right)\right]. \qquad (253)$$

Now when the potential walls are approached, the Kasner solutions (eq. (251)) lose its validity of approximation and are no longer true. But since we are dealing with free & massless homogeneous scalar field, we know that there is no interaction between the potential walls and the scalar field, thereby ensuring that the conjugate momenta to the field $\pi_\Phi = p_3$ is conserved throughout the motion. But the same is not true for $p_1$ and $p_2$. As we showed in eq. (245), the potential term contains a pre-factor of $v^{4/3}$ and therefore is growing monotonically smaller with time as $v \to 0$ because the future big-bang singularity of Bianchi IX universe is approached (see Fig. (7)). Thus $k_x$ & $k_y$ (accordingly $p_1$ & $p_2$) diminish after each collision with the potential walls because the collisions are inelastic. Hence the entity $\sqrt{p_1^2 + p_2^2}$ reduces with time as singularity is approached just like the 3−volume (which goes to 0 at the singularity). Therefore $\sqrt{1 + \frac{p_1^2}{p_2^2 + p_3^2}}$ becomes smaller and smaller with time. Also from Fig. (6), we know that all the six functions $\chi_i(\phi_0)$ are bounded. Thus eventually $\left(\chi_i(\phi_0) - \frac{2}{\sqrt{3}}\sqrt{1 + \frac{p_3^2}{p_1^2 + p_2^2}}\right)$ in eq. (253) becomes negative after a certain point of time for each of the $\zeta_i$. Hence there will be one last Kasner epoch where the particle will set off on a straight line motion for eternity before hitting singularity as the potential walls will recede exponentially fast (and exponentially decay with coordinate time $t$) for the particle to be never able to catch up to the them. So there will be only a finite number of bounces off the potential walls before the particle reaches singularity in infinite coordinate time (but finite proper time [1, 27]). *Thus quiescence is established.*

---

[11]As argued in footnote 10 in Section 5.2.4, we chose the negative sign for $p_0$ here as well. The argument is the same. Here a negative $p_0 = -\sqrt{p_1^2 + p_2^2 + p_3^2}$ implies a shrinking solution as we realize from the transformations in eq. (246) that we need $x^0 \to +\infty$ for $v \to 0$. The corresponding Hamiltonian constraint in eq. (248) for the final Kasner epoch is $\mathbb{H} = \frac{1}{2}\left(-p_0^2 + p_1^2 + p_2^2 + p_3^2\right)$. Thus the equations of motion for $x^0$ & $p_0$ are: $\dot{x}^0 = -p_0$ & $\dot{p}_0 = 0$, thereby implying the solutions: $p_0 = c$ (constant) & $x^0 = -ct + \xi^0$. Thus for $x^0$ to be positive at large time $t \to +\infty$, $c < 0$ and therefore $p_0 = c < 0$.

## 5.4 Generalization to scalar field with a potential term

We now generalize the system in Section 5.3 to the case of Bianchi IX universe coupled to a classical scalar field with a potential term $V = V(\Phi)$. With this additional term, we can again redo the calculations like we did in Chapter 3.2 and again in this Chapter 5.3 to get for the ADM action ($\pi_\Phi$ is the conjugate variable corresponding to the scalar field $\Phi$):

$$S_{\text{ADM}+\Phi} = \int_{t_1}^{t_2} dt \int_{\Sigma_t} d^3x \left( \pi^{ij} \dot{\gamma}_{ij} + \pi_\Phi \dot{\Phi} - \mathcal{H}_{\text{ADM}+\Phi} \right), \tag{254}$$

where the symplectic potential is given by:

$$\boxed{\mathbb{S}_{\text{ADM}+\Phi} \equiv \int_{\Sigma_t} d^3x \left( \pi^{ij} \dot{\gamma}_{ij} + \pi_\Phi \dot{\Phi} \right),} \tag{255}$$

and the total Hamiltonian $H_{\text{ADM}+\Phi} = \int d^3x \mathcal{H}_{\text{ADM}+\Phi} = \mathbb{H}_{\text{ADM}+\Phi}[N] + \mathbb{D}_{\text{ADM}+\Phi}[N^i]$ is given by:

$$\boxed{\begin{aligned}
\mathbb{H}_{\text{ADM}+\Phi}[N] &\equiv \int_{\Sigma_t} d^3x N \left[ -\sqrt{\gamma}{}^{(3)}R - \frac{1}{\sqrt{\gamma}} \left( \frac{\pi^2}{2} - \pi^{ij}\pi_{ij} - \frac{1}{2}\pi_\Phi^2 + \sqrt{\gamma}V(\Phi) \right) + \frac{1}{2}\sqrt{\gamma}D^i\Phi D_i\Phi \right], \\
\mathbb{D}_{\text{ADM}+\Phi}[\vec{N}] &\equiv \int_{\Sigma_t} d^3x N^i \left[ -2D^j\pi_{ij} + \pi_\Phi D_i\Phi \right],
\end{aligned}} \tag{256}$$

where terms marked in green are the additional terms apart from the pure Bianchi IX expressions. Note that these equations hold true in general when a scalar field in a potential is coupled with gravity. We will specialize to homogeneous cosmologies, in particular Bianchi IX universe, below.

Assumption: Just like Section 5.3, we again impose a (spatially) homogeneous scalar field. Then they simplify to (we will denote the entities as $S_\Phi, \mathbb{S}_\Phi, \mathbb{H}_\Phi[N] = N\mathbb{H}_\Phi$ and $\mathbb{D}_\Phi[N_i] = N_i\mathbb{D}_\Phi^i$ for this assumption):

$$\begin{aligned}
\mathbb{S}_\Phi &= \int_{\Sigma_t} d^3x \left( \pi^{ij}\dot{\gamma}_{ij} + \pi_\Phi \dot{\Phi} \right), \\
\mathbb{H}_\Phi[N] &= \int_{\Sigma_t} d^3x N \left[ -\sqrt{\gamma}{}^{(3)}R - \frac{1}{\sqrt{\gamma}} \left( \frac{\pi^2}{2} - \pi^{ij}\pi_{ij} - \frac{1}{2}\pi_\Phi^2 \right) + \sqrt{\gamma}V(\Phi) \right], \\
\mathbb{D}_\Phi[\vec{N}] &= \int_{\Sigma_t} d^3x N^i \left[ -2D^j\pi_{ij} \right].
\end{aligned} \tag{257}$$

Now we specialize to homogeneous cosmologies to impose the Bianchi IX homogeneous ansatz for the metric just like we did in Sections 5.2 & 5.3. We realize that since the diffeomorphism constraints in eq. (257) are again unaffected even in presence of a homogeneous scalar field in a potential, the ansatz that we imposed after solving the diffeomorphism constraint (eq. (209)) remains the same and we still resort to the same AHS variables we did in eq. (212). Then again we keep the variables for $\Phi$ and its conjugate $\pi_\Phi$ as they are and impose the change of variables to Jacobi variables (eq. (218)). We show explicitly for the additional

potential term that we have here where we use eq. (197).

$$
\begin{aligned}
\int_{\Sigma_t} d^3x N \sqrt{\gamma} V(\Phi) &= \left( \int_{\Sigma_t} d^3x N \sin(\theta) \right) \sqrt{h} V(\Phi) \\
&= \frac{n}{\sqrt{h}} h V(\Phi) \\
&= n' Q_1 Q_2 Q_3 V(\Phi) \\
&= n' \alpha^3 V(\Phi) \\
&= n' v^2 V(\Phi),
\end{aligned}
\tag{258}
$$

where we introduced $n' = n/\sqrt{h}$ in the third line which can be set to unity (see the paragraph above eq. (223), used the AHS variables in the fourth line to substitute for the determinant of 3−metric $h_{\alpha\beta}$ from eq. (212) and finally shifted to Jacobi variables in the last two lines using eq. (218).

We have already showed for the remaining terms to reach eq. (245). Thus we get for Bianchi IX universe coupled to a homogeneous scalar field with a potential term the following Hamiltonian constraint (where we have again set the Lagrange multiplier $n'$ to unity):

$$
\boxed{\mathbb{H}_{\text{BIX}-\Phi} = k_x^2 + k_y^2 + \frac{\pi_\Phi^2}{2} - \frac{3}{8} v^2 \tau^2 + v^{4/3} U(x,y) + v^2 V(\Phi).}
\tag{259}
$$

Thus we have "$\mathbb{H}_{\text{BIX}-\Phi} = \mathbb{H}_{\text{BIX}}$+Homogeneous Scalar Field with a Potential".

Now we again explore the phenomenon of quiescence (i.e., averting the Mixmaster dynamics) in this system, just like we did in Subsections 5.2.4 & 5.3.1. As shown in Section 5.4.1, not every form of potential $V$ can lead to quiescence and demanding the condition for quiescence puts constraints on the type of potentials allowed.

### 5.4.1 Restriction on the potential to fulfil quiescence

The physical idea is that the potential $V$ should decay fast with coordinate time $t$ so that the quiescence is achieved. We now make this statement mathematically precise. We start by change of variables as given in eq. (246) of the Hamiltonian constraint in eq. (259). We get something very similar to eq. (248) with the addition of a potential term. The Hamiltonian constraint looks like:

$$
\mathbb{H}_{\text{BIX}-\Phi} = \frac{1}{2} \left( -p_0^2 + p_1^2 + p_2^2 + p_3^2 \right) + W \left( x^0, x^1, x^2, x^3 \right) \approx 0,
\tag{260}
$$

where $W \left( x^0, x^1, x^2, x^3 \right)$ is the new potential term dependent on $x^3 \, (= \Phi)$ as well. Its expression is given by:

$$
W \left( x^0, x^1, x^2, x^3 \right) = \underbrace{v_{(0)}^{\frac{4}{3}} e^{\sqrt{-\frac{2}{3}} x^0} U \left( x^1, x^2 \right)}_{\text{Term } \text{(a)}} + \underbrace{v_{(0)}^2 e^{-\sqrt{3} x^0} V(x^3)}_{\text{Term } \text{(b)}},
\tag{261}
$$

We first briefly discuss term (a). The expression appearing here, namely $U(x^1, x^2)$, can further be simplified as done in Subsection 5.2.2 in terms of polar coordinates to get eq. (232). We have already studied in detail in Section 5.3.1 where we established quiescence corresponding to the shape potential $U$. There we got an expression for this in eq. (252) and the final condition was obtained in eq. (253). Eq. (253) implies that if:

$$\chi_i(\phi_0) < \frac{2}{\sqrt{3}}\sqrt{1 + \frac{p_3^2}{p_1^2 + p_2^2}} \qquad \text{[Condition 1 for quiescence]}, \qquad (262)$$

then the potential term will decay exponentially fast with coordinate time $t \to +\infty$ in eq. (253). Here the six $\chi_i$ functions are provided in eq. (233).

Now we focus on term $\textcircled{b}$ which is the main topic of this subsection. Suppose we are in a Kasner regime where eqs. (250, 251) hold. Then at large time $t$, we have $x^0 \sim -p_0 t$ and the Hamiltonian constraint tells us that $p_0 = \pm\sqrt{p_1^2 + p_2^2 + p_3^2}$. We know from the discussions surrounding quiescence in Sections 5.2.4 (footnote 10) & 5.3.1 that $p_0$ needs to be negative. Also $x^i \sim p_i t$ for $i = \{1, 2, 3\}$. We wish for the potential to decay with time and this gives us the condition for the type of potential that can allow quiescence to occur:

$$\lim_{t \to \infty} e^{-\sqrt{3}\sqrt{p_1^2 + p_2^2 + p_3^2}t} V(p_3 t) = 0 \qquad \text{[Condition 2 for quiescence]}. \qquad (263)$$

Thus for the case of Bianchi IX universe coupled with a homogeneous scalar field in a potential, we have conditions 1 and 2 to be satisfied by the potential terms (namely, the Bianchi potential & the scalar potential terms) occurring in the Hamiltonian constraint eq. (260), namely eq. (261). Note that condition 1 is the same as that for the case of Bianchi IX universe coupled with a free & massless homogeneous scalar field which we showed in Section 5.3.1.

Let us give examples for both a bad choice as well as a good choice for the potential $V(x^3)$. An example for a bad choice is $V(x^3) = e^{c(x^3)^{1+\epsilon}}$ where $\{c, \epsilon\} > 0$. Then the term in eq. (263) looks like:

$$e^{-\sqrt{3}x^0} V\left(x^3\right) \xrightarrow{\text{large } t} e^{-\sqrt{3}\sqrt{p_1^2 + p_2^2 + p_3^2}t} e^{c(p_3 t)^{1+\epsilon}}. \qquad (264)$$

Thus for large $t$, the time dependence on time will overpower the time decaying factor in the first exponential, thereby making this a monotonically *increasing* function. Quiescence can never be achieved with this potential term.

Now we present an example for the potential term which satisfies eq. (263). One of the plausible candidates for inflationary model, namely Starobinsky potential which is within the current cosmological constraints, satisfies the phenomenon of quiescence. This has already been noticed in the literature that Starobinsky potential leads to quiescent solutions [30]. The Starobinsky potential is given as follows:

$$V\left(x^3\right) = \left(1 - ce^{-\sqrt{\frac{2}{3}}x^3}\right)^2, \qquad (265)$$

where $c > 0$ is a constant. Thus at large time $t$, this behaves as $\left(1 - ce^{-\sqrt{\frac{2}{3}}p_3 t}\right)^2$ which goes to a constant value when $t \to \infty$. Thus plugging this form of $V$ in eq. (263) shows that Starobinsky potential allows for quiescence to happen.

# 6 Extension to Einstein/Bianchi IX-Maxwell-scalar field system

We extend the ADM analysis to the case of classical electromagnetic field coupled to gravity. As is true for the rest of the lecture notes, topics covered here are already known [31, 32]. Although the methodology to obtain the results presented in Sections 6.2 & 6.3 have been known in the literature, to the best of our knowledge, this has never been reported explicitly in complete details like we do here in these two sections. Therefore these two sections can be considered as a new component in this work, albeit not original.

In Section 6.1, we perform a $3+1$-decomposition of Maxwell's equations of motion and in Subsection 6.1.1, we present the full Einstein-Maxwell equations of motion in the $3+1$-formalism without specializing to any particular cosmological solution. In Section 6.2, we perform an ADM analysis of the Einstein-Maxwell system and get the ADM action whose variation leads to equations of motion which are already derived in Subsection 6.1.1. We specialize to Bianchi IX cosmology in Subsection 6.2.1. In Section 6.3, we couple a free & massless homogeneous scalar field (as done in Chapter 5.3) to this Bianchi IX-Maxwell system. Based on the diffeomorphism and Hamiltonian constraints obtained therein, we proceed to solve the diffeomorphism constraints in Subsection 6.3.1, just like the way we did in Chapter 5.2 (eq. (209)). In Subsection 6.3.2, we then simplify the Hamiltonian constraint based on the 3-metric and its conjugate momenta obtained therein. Once we have the Hamiltonian constraint, we proceed to calculate the complete set of equations of motion for the Bianchi IX-Maxwell-scalar field system in Subsection 6.3.4, thereby concluding this manuscript about the ADM formulation of general relativity.

## 6.1 $3+1$-decomposition of Maxwell's equations

For this section, we rely extensively on the resources [31, 32]. We start with the Maxwell's equations which we decompose in $3+1$-form:

$$
\begin{aligned}
\nabla_\mu F^{\mu\nu} &= -4\pi j^\nu, \\
\nabla_\mu F^{*\mu\nu} &= 0,
\end{aligned}
\tag{266}
$$

where $j^\nu$ is the 4-current, $F_{\mu\nu} \equiv \nabla_\mu A_\nu - \nabla_\nu A_\mu = \partial_\mu A_\nu - \partial_\nu A_\mu$ (identically) and $F^{*\mu\nu} \equiv -\frac{1}{2}\epsilon^{\mu\nu\delta\sigma}F_{\delta\sigma}$. As per our convention, we define for the Levi-Civita tensor $\epsilon^{0123} = -1/\sqrt{-g}$ & $\epsilon_{0123} = +\sqrt{-g}$. The first Maxwell's equation also gives the *continuity equation*:

$$
\nabla_\mu j^\mu = 0.
\tag{267}
$$

The plan of this section is to first decompose the Faraday tensor and then proceed to decompose the Maxwell's equations. Recall that Einstein field equations have two free indices and therefore we had to take projections in mathematically 3 possible ways: (i) both projections along $n^\mu$, (ii) both along $\Sigma_t$ as well as (iii) mixed projections along $n^\mu$ and $\Sigma_t$. This is done in Appendix D. As clear from eq. (266), there is one free index, hence we can take one projection only. This leads to mathematically 2 possible ways of projecting in $3+1$-form, namely (a) projection along $n^\mu$ and (b) projection along $\Sigma_t$. We present both the cases here. Once this is done, we show the $3+1$-decomposition of the continuity equation and derive the stress-energy-momentum tensor for the electromagnetic field that appears on the RHS of the Einstein field equations. With this derived, we finally present the full Einstein-Maxwell equations of motion in $3+1$-variables in Subsection 6.1.1.

### $3+1$-decomposition of Faraday tensor

We give the procedure to decompose any arbitrary rank$-2$ tensor, say $T^{\mu\nu}$ as follows:

$$
\boxed{T^{\mu\nu} = {}^{(3)}T^{\mu\nu} + n^{\mu(3)}T^{\perp\nu} + {}^{(3)}T^{\mu\perp}n^\nu + T^{\perp\perp}n^\mu n^\nu,}
\tag{268}
$$

where

$$
\boxed{
\begin{aligned}
{}^{(3)}T^{\mu\nu} &\equiv \gamma^\mu_\alpha \gamma^\nu_\beta T^{\alpha\beta}, \\
{}^{(3)}T^{\perp\nu} &\equiv -\gamma^\mu_\alpha \gamma^\nu_\beta T^{\alpha\beta}, \\
{}^{(3)}T^{\mu\perp} &\equiv -n_\alpha \gamma^\mu_\beta T^{\alpha\beta}, \\
T^{\perp\perp} &\equiv n_\mu n_\nu T^{\mu\nu}.
\end{aligned}
}
\tag{269}
$$

We apply this procedure to the Faraday tensor where $T^{\mu\nu} = F^{\mu\nu}$. Due to the anti-symmetry properties of $F^{\mu\nu} = -F^{\nu\mu}$, we have $F^{\perp\perp} = 0$ as contraction is happening between symmetric and anti-symmetric indices (see eq. (269)). Thus we have:

$$F^{\mu\nu} = {}^{(3)}F^{\mu\nu} + n^{\mu(3)}F^{\perp\nu} + {}^{(3)}F^{\mu\perp}n^{\nu}, \tag{270}$$

where we *define electric and magnetic fields as measured by an observer with 4-velocity* $n^{\mu}$ (also known as the *Eulerian observer*) as:

$$\begin{aligned} E^{\mu} &\equiv -n_{\nu}F^{\nu\mu} \equiv {}^{(3)}F^{\perp\mu}, \\ B^{\mu} &\equiv -n_{\nu}F^{*\nu\mu} \equiv {}^{(3)}F^{*\perp\mu}. \end{aligned} \tag{271}$$

Note that the electric and magnetic fields are tangential to the spacelike hypersurface $\Sigma_t$ (since contraction with $n_{\mu}$ vanishes for both) and thus are 3-vector fields. Accordingly their indices can be raised or lowered using the 3-metric, for example $E_{\mu} = \gamma_{\mu\nu}E^{\nu}$. We could have also used Latin indices instead of Greek without loss of generality.

Thus we have for the Faraday tensor the following $3+1$-decomposition:

$$F^{\mu\nu} = {}^{(3)}F^{\mu\nu} + n^{\mu}E^{\nu} - E^{\mu}n^{\nu}. \tag{272}$$

We can further simplify this by expressing ${}^{(3)}F^{\mu\nu}$ in terms of the magnetic field defined in eq. (271). By plugging the definition of $F^{*\nu\mu}$ in the definition of $B^{\mu}$, we get:

$$\begin{aligned} B^{\mu} &\equiv -n_{\nu}F^{*\nu\mu} \\ &= +\frac{1}{2}n_{\nu}\epsilon^{\nu\mu\alpha\beta}F_{\alpha\beta} \\ &= +\frac{1}{2}n_{\nu}\epsilon^{\nu\mu\alpha\beta}\left({}^{(3)}F_{\alpha\beta} + n_{\alpha}E_{\beta} - E_{\alpha}n_{\beta}\right) \\ &= +\frac{1}{2}n_{\nu}\epsilon^{\nu\mu\alpha\beta(3)}F_{\alpha\beta}, \end{aligned} \tag{273}$$

where we used eq. (272) and $n_{\mu}n_{\nu}\epsilon^{\mu\nu\alpha\beta} = 0$ due to completely anti-symmetric properties of the Levi-Civita tensor. We define the 3-Levi-Civita tensor living on the hypersurface $\Sigma_t$:

$$\boxed{{}^{(3)}\epsilon^{\mu\alpha\beta} \equiv n_{\nu}\epsilon^{\nu\mu\alpha\beta},} \tag{274}$$

where as per the convention set for 4-Levi-Civita tensor and making use of eq. (62) where $n_{\mu} = (-N, 0, 0, 0)$, we get:

$$\boxed{{}^{(3)}\epsilon^{123} = n_0\epsilon^{0123} = -N\left(\frac{-1}{\sqrt{-g}}\right) = N\frac{1}{N\sqrt{\gamma}} = \frac{1}{\sqrt{\gamma}}.} \tag{275}$$

Observe that $\sqrt{\gamma}{}^{(3)}\epsilon^{123} = 1$ and thus there is a resemblance between the general 3-Levi-Civita tensor and the Levi-Civita tensor in a Euclidean (flat) space, namely $\sqrt{\gamma}{}^{(3)}\epsilon^{abc} = \epsilon_{\text{Flat}}^{abc}$.

Thus we have from eq. (273):

$$\boxed{{}^{(3)}F^{\alpha\beta} = {}^{(3)}\epsilon^{\alpha\beta\mu}B_{\mu}.} \tag{276}$$

Accordingly the final expression for the $3+1$-decomposition of Faraday tensor in eq. (272) using eq. (276) becomes as follows:

$$\boxed{F^{\mu\nu} = {}^{(3)}\epsilon^{\mu\nu\sigma}B_{\sigma} + n^{\mu}E^{\nu} - E^{\mu}n^{\nu},} \tag{277}$$

and its dual is given by:

$$F^{*\mu\nu} \equiv -\frac{1}{2}\epsilon^{\mu\nu\delta\sigma}F_{\delta\sigma} = -^{(3)}\epsilon^{\mu\nu\sigma}E_\sigma + n^\mu B^\nu - B^\mu n^\nu\,. \tag{278}$$

We observe that the *duality* $\vec{E} \to \vec{B}$ & $\vec{B} \to -\vec{E}$ that exists in Minkowski spacetime also exists here. This is a useful knowledge to derive equations of motion for $\vec{B}$ if the equations of motion for $\vec{E}$ are known.

We finally take $3+1$-decomposition of the Maxwell's equations in eq. (266) by first projecting them along $n^\mu$ and then along $\Sigma_t$. We will get a total of 4 equations that are equivalent to the ones in eq. (266).

**Projection along $n^\mu$**

We project the first Maxwell equation along $n^\mu$:

$$n_\nu \nabla_\mu F^{\mu\nu} = 4\pi\rho\,, \tag{279}$$

where we define the charge density as the temporal component of the 4-current as measured by an Eulerian observer (an observer with 4-velocity $n^\mu$):

$$\rho \equiv -n_\nu j^\nu\,. \tag{280}$$

Then we have for the LHS:

$$\text{LHS} = \nabla_\mu(n_\nu F^{\mu\nu}) - F^{\mu\nu}\nabla_a n_b = \underbrace{\nabla_\mu E^\mu}_{\text{Term I}} - \underbrace{F^{\mu\nu}\nabla_\mu n_\nu}_{\text{Term II}}\,, \tag{281}$$

where we used the definition of $E^\mu$ from eq. (271).

We use the identity for 4-divergence in eq. (A.4) from Appendix A to get for term I:

$$\begin{aligned}
\nabla_\mu E^\mu &= \frac{1}{\sqrt{-g}}\partial_\alpha\left(\sqrt{-g}E^\alpha\right) = \frac{1}{N\sqrt{\gamma}}\partial_\mu(N\sqrt{\gamma}E^\mu)\\
&= E^\mu \partial_\mu \ln(N) + \frac{1}{\sqrt{\gamma}}\partial_\mu\left(\sqrt{\gamma}E^\mu\right)\,.
\end{aligned} \tag{282}$$

But $E^\mu$ is a 3-vector, thus projecting it onto $\Sigma_t$ should given the same vector, i.e. $E^\mu = \gamma^\mu_\nu E^\nu$. Also we have the relation between 3-covariant derivative and 4-covariant derivative in eq. (B.4) that gives $\gamma^\mu_\nu \nabla_\mu = D_\nu$. Moreover, since $E^\mu$ is a 3-vector, we have the time component $E^0 = 0$, which reduces the term $\frac{1}{\sqrt{\gamma}}\partial_\mu\left(\sqrt{\gamma}E^\mu\right)$ to $D_i E^i$. So we have for term I[12]

$$\nabla_\mu E^\mu = D_\mu E^\mu + E_\mu a^\mu\,, \tag{283}$$

where we used the definition of the acceleration of the foliation $a^\mu \equiv n^\nu \nabla_\nu n^\mu = D^\mu(\ln(N))$ from eq. (63) whose proof is provided in Appendix C.

Alternate Proof: We use the relation in eq. (B.4), idempotent property of 3-metric ($\gamma^j_a \gamma^a_i = \gamma^j_i = \delta^j_i + n^j n_i$) and $\vec{E}$ is a 3-vector ($n_\mu E^\mu = 0$), to get:

$$\begin{aligned}
D_\mu E^\mu &= \gamma^\sigma_\mu \gamma^\mu_\nu \nabla_\sigma E^\nu\\
&= \gamma^\sigma_\nu \nabla_\sigma E^\nu = \left(\delta^\sigma_\nu + n^\sigma n_\nu\right)\nabla_\sigma E^\nu\\
&= \nabla_\nu E^\nu + n^\sigma n_\nu \nabla_\sigma E^\nu\\
&= \nabla_\nu E^\nu + n^\sigma \nabla_\sigma(n_\nu E^\nu) - n^\sigma E^\nu \nabla_\sigma n_\nu\\
&= \nabla_\nu E^\nu - E_\mu a^\mu\,,
\end{aligned} \tag{284}$$

---

[12]Eq. (283) is *true for any 3-vector*.

where we used eq. (47) in the last step. We never used any property of the electric field. Thus this proof shows that this relation is true for any arbitrary 3-vector.

Now we focus on term II in eq. (281) where we plug the expression for $F^{\mu\nu}$ from eq. (277) to get:

$$F^{\mu\nu}\nabla_\mu n_\nu = \left(^{(3)}\epsilon^{\mu\nu\sigma}B_\sigma + n^\mu E^\nu - E^\mu n^\nu\right)\nabla_\mu n_\nu, \tag{285}$$

but $a_\mu = n^\nu\nabla_\nu n_\mu$ from eq. (47) and $n^\mu\nabla_\nu n_\mu = \frac{1}{2}\nabla_\nu\left(n^\mu n_\mu\right) = 0$ as $n^\mu n_\mu = -1$. Thus we have for term II:

$$F^{\mu\nu}\nabla_\mu n_\nu = E^\mu a_\mu = E_\mu a^\mu, \tag{286}$$

which exactly cancels one of the terms in eq. (283).

Thus we plug eqs. (283, 286) into the LHS of eq. (279) to finally get:

$$\boxed{D_\mu E^\mu = 4\pi\rho,} \tag{287}$$

where $\rho$ is defined in eq. (280). We could have used Latin indices instead of Greek too without loss of any generality on the LHS as the LHS contains 3-objects only.

To project the second Maxwell equation in eq. (266) along $n^\mu$, we can repeat the above procedure and use the expression for $F^{*\mu\nu}$ from eq. (278) or simply use the duality $\vec{E}\to\vec{B}$ & $\vec{B}\to-\vec{E}$ and realize that there are no magnetic charges. Needless to say, both procedure gives the same result as follows:

$$\boxed{D_\mu B^\mu = 0,} \tag{288}$$

where again we could have used Latin indices without loss of generality as this equation contains 3-objects only.

**Projection onto $\Sigma_t$**

We start with the first Maxwell's equation in eq. (266) and take its projection onto the spacelike hypersurface $\Sigma_t$:

$$\gamma^\alpha_\nu\nabla_\mu F^{\mu\nu} = -4\pi\,^{(3)}j^\alpha, \tag{289}$$

where we define the spatial 3-current vector flowing over $\Sigma_t$ as:

$$\boxed{^{(3)}j^\alpha \equiv \gamma^\alpha_\beta j^\beta.} \tag{290}$$

We plug eq. (272) for $F^{\mu\nu}$ in eq. (289) to get for the LHS (using $\gamma^\alpha_\beta n^\beta = 0$):

$$\begin{aligned}
\text{LHS} &= \gamma^\alpha_\nu\nabla_\mu F^{\mu\nu} = \gamma^\alpha_\nu\nabla_\mu\left(^{(3)}F^{\mu\nu} + n^\mu E^\nu - E^\mu n^\nu\right)\\
&= \gamma^\alpha_\nu\nabla_\mu\,^{(3)}F^{\mu\nu} + \gamma^\alpha_\nu n^\mu\nabla_\mu E^\nu + \gamma^\alpha_\nu E^\nu\nabla_\mu n^\mu - \gamma^\alpha_\nu E^\mu\nabla_\mu n^\nu\\
&= \gamma^\alpha_\nu\nabla_\mu\,^{(3)}F^{\mu\nu} + \gamma^\alpha_\nu n^\mu\nabla_\mu E^\nu - E^\alpha K + E^\mu K^\alpha_\mu,
\end{aligned} \tag{291}$$

where we identified extrinsic curvature scalar as $\nabla_\mu n^\mu = -K$ from eq. (48) and extrinsic curvature scalar as $K^\alpha_\mu = -\gamma^\alpha_\nu\nabla_\mu n^\nu$ from eq. (46).

We rely on the corollary deduced from eq. (64) where we noticed that if 3-metric is contracted with a 3-object whose derivative is being taken, then we can commute the 3-metric with the derivative operation. Making use of (i) this, (ii) $^{(3)}F^{\mu\nu}n_\mu = 0$, and (iii) the idempotent

property of 3-metric, the relation between derivatives in eq. (B.4) gives us:

$$
\begin{aligned}
D_\mu {}^{(3)}F^{\mu\nu} &= \gamma^\alpha_\mu \gamma^\mu_\beta \gamma^\nu_\sigma \nabla_\alpha {}^{(3)}F^{\beta\sigma} = \gamma^\alpha_\beta \gamma^\nu_\sigma \nabla_\alpha {}^{(3)}F^{\beta\sigma} \\
&= \left( \delta^\alpha_\beta + n^\alpha n_\beta \right) \gamma^\nu_\sigma \nabla_\alpha {}^{(3)}F^{\beta\sigma} \\
&= \gamma^\nu_\sigma \nabla_\alpha {}^{(3)}F^{\alpha\sigma} - {}^{(3)}F^{\beta\nu} \nabla_\alpha \left( n^\alpha n_\beta \right) \\
&= \gamma^\nu_\sigma \nabla_\alpha {}^{(3)}F^{\alpha\sigma} - {}^{(3)}F^{\beta\nu} \nabla_\alpha (n^\alpha) n_\beta - {}^{(3)}F^{\beta\nu} \nabla_\alpha \left( n_\beta \right) n^\alpha \\
&= \gamma^\nu_\sigma \nabla_\alpha {}^{(3)}F^{\alpha\sigma} - {}^{(3)}F^{\beta\nu} \nabla_\alpha \left( n_\beta \right) n^\alpha \\
&= \gamma^\nu_\sigma \nabla_\alpha {}^{(3)}F^{\alpha\sigma} - {}^{(3)}F^{\beta\nu} a_\beta,
\end{aligned}
\tag{292}
$$

where we used integration by parts in the third line and ignored the boundary terms.

We use this relation to replace $\gamma^\alpha_\nu \nabla_\mu {}^{(3)}F^{\mu\nu}$ in eq. (291) to get for the LHS:

$$
\Rightarrow \text{LHS} = D_\mu {}^{(3)}F^{\mu\alpha} + {}^{(3)}F^{\beta\alpha} a_\beta + \gamma^\alpha_\nu n^\mu \nabla_\mu E^\nu - E^\alpha K + E^\mu K^\alpha_\mu.
\tag{293}
$$

We use the definition of Lie derivative for the case of torsion-free (eq. (11)) and the definition of extrinsic curvature tensor $K^\alpha_\mu = -\gamma^\alpha_\nu \nabla_\mu n^\nu$ from eq. (46) to get:

$$
\begin{aligned}
\gamma^\alpha_\beta \mathcal{L}_n E^\beta &= \gamma^\alpha_\beta \left( n^\mu \nabla_\mu E^\beta - E^\mu \nabla_\mu n^\beta \right) \\
&= \gamma^\alpha_\nu n^\mu \nabla_\mu E^\nu + K^\alpha_\mu E^\mu.
\end{aligned}
\tag{294}
$$

We use this relation to replace $\gamma^\alpha_\nu n^\mu \nabla_\mu E^\nu$ in eq. (293) to get ($K^\alpha_\mu E^\mu$ cancels out):

$$
\Rightarrow \text{LHS} = D_\mu {}^{(3)}F^{\mu\alpha} + {}^{(3)}F^{\beta\alpha} a_\beta + \gamma^\alpha_\beta \mathcal{L}_n E^\beta - E^\alpha K.
\tag{295}
$$

We finally simplify $D_\mu {}^{(3)}F^{\mu\alpha}$ by using eq. (276) and realizing that $\epsilon^{\alpha\beta\sigma}_{\text{Flat}} = \sqrt{\gamma} {}^{(3)}\epsilon^{\alpha\beta\sigma}$ (see text below eq. (275)) which is a constant tensor. We use the 3-divergence formula similar to eq. (A.3) to get:

$$
\begin{aligned}
D_\mu {}^{(3)}F^{\mu\alpha} &= \frac{1}{\sqrt{\gamma}} \partial_\mu \left( \sqrt{\gamma} {}^{(3)}F^{\mu\alpha} \right) \\
&= \frac{1}{\sqrt{\gamma}} \partial_\mu \left( {}^{(3)}\epsilon^{\mu\alpha\sigma}_{\text{Flat}} B_\sigma \right) \\
&= \frac{{}^{(3)}\epsilon^{\mu\alpha\sigma}_{\text{Flat}}}{\sqrt{\gamma}} \partial_\mu B_\sigma \\
&= +{}^{(3)}\epsilon^{\mu\alpha\sigma} \partial_\mu B_\sigma = -{}^{(3)}\epsilon^{\alpha\mu\sigma} \partial_\mu B_\sigma.
\end{aligned}
\tag{296}
$$

We plug this into LHS and utilize the RHS from eq. (289) to get the first Maxwell's equation onto $\Sigma_t$ in *component form* as:

$$
\boxed{\gamma^\alpha_\beta \mathcal{L}_n E^\beta - {}^{(3)}\epsilon^{\alpha\mu\sigma} \partial_\mu B_\sigma + {}^{(3)}\epsilon^{\alpha\mu\sigma} B_\mu a_\sigma - E^\alpha K = -4\pi {}^{(3)}j^\alpha.}
\tag{297}
$$

Since we are dealing with 3-D spatial hypersurfaces $\Sigma_t$, we can introduce a vector notation and re-express eq. (297) in terms of vector notation. We begin with using the definition of vector cross-product to get:

$$
\begin{aligned}
(D \times B)^\alpha &= {}^{(3)}\epsilon^{\alpha\mu\sigma} \partial_\mu B_\sigma, \\
(B \times a)^\alpha &= {}^{(3)}\epsilon^{\alpha\mu\sigma} B_\mu a_\sigma.
\end{aligned}
\tag{298}
$$

Accordingly we have (using $a_\sigma = D_\sigma \ln N$ from eq. (63)):

$$
\begin{aligned}
N (D \times B)^\alpha - N (B \times a)^\alpha &= N (D \times B)^\alpha - N (B \times D \ln N)^\alpha \\
&= N (D \times B)^\alpha - (B \times DN)^\alpha \\
&= N (D \times B)^\alpha + (DN \times B)^\alpha \\
&= (D \times (NB))^\alpha.
\end{aligned}
\tag{299}
$$

We also simplify the Lie derivative term where we use the expressions for $n^\mu$ & $n_\mu$ from eq. (41) to get:

$$
\begin{aligned}
\gamma^\alpha_\beta \mathcal{L}_n E^\beta &= \gamma^\alpha_\beta \left( n^\sigma \partial_\sigma E^\beta - E^\sigma \partial_\sigma n^\beta \right) \\
&= \frac{1}{N} \partial_0 E^\alpha + \frac{N^j}{N} \partial_j E^\alpha - \frac{E^j}{E} \partial_j N^\alpha \\
&= \frac{1}{N} \partial_0 E^\alpha + \frac{1}{N} \mathcal{L}_{\vec{N}} E^\alpha .
\end{aligned}
\tag{300}
$$

Thus we get in *vector form* the first Maxwell's equation onto $\Sigma_t$ by plugging eqs. (298, 299, 300) in eq. (297) (we could also use Latin indices without loss of generality as all terms involve 3-objects):

$$
\boxed{\partial_t E^\alpha + \mathcal{L}_{\vec{N}} E^\alpha = (D \times (NB))^\alpha + NKE^\alpha - 4\pi N^{(3)} j^\alpha ,}
\tag{301}
$$

where $t$ is the coordinate time.

We can redo this exercise to project the second Maxwell's equation in eq. (266) onto $\Sigma_t$ or simply use the duality $\vec{E} \to \vec{B}$ & $\vec{B} \to -\vec{E}$ and recognize that there are no magnetic charges to get:

$$
\boxed{\partial_t B^\alpha + \mathcal{L}_{\vec{N}} B^\alpha = -(D \times (NE))^\alpha + NKB^\alpha .}
\tag{302}
$$

**3+1-decomposition of continuity equation**

The continuity equation reads (eq. (267)):

$$
\nabla_\mu j^\mu = 0 ,
\tag{303}
$$

where we have already projected $j^\mu$ in eqs. (280, 290) to have:

$$
j^\mu = \rho n^\alpha + {}^{(3)} j^\alpha .
\tag{304}
$$

We plug eq. (304) into eq. (303) and identify the extrinsic curvature scalar as $\nabla_\mu n^\mu = -K$ from eq. (48) to get:

$$
n^\alpha \nabla_\alpha \rho - \rho K + \nabla_\alpha {}^{(3)} j^\alpha = 0 .
\tag{305}
$$

Now we use the identity we derived in eq. (283) for any 3-vector and apply it to ${}^{(3)} j^\alpha$ to get for the continuity equation in $3 + 1$-variables:

$$
\boxed{\mathcal{L}_n \rho + D_\alpha {}^{(3)} j^\alpha + {}^{(3)} j^\alpha a_\alpha - \rho K = 0 ,}
\tag{306}
$$

where we identified the Lie derivative as $\mathcal{L}_n \rho = n^\alpha \nabla_\alpha \rho = \frac{1}{N} \left[ \partial_t \rho + N^j \partial_j \rho \right]$.

**Stress-energy-momentum tensor of the electromagnetic field**

The Faraday tensor is anti-symmetric in its indices but the stress-energy-moment tensor $T_{\mu\nu}$ that appears on the RHS of the Einstein field equations is symmetric. Thus for the electromagnetic field, the $T_{\mu\nu}$ is defined as:

$$
\boxed{T_{\mu\nu} = \frac{1}{4\pi} \left[ F_{\mu\alpha} F^\alpha_\nu - \frac{1}{4} g_{\mu\nu} F_{\alpha\beta} F^{\alpha\beta} \right] ,}
\tag{307}
$$

which is symmetric and traceless $T^\mu_\mu = 0$.

Using eq. (277) for the Faraday tensor, we get the following two results:

$$F_{\mu\alpha}F_{\nu}^{\alpha} = -(E_\mu E_\nu + B_\mu B_\nu) + B^2\gamma_{\mu\nu} + E^2 n_\alpha n_\beta + 2E^\sigma B^{\delta(3)}\epsilon_{\sigma\delta(\mu}n_{\nu)},$$
$$F_{\mu\nu}F^{\mu\nu} = -2(E^2 - B^2),$$
(308)

where $E^2 \equiv E^\mu E_\mu$ and $B^2 = B^\mu B_\mu$. We plug it back in eq. (307) to get (after using $g_{\mu\nu} = \gamma_{\mu\nu} - n_\mu n_\nu$):

$$T_{\mu\nu} = \frac{1}{4\pi}\left[-(E_\mu E_\nu + B_\mu B_\nu) + \frac{1}{2}\gamma_{\mu\nu}(E^2 + B^2) + \frac{1}{2}n_\mu n_\nu(E^2 + B^2) + 2E^\sigma B^{\delta(3)}\epsilon_{\sigma\delta(\mu}n_{\nu)}\right],$$
(309)

where parentheses around the indices imply symmetrization procedure (square brackets imply anti-symmetrization procedure). For example, for any matrix $M$, we have $M_{(\alpha\beta)} = \frac{1}{2}\left(M_{\alpha\beta} + M_{\beta\gamma}\right)$ and $M_{[\alpha\beta]} = \frac{1}{2}\left(M_{\alpha\beta} - M_{\beta\gamma}\right)$. In general, we have:

$$M_{(\alpha\beta\gamma...)} = \frac{1}{n!}\sum_{p\in\text{ permutations}} M_{p(\alpha\beta\gamma...)},$$
$$M_{[\alpha\beta\gamma...]} = \frac{1}{n!}\sum_{p\in\text{ permutations}} (-1)^{n_p} M_{p(\alpha\beta\gamma...)}.$$

We now decompose this tensor $T_{\mu\nu}$ in eq. (309) using the general prescription provided in eqs. (268, 269). We also make use of eq. (69) which we reproduce here for convenience:

$$\mathcal{E} \equiv T_{\mu\nu}n^\mu n^\nu \qquad \text{(Energy Density)},$$
$$p_\alpha \equiv -T_{\mu\nu}n^\mu\gamma_\alpha^\nu \qquad \text{(Momentum Density)},$$
$$S_{\alpha\beta} \equiv T_{\mu\nu}\gamma_\alpha^\mu\gamma_\beta^\nu \qquad \text{(Stress Tensor)},$$
(310)

where we have replaced the notation for energy density $E \to \mathcal{E}$ in order to not confuse it for the electric field. By plugging eq. (309) into eq. (310), we get the expressions:

$$\boxed{\begin{aligned}
\mathcal{E} &= \frac{1}{8\pi}\left(E^2 + B^2\right), \\
p_\alpha &= \frac{1}{4\pi}{}^{(3)}\epsilon_{\alpha\mu\nu}E^\mu B^\nu, \\
S_{\alpha\beta} &= \frac{1}{8\pi}\left[\gamma_{\alpha\beta}(E^2 + B^2) - 2(E_\alpha E_\beta + B_\alpha B_\beta)\right].
\end{aligned}}$$
(311)

The expression of $p_\alpha$ identifies itself as the *Poynting vector*. It is the momentum density as measured by an Eulerian observer. Using the expressions in eq. (311) in eq. (309), we have:

$$\boxed{T_{\mu\nu} = \mathcal{E}n_\mu n_\nu + n_\mu p_\nu + p_\mu n_\nu + S_{\mu\nu}.}$$
(312)

Also using eq. (70), namely $T = S - \mathcal{E}$ where $T$ and $S$ are traces $T_\mu^\mu$ & $S_\mu^\mu$ respectively, and $T = 0$ corresponding to eq. (307), we have:

$$\boxed{S = \mathcal{E},}$$
(313)

for the electromagnetic field. This is a well known result in Maxwellian electrodynamics [33].

### 6.1.1 Einstein-Maxwell equations of motion in $3+1$-form

We have already decomposed the full Einstein field equations in Chapter 3.1 (eqs. (71, 72, 73)) whose details can be found in Appendix D. Then we use eqs. (311, 313) to obtain:

- Both Projections along $n^{mu}$:

$$\boxed{{}^{(3)}R - K_{ij}K^{ij} + K^2 = 16\pi\mathcal{E} = 2\left(E^2 + B^2\right).}$$

(314)

- Both Projections along $\Sigma_t$:

$$\boxed{D_\beta K_\alpha^\beta - D_\alpha K = 8\pi p_\alpha = 2^{(3)}\epsilon_{\alpha\mu\nu}E^\mu B^\nu.}$$

(315)

- Mixed Projection along $n^\mu$ and $\Sigma_t$:

$$\boxed{\begin{aligned}
-\partial_t K_{\alpha\beta} &- \mathcal{L}_{\vec{N}}K_{\alpha\beta} + N\left({}^{(3)}R_{\alpha\beta} - 2K_{\alpha\lambda}K_\beta^\lambda + KK_{\alpha\beta}\right) + D_\alpha D_\beta N \\
&= 8\pi N\left[S_{\alpha\beta} - \frac{1}{2}\gamma_{\alpha\beta}(S - \mathcal{E})\right] \\
&= 8\pi N S_{\alpha\beta} \\
&= N\left[\gamma_{\alpha\beta}(E^2 + B^2) - 2(E_\alpha E_\beta + B_\alpha B_\beta)\right],
\end{aligned}}$$

(316)

where we used the manipulation for the Lie derivative in eq. (300).

Summary: Maxwell's equations in covariant form are given in eq. (266). Faraday tensor & its dual in $3+1$-variables are expressed in eqs. (277) & (278) respectively, where 3-Levi-Civita tensor is defined in eq. (274). The projections of Maxwell's equations along the normal $n^\mu$ are given in eqs. (287) & (288). The projections of Maxwell's equations along $\Sigma_t$ are given in eq. (301) (or eq. (297) in component form) & eq. (302). Thus eqs. (287, 288, 301, 302) are the $3+1$-decomposition of the covariant Maxwell's equations (eq. (266)). The stress-energy-momentum tensor corresponding to the electromagnetic field is given in eq. (312) where eqs. (311) & (313) are used. Using this on the RHS of the Einstein field equations, we have eqs. (314, 315, 316) as the full $3+1$-decomposition of the Einstein-Maxwell system.

Now we go one step back and calculate the ADM action of the Einstein-Maxwell system whose variations (as done in Appendix F) lead us back to these equations of motion (eqs. (314, 315, 316)). Having done that, we then specialize to the case of Bianchi IX-Maxwell system and do its ADM analysis just like we did for the case of scalar field in Chapters 5.3 & 5.4.

## 6.2 ADM formulation of Einstein-Maxwell system

We present the ADM analysis of the Einstein-Maxwell system in general without any assumption of homogeneous cosmologies. Then in Section 6.2.1, we specialize to the case of Bianchi IX cosmology, thereby getting the ADM Hamiltonian for the Bianchi IX-Maxwell system which we will use in Section 6.3 to derive the equations of motion.

The Einstein-Maxwell system for the vacuum case without the cosmological constant is defined by the following action:

$$S_{\text{Einstein-Maxwell}} = \int d^4x\sqrt{-g}\left[{}^{(4)}R - \frac{1}{4}F_{\mu\nu}F^{\mu\nu}\right].$$

(317)

Thus we see that the action can be written as "Action = Einstein-Hilbert + Maxwell". We have already dealt with the Einstein-Hilbert action and done its ADM analysis in details in Chapter 3.2. We now focus on the electromagnetic Lagrangian density.

**Maxwell system**

We start with the electromagnetic Lagrangian density:

$$\mathcal{L}_{\text{EM}} = -\frac{1}{4}\sqrt{-g}\,g^{\mu\alpha}g^{\nu\beta}F_{\mu\nu}F_{\alpha\beta}\,. \tag{318}$$

But we have in eq. (60) the $3+1$-decomposition of the 4-metric which we use to expand the summation in eq. (318) along with eq. (61) to get:

$$\Rightarrow \mathcal{L}_{\text{EM}} = -\frac{1}{4N}\sqrt{\gamma}\Big[4N^i\dot{A}^jF_{ij} + 4\dot{A}^i\partial_iA_0 + -2\dot{A}^i\dot{A}_i \\ -2\big(\partial^iA_0\big)\big(\partial_iA_0\big) - 4N^i\partial^jA_0F_{ij} + N^2F_{ij}Fij - 2F^i_jF_{ik}N^jN^k\Big]\,. \tag{319}$$

Having this expanded expression for the electromagnetic Lagrangian density in terms of $3+1$-variables, in order to calculate its Hamiltonian, we need to calculate the conjugate momenta $\Pi^i$ corresponding to to the vector potential $A_i$. We realize that only the first line in eq. (319) survives:

$$\Pi^i = \frac{\partial\mathcal{L}_{\text{EM}}}{\partial\dot{A}_i} \\ = -\frac{1}{4N}\sqrt{\gamma}\Big[4N_jF^j_i + 4\partial^iA_0 - 4\dot{A}^i\Big] \\ = -\frac{1}{N}\sqrt{\gamma}\big[F_{k0} + N^jF_{jk}\big]\gamma^{ki}\,, \tag{320}$$

where the presence of $\sqrt{\gamma}$ tells that $\Pi^i$ is a tensor density with rank $W=1$ (see eq. (3)).

The Legendre transform of the electromagnetic Lagrangian density in eq. (318) gives the Hamiltonian density of the electromagnetic field:

$$\boxed{\begin{aligned}\mathcal{H}_{\text{EM}} &= \Pi^i\dot{A}_i - \mathcal{L}_{\text{EM}} \\ &= N\left(\frac{1}{2\sqrt{\gamma}}\Pi^i\Pi_i + \frac{1}{4}\sqrt{\gamma}F_{ij}F^{ij}\right) + N^i\Pi^jF_{ij} + \Pi^iD_iA_0\,.\end{aligned}} \tag{321}$$

We integrate by parts the last term $\Pi^iD_iA_0$ to get $-A_0D_i\Pi^i$ and ignore the boundary terms. Thus the ADM Hamiltonian of the electromagnetic field becomes:

$$H_{\text{EM}} = \int_{\Sigma_t}d^3x\left[N\left(\frac{1}{2\sqrt{\gamma}}\Pi^i\Pi_i + \frac{1}{4}\sqrt{\gamma}F_{ij}F^{ij}\right) + N^i\big(\Pi^jF_{ij}\big) - A_0\big(D_i\Pi^i\big)\right], \tag{322}$$

while the ADM action is given by using eq. (321):

$$\boxed{S_{\text{EM}} = \int_{t_1}^{t_2}dt\int_{\Sigma_t}d^3x\big(\Pi^i\dot{A}_i - \mathcal{H}_{\text{EM}}\big)\,,} \tag{323}$$

where the symplectic potential is:

$$\boxed{\mathcal{S}_{\text{EM}} = \int_{\Sigma_t}d^3x\big(\Pi^i\dot{A}_i\big)\,.} \tag{324}$$

Using eq. (322), we get the following Hamiltonian constraint, diffeomorphism constraint as well as *Gauss constraint*:

$$\mathbb{H}_{\text{EM}}[N] = N\mathbb{H}_{\text{EM}} = \int_{\Sigma_t} d^3x \left[ N \left( \frac{1}{2\sqrt{\gamma}} \Pi^i \Pi_i + \frac{1}{4} \sqrt{\gamma} F_{ij} F^{ij} \right) \right],$$

$$\mathbb{D}_{\text{EM}}[N^i] = N^i \mathbb{D}_{(\text{EM})i} = \int_{\Sigma_t} d^3x \left[ N^i \left( \Pi^j F_{ij} \right) \right],$$

$$\mathbb{G}[A_0] = A_0 \mathbb{G} = \int_{\Sigma_t} d^3x \left[ -A_0 \left( D_i \Pi^i \right) \right],$$

(325)

where the total Hamiltonian in eq. (322) becomes:

$$H_{\text{EM}} = \mathbb{H}_{\text{EM}}[N] + \mathbb{D}_{\text{EM}}[N^i] + \mathbb{G}[A_0] = N\mathbb{H}_{\text{EM}} + N^i \mathbb{D}_{(\text{EM})i} + A_0 \mathbb{G}.$$

(326)

As shown explicitly in Appendix F, we can vary the ADM action in eq. (323) with respect to $N$, $N^i$ and $A_0$ to get the *five constraint relations* (thereby implying that $N$, $N^i$ & $A_0$ are Lagrange multipliers):

$$\mathbb{H}_{\text{EM}} \approx 0, \qquad \mathbb{D}_{(\text{EM})i} \approx 0, \qquad \mathbb{G} \approx 0.$$

(327)

**Einstein-Maxwell system**

We now go back to eq. (317) where we just did the ADM analysis for the electromagnetic part. The ADM analysis of the pure gravity part is already done in Chapter 3.2 where we use the results from eqs. (96, 97, 98) as well as eq. (F.21) to get for the combined Einstein-Maxwell system the following ADM action:

$$S_{\text{Einstein-Maxwell}} = \int_{t_1}^{t_2} dt \int_{\Sigma_t} d^3x \left( \pi^{ij} \dot{\gamma}_{ij} + \Pi^i \dot{A}_i - \mathcal{H}_{\text{Einstein-Maxwell}} \right),$$

(328)

where the Hamiltonian density for the Einstein-Maxwell system is:

$$\mathcal{H}_{\text{Einstein-Maxwell}} = N \left[ -\sqrt{\gamma}^{(3)}R - \frac{1}{\sqrt{\gamma}} \left( \frac{\pi^2}{2} - \pi^{ij} \pi_{ij} \right) + \frac{1}{2\sqrt{\gamma}} \Pi^i \Pi_i + \frac{1}{4} \sqrt{\gamma} F_{ij} F^{ij} \right]$$
$$+ N^i \left( -2D^j \pi_{ij} + \Pi^j F_{ij} \right) - A_0 \left( D_i \Pi^i \right),$$

(329)

and symplectic potential is:

$$\mathbb{S} = \int_{\Sigma_t} d^3x \left( \pi^{ij} \dot{\gamma}_{ij} + \Pi^i \dot{A}_i \right).$$

(330)

The Hamiltonian can then be written as:

$$H_{\text{Einstein-Maxwell}} = \int_{\Sigma_t} d^3x \, \mathcal{H}_{\text{Einstein-Maxwell}} = \left( \mathbb{H}[N] + \mathbb{D}[N^i] + \mathbb{G}[A_0] \right)_{\text{Einstein-Maxwell}},$$

(331)

to get for the Hamiltonian, diffeomorphism & Gauss constraints as follows:

$$\mathbb{H}[N] = N\mathbb{H} = \int_{\Sigma_t} d^3x \left[ N \left( -\sqrt{\gamma}\,{}^{(3)}R - \frac{1}{\sqrt{\gamma}} \left( \frac{\pi^2}{2} - \pi^{ij}\pi_{ij} \right) + \frac{1}{2\sqrt{\gamma}}\Pi^i\Pi_i + \frac{1}{4}\sqrt{\gamma}\,F_{ij}F^{ij} \right) \right],$$

$$\mathbb{D}[N^i] = N^i\mathbb{D}_i = \int_{\Sigma_t} d^3x \left[ N^i \left( -2D^j\pi_{ij} + \Pi^j F_{ij} \right) \right],$$

$$\mathbb{G}[A_0] = A_0\mathbb{G} = \int_{\Sigma_t} d^3x \left[ -A_0 \left( D_i\Pi^i \right) \right],$$

(332)

where $N$, $N^i$ & $A_0$ are the Lagrange multipliers causing the variation of action (eq. (328)) with respect to them lead to five constraint relations:

$$\mathbb{H} \approx 0, \qquad \mathbb{D}_i \approx 0, \qquad \mathbb{G} \approx 0.$$

(333)

Thus we have found the ADM action of the Einstein-Maxwell system (eq. (328)) which leads to the equations of motion provided in eqs. (314, 315, 316) where we have to use the definitions of electric and magnetic fields from eq. (271). The readers will notice that every boxed equation of the Einstein-Maxwell system is of the form "Einstein + Maxwell".

### 6.2.1 ADM formulation of Bianchi IX-Maxwell system

Now we specialize to the case of Bianchi IX cosmology where we need to impose the homogeneous ansatz on the Hamiltonian, diffeomorphism & Gauss constraints in eq. (332), just like we did in Chapter 5.2. We impose the following ansatz in invariant basis on top of what we imposed in Chapter 5.2 in eq. (197) (recall $\Pi^i$ is a tensor density just like $\pi^{ij}$):

$$\gamma_{ij} = h_{\alpha\beta}(t)e_i^\alpha e_j^\beta \Rightarrow \gamma = h\sin^2(\theta),$$

$$\pi^{ij} = \sin(\theta)\pi^{\alpha\beta}(t)e_\alpha^i e_\beta^j,$$

$$\Pi^i = \sin(\theta)\Pi^\alpha(t)e_\alpha^i,$$

$$A_i = A_\alpha(t)e_i^\alpha.$$

(334)

We have already imposed this homogeneous ansatz in Chapter 5.2 for the pure gravity parts in eq. (332) and we need to impose on the electromagnetic components here. So we focus on the Hamiltonian, diffeomorphism & Gauss constraints provided in eq. (325).

Detailed calculations of imposing the homogeneous ansatz in eq. (334) on the electromagnetic components provided in eq. (325) are given in Appendix G. We present the results here (eqs. (G.13, G.17, G.20)):

$$\mathbb{H}_{\text{EM}}[N] = N\mathbb{H}_{\text{EM}} = \frac{n}{\sqrt{h}} \left[ \frac{1}{2}\Pi^\alpha\Pi_\alpha + \frac{h}{4}h^{\mu\alpha}h^{\nu\sigma}A_\beta A_\tau \epsilon^\beta_{\alpha\sigma}\epsilon^\tau_{\mu\nu} \right],$$

$$\mathbb{D}_{\text{EM}}[N^i] = N^i\mathbb{D}_{(\text{EM})i} = n^\delta \Pi^\alpha A_\beta \epsilon^\beta_{\delta\alpha},$$

$$\mathbb{G}[A_0] = A_0\mathbb{G} = 0 \qquad \text{(identically)}.$$

(335)

Thus we see that the Gauss constraint is identically zero and while solving the constraint equations, we again simply need to take care of the diffeomorphism constraints only. In a general classical Yang-Mills gauge field, the Gauss constraint is not identically zero and we need to take some combination of the diffeomorphism & Gauss constraints to solve for the 3-metric (by choosing a suitable gauge that will make the combination a second class constraint) and its conjugate momenta.

We have already derived the Hamiltonian and diffeomorphism constraints for the pure Bianchi IX universe in Chapter 5.2 (eq. (206)) which we use here and get for the eq. (332) in homogeneous ansatz eq. (334) for the Bianchi IX-Maxwell system:

$$H_{\text{Bianchi IX-Maxwell}} = \left(\mathbb{H}[N] + \mathbb{D}[N^i] + \mathbb{G}[A_0]\right)_{\text{Bianchi IX-Maxwell}},$$

(336)

where the Hamiltonian, diffeomorphism & Gauss constraints are:

$$
\begin{aligned}
\mathbb{H}_{\text{BIX-EM}}[N] &= N\mathbb{H} \\
&= \frac{n}{\sqrt{h}}\Bigg[\text{Tr}\left(h^2\right) - \frac{1}{2}\left(\text{Tr}(h)\right)^2 - \frac{1}{2}\left(\pi^{\alpha\beta}h_{\alpha\beta}\right)^2 + \pi^{\alpha\beta}\pi_{\alpha\beta} \\
&\qquad\qquad + \frac{1}{2}\Pi^{\alpha}\Pi_{\alpha} + \frac{h}{4}h^{\mu\alpha}h^{\nu\sigma}A_{\beta}A_{\tau}\epsilon^{\beta}_{\alpha\sigma}\epsilon^{\tau}_{\mu\nu}\Bigg] \approx 0, \\
\mathbb{D}_{\text{BIX-EM}}[N^i] &= N^i D_{(\text{BIX-EM})i} \\
&= n^{\delta}\left[2\epsilon^{\gamma}_{\delta\beta}\pi^{\beta\alpha}h_{\alpha\gamma} + \Pi^{\alpha}A_{\beta}\epsilon^{\beta}_{\delta\alpha}\right] \approx 0, \\
\mathbb{G}_{\text{BIX-EM}}[A_0] &= A_0\mathbb{G}_{\text{BIX-EM}} \\
&= 0 \qquad\qquad\text{(identically)}.
\end{aligned}
$$

(337)

Here we keep in mind eq. (G.8) where we saw that in the invariant basis, the Levi-Civita tensor ($\epsilon_{\mu\nu\sigma}$) which acts as a structure constant ($C^{\tau}_{\mu\nu}$) for Bianchi IX universe can be raised/lowered using a Kronecker delta function ($\delta^{\sigma\tau}$) and not the 3-metric $h_{\alpha\beta}$. Appendix G contains the detailed calculations.

## 6.3 Bianchi IX-3D Maxwell-scalar field system

In order to calculate the equations of motion corresponding to the Hamiltonian constraint in eq. (337) plus the free & massless homogeneous scalar field, we need to first solve the diffeomorphism constraints like we did in Chapter 5.2 (eq. (209)). We don't need to worry about the Gauss constraint as it is identically zero for a Bianchi IX-Maxwell-scalar field system.

### 6.3.1 Solving the diffeomorphism constraints

Coupling a free & massless homogeneous scalar field does not change the diffeomorphism constraints as we proved in eq. (244). The electromagnetic field does, so we consider the diffeomorphism constraints in eq. (337). We already know the contribution for the pure Bianchi IX case from eqs. (207)-(209). For the electromagnetic case, we have:

$$
\begin{aligned}
n^{\delta}\Pi^{\alpha}A_{\beta}\epsilon_{\tau\delta\alpha}\delta^{\tau\beta} &= n^3\Pi^1 A_2\epsilon_{231}\delta^{22} + n^3\Pi^2 A_1\epsilon_{132}\delta^{11} + n^2\Pi^1 A_3\epsilon_{321}\delta^{33} \\
&\quad + n^2\Pi^3 A_1\epsilon_{123}\delta^{11} + n^1\Pi^2 A_3\epsilon_{312}\delta^{33} + n^1\Pi^3 A_2\epsilon_{213}\delta^{22} \\
&= n^1\left(\Pi^2 A_3 - \Pi^3 A_2\right) + n^2\left(\Pi^3 A_1 - \Pi^1 A_3\right) + n^3\left(\Pi^1 A_2 - \Pi^2 A_1\right).
\end{aligned}
$$

Thus we have for the overall diffeomorphism constraints in eq. (337) (Bianchi IX + electromagnetic field) the following:

$$2\left(h_{\alpha\tau}\pi^{\tau\beta} - h_{\beta\tau}\pi^{\tau\alpha}\right) + \left(\Pi^{\alpha}A_{\beta} - \Pi^{\beta}A_{\alpha}\right) \approx 0.$$

(338)

We identify the matrix commutation just like in eq. (208) and if we define:

$$W^{\alpha}_{\beta} \equiv -\frac{1}{2}\left(A_{\beta}\Pi^{\alpha} - A_{\alpha}\Pi^{\beta}\right),$$

(339)

then eq. (338) can be written as follows:

$$[h, \pi]^\alpha_\beta \approx W^\alpha_\beta. \tag{340}$$

Just like in Chapter 5.2, we need to make this set of constraints a second class constraint.[13] To motivate the choice of gauge fixing, we evaluate the Poisson brackets using the definition provided in eq. (188). We again get the same as eqs. (210, 211). Thus we again choose as an ansatz for the 3-metric $h_{\alpha\beta}$ as:

$$h_{\alpha\beta} = \text{diag}(Q_1, Q_2, Q_3), \tag{341}$$

where again, eq. (211) suggests that $Q_1 \neq Q_2$, $Q_2 \neq Q_3$ and $Q_3 \neq Q_1$ for the diffeomorphism constraints to be a true set of second class constraints. But unlike eq. (209), here the conjugate metric $\pi^{\alpha\beta}$ does not commute with the 3-metric $h_{\delta\sigma}$ as clear from eq. (340). Thus in this case the conjugate momenta is not in the diagonal form unlike eq. (212). We need to solve for the off-diagonal terms of $\pi^{\alpha\beta}$.

We choose $\alpha = 1, \beta = 2$ in eq. (340) to solve for the strong equality case and use the ansatz in eq. (341) along with the symmetric property of the conjugate momenta ($\pi^{\alpha\beta} = +\pi^{\beta\alpha}$) to get:

$$\left(h_{1\tau}\pi^{\tau 2} - h_{2\tau}\pi^{\tau 1}\right) = -\frac{1}{2}\left(A_2 \Pi^1 - A_1 \Pi^2\right),$$

$$\Rightarrow \left(h_{11}\pi^{12} - h_{22}\pi^{21}\right) = -\frac{1}{2}\left(A_2 \Pi^1 - A_1 \Pi^2\right), \tag{342}$$

$$\Rightarrow \left(Q_1 \pi^{12} - Q_2 \pi^{21}\right) = -\frac{1}{2}\left(A_2 \Pi^1 - A_1 \Pi^2\right).$$

Thus $\{\alpha, \beta\} = \{1, 2\}$ in eq. (340) using the ansatz in eq.(341) leads to:

$$\Rightarrow \pi^{12} = \pi^{21} = -\frac{1}{2}\frac{\left(A_2 \Pi^1 - A_1 \Pi^2\right)}{(Q_1 - Q_2)}. \tag{343}$$

Similarly for the choices of $\{\alpha, \beta\} = \{1, 3\}$ and $\{\alpha, \beta\} = \{2, 3\}$ respectively, we get:

$$\pi^{13} = \pi^{31} = -\frac{1}{2}\frac{\left(A_3 \Pi^1 - A_1 \Pi^3\right)}{(Q_1 - Q_3)},$$

$$\pi^{23} = \pi^{32} = -\frac{1}{2}\frac{\left(A_3 \Pi^2 - A_2 \Pi^3\right)}{(Q_2 - Q_3)}. \tag{344}$$

The cases $\{\alpha, \beta\} = \{2, 1\}$, $\{\alpha, \beta\} = \{3, 1\}$ & $\{\alpha, \beta\} = \{3, 2\}$ lead to the same expressions above and are not independent. For the diagonal terms $\{\pi^{11}, \pi^{22}, \pi^{33}\}$, we use the AHS variables just like in eq. (212).

Thus we have solved the diffeomorphism constraints to get for the 3-metric and its conjugate momenta the following:

$$h_{\alpha\beta} = \begin{pmatrix} Q_1 & 0 & 0 \\ 0 & Q_2 & 0 \\ 0 & 0 & Q_3 \end{pmatrix},$$

$$\pi^{\delta\sigma} = \begin{pmatrix} \frac{P_1}{Q_1} & \pi^{12} & \pi^{13} \\ \pi^{21} = \pi^{12} & \frac{P_2}{Q_2} & \pi^{23} \\ \pi^{31} = \pi^{13} & \pi^{32} = \pi^{23} & \frac{P_3}{Q_3} \end{pmatrix}, \tag{345}$$

where $\{\pi^{12}, \pi^{13}, \pi^{23}\}$ are provided in eqs. (343, 344).

---

[13]See footnote 3 in Chapter 3.3 for the definitions of *first class* and *second class constraints*.

### 6.3.2 Simplifying the Hamiltonian constraint

Recall that we are coupling a free & massless homogeneous classical scalar field (the case that we solved in Chapter 5.3) to the Bianchi IX-Maxwell system. The complete Hamiltonian constraint of the Bianchi IX-Maxwell-scalar field system is:

$$
\mathbb{H} = \text{Tr}\left(h^2\right) - \frac{1}{2}\left(\text{Tr}(h)\right)^2 + \frac{1}{2}\left(\pi_\Phi\right)^2 - \frac{1}{2}\left(\pi^{\alpha\beta}h_{\alpha\beta}\right)^2
$$
$$
+ \underbrace{\pi^{\alpha\beta}\pi_{\alpha\beta}}_{\text{x}} + \underbrace{\frac{1}{2}\Pi^\alpha\Pi_\alpha}_{\text{y}} + \underbrace{\frac{h}{4}h^{\mu\alpha}h^{\nu\sigma}A_\beta A_\tau \epsilon^\beta_{\alpha\sigma}\epsilon^\tau_{\mu\nu}}_{\text{z}}, \tag{346}
$$

where we made use of eqs. (244, 337) because "Bianchi IX-Maxwell-Scalar = Bianchi IX + Maxwell + Scalar field". Also we set the Lagrange multiplier $n$ to scale like $\sqrt{h}$ so that $n' = \frac{n}{\sqrt{h}} = 1$ without loss of generality.

The first three terms are already evaluated in eq. (214) as the ansatz for the 3-metric here in eq. (341) is the same as in eq. (212). Similarly for the term $\left(-\frac{1}{2}\left(\pi^{\alpha\beta}h_{\alpha\beta}\right)^2\right)$ in eq. (346), we have the same result as in eq. (215) because the off-diagonal terms in $\pi^{\alpha\beta}$ in eq. (345) do not contribute. Thus we are left with terms (x), (y) and (z).

Terms (x), (y) & (z) become:

$$
\begin{aligned}
\text{Term (x)} &= \pi^{\alpha\beta}\pi_{\alpha\beta} = \pi^{\alpha\beta}\pi^{\tau\sigma}h_{\alpha\tau}h_{\beta\sigma} \\
&= h_{11}h_{11}\pi^{11}\pi^{11} + h_{22}h_{22}\pi^{22}\pi^{22} + h_{33}h_{33}\pi^{33}\pi^{33} \\
&\quad + 2h_{11}h_{22}\pi^{12}\pi^{12} + 2h_{11}h_{33}\pi^{13}\pi^{13} + 2h_{22}h_{33}\pi^{23}\pi^{23} \\
&= \left(P_1^2 + P_2^2 + P_3^2\right)^2 + 2\left(Q_1 Q_2 (\pi^{12})^2 + Q_1 Q_3 (\pi^{13})^2 + Q_2 Q_3 (\pi^{23})^2\right), \\
\text{Term (y)} &= \frac{1}{2}\Pi^\alpha\Pi_\alpha = \frac{1}{2}\Pi^\alpha\Pi^\beta h_{\alpha\beta} \\
&= \frac{1}{2}\left[(P^1)^2 h_{11} + (P^2)^2 h_{22} + (P^3)^2 h_{33}\right] \\
&= \frac{1}{2}\left[(P^1)^2 Q_1 + (P^2)^2 Q_2 + (P^3)^2 Q_3\right], \\
\text{Term (z)} &= \frac{h}{4}h^{\mu\alpha}h^{\nu\sigma}A_\beta A_\tau \epsilon_{\gamma\alpha\sigma}\delta^{\gamma\beta}\epsilon_{\omega\mu\nu}\delta^{\omega\tau} \\
&= \frac{h}{4}2\left[h^{11}h^{22}A_3 A_3 \epsilon_{312}\epsilon_{312} + h^{11}h^{33}A_2 A_2 \epsilon_{213}\epsilon_{213} + h^{22}h^{33}A_1 A_1 \epsilon_{123}\epsilon_{123}\right] \\
&= \frac{1}{2}Q_1 Q_2 Q_3 \left[\frac{1}{Q_1 Q_2}(A_3)^2 + \frac{1}{Q_1 Q_3}(A_2)^2 + \frac{1}{Q_2 Q_3}(A_1)^2\right] \\
&= \frac{1}{2}\left[Q_3(A_3)^2 + Q_2(A_2)^2 + Q_1(A_1)^2\right].
\end{aligned} \tag{347}
$$

Therefore plugging eqs. (214, 215, 347) into eq. (346) as well as using the definitions for $\{\pi^{12}, \pi^{13}, \pi^{23}\}$ from eqs. (343, 344), we get for the Hamiltonian constraint:

$$
\begin{aligned}
\mathbb{H} = &\left(Q_1^2 + Q_2^2 + Q_3^2\right) + \left(P_1^2 + P_2^2 + P_3^2\right) - \frac{1}{2}\left(Q_1 + Q_2 + Q_3\right)^2 - \frac{1}{2}\left(P_1 + P_2 + P_3\right)^2 + \frac{\pi_\Phi^2}{2} \\
&+ Q_1 Q_2 \frac{\left(A_2\Pi^1 - A_1\Pi^2\right)^2}{2\left(Q_1 - Q_2\right)^2} + Q_1 Q_3 \frac{\left(A_3\Pi^1 - A_1\Pi^3\right)^2}{2\left(Q_1 - Q_3\right)^2} + Q_2 Q_3 \frac{\left(A_3\Pi^2 - A_2\Pi^3\right)^2}{2\left(Q_2 - Q_3\right)^2} \\
&+ \frac{Q_1}{2}\left((A_1)^2 + (\Pi^1)^2\right) + \frac{Q_2}{2}\left((A_2)^2 + (\Pi^2)^2\right) + \frac{Q_3}{2}\left((A_3)^2 + (\Pi^3)^2\right),
\end{aligned} \tag{348}
$$

where the first line represents the Bianchi IX + scalar field system while the second & third lines contain the Maxwell's contributions.

### 6.3.3 Switching to Jacobi variables

We now switch to Jacobi variables like we did in eqs. (218, 219) which we reproduce here for convenience:

$$
P_1 = -\frac{k_x}{\sqrt{2}} + \frac{k_y}{\sqrt{6}} + D, \quad P_2 = \frac{k_x}{\sqrt{2}} + \frac{k_y}{\sqrt{6}} + D, \quad P_3 = -\sqrt{\frac{2}{3}}k_y + D,
$$
$$
Q_1 = \alpha e^{-\frac{x}{\sqrt{2}} + \frac{y}{\sqrt{6}}}, \qquad Q_2 = \alpha e^{\frac{x}{\sqrt{2}} + \frac{y}{\sqrt{6}}}, \qquad Q_3 = \alpha e^{-\sqrt{\frac{2}{3}}y},
\tag{349}
$$

and

$$
\alpha = v^{\frac{2}{3}}, \qquad D = \frac{v\tau}{2}.
\tag{350}
$$

We have already done this transformation for the first line in eq. (245), so we just need to consider the transformations of the Maxwell's terms in eq. (348).

We start with the second line of eq. (348) containing terms of the form $(A_\beta \Pi^\alpha - A_\alpha \Pi^\beta)$:

$$
\begin{aligned}
S_{33} &\equiv \frac{2Q_1 Q_2}{(Q_1 - Q_2)^2} = \frac{2\alpha^2 e^{2y/\sqrt{6}}}{\alpha^2 \left(e^{-\sqrt{2}x + 2y/\sqrt{6}} + e^{+\sqrt{2}x + 2y/\sqrt{6}} - 2e^{2y/\sqrt{6}}\right)} \\
&= \frac{2}{\left(e^{-x/\sqrt{2}} - e^{x/\sqrt{2}}\right)^2} = \frac{1}{2} \operatorname{csch}^2\left(\frac{x}{\sqrt{2}}\right), \\
S_{22} &\equiv \frac{2Q_1 Q_3}{(Q_1 - Q_3)^2} = \frac{1}{2} \operatorname{csch}^2\left(\frac{x - \sqrt{3}y}{2\sqrt{2}}\right), \\
S_{11} &\equiv \frac{2Q_2 Q_3}{(Q_2 - Q_3)^2} = \frac{1}{2} \operatorname{csch}^2\left(\frac{x + \sqrt{3}y}{2\sqrt{2}}\right).
\end{aligned}
\tag{351}
$$

We further define:

$$
\begin{aligned}
G^1 &\equiv \frac{1}{2}\left(A_3 \Pi^2 - A_2 \Pi^3\right), \\
G^2 &\equiv \frac{1}{2}\left(A_1 \Pi^3 - A_3 \Pi^1\right), \\
G^3 &\equiv \frac{1}{2}\left(A_2 \Pi^1 - A_1 \Pi^2\right).
\end{aligned}
\tag{352}
$$

Thus we get for the second line of eq. (348) the following in Jacobi variables:

$$
\text{Second Line} = S_{\mu\nu}(x, y)G^\mu(A, \Pi) G^\nu(A, \Pi),
\tag{353}
$$

where $S_{\mu\nu} = \operatorname{diag}(S_{11}, S_{22}, S_{33})$ and $\{S_{11}, S_{22}, S_{33}, G^1, G^2, G^3\}$ are defined in eqs. (351, 352).

Now we focus on the third line of eq. (348) containing terms of the form $\left((A_\alpha)^2 + (\Pi^\alpha)^2\right)$:

$$
\text{Third Line} = v^{2/3} T_{\mu\nu}(x, y)\left[\Pi^\mu \Pi^\nu + A_\alpha A_\beta \delta^{\alpha\mu} \delta^{\beta\nu}\right],
\tag{354}
$$

where $T_{\mu\nu} \equiv \frac{1}{2}\operatorname{diag}\left(e^{-x/\sqrt{2} + y/\sqrt{6}}, e^{x/\sqrt{2} + y/\sqrt{6}}, e^{-\sqrt{2/3}y}\right)$

Thus as a summary, the Hamiltonian constraint for Bianchi IX-Maxwell-scalar field system after solving the diffeomorphism constraints in Section 6.3.1 is as follows:

$$
\begin{aligned}
\mathbb{H} = {}& k_x^2 + k_y^2 + \frac{\pi_\Phi^2}{2} - \frac{3}{8}v^2\tau^2 + v^{4/3}U(x, y) \\
&+ S_{\mu\nu}(x, y)G^\mu(A, \Pi) G^\nu(A, \Pi) + v^{2/3}T_{\mu\nu}(x, y)\left[\Pi^\mu \Pi^\nu + A_\alpha A_\beta \delta^{\alpha\mu}\delta^{\beta\nu}\right],
\end{aligned}
\tag{355}
$$

where we reproduce the shape potential $U(x, y)$ from eqs. (224, 225) here:

$$U(x, y) \equiv f(-\sqrt{3}x + y) + f(\sqrt{3}x + y) + f(-2y), \tag{356}$$

having $\boxed{f(z) \equiv \frac{1}{2}e^{2z/\sqrt{6}} - e^{-z/\sqrt{6}}.}$ Also:

$$
\begin{aligned}
S_{\mu\nu} &\equiv \frac{1}{2}\mathrm{diag}\left(\mathrm{csch}^2\left(\frac{x + \sqrt{3}y}{2\sqrt{2}}\right), \mathrm{csch}^2\left(\frac{x - \sqrt{3}y}{2\sqrt{2}}\right), \mathrm{csch}^2\left(\frac{x}{\sqrt{2}}\right)\right), \\
G^{\mu} &\equiv \frac{1}{2}\left(A_3\Pi^2 - A_2\Pi^3, A_1\Pi^3 - A_3\Pi^1, A_2\Pi^1 - A_1\Pi^2\right), \\
T_{\mu\nu} &\equiv \frac{1}{2}\mathrm{diag}\left(e^{-\frac{x}{\sqrt{2}} + \frac{y}{\sqrt{6}}}, e^{\frac{x}{\sqrt{2}} + \frac{y}{\sqrt{6}}}, e^{-\sqrt{\frac{2}{3}}y}\right),
\end{aligned}
\tag{357}
$$

where for brevity, we denote the components as $S_{\mu\nu} = \mathrm{diag}(S_{11}, S_{22}, S_{33})$, $G^{\mu} = (G^1, G^2, G^3)$ and $T_{\mu\nu} = \mathrm{diag}(T_{11}, T_{22}, T_{33})$. The electromagnetic contributions are called the $S$-potential & the $T$-potential, in addition to the (shape) $U$-potential of the pure Bianchi IX universe. We plot the the matrix elements of $S_{\mu\nu}$ & $T_{\mu\nu}$ in Fig. (12) and Fig. (13), respectively.

Here the list of conjugate variables is:

$$
\begin{aligned}
\{x, k_x\} = \{y, k_y\} = \{v, \tau\} &= 1, \\
\{\Phi, \pi_\Phi\} &= 1, \\
\{A_\alpha, \Pi^\beta\} &= \delta_\alpha^\beta.
\end{aligned}
\tag{358}
$$

### 6.3.4 Equations of motions

We evaluate the complete set of equations of motion for the dynamical variables of the Bianchi IX-Maxwell-scalar field system, namely the ones contained in eq. (358).

The Hamilton's equations of motion (with respect to the coordinate time $t$) corresponding to the Hamiltonian constraint in eq. (357) and the conjugate variables in eq. (358) are:

$$
\begin{aligned}
\dot{x} &= +\frac{\partial \mathbb{H}}{\partial k_x}, & \dot{k}_x &= -\frac{\partial \mathbb{H}}{\partial x}, & \dot{y} &= +\frac{\partial \mathbb{H}}{\partial k_y}, & \dot{k}_y &= -\frac{\partial \mathbb{H}}{\partial y}, & \dot{v} &= +\frac{\partial \mathbb{H}}{\partial \tau}, \\
\dot{\tau} &= -\frac{\partial \mathbb{H}}{\partial v}, & \dot{\Phi} &= +\frac{\partial \mathbb{H}}{\partial \pi_\Phi}, & \dot{\pi}_\Phi &= -\frac{\partial \mathbb{H}}{\partial \Phi}, & \dot{A}_\alpha &= +\frac{\partial \mathbb{H}}{\partial \Pi^\alpha}, & \dot{\Pi}^\beta &= -\frac{\partial \mathbb{H}}{\partial A_\beta}.
\end{aligned}
\tag{359}
$$

We now proceed to show their explicit expressions:

$$
\begin{aligned}
\dot{x} =\,& 2k_x, \\
\dot{k}_x =\,& -v^{4/3}\frac{\partial U(x, y)}{\partial x} - \left[(G^1)^2\frac{\partial S_{11}}{\partial x} + (G^2)^2\frac{\partial S_{22}}{\partial x} + (G^3)^2\frac{\partial S_{33}}{\partial x}\right] \\
& -v^{2/3}\left[\left((A_1)^2 + (\Pi^1)^2\right)\frac{\partial T_{11}}{\partial x} + \left((A_2)^2 + (\Pi^2)^2\right)\frac{\partial T_{22}}{\partial x}\right],
\end{aligned}
\tag{360}
$$

$$
\begin{aligned}
\dot{y} =\,& 2k_y, \\
\dot{k}_y =\,& -v^{4/3}\frac{\partial U(x, y)}{\partial y} - \left[(G^1)^2\frac{\partial S_{11}}{\partial y} + (G^2)^2\frac{\partial S_{22}}{\partial y}\right] \\
& -v^{2/3}\left[\left((A_1)^2 + (\Pi^1)^2\right)\frac{\partial T_{11}}{\partial y} + \left((A_2)^2 + (\Pi^2)^2\right)\frac{\partial T_{22}}{\partial y} + \left((A_3)^2 + (\Pi^3)^2\right)\frac{\partial T_{33}}{\partial y}\right],
\end{aligned}
\tag{361}
$$

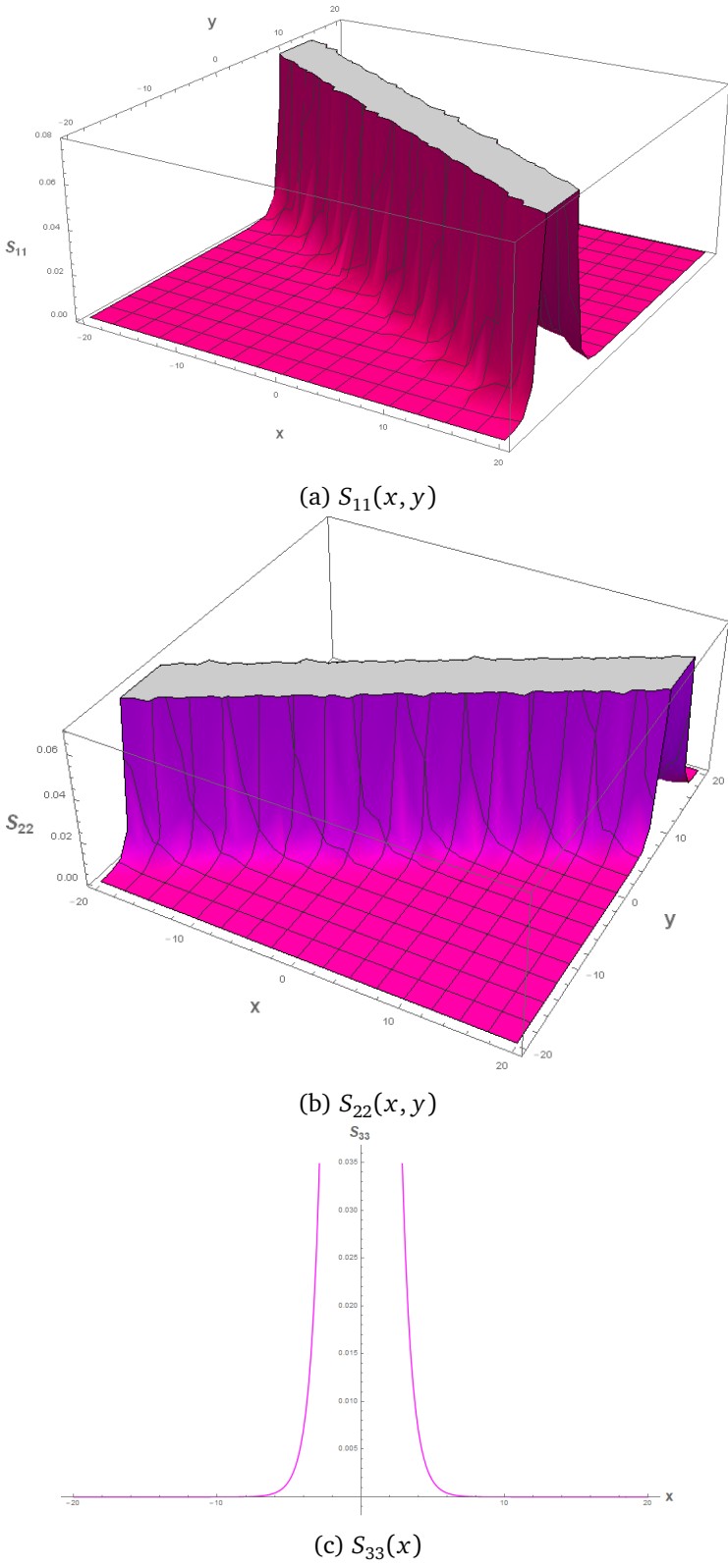

(a) $S_{11}(x, y)$

(b) $S_{22}(x, y)$

(c) $S_{33}(x)$

Figure 12: $S$-Potential for the Bianchi IX-Maxwell system

(a) $T_{11}(x, y)$

(b) $T_{22}(x, y)$

(c) $T_{33}(y)$

Figure 13: $T$-Potential for the Bianchi IX-Maxwell system

$$\dot{v} = \frac{-3}{4} v^2 \tau \,,$$

$$\dot{\tau} = \frac{3}{4} v \tau^2 - \frac{4}{3} v^{1/3} U(x, y)$$

$$- \frac{2}{3 v^{1/3}} \left[ \left( (A_1)^2 + (\Pi^1)^2 \right) T_{11} + \left( (A_2)^2 + (\Pi^2)^2 \right) T_{22} + \left( (A_3)^2 + (\Pi^3)^2 \right) T_{33} \right], \tag{362}$$

$$\dot{\Phi} = \pi_\Phi \,,$$
$$\dot{\pi}_\Phi = 0 \,, \tag{363}$$

$$\dot{A}_1 = 2 v^{\frac{2}{3}} T_{11}(x, y) \Pi^1 - \frac{1}{2} S_{22} \left( A_1 \Pi^3 - A_3 \Pi^1 \right) A_3 + \frac{1}{2} S_{33} \left( A_2 \Pi^1 - A_1 \Pi^2 \right) A_2 \,,$$
$$\dot{\Pi}^1 = -2 v^{\frac{2}{3}} T_{11}(x, y) A_1 - \frac{1}{2} S_{22} \left( A_1 \Pi^3 - A_3 \Pi^1 \right) \Pi^3 + \frac{1}{2} S_{33} \left( A_2 \Pi^1 - A_1 \Pi^2 \right) \Pi^2 \,, \tag{364}$$

$$\dot{A}_2 = 2 v^{\frac{2}{3}} T_{22}(x, y) \Pi^2 - \frac{1}{2} S_{33} \left( A_2 \Pi^1 - A_1 \Pi^2 \right) A_1 + \frac{1}{2} S_{11} \left( A_3 \Pi^2 - A_2 \Pi^3 \right) A_3 \,,$$
$$\dot{\Pi}^2 = -2 v^{\frac{2}{3}} T_{22}(x, y) A_2 - \frac{1}{2} S_{33} \left( A_2 \Pi^1 - A_1 \Pi^2 \right) \Pi^1 + \frac{1}{2} S_{11} \left( A_3 \Pi^2 - A_2 \Pi^3 \right) \Pi^3 \,, \tag{365}$$

$$\dot{A}_3 = 2 v^{\frac{2}{3}} T_{33}(x, y) \Pi^3 - \frac{1}{2} S_{11} \left( A_3 \Pi^2 - A_2 \Pi^3 \right) A_2 + \frac{1}{2} S_{22} \left( A_1 \Pi^3 - A_3 \Pi^1 \right) A_1 \,,$$
$$\dot{\Pi}^3 = -2 v^{\frac{2}{3}} T_{33}(x, y) A_3 - \frac{1}{2} S_{11} \left( A_3 \Pi^2 - A_2 \Pi^3 \right) \Pi^2 + \frac{1}{2} S_{22} \left( A_1 \Pi^3 - A_3 \Pi^1 \right) \Pi^1 \,, \tag{366}$$

where we use eqs. (356, 357) for explicit forms of components.

# 7 Conclusion & Outlook

We introduced the Hamiltonian formulation of general relativity and homogeneous cosmologies through this work where we tried to be detailed and self-contained in our approach. We presented a variety of examples and did their ADM analysis such as a scalar field coupled to Bianchi IX universe, electromagnetic field field coupled to gravity, & so on. The idea was to acquaint the readers with the canonical formalism and provide them a hands-on practice of implementing it. But in order to keep this introductory and detailed, we had to overlook some topics such as ADM mass & ADM momentum.

The concepts we developed in Chapter 5 such as Mixmaster dynamics and quiescence play a significant role at the research frontier while trying to resolve big-bang singularities in various cosmological models. We already know about the inevitability of singular solutions in general relativity which is captured by the famous Penrose-Hawking singularity theorems [34–37]. As Stephen Hawking put it, "A singularity is a place where the classical concepts of space and time break down as do all the known laws of physics..." [38]. But it has been shown in the literature [28, 29] that for the systems considered in Sections 5.3 & 5.4, the classical degrees of freedom can be evolved uniquely through a timelike singularity, namely the big-bang. Clearly this contradicts the common understanding of gravitational singularities, which so far have been considered as regions where the known laws of classical field theory cease to be valid. The methodology and concepts introduced in this work play a significant role towards resolving singularities such as these. The arguments detailed in [28, 29] do not

provide a general statement, but showing counter-examples is a beginning towards what we hope could be, in the future, a general theorem on the continuation through singularities (similar in spirit to the Hawking-Penrose theorems on the inevitability of singularities). There is a large amount of work that remains to be done as future research projects and the concepts introduced here will continue to play a fundamental role towards this goal.

Just as in Chapter 6 where we generalized to the case of Einstein-Maxwell system, the next natural generalization is to consider a general Yang-Mills classical gauge field. This means adding an extra Lagrangian density term to the action, just like what we did in eq. (317):

$$\mathcal{L}_{\text{Yang-Mills}} = -\frac{1}{4}\sqrt{-g}\,\mathcal{F}^{\mu\nu(i)}\mathcal{F}^{(j)}_{\mu\nu}\delta_{(i)(j)}\,, \tag{367}$$

where $\{(i),(j)\}$ denote an internal index that runs over $\{1,2,\dots,n\}$ with $n$ being the dimension of the Lie algebra of the Yang-Mills gauge field [39]. The generalized Faraday tensor is defined as:

$$\mathcal{F}^{(a)}_{\mu\nu} = \partial_\mu\mathcal{A}^{(a)}_\nu - \partial_\nu\mathcal{A}^{(a)}_\mu + g f^{(a)}_{(b)(c)}\mathcal{A}^{(b)}_\mu\mathcal{A}^{(c)}_\nu\,, \tag{368}$$

where $\mathcal{A}^{(a)}_\mu$ is the gauge field, $g$ is the coupling constant and $f^{(a)}_{(b)(c)}$ are the structure constants of the Lie algebra. Then just like what we did with the electromagnetic field in Chapter 6, we need to find the corresponding Hamiltonian of the Yang-Mills field in $3+1-$variables and show the full Hamiltonian of the Einstein−Yang-Mills system to be of the form "Einstein + Yang-Mills". Then again we need to impose the homogeneous ansatz on the Hamiltonian so-obtained and in doing so, we will observe that, on top of the gauge field having more than one component, this case introduces also cubic and quartic terms in the Lagrangian, which will correspond to cubic and quartic terms in the Hamiltonian. Moreover, in this case, even after imposing the homogeneous ansatz, the non-linearity of Yang-Mills theory implies that the Gauss constraint won't be identically zero (unlike the electromagnetic case). Thus there will be several complications when compared to the Maxwell case and solving all the constraints of the theory will be significantly more challenging.

The next interesting case (or generalization) is that even though we exclusively focused on the homogeneity assumption, our universe is obviously non-homogeneous on small scales. Having a Hamiltonian approach for inhomogeneous cosmologies will go a long way in understanding the (classical) micro-structures of the cosmos. This generalization is expected to present significant additional challenges, with respect to all the cases mentioned previously.

Finally, we would like to remind that this entire work is grounded in classical Physics. The Hamiltonian formulation of any classical field theory is a pre-cursor to its quantization as is evident from the canonical approach to quantum field theory [40]. Accordingly, the Hamiltonian approach to classical gravity is seen as an imperative step in the grand attempt to quantize gravity. Various approaches towards this goal, such as Wheeler-DeWitt quantization & loop quantum gravity, are grounded in the concepts introduced in this work [13]. Then a natural extension of the models considered here which still lies within the limits of homogeneous cosmologies, is that of fermionic fields. Spinors do not couple directly to the metric variables, but rather to the frame fields ("vielbeins"), which take into account the orientation of spatial slices. Providing a canonical formalism for such a system will inevitably lead to some interesting consequences especially with the realization that a spinor is fundamentally a quantum object.

# Acknowledgements

I would like to thank Prof. Dr. Flavio Mercati and Prof. Dr. Laura Covi for taking their time out of their schedule to read parts of the manuscript as well as their support & insightful

discussions. I am also grateful to Ms. Martina Adamo for proof-reading parts of the manuscript and discussions. Naturally, I am responsible for the remaining mistakes.

## A Basic definitions & formulae in general relativity

We consider a spacetime manifold $\{\mathcal{M}, g_{\mu\nu}\}$ where $g_{\mu\nu}$ is the corresponding metric defined on the manifold which is symmetric in both its indices. Then we define *Christoffel symbols* as:

$$\Gamma^{\mu}_{\alpha\nu} = \frac{1}{2} g^{\mu\sigma} \left( \partial_{\alpha} g_{\nu\sigma} + \partial_{\nu} g_{\sigma\alpha} - \partial_{\sigma} g_{\alpha\nu} \right), \tag{A.1}$$

where we have $\Gamma^{\lambda}_{\mu\nu} = \Gamma^{\lambda}_{\nu\mu}$. The *covariant derivative* is defined accordingly for a tensor of arbitrary rank $(k, l)$ as:

$$\begin{aligned}
\nabla_{\sigma} T^{\mu_1\mu_2\cdots\mu_k}{}_{\nu_1\nu_2\cdots\nu_l} =& \partial_{\sigma} T^{\mu_1\mu_2\cdots\mu_k}{}_{\nu_1\nu_2\cdots\nu_l} \\
&+ \Gamma^{\mu_1}_{\sigma\lambda} T^{\lambda\mu_2\cdots\mu_k}{}_{\nu_1\nu_2\cdots\nu_l} + \Gamma^{\mu_2}_{\sigma\lambda} T^{\mu_1\lambda\cdots\mu_k}{}_{\nu_1\nu_2\cdots\nu_l} + \cdots \\
&- \Gamma^{\lambda}_{\sigma\nu_1} T^{\mu_1\mu_2\cdots\mu_k}{}_{\lambda\nu_2\cdots\nu_l} - \Gamma^{\lambda}_{\sigma\nu_2} T^{\mu_1\mu_2\cdots\mu_k}{}_{\nu_1\lambda\cdots\nu_l} - \cdots,
\end{aligned} \tag{A.2}$$

where shorthand notation for covariant derivative is $\nabla_{\mu} V^{\nu} \equiv V^{\nu}_{;\mu}$ and simple partial derivative is $\partial_{\mu} V^{\nu} \equiv V^{\nu}_{,\mu}$. By definition, covariant derivative of the metric is 0, $\nabla_{\alpha} g_{\mu\nu} = 0$. The 4−divergence of a vector $V^{\mu}$ is given by:

$$\nabla_{\mu} V^{\mu} = \frac{1}{\sqrt{-g}} \partial_{\alpha} \left( \sqrt{-g} V^{\alpha} \right). \tag{A.3}$$

Also the 4−Laplacian of a scalar function $f$ is given by:

$$\nabla_{\mu} \nabla^{\mu} f = \frac{1}{\sqrt{-g}} \frac{\partial}{\partial x^{\mu}} \left( \sqrt{-g} \frac{\partial f}{\partial x_{\mu}} \right). \tag{A.4}$$

The *Riemann curvature tensor* is defined as:

$$R^{\rho}_{\sigma\mu\nu} = \partial_{\mu} \Gamma^{\rho}_{\nu\sigma} - \partial_{\nu} \Gamma^{\rho}_{\mu\sigma} + \Gamma^{\rho}_{\mu\lambda} \Gamma^{\lambda}_{\nu\sigma} - \Gamma^{\rho}_{\nu\lambda} \Gamma^{\lambda}_{\mu\sigma}, \tag{A.5}$$

whose physical meaning can be captured by the action of $\left[ \nabla_{\mu}, \nabla_{\nu} \right]$ on a tensor $X$ of arbitrary rank $(k, l)$ as:

$$\begin{aligned}
\left[ \nabla_{\rho}, \nabla_{\sigma} \right] X^{\mu_1\cdots\mu_k}{}_{\nu_1\cdots\nu_l} =& -T^{\lambda}_{\rho\sigma} \nabla_{\lambda} X^{\mu_1\cdots\mu_k}{}_{\nu_1\cdots\nu_l} \\
&+ R^{\mu_1}{}_{\lambda\rho\sigma} X^{\lambda\mu_2\cdots\mu_k}{}_{\nu_1\cdots\nu_l} + R^{\mu_2}{}_{\lambda\rho\sigma} X^{\mu_1\lambda\cdots\mu_k}_1{}_{\nu_1\cdots\nu_l} + \cdots \\
&- R^{\lambda}{}_{\nu_1\rho\sigma} X^{\mu_1\cdots\nu_2\cdots\nu_l}{}_{\lambda_1\cdots\mu_k} - R^{\lambda}{}_{\nu_2\rho\sigma} X^{\mu_1\cdots\mu_k}{}_{\nu_1\lambda\cdots\nu_l} - \cdots,
\end{aligned} \tag{A.6}$$

where $T$ is the torsion tensor defined as a map from two vector fields $X$ and $Y$ to a third vector field:

$$T(X, Y) = \nabla_X Y - \nabla_Y X - [X, Y]. \tag{A.7}$$

Defining $R_{\rho\sigma\mu\nu} = g_{\rho\lambda} R^{\lambda}_{\sigma\mu\nu}$, we have the following properties of the Riemann curvature tensor: (i) $R_{\rho\sigma\mu\nu} = -R_{\sigma\rho\mu\nu}$, (ii) $R_{\rho\sigma\mu\nu} = -R_{\rho\sigma\nu\mu}$, (iii) $R_{\rho\sigma\mu\nu} = R_{\mu\nu\rho\sigma}$, (iv) $R_{\rho\sigma\mu\nu} + R_{\rho\mu\nu\sigma} + R_{\rho\nu\sigma\mu} = 0 \Leftrightarrow R_{\rho[\sigma\mu\nu]} = 0$, (v) $R_{[\rho\sigma\mu\nu]} = 0$, and (vi) $\nabla_{[\lambda} R_{\rho\sigma]\mu\nu} = 0$ (Bianchi identity) $\Leftrightarrow \left[ \left[ \nabla_{\lambda}, \nabla_{\rho} \right], \nabla_{\sigma} \right] + \left[ \left[ \nabla_{\rho}, \nabla_{\sigma} \right], \nabla_{\lambda} \right] + \left[ \left[ \nabla_{\sigma}, \nabla_{\lambda} \right], \nabla_{\rho} \right] = 0$. Clearly, there are $\frac{1}{12} n^2 \left( n^2 - 1 \right)$ independent components of the Riemann curvature tensor where, for example, $n = 4$ in $3 + 1$−dimensions give 20 independent components.

The *Ricci tensor* is defined as:

$$R_{\mu\nu} = R^{\lambda}_{\mu\lambda\nu}, \tag{A.8}$$

which has the property $R_{\mu\nu} = R_{\nu\mu}$. It can be shown that this can be written as:

$$R_{\mu\nu} = \frac{1}{\sqrt{|g|}} \partial_\lambda \left[ \sqrt{|g|} \Lambda^{\lambda}_{\mu\nu} \right] - \Lambda^{\rho}_{\mu\lambda} \Lambda^{\lambda}_{\rho\nu} - \partial_\mu \partial_\nu \ln\left( \sqrt{|g|} \right). \tag{A.9}$$

The *Ricci scalar* is defined as:

$$R = R^{\mu}_{\mu} = g^{\mu\nu} R_{\mu\nu}. \tag{A.10}$$

The *Weyl tensor* is defined as the Riemann curvature tensor minus its contractions as follows:

$$C_{\rho\sigma\mu\nu} = R_{\rho\sigma\mu\nu} - \frac{2}{(n-2)} \left( g_{\rho[\mu}R_{\nu]\sigma} - g_{\sigma[\mu}R_{\nu]\rho} \right) + \frac{2}{(n-1)(n-2)} g_{\rho[\mu}g_{\nu]\sigma}R, \tag{A.11}$$

where $n$ is the full spacetime dimensions. It has the properties: (i) $C_{\rho\sigma\mu\nu} = C_{[\rho\sigma][\mu\nu]}$, (ii) $C_{\rho\sigma\mu\nu} = C_{\mu\nu\rho\sigma}$, and (iii) $C_{\rho[\sigma\mu\nu]} = 0$.

The *Einstein tensor*, which we will also encounter in Chapter 2.2, is defined as:

$$G_{\mu\nu} = R_{\mu\nu} - \frac{1}{2} R g_{\mu\nu}, \tag{A.12}$$

which satisfies the symmetric property $G_{\mu\nu} = G_{\nu\mu}$ as well as $\nabla^{\mu} G_{\mu\nu} = 0$ (which follows from the Bianchi identity mentioned above) which will be of immense importance concerning the conservation of energy-momentum tensor as explained in Chapter 2.2.

Finally, we define a *geodesic* which is the generalization of the notion of a straight line in Euclidean plane to a general curved manifold. Any parametrized curve $x^{\mu}(\lambda)$ is a geodesic if it obeys:

$$\frac{d^2 x^{\mu}}{d\lambda^2} + \Gamma^{\mu}_{\rho\sigma} \frac{dx^{\rho}}{d\lambda} \frac{dx^{\sigma}}{d\lambda} = 0, \tag{A.13}$$

which is also known as the *geodesic equation*.

# B   Derivation of Gauss, Codazzi, Mainardi relations

The Gauss, Codazzi, Mainardi relations form the basis of $3+1-$formalism of general relativity. We assume the spacetime manifold $(\mathcal{M}, g_{\mu\nu})$ to be *globally hyperbolic* which admits foliation by a family of *spacelike* hypersurfaces $\Sigma_t$ ($t \in \mathbb{R}$ is the time parametrization of each hypersurface) as follows:

$$\mathcal{M} = \bigcup_t \Sigma_t \qquad \left( \Sigma_t \bigcap \Sigma_{t'} = \emptyset \right). \tag{B.1}$$

Let $n^{\mu}$ be the normal vector to the spacelike hypersurface $\Sigma_t$. Then the 3$-$*metric* induced on each of the hypersurface ($\gamma_{\mu\nu}$) is given by:

$$\gamma_{\mu\nu} = g_{\mu\nu} + n_\mu n_\nu \iff \gamma^{\mu\nu} = g^{\mu\nu} + n^\mu n^\nu, \tag{B.2}$$

and

$$\gamma^{\alpha}_{\beta} = \delta^{\alpha}_{\beta} + n^{\alpha} n_{\beta}. \tag{B.3}$$

Accordingly, the covariant derivative induced by $\gamma_{\mu\nu}$ is denoted by $D_\mu$ that satisfies $D_\alpha \gamma_{\mu\nu} = 0$ (or $D_a \gamma_{ij} = 0$), just like in full $3+1-$dimensions we have $\nabla_\alpha g_{\mu\nu} = 0$. The relation between 3$-$covariant derivative and $3+1-$covariant derivative is:

$$D_\rho T^{\alpha_1 \dots \alpha_p}_{\phantom{\alpha_1 \dots \alpha_p}\beta_1 \dots \beta_q} = \gamma^{\alpha_1}_{\mu_1} \cdots \gamma^{\alpha_p}_{\mu_p} \gamma^{\nu_1}_{\beta_1} \cdots \gamma^{\nu_q}_{\beta_q} \gamma^{\sigma}_{\rho} \nabla_\sigma T^{\mu_1 \dots \mu_p}_{\phantom{\mu_1 \dots \mu_p}\nu_1 \dots \nu_q}. \tag{B.4}$$

*Intrinsic curvature* is the Riemann curvature of each hypersurface induced by the 3−metric $\gamma_{\mu\nu}$, denoted by $^{(3)}R$, just like the 4−metric induces the Riemann curvature of the spacetime manifold $\mathcal{M}$, denoted by $^{(4)}R$. Intrinsic curvature of a hypersurface is independent of other hypersurfaces. But we are also interested in knowing that how the curvature itself changes as one proceeds from one hypersurface to the next. This is what is known as *extrinsic curvature $K$*. The defining relation for the extrinsic curvature tensor can be taken in terms of Lie derivative along the normal vector $n^\mu$ as follows:

$$K_{\mu\nu} = -\frac{1}{2}\mathcal{L}_n\gamma_{\mu\nu}, \tag{B.5}$$

which can be shown to be equivalent to (as done in Chapter 2.3):

$$K_{\mu\nu} = -\gamma_\mu^\alpha\gamma_\nu^\beta\nabla_\alpha n_\beta. \tag{B.6}$$

Similar to Ricci scalar $R$ in $3+1-$D, we have extrinsic curvature scalar defined by $K = \gamma^{\mu\nu}K_{\mu\nu}$. Thus, $\{\gamma_{\mu\nu}, D_\mu, {}^{(3)}R^\mu_{\nu\alpha\beta}, K_{\mu\nu}\}$ are 3−objects (as their respective contractions with the normal vector $n^\mu$ are zero), living on $\Sigma_t$ and accordingly the Greek indices can be replaced with Latin ones as only the spatial components are relevant.

The relations we are about to prove decomposes the $3+1-$objects (such as tensor $^{(4)}R^\mu_{\nu\alpha\beta}$, scalar $^{(4)}R$, etc.) living on full spacetime manifold $\mathcal{M}$ in terms of 3−objects (summarized above) residing on the spacelike hypersurface $\Sigma_t$.

## Gauss identities

### Gauss relation

Just like in $3+1-$D, we have in 3D the following identity valid for any arbitrary vector $V^\mu$ living on the hypersurface $\Sigma_t$ (such that $n_\mu V^\mu = 0$):

$$\left[D_\alpha, D_\beta\right]V^\gamma = {}^{(3)}R^\gamma_{\mu\alpha\beta}V^\mu. \tag{B.7}$$

Then we use eq. (B.4) to simplify the LHS as follows:

$$\begin{aligned}D_\alpha D_\beta V^\gamma = D_\alpha\left(D_\beta V^\gamma\right) &= \gamma^\mu_\alpha\gamma^\nu_\beta\gamma^\gamma_\rho\nabla_\mu\left(D_\nu V^\rho\right)\\ &= \gamma^\mu_\alpha\gamma^\nu_\beta\gamma^\gamma_\rho\nabla_\mu\left(\gamma^\sigma_\nu\gamma^\rho_\lambda\nabla_\sigma V^\lambda\right).\end{aligned} \tag{B.8}$$

Using the idempotent relation $\gamma^\alpha_\beta\gamma^\beta_\tau = \gamma^\alpha_\tau$ and the definition of the extrinsic curvature tensor, we get:

$$\begin{aligned}\Rightarrow D_\alpha D_\beta V^\gamma &= \gamma^\mu_\alpha\gamma^\nu_\beta\gamma^\gamma_\rho(n^\sigma\nabla_\mu n_\nu\gamma^\rho_\lambda\nabla_\sigma V^\lambda + \gamma^\sigma_\nu\nabla_\mu n^\rho\underbrace{n_\lambda\nabla_\sigma V^\lambda}_{-V^\lambda\nabla_\sigma n_\lambda} + \gamma^\sigma_\nu\gamma^\rho_\lambda\nabla_\mu\nabla_\sigma V^\lambda)\\ &= \gamma^\mu_\alpha\gamma^\nu_\beta\gamma^\gamma_\lambda\nabla_\mu n_\nu n^\sigma\nabla_\sigma V^\lambda - \gamma^\mu_\alpha\gamma^\sigma_\beta\gamma^\gamma_\rho V^\lambda\nabla_\mu n^\rho\nabla_\sigma n_\lambda + \gamma^\mu_\alpha\gamma^\sigma_\beta\gamma^\gamma_\lambda\nabla_\mu\nabla_\sigma V^\lambda\\ &= -K_{\alpha\beta}\gamma^\gamma_\lambda n^\sigma\nabla_\sigma V^\lambda - K^\gamma_\alpha K_{\beta\lambda}V^\lambda + \gamma^\mu_\alpha\gamma^\sigma_\beta\gamma^\gamma_\lambda\nabla_\sigma V^\lambda.\end{aligned} \tag{B.9}$$

Then we $\alpha\leftrightarrow\beta$ to get $D_\beta D_\alpha V^\gamma$ and subtract from $D_\alpha D_\beta V^\gamma$ to get:

$$\left[D_\alpha, D_\beta\right]V^\gamma = \left(K_{\alpha\mu}K^\gamma_\beta - K_{\beta\mu}K^\gamma_\alpha\right)V^\mu + \gamma^\rho_\alpha\gamma^\sigma_\beta\gamma^\gamma_\lambda\underbrace{\left(\nabla_\rho\nabla_\sigma V^\lambda - \nabla_\sigma\nabla_\rho V^\lambda\right)}_{{}^{(4)}R^\lambda_{\mu\rho\sigma}V^\mu}. \tag{B.10}$$

Rearranging and using $V^\mu = \gamma^\mu_\sigma V^\sigma$ (since $V^\mu$ is a 3−vector), we get:

$$\gamma^\mu_\alpha\gamma^\nu_\beta\gamma^\gamma_\rho\gamma^{\sigma\,(4)}_\lambda R^\rho_{\sigma\mu\nu}V^\lambda = {}^{(3)}R^\gamma_{\lambda\alpha\beta}V^\lambda + \left(K^\gamma_\alpha K_{\lambda\beta} - K^\gamma_\beta K_{\alpha\lambda}\right)V^\lambda. \tag{B.11}$$

But $V^\mu$ is an arbitrary 3−vector. We finally get the *Gauss relation*:

$$\boxed{\gamma^\mu_\alpha\gamma^\nu_\beta\gamma^\gamma_\rho\gamma^{\sigma\,(4)}_\delta R^\rho_{\sigma\mu\nu} = {}^{(3)}R^\gamma_{\delta\alpha\beta} + K^\gamma_\alpha K_{\delta\beta} - K^\gamma_\beta K_{\alpha\delta}.} \tag{B.12}$$

**Contracted Gauss relation**

We simply contract the indices $\gamma$ and $\alpha$ in the Gauss relation (eq. (B.12)) and use the idempotent relation of 3−metric, namely $\gamma^\alpha_\beta \gamma^\beta_\tau = \gamma^\alpha_\tau = \delta^\alpha_\tau + n^\alpha n_\tau$, to get the *contracted Gauss relation*:

$$\gamma^\mu_\alpha \gamma^\nu_\beta {}^{(4)}R_{\mu\nu} + \gamma_{\alpha\mu} n^\nu \gamma^\rho_\beta n^{\sigma (4)}R^\mu_{\nu\rho\sigma} = {}^{(3)}R_{\alpha\beta} + K K_{\alpha\beta} - K_{\alpha\mu} K^\mu_\beta . \tag{B.13}$$

**Scalar Gauss relation − generalization of *Theorema Egregium***

We take the trace of the contracted Gauss relation (eq. (B.13)) with respect to the 3−metric $\gamma^{\alpha\beta}$ and use the idempotent property of the 3−metric, $K^\mu_\mu = K^i_i = K$ and $K_{\mu\nu} K^{\mu\nu} = K_{ij} K^{ij}$ to get:

$$\gamma^{\alpha\beta} \left( \gamma^\mu_\alpha \gamma^\nu_\beta {}^{(4)}R_{\mu\nu} + \gamma_{\alpha\mu} n^\nu \gamma^\rho_\beta n^{\sigma (4)}R^\mu_{\nu\rho\sigma} \right) = \gamma^{\alpha\beta} \left( {}^{(3)}R_{\alpha\beta} + K K_{\alpha\beta} - K_{\alpha\mu} K^\mu_\beta \right) \tag{B.14}$$

$$\Rightarrow {}^{(4)}R + \gamma^\rho_\mu n^\nu n^{\sigma (4)}R^\mu_{\nu\rho\sigma} = {}^{(3)}R + K^2 - K_{ij} K^{ij} . \tag{B.15}$$

Then using $\gamma^\rho_\mu = \delta^\rho_\mu + n^\rho n_\mu$ and ${}^{(4)}R^\mu_{\nu\rho\sigma} n^\rho n_\mu n^\nu n^\sigma = 0$ as contractions are done between symmetric and anti-symmetric pair of indices, we finally get the *scalar Gauss relation*:

$$^{(4)}R + 2 {}^{(4)}R_{\mu\nu} n^\mu n^\nu = {}^{(3)}R + K^2 - K_{ij} K^{ij} . \tag{B.16}$$

This is a generalization of the famous *Theorema Egregium* which was originally proposed for 2−D surfaces embedded in Euclidean space $\mathbb{R}^3$ whose curvature is 0. Accordingly the LHS vanishes. Moreover the metric $g_{\mu\nu}$ of $\mathbb{R}^3$ is Riemannian and not Lorentzian, so $\gamma^\alpha_\tau = \delta^\alpha_\tau - n^\alpha n_\tau$ instead of what we used above, namely $\gamma^\alpha_\tau = \delta^\alpha_\tau + n^\alpha n_\tau$. Thus $K^2 - K_{ij} K^{ij}$ will have signs reversed and we get the original *Theorema Egregium*:

$$^{(2)}R - K^2 + K_{ij} K^{ij} = 0 . \tag{B.17}$$

We can further simplify this for the special case of 2−D where $K_{ij}$ can be diagonalized in an orthonormal basis with respect to 2−metric $\gamma_{ij}$ (remember $g_{\mu\nu}$ is Euclidean), so that $K_{ij} = \text{diag}(\kappa_1, \kappa_2)$ where $\kappa_1$ and $\kappa_2$ are principal curvatures of the 2−D hypersurface $\Sigma$. Obviously, $K^{ij} = \text{diag}(\kappa_1, \kappa_2)$. Thus $K = \kappa_1 + \kappa_2$ and $K_{ij} K^{ij} = (\kappa_1)^2 + (\kappa_2)^2$. The original Theorema Egregium simplifies to:

$$^{(2)}R = 2\kappa_1 \kappa_2 \quad \text{(Special Case of 2D)}. \tag{B.18}$$

## Codazzi-Mainardi identities

### Codazzi-Mainardi relation

We have for the normal vector $n^\mu$:

$$\left[ \nabla_\alpha, \nabla_\beta \right] n^\mu = {}^{(4)}R^\mu_{\nu\alpha\beta} n^\nu . \tag{B.19}$$

We project this relation onto the hypersurface $\Sigma_t$ which simply means contracting each of the free indices with the 3−metric:

$$\gamma^\rho_\alpha \gamma^\tau_\beta \gamma^\mu_\gamma \left[ \nabla_\rho, \nabla_\tau \right] n^\gamma = \gamma^\rho_\alpha \gamma^\tau_\beta \gamma^\mu_\sigma {}^{(4)}R^\sigma{}_{\gamma\rho\tau} n^\gamma . \tag{B.20}$$

Then using the identity for the extrinsic curvature tensor proved in Chapter 2.3, namely $K_{\mu\nu} = -\nabla_\mu n_\nu - n_\mu a_\nu$ where $a_\mu \equiv n^\nu \nabla_\nu n_\mu$, we replace $\nabla_\nu n^\gamma$ to get for the first term on the LHS:

$$
\begin{aligned}
\gamma_\alpha^\mu \gamma_\beta^\nu \gamma_\rho^\gamma \nabla_\mu \nabla_\nu n^\rho &= \gamma_\alpha^\mu \gamma_\beta^\nu \gamma_\rho^\gamma \nabla_\mu \left( -K_\nu^\rho - a^\rho n_\nu \right) \\
&= -\gamma_\alpha^\mu \gamma_\beta^\nu \gamma_\rho^\gamma \left( \nabla_\mu K_\nu^\rho + \nabla_\mu a^\rho n_\nu + a^\rho \nabla_\mu n_\nu \right).
\end{aligned}
\tag{B.21}
$$

Then we use the relation between 3– and 4–covariant derivatives (eq. (B.4)), namely $D_\mu T_\beta^\gamma = \gamma_\mu^\alpha \gamma_\lambda^\nu \gamma_\beta^\rho \nabla_\alpha T_\rho^\lambda$, as well as $\gamma_\beta^\nu n_\nu = 0$ (since $n_\mu$ is a timelike vector, so there is no projection on a spacelike hypersurface $\Sigma_t$), $\gamma_\beta^\nu a^\beta = a^\nu$ (since $a^\nu$ is a spacelike vector, so projection onto $\Sigma_t$ will give the same vector) and the definition of the extrinsic curvature tensor (eq. B.6), to get:

$$
\gamma_\alpha^\mu \gamma_\beta^\nu \gamma_\rho^\gamma \nabla_\mu \nabla_\nu n^\rho = -D_\alpha K^\gamma \beta + a^\gamma K_{\alpha\beta}.
\tag{B.22}
$$

With this result obtained, we permute $\alpha \leftrightarrow \beta$ (and not $\mu \leftrightarrow \nu$ as these are contracted indices) and then subtract from this result to get the RHS of eq. (B.20) (keeping in mind $K_{\mu\nu} = +K_{\nu\mu}$):

$$
\boxed{\gamma_\rho^\gamma n^\sigma \gamma_\alpha^\mu \gamma_\beta^\nu {}^{(4)}R_{\sigma\mu\nu}^\rho = D_\beta K_\alpha^\gamma - D_\alpha K_\beta^\gamma.}
\tag{B.23}
$$

This is the *Codazzi-Mainardi relation*. The point to be noted about this relation is that on the LHS, we have contracted $n^\sigma$ with ${}^{(4)}R_{\sigma\mu\nu}^\rho$. Had we contracted $n_\rho$ with ${}^{(4)}R_{\sigma\mu\nu}^\rho$ or $n^\mu$ with ${}^{(4)}R_{\sigma\mu\nu}^\rho$, we would *not* have obtained an independent relation due to the symmetries of the Riemann curvature tensor and the RHS would at most be different by a minus sign.

**Contracted Codazzi relation**

In the Codazzi-Mainardi relation (eq. B.23), we contract the indices $\alpha$ and $\gamma$ to get:

$$
\gamma_\rho^\mu n^\sigma \gamma_\beta^\nu {}^{(4)}R_{\sigma\mu\nu}^\rho = D_\beta K - D_\mu K_\beta^\mu.
\tag{B.24}
$$

Then using $\gamma_\rho^\mu = \delta_\rho^\mu + n^\mu n_\rho$, we simplify the LHS as follows:

$$
\gamma_\rho^\mu n^\sigma \gamma_\beta^\nu {}^{(4)}R_{\sigma\mu\nu}^\rho = \left( \delta_\rho^\mu + n^\mu n_\rho \right) n^\sigma \gamma_\beta^\nu {}^{(4)}R_{\sigma\mu\nu}^\rho = n^\sigma \gamma_\beta^\nu {}^{(4)}R_{\sigma\nu} + \underbrace{\gamma_\beta^\nu {}^{(4)}R_{\sigma\mu\nu}^\rho n_\rho n^\sigma n^\mu}_{=0},
\tag{B.25}
$$

where the last term is zero because symmetric-antisymmetric indices $\{\rho, \sigma\}$ are contracted.

Thus we get the *contracted Codazzi relation*:

$$
\boxed{\gamma_\alpha^\mu n^\nu {}^{(4)}R_{\mu\nu} = D_\alpha K - D_\mu K_\alpha^\mu.}
\tag{B.26}
$$

# C Proofs of some results in 3+1-formalism

In this appendix, we prove the following results:

$$
\begin{aligned}
a_\mu &= D_\mu \ln(N), & \mathcal{L}_m \gamma_\nu^\mu &= 0, \\
\nabla_\mu n_\nu &= -K_{\mu\nu} - n_\mu D_\nu \ln(N), & \nabla_\mu m^\nu &= -NK_\mu^\nu - n_\mu D^\nu N + n^\nu \nabla_\mu N, \\
\mathcal{L}_m \gamma^{\mu\nu} &= +2NK^{\mu\nu}, & \mathcal{L}_m \gamma_{\mu\nu} &= -2NK_{\mu\nu}, \\
\mathcal{L}_m K_{\mu\nu} &= N\gamma_\mu^\alpha \gamma_\nu^\beta \nabla_n K_{\alpha\beta} - 2NK_{\mu\rho}K_\nu^\rho, & K_{\mu\nu} &= \frac{1}{2N} \left[ D_\mu N_\nu + D_\nu N_\mu - \dot{\gamma}_{\mu\nu} \right].
\end{aligned}
\tag{C.1}
$$

We start from the definition of the acceleration of a foliation eq. (47) and use $n_\mu = \Omega \nabla_\mu t$ where $t \in \mathbb{R}$ is a scalar field and $\Omega = -\frac{1}{\sqrt{-g^{00}}} = -N$ (see the paragraph above eq. (41)) to get:

$$a_\mu = n^\sigma \nabla_\sigma n_\mu = -n^\sigma \nabla_\sigma \left( N \nabla_\mu t \right),$$
$$\Rightarrow a_\mu = -n^\sigma \left( \nabla_\sigma N \right) \left( \nabla_\mu t \right) - n^\sigma N \nabla_\sigma \nabla_\mu t. \tag{C.2}$$

But we realize that $\nabla$ is torsion free and therefore when applied to any scalar field, such as $t$ here, we always have $\left[ \nabla_\sigma, \nabla_\mu \right] t = 0$. We also use $\nabla_\mu t = -\frac{n_\mu}{N}$ and $n^\sigma n_\sigma = -1$ to get:

$$a_\mu = \frac{1}{N} n_\mu n^\sigma \nabla_\sigma N + n^\sigma N \nabla_\mu \left( \frac{n_\sigma}{N} \right)$$
$$= n_\mu n^\sigma \nabla_\sigma \ln(N) + \underbrace{n^\sigma \nabla_\mu n_\sigma}_{\frac{1}{2} \nabla_\mu (n^\sigma n_\sigma) = 0} + n^\sigma N n_\sigma \left( \frac{-1}{N^2} \right) \nabla_\mu N$$
$$= n_\mu n^\sigma \nabla_\sigma \ln(N) - n_\sigma n^\sigma \nabla_\mu \ln N + n^\sigma \nabla_\mu n_\sigma \tag{C.3}$$
$$= \underbrace{\left( n^\sigma n_\mu + \delta_\mu^\sigma \right)}_{= \gamma_\mu^\sigma} \nabla_\sigma \ln(N) = \gamma_\mu^\sigma \nabla_\sigma \ln N = D_\mu \ln(N)$$

$$\Rightarrow \boxed{a_\mu = D_\mu \ln(N).}$$

After proving this, we use it in the alternative expression obtained for extrinsic curvature tensor in Chapter 2.3 (below eq. (47)), namely $K_{\mu\nu} = -\nabla_\mu n_\nu - n_\mu a_\nu$ to get the next result:

$$\boxed{\nabla_\mu n_\nu = -K_{\mu\nu} - n_\mu D_\nu \ln(N).} \tag{C.4}$$

Next we realize that from the definition of normal evolution vector $m^{mu}$ in eq. (56) that $m^\mu = N n^\mu$ and this gives the next result:

$$\nabla_\mu m^\nu = \nabla_\mu (N n^\nu) = n^\nu \nabla_\mu N + N \underbrace{\nabla_\mu n^\nu}_{\text{use the above result}} \tag{C.5}$$
$$\Rightarrow \boxed{\nabla_\mu m^\nu = -N K_\mu^\nu - n_\mu D^\nu N + n^\nu \nabla_\mu N.}$$

Next we proceed to calculate the Lie derivatives. If we have a 3−object living on $\Sigma_t$, then it is invariant under the projection onto $\Sigma_t$, namely:

$$T^{\alpha_1 \ldots \alpha_r}{}_{\beta_1 \ldots \beta_s} = \gamma_{\mu_1}^{\alpha_1} \cdots \gamma_{\mu_r}^{\alpha_r} \gamma_{\beta_1}^{\nu_1} \cdots \gamma_{\beta_s}^{\nu_s} T^{\mu_1 \ldots \mu_r}{}_{\nu_1 \ldots \nu_s}. \tag{C.6}$$

Accordingly, as seen in Chapter 2.1, the Lie derivative is a map from tensor field of rank $(r,s)$ to another tensor field of rank $(r,s)$. For any 3−object on $\Sigma_t$, the Lie derivative then acts as an endomorphism of the space of tangent vectors living on $\Sigma_t$ and thus we get:

$$\boxed{\mathcal{L}_m \gamma_\nu^\mu = 0.} \tag{C.7}$$

With this obtained, we can generalize eq. (C.6) for the case of Lie derivatives of 3−objects as follows:

$$(\mathcal{L}_m T)^{\alpha_1 \ldots \alpha_r}{}_{\beta_1 \ldots \beta_s} = \gamma_{\mu_1}^{\alpha_1} \cdots \gamma_{\mu_r}^{\alpha_r} \gamma_{\beta_1}^{\nu_1} \cdots \gamma_{\beta_s}^{\nu_s} (\mathcal{L}_m T)^{\mu_1 \ldots \mu_r}{}_{\nu_1 \ldots \nu_s}. \tag{C.8}$$

We realize that $\nabla$ is torsion free, so we can use the special case of Lie derivatives as in eq. (11) to get:

$$\mathcal{L}_m \gamma_{\mu\nu} = m^\alpha \nabla_\alpha \gamma_{\mu\nu} + \gamma_{\alpha\nu} \nabla_\mu m^\alpha + \gamma_{\mu\alpha} \nabla_\nu m^\alpha. \tag{C.9}$$

But we have already obtained $\nabla_\mu m^\nu$ above which we plug it here to get (using $\gamma_{\mu\nu} = g_{\mu\nu} + n_\mu n_\nu$, $\nabla_\alpha g_{\mu\nu} = 0$ and $\gamma_{\mu\nu} n^\nu = 0$):

$$\Rightarrow \mathcal{L}_m \gamma_{\mu\nu} = m^\alpha \nabla_\alpha \left(g_{\mu\nu} + n_\mu n_\nu\right) + \gamma_{\alpha\nu} \left(-NK_\mu^\alpha - n_\mu D^\alpha N + n^\alpha \nabla_\mu N\right) \\ + \gamma_{\mu\alpha} \left(-NK_\nu^\alpha - n_\nu D^\alpha N + n^\alpha \nabla_\nu N\right) \tag{C.10}$$

$$\Rightarrow \mathcal{L}_m \gamma_{\mu\nu} = N n^\alpha \nabla_\alpha \left(n_\mu n_\nu\right) - n_\mu D_\nu N - n_\nu D_\mu N - 2NK_{\mu\nu} \\ = N n^\alpha \left(\nabla_\alpha n_\mu\right) n_\nu + N n^\alpha \left(\nabla_\alpha n_\nu\right) n_\mu - n_\mu D_\nu N - n_\nu D_\mu N - 2NK_{\mu\nu}. \tag{C.11}$$

But we identify $n^\alpha \nabla_\alpha n_\mu = a_\mu$ and use the result from above $a_\mu = D_\mu \ln(N)$ to further simplify this to:

$$\Rightarrow \boxed{\mathcal{L}_m \gamma_{\mu\nu} = -2NK_{\mu\nu}.} \tag{C.12}$$

As a redundant check, since $m^\mu = N n^\mu$, we get that $K_{\mu\nu} = -\frac{1}{2}\mathcal{L}_n \gamma_{\mu\nu}$ which matches with the result obtained in Chapter 2.3 (see below eq. (47)).

Then we use the idempotent identity $\gamma^{ik}\gamma_{kj} = \gamma_j^i$ and $\mathcal{L}_m \gamma_j^i = 0$ to get:

$$\mathcal{L}_m \left(\gamma^{ik}\gamma_{kj}\right) = 0, \\ \Rightarrow \left(\mathcal{L}_m \gamma^{ik}\right)\gamma_{kj} + \gamma^{ik}\left(\mathcal{L}_m \gamma_{kj}\right) = 0, \\ \Rightarrow \left(\mathcal{L}_m \gamma^{ik}\right)\gamma_{kj} = -\gamma^{ik}\left(-2NK_{kj}\right). \tag{C.13}$$

We multiply on both sides by $\gamma^{mj}$ and contract on the index $j$ where we make use of the fact that $\gamma^{mj}\gamma_{kj} = \gamma_k^m = \delta_k^m + n^m n_k$ but $n^m, n_k$ do not have spatial components ($n_k \gamma^{kj} = 0$). We get:

$$\Rightarrow \gamma^{mj}\gamma_{kj}\left(\mathcal{L}_m \gamma^{ik}\right) = 2\gamma^{mj}\gamma^{ik}NK_{kj} \\ \Rightarrow \boxed{\mathcal{L}_m \gamma^{\mu\nu} = +2NK^{\mu\nu}.} \tag{C.14}$$

Next we evaluate the the Lie derivative of extrinsic curvature tensor where we will again use the fact that $\nabla$ is torsion free and thus be able to use eq. (11) to get:

$$\mathcal{L}_m K_{\alpha\beta} = N \left(\nabla_n K_{\alpha\beta} + K_{\alpha\rho}\nabla_\beta n^\rho + K_{\rho\beta}\nabla_\alpha n^\rho\right). \tag{C.15}$$

Recognizing that $K_{\mu\nu}$ is a tangent vector to $\Sigma_t$ (i.e. a 3−object), so we can use eq. (C.8) to get:

$$\Rightarrow \mathcal{L}_m K_{\mu\nu} = \gamma_\mu^\alpha \gamma_\nu^\beta \mathcal{L}_m K_{\alpha\beta} \\ = N \gamma_\mu^\alpha \gamma_\nu^\beta \left(\nabla_n K_{\alpha\beta} + K_{\alpha\rho}\nabla_\beta n^\rho + K_{\rho\beta}\nabla_\alpha n^\rho\right). \tag{C.16}$$

Then we use the alternative expression obtained for extrinsic curvature tensor in Chapter 2.3 (below eq. (47)), namely $K_{\mu\nu} = -\nabla_\mu n_\nu - n_\mu a_\nu$, to replace $\nabla_\mu n^\nu$ and use $\gamma_\nu^\beta n_\beta = 0$ to get:

$$\Rightarrow \mathcal{L}_m K_{\mu\nu} = N \gamma_\mu^\alpha \gamma_\nu^\beta \nabla_n K_{\alpha\beta} + NK_{\mu\rho}\gamma_\nu^\beta \left(-K_\beta^\rho - n_\beta a^\rho\right) + NK_{\nu\rho}\gamma_\mu^\alpha \left(-K_\alpha^\rho - n_\alpha a^\rho\right) \\ = N \gamma_\mu^\alpha \gamma_\nu^\beta \nabla_n K_{\alpha\beta} - NK_{\mu\rho}K_\nu^\rho - NK_{\nu\rho}K_\mu^\rho. \tag{C.17}$$

But $K_{\mu\rho}K_\nu^\rho = \gamma^{\rho\alpha}K_{\mu\rho}K_{\nu\alpha} = K_\mu^\alpha K_{\nu\alpha} = K_\mu^\rho K_{\nu\rho}$. Thus we finally get:

$$\Rightarrow \boxed{\mathcal{L}_m K_{\mu\nu} = N \gamma_\mu^\alpha \gamma_\nu^\beta \nabla_n K_{\alpha\beta} - 2NK_{\mu\rho}K_\nu^\rho.} \tag{C.18}$$

Finally we prove the only remaining result. We start from $\mathcal{L}_m \gamma_{ij} = -2NK_{ij}$ and realize $m^\alpha = N n^\alpha$ where $n^\alpha = \left(\frac{1}{N}, -\frac{N^i}{N}\right)$ (as obtained in eq. (62) in Chapter 3.1) which enables us to split the derivative with respect to $m$ into $\left(\partial_t - N^k \partial_k\right)$ and we get for the LHS:

$$\mathcal{L}_m \gamma_{ij} = \partial_t \gamma_{ij} - \left(\gamma_{kj}D_i N^k + \gamma_{ik}D_j N^k + N^k D_k \gamma_{ij}\right). \tag{C.19}$$

But $D_k \gamma_{ij} = 0$ and we finally get for the LHS:

$$\text{LHS} = \dot{\gamma}_{ij} - D_i N_j - D_j N_i = \text{RHS} = -2NK_{ij}$$

$$\Rightarrow \boxed{K_{\mu\nu} = \frac{1}{2N} \left[ D_\mu N_\nu + D_\nu N_\mu - \dot{\gamma}_{\mu\nu} \right].} \tag{C.20}$$

# D   Projection of Einstein field equations in 3+1-variables

We derive all the results presented in Chapter 3.1. We know the full Einstein field equations without the cosmological constant term to be $^{(4)}R_{\mu\nu} - \frac{1}{2} g_{\mu\nu}{}^{(4)}R = 8\pi T_{\mu\nu}$. Accordingly we need to take projection of the LHS, namely the 4−curvature as well as the RHS, namely the 4−stress-energy-momentum tensor in terms of 3−variables. We will take the projection of the LHS & the RHS separately and then combine them to project the full Einstein field equations.

**Projection of 4-curvature**

We start with the definition of the 4−Riemann tensor when applied to normal vector $n^\mu$, namely:

$$^{(4)}R^\rho_{\sigma\mu\nu} n^\sigma = \left[ \nabla_\mu, \nabla_\nu \right] n^\rho. \tag{D.1}$$

Then we project twice on $\Sigma_t$ and once along $n^\mu$ to get:

$$\gamma_{\rho\alpha} \gamma^\mu_\beta n^\nu \left( {}^{(4)}R^\rho_{\sigma\mu\nu} n^\sigma \right) = \gamma_{\rho\alpha} \gamma^\mu_\beta n^\nu \left[ \nabla_\mu, \nabla_\nu \right] n^\rho. \tag{D.2}$$

Then we focus on the RHS to get:

$$\text{RHS} = \gamma_{\rho\alpha} \gamma^\mu_\beta n^\nu \left( \nabla_\mu \nabla_\nu n^\rho - \nabla_\nu \nabla_\mu n^\rho \right). \tag{D.3}$$

Then we use eq. (63) to replace $\nabla_\nu n^\rho$ and $\nabla_\mu n^\rho$ as well as $n^\mu n_\mu = -1$, $\gamma^\nu_\alpha \gamma^\mu_\beta n^\sigma \nabla_\sigma K_{\mu\nu} = \gamma^\nu_\alpha \gamma^\mu_\beta \nabla_n K_{\mu\nu}$ (since $K_{\mu\nu}$ is a 3−object) and $n^\mu \nabla_\nu n^\mu = \frac{1}{2} \nabla_\nu \left( n^\mu n_\mu \right) = 0$ to get:

$$\Rightarrow \text{RHS} = \gamma^\nu_\alpha \gamma^\mu_\beta \nabla_n K_{\mu\nu} \underbrace{- \gamma_{\rho\alpha} \gamma^\mu_\beta n^\nu \nabla_\mu K^\rho_\nu}_{\text{Term A}}$$
$$\underbrace{+ \gamma_{\rho\alpha} \gamma^\mu_\beta n^\nu \left( \nabla_\nu n_\mu \right) D^\rho \ln(N)}_{\text{Term B}} \underbrace{+ \gamma_{\rho\alpha} \gamma^\mu_\beta \left( \nabla_\mu D^\rho \ln(N) \right)}_{\text{Term C}}. \tag{D.4}$$

Then term A can be simplified as:

$$-\gamma_{\rho\alpha} \gamma^\mu_\beta n^\nu \nabla_\mu K^\rho_\nu = -\gamma_{\rho\alpha} \gamma^\mu_\beta \nabla_\mu \underbrace{\left( n^\nu K^\rho_\nu \right)}_{=0} + \gamma_{\rho\alpha} \gamma^\mu_\beta K^\rho_\nu \nabla_\mu n^\nu$$
$$= \gamma_{\rho\alpha} \gamma^\mu_\beta K^\rho_\sigma \gamma^\sigma_\nu \nabla_\mu n^\nu = -\gamma_{\rho\alpha} K^\rho_\sigma K^\sigma_\beta \tag{D.5}$$
$$= -K_{\alpha\sigma} K^\sigma_\beta,$$

where we used the definition of $K_{\mu\nu}$ from eq. (46).

Term B simplifies to (using eq. (63) to replace $\nabla_\nu n_\mu$, $n^\nu_\nu = -1$ and $K_{\mu\nu} n^\nu = 0$):

$$\gamma_{\rho\alpha} \gamma^\mu_\beta n^\nu \left( \nabla_\nu n_\mu \right) D^\rho \ln(N) = -\gamma_{\rho\alpha} \gamma^\mu_\beta n^\nu n_\nu \left( D_\mu \ln(N) \right) \left( D^\rho \ln(N) \right)$$
$$= \frac{1}{N^2} \left( D_\alpha N \right) \left( D_\beta N \right). \tag{D.6}$$

Term C becomes (using eq. (B.4) so that $\gamma^{\mu}_{\beta}\nabla_{\mu}(D^{\rho}\ln(N)) = D_{\beta}(D^{\rho}\ln(N))$ as well as $\gamma_{\rho\alpha}D^{\rho} = D_{\alpha}$) to get:

$$
\begin{aligned}
\gamma_{\rho\alpha}\gamma^{\mu}_{\beta}\left(\nabla_{\mu}D^{\rho}\ln(N)\right) &= D_{\beta}D_{\alpha}\ln(N) = D_{\alpha}D_{\beta}\ln(N) \\
&= D_{\alpha}\left(\frac{1}{N}D_{\beta}N\right) \\
&= \frac{1}{N}D_{\alpha}D_{\beta}N - \frac{1}{N^2}\left(D_{\alpha}N\right)\left(D_{\beta}N\right),
\end{aligned}
\tag{D.7}
$$

where we used in the first line the fact that for any scalar function $f$, we have $\left[\nabla_{\alpha},\nabla_{\beta}\right]f = 0$ where $f = \ln(N)$ here. The second term on the last line exactly cancels term B above. Thus combining terms A, B, C and plugging them back in eq. (D.4), we get the desired result:

$$
\boxed{\gamma_{\rho\alpha}\gamma^{\mu\,(4)}_{\beta}R^{\rho}_{\sigma\mu\nu}n^{\sigma}n^{\nu} = -K_{\alpha\lambda}K^{\lambda}_{\beta} + \gamma^{\mu}_{\alpha}\gamma^{\nu}_{\beta}\nabla_{n}K_{\mu\nu} + \frac{1}{N}D_{\alpha}D_{\beta}N\,.}
\tag{D.8}
$$

Now to obtain the result for 4−Ricci tensor, we make use of the contracted Gauss relation (eq. (B.13)) which we reproduce here for convenience:

$$
\gamma^{\mu}_{\alpha}\gamma^{\nu\,(4)}_{\beta}R_{\mu\nu} + \gamma_{\alpha\mu}n^{\nu}\gamma^{\rho}_{\beta}n^{\sigma\,(4)}R^{\mu}_{\nu\rho\sigma} = {}^{(3)}R_{\alpha\beta} + KK_{\alpha\beta} - K_{\alpha\mu}K^{\mu}_{\beta}\,.
\tag{D.9}
$$

Comparing with eq. (D.8), we see that there are two common terms and subtracting these two equations lead to the desired result for 4−Ricci tensor:

$$
\boxed{\gamma^{\alpha}_{\mu}\gamma^{\beta\,(4)}_{\nu}R_{\alpha\beta} = {}^{(3)}R_{\mu\nu} + KK_{\mu\nu} - \gamma^{\alpha}_{\mu}\gamma^{\beta}_{\ \nu}\nabla_{n}K_{\alpha\beta} - \frac{1}{N}D_{\mu}D_{\nu}N\,.}
\tag{D.10}
$$

Finally, to get the result for 4−Ricci scalar, we contract eq. (D.10) with $\gamma^{\mu\nu}$ and replace Greek indices with Latin indices in terms containing 3−objects to get:

$$
\begin{aligned}
\gamma^{\mu\nu\,(4)}R_{\mu\nu} &= {}^{(3)}R + K^2 - \underbrace{\gamma^{ij}\nabla_{n}K_{ij}}_{=\nabla_{n}K} - \frac{1}{N}\gamma^{ij}D_{i}D_{j}N \\
&= {}^{(3)}R + K^2 - \nabla_{n}K - \frac{1}{N}\gamma^{ij}D_{i}D_{j}N\,,
\end{aligned}
\tag{D.11}
$$

where we used the corollary deduced from eq. (64) in Chapter 3.1. Then we use $\gamma^{\mu\nu} = g^{\mu\nu} + n^{\mu}n^{\nu}$ to split the LHS and get:

$$
{}^{(4)}R + {}^{(4)}R_{\mu\nu}n^{\mu}n^{\nu} = {}^{(3)}R + K^2 - \nabla_{n}K - \frac{1}{N}\gamma^{ij}D_{i}D_{j}N\,.
\tag{D.12}
$$

Now we compare this equation with the scalar Gauss relation (eq. (B.16) derived in Appendix B) which we reproduce here for convenience:

$$
{}^{(4)}R + {}^{(4)}R_{\mu\nu}n^{\mu}n^{\nu} = {}^{(3)}R + K^2 - K_{ij}K^{ij}\,.
\tag{D.13}
$$

We use this equation to replace ${}^{(4)}R_{\mu\nu}n^{\mu}n^{\nu}$ in eq. (D.12) to finally get the desired result for 4−Ricci scalar:

$$
\boxed{{}^{(4)}R = {}^{(3)}R + K^2 + K^{ij}K_{ij} - 2\nabla_{n}K - \frac{2}{N}D^{i}D_{i}N\,.}
\tag{D.14}
$$

**Projection of 4-stress-energy-momentum tensor**

There are three possible types of projection possible for $T_{\mu\nu}$, all of them defined in eq. (69). The relation between $T$ and $S$ is also derived in eq. (70). One extra relation to show is between stress scalar $S$ and the stress-energy-momentum tensor $T_{\mu\nu}$. We have (using $\gamma^\mu_\nu = \delta^\mu_\nu + n^\mu n_\nu$):

$$
\begin{aligned}
S_{\alpha\beta} &\equiv T_{\mu\nu}\gamma^\mu_\alpha\gamma^\nu_\beta \\
&= T_{\alpha\beta} + E n_\alpha n_\beta + n^\rho \left( T_{\alpha\rho} n_\beta + T_{\rho\beta} n_\alpha \right).
\end{aligned}
\tag{D.15}
$$

Then the corresponding trace of $S_{\alpha\beta}$ becomes:

$$
\begin{aligned}
S &\equiv S_{\alpha\beta}\gamma^{\alpha\beta} \\
&= \left( T_{\alpha\beta} + E n_\alpha n_\beta + n^\rho \left( T_{\alpha\rho} n_\beta + T_{\rho\beta} n_\alpha \right) \right)\gamma^{\alpha\beta} \\
\Rightarrow \boxed{S = T_{\mu\nu}\gamma^{\mu\nu}.}
\end{aligned}
\tag{D.16}
$$

Thus, taking trace of 4−stress-energy-momentum tensor $T_{\mu\nu}$ with respect to 4−metric gives 4−stress-energy-momentum scalar $T$ while taking its trace with respect to 3−metric gives 3−stress scalar $S$.

**Projection of Einstein field equations**

The Einstein field equations with $\Lambda = 0$ are given by:

$$
{}^{(4)}R_{\mu\nu} - \frac{1}{2}g_{\mu\nu}{}^{(4)}R = 8\pi T_{\mu\nu},
\tag{D.17}
$$

where $T_{\mu\nu} = T_{\nu\mu}$. We can recast this equation in terms of trace of $T_{\mu\nu}$ by contracting this equation with $g^{\mu\nu}$ to get (using $g^{\mu\nu}g_{\mu\nu} = 4$ and $T_{\mu\nu}g^{\mu\nu} = T$):

$$
{}^{(4)}R = -8\pi T.
\tag{D.18}
$$

We substitute this relation back into eq. (D.17) to get:

$$
{}^{(4)}R_{\alpha\beta} = 8\pi \left( T_{\alpha\beta} - \frac{1}{2}g_{\alpha\beta} T \right).
\tag{D.19}
$$

The RHS is basically the traceless part of the stress-energy-momentum tensor.

Now we start projecting the Einstein field equations in 3 ways possible (as discussed in Chapter 3.1).

**Total projection onto $\Sigma_t$**

We start with eq. (D.19) and project it twice (since there are two indices) onto $\Sigma_t$ by using the 3−metric. Then the LHS is given by:

$$
\text{LHS} = \gamma^\mu_\alpha\gamma^\nu_\beta {}^{(4)}R_{\mu\nu}.
\tag{D.20}
$$

But this is exactly the quantity that we evaluated in eq. (D.10). We further use eq. (63), in particular $\mathcal{L}_m K_{\mu\nu} = N\gamma^\alpha_\mu\gamma^\beta_\nu \nabla_n K_{\alpha\beta} - 2N K_{\mu\rho} K^\rho_\nu$, to get:

$$
\Rightarrow \text{LHS} = {}^{(4)}R_{\alpha\beta} - 2K_{\alpha\lambda}K^\lambda_\beta + K K_{\alpha\beta} - \frac{1}{N}\left[ \mathcal{L}_m K_{\alpha\beta} + D_\alpha D_\beta N \right].
\tag{D.21}
$$

Next we consider the RHS:

$$
\text{RHS} = \gamma^\mu_\alpha\gamma^\nu_\beta \left[ 8\pi \left( T_{\alpha\beta} - \frac{1}{2}g_{\alpha\beta} T \right) \right].
\tag{D.22}
$$

But using the definitions from eq. (69) and $T = S - E$, we get:

$$\Rightarrow \text{RHS} = 8\pi\left(S_{\alpha\beta} - \frac{1}{2}g_{\alpha\beta}(S - E)\right). \tag{D.23}$$

Thus the total projection of Einstein field equations along spacelike hypersurface $\Sigma_t$ is given by:

$$\boxed{\begin{aligned} &^{(3)}R_{\alpha\beta} - 2K_{\alpha\lambda}K_{\beta}^{\lambda} + KK_{\alpha\beta} - \frac{1}{N}\left[\mathcal{L}_m K_{\alpha\beta} + D_\alpha D_\beta N\right] \\ &= 8\pi\left[S_{\alpha\beta} - \frac{1}{2}\gamma_{\alpha\beta}(S - E)\right]. \end{aligned}} \tag{D.24}$$

Note that all tensors involved in this equation are tangent to $\Sigma_t$ as expected. Furthermore this equation can be re-arranged to provide for an *equation of evolution of $K_{ij}$* along the normal to the hypersurface $\Sigma_t$ as follows:

$$\begin{aligned} \mathcal{L}_m K_{ij} = -D_i D_j N &+ N\left[R_{ij} - 2K_{il}K_j^l + KK_{ij}\right] \\ &+ 4\pi N\left[\gamma_{ij}(S - E) - 2S_{ij}\right]. \end{aligned} \tag{D.25}$$

### Total projection along $n^\mu$

This time we start from eq. (D.17) and project it twice along $n^\mu$ (so as to obtain an equation orthogonal to $\Sigma_t$) as follows (using $n^\mu n^\nu g_{\mu\nu} = n^\mu n_\mu = -1$):

$$\begin{aligned} n^\mu n^{\nu(4)}R_{\mu\nu} - \frac{1}{2}n^\mu n^\nu g_{\mu\nu}{}^{(4)}R &= 8\pi n^\mu n^\nu T_{\mu\nu}, \\ n^\mu n^{\nu(4)}R_{\mu\nu} + \frac{1}{2}{}^{(4)}R &= 8\pi E. \end{aligned} \tag{D.26}$$

We realize that the LHS is the same as the the LHS of scalar Gauss relation (eq. (B.16)) which we use to eliminate $^{(4)}R_{\mu\nu}$ and $^{(4)}R$ to finally get:

$$\boxed{^{(3)}R - K_{ij}K^{ij} + K^2 = 16\pi E.} \tag{D.27}$$

This is also known as the *Hamiltonian constraint*.

### Mixed Projection along $\Sigma_t$ and $n^\mu$

We again start from eq. (D.17) and project once along $\Sigma_t$ using the 3−metric as well as along the normal using $n^\mu$ as follows:

$$\begin{aligned} n^\mu \gamma_\alpha^{\nu(4)}R_{\mu\nu} &= 8\pi \underbrace{n^\mu \gamma_\alpha^\nu T_{\mu\nu}}_{=-p_\alpha} \\ \Rightarrow n^\mu \gamma_\alpha^{\nu(4)}R_{\mu\nu} &= -8\pi p_\alpha, \end{aligned} \tag{D.28}$$

where we made use of the fact that $g_{\mu\nu}n^\nu\gamma_\alpha^\mu = n_\mu\gamma_\alpha^\mu = 0$.

Now we compare this with the contracted Codazzi relation (eq. (B.26)) which is reproduced here for convenience:

$$\gamma_\alpha^\mu n^{\nu(4)}R_{\mu\nu} = D_\alpha K - D_\mu K_\alpha^\mu, \tag{D.29}$$

and use this to eliminate $n^\mu \gamma_\alpha^{\nu(4)}R_{\mu\nu}$ to finally get:

$$\boxed{D_\beta K_\alpha^\beta - D_\alpha K = 8\pi p_\alpha.} \tag{D.30}$$

This is also known as the *momentum constraint* or the *diffeomorphism constraint*.

# E  Alternate derivation of the ADM action

We begin with the Einstein-Hilbert action for pure gravity without the cosmological constant:

$$S_H = \frac{1}{16\pi} \int {}^{(4)}R \sqrt{-g} d^4x = \int d^4x \mathcal{L}_H. \tag{E.1}$$

Then we make use of scalar Gauss relation (eq.(B.16)) which is reproduced here for convenience:

$$^{(4)}R = {}^{(3)}R + K^2 - K_{ij}K^{ij} - 2{}^{(4)}R_{\mu\nu}n^\mu n^\nu. \tag{E.2}$$

But we make use of the defining relation for the curvature tensor in terms of commutators and contract with $n^\mu$ and $n^\nu$:

$$^{(4)}R_{\mu\nu}n^\mu n^\nu = n^\nu \left[\nabla_\mu, \nabla_\nu\right] n^\mu. \tag{E.3}$$

Thus we have:

$$^{(4)}R = {}^{(3)}R + K^2 - K_{ij}K^{ij} - 2n^\nu \left[\nabla_\mu, \nabla_\nu\right] n^\mu. \tag{E.4}$$

But now we have the following relation:

$$n^\nu \left[\nabla_\mu, \nabla_\nu\right] n^\mu = \nabla_\alpha \left(n^\mu \nabla_\mu n^\alpha - n^\alpha \nabla_\mu n^\mu\right) - \left(\nabla_\alpha n^\mu\right) \nabla_\mu n^\alpha + \left(\nabla_\alpha n^\alpha\right)^2. \tag{E.5}$$

Proof: Ignoring any boundary term arising due to integration by parts, we have for the RHS :

$$\begin{aligned}
\text{RHS} &= \left(\nabla_\alpha n^\mu\right)\left(\nabla_\mu n^\alpha\right) + n^\mu \left(\nabla_\alpha \nabla_\mu n^\alpha\right) \\
&\quad - \left(\nabla_\alpha n^\alpha\right)\left(\nabla_\mu n^\mu\right) - n^\alpha \left(\nabla_\alpha \nabla_\mu n^\mu\right) \\
&\quad - \left(\nabla_\alpha n^\mu\right)\left(\nabla_\mu n^\alpha\right) + \left(\nabla_\alpha n^\alpha\right)^2
\end{aligned} \tag{E.6}$$

$$\begin{aligned}
\Rightarrow \text{RHS} &= n^\mu \left(\nabla_\alpha \nabla_\mu n^\alpha\right) - n^\alpha \left(\nabla_\alpha \nabla_\mu n^\mu\right) \\
\Rightarrow \text{RHS} &= n^\mu \left(\nabla_\alpha \nabla_\mu n^\alpha\right) - n^\mu \left(\nabla_\mu \nabla_\alpha n^\alpha\right) = \text{LHS},
\end{aligned} \tag{E.7}$$

where we applied integration by parts twice on the second term that led us to the next line in eq. (E.7) (ignoring boundary terms).

Now we plug eq. (E.5) in eq. (E.4), make use of $\nabla_\mu n^\mu = -K$ (below eq. (48)), rewrite $\left(\nabla_\alpha n^\mu\right)\nabla_\mu n^\alpha = \left(\nabla^\alpha n^\mu\right)\nabla_\mu n_\alpha$, use the definition of $K_{\mu\nu}$ in terms of acceleration $a_\mu$ (see below eq. (46)) and realize $K_{\mu\nu}n^\nu = a_\mu n^\mu = 0$ to get:

$$\Rightarrow {}^{(4)}R = {}^{(3)}R + K^2 - K_{ij}K^{ij} - 2\left[\nabla_\alpha \left(n^\mu \nabla_\mu n^\alpha - n^\alpha \nabla_\mu n^\mu\right) - K^{\alpha\mu}K_{\alpha\mu} + K^2\right]. \tag{E.8}$$

If we ignore the total divergence, then we get:

$$\Rightarrow {}^{(4)}R = {}^{(3)}R - K^2 + K_{ij}K^{ij}. \tag{E.9}$$

We plug this back into the action to finally get the Einstein-Hilbert action for pure gravity in $3+1$−variables (ignoring the pre-factor $\frac{1}{16\pi}$ without loss of generality):

$$\boxed{S_H = \int_{t_1}^{t_2} dt \int_{\Sigma_t} d^3x N \sqrt{\gamma} \left({}^{(3)}R - K^2 + K^{ij}K_{ij}\right).} \tag{E.10}$$

This exactly matches with eq. (81) and thus concludes our alternate derivation of this result.

# F    Variation of the ADM action

We start with the ADM action, either in the form provided in eq. (81) or eq. (101), and extremize it with respect to $\{N, N_j, \pi^{ij}, \gamma_{ij}\}$ which we equate to zero to get the corresponding equations of motion. As we show below, the equations of motion corresponding to $N$ and $N_j$ are the constraint equations which simply imply that $N$ and $\vec{N}$ serve the role of Lagrange multipliers (& thus are not dynamical variables) while the ones corresponding $\pi^{ij}$ and $\gamma_{ij}$ are the actual evolution equations governing time evolution of tensor fields (namely $\gamma_{ij}$ and $\pi^{ij}$ on spacelike hypersurfaces $\Sigma_t$).

In order to calculate the equations of motion, we need to impose the *boundary conditions* (where $\partial \Sigma_t$ denotes the boundary of the hypersurface $\Sigma_t$):

$$\delta N|_{\partial \Sigma_t} = \delta N^i|_{\partial \Sigma_t} = \delta \gamma_{ij}|_{\partial \Sigma_t} = 0. \tag{F.1}$$

However there are no restrictions on the conjugate momenta $\pi^{ij}$ which are treated as independent variables. The set $\{\gamma_{ij}, \pi^{ij}, N, \vec{N}\}$ are also taken as an independent set of variables.

We now vary the ADM action with respect to $\{N, \vec{N}, \pi^{ij}, \gamma_{ij}\}$.

**Variation with respect to Lapse function $N$**

We start with the ADM action provided in eq. (81) which is reproduced here for convenience:

$$S_{\text{ADM}} = \int_{t_1}^{t_2} dt \int_{\Sigma_t} d^3x \, N \sqrt{\gamma} \left( {}^{(3)}R - K^2 + K^{ij} K_{ij} \right). \tag{F.2}$$

For the sake of convenience, let's define $S \equiv N \sqrt{\gamma} \left( {}^{(3)}R - K^2 + K^{ij} K_{ij} \right)$. We realize that ${}^{(3)}R$ does not depend on $N$ and $\gamma$ is taken independent of $N$. Thus we have:

$$\frac{\delta S}{\delta N} = \sqrt{\gamma} \left( {}^{(3)}R - K^2 + K^{ij} K_{ij} \right) + N \sqrt{\gamma} \left( -2K \frac{\partial K}{\partial N} + 2K^{ij} \frac{\partial K_{ij}}{\partial N} \right). \tag{F.3}$$

We make use of the relation in eq. (63), namely $K_{ij} = \frac{1}{2N} \left[ D_i N_j + D_i N_j - \dot{\gamma}_{ij} \right]$ and realize that $N$ is independent from its spatial derivatives, just like we have in classical field theory where we take $\phi$ and $\phi_{,i}$ as independent, to get:

$$\frac{\partial K_{ij}}{\partial N} = -\frac{1}{2N^2} \left[ D_i N_j + D_i N_j - \dot{\gamma}_{ij} \right] = -\frac{1}{N} K_{ij},$$

$$\frac{\partial K}{\partial N} = \frac{\partial}{\partial N} \left( \gamma^{ij} K_{ij} \right) = \gamma^{ij} \frac{\partial K_{ij}}{\partial N} = -\frac{1}{N} K. \tag{F.4}$$

Thus we have:

$$\Rightarrow \frac{\delta S}{\delta N} = \sqrt{\gamma} \left( {}^{(3)}R - K^2 + K^{ij} K_{ij} \right) + N \sqrt{\gamma} \left( \frac{2}{N} K^2 - \frac{2}{N} K^{ij} K_{ij} \right)$$

$$= \sqrt{\gamma} \left( {}^{(3)}R + K^2 - K^{ij} K_{ij} \right) \overset{!}{=} 0. \tag{F.5}$$

We compare with eq. (72) to realize that the Hamiltonian constraint in the vacuum case vanishes:

$$E = 0. \tag{F.6}$$

Now we will show that $E = 0 \Leftrightarrow \mathbb{H} = 0$. For this, we need to use eqs. (88, 89) to get for $\left( {}^{(3)}R + K^2 - K^{ij}K_{ij} \right)$:

$$
\begin{aligned}
{}^{(3)}R + K^2 - K^{ij}K_{ij} &= {}^{(3)}R + \frac{\pi^2}{4\gamma} - \frac{1}{4\gamma}\left(\pi\gamma^{ij} - 2\pi^{ij}\right)\left(\pi\gamma_{ij} - 2\pi_{ij}\right) \\
&= {}^{(3)}R + \frac{\pi^2}{4\gamma} - \frac{1}{4\gamma}\left(3\pi^2 - 2\pi^2 - 2\pi^2 + 4\pi^{ij}\pi_{ij}\right) \\
&= {}^{(3)}R + \frac{1}{\gamma}\left(\frac{\pi^2}{2} - \pi^{ij}\pi_{ij}\right),
\end{aligned} \tag{F.7}
$$

which is basically the integrand of the Hamiltonian constraint in eq. (97). Thus we have established *a constraint relation*:

$$
\boxed{\frac{\delta S_{ADM}}{\delta N} \overset{!}{=} 0 \quad \Leftrightarrow \quad E = 0 \quad \Leftrightarrow \quad \mathbb{H} = 0.} \tag{F.8}
$$

**Variation with respect to Shift functions $N_j$**

In order to find the variation with respect to $N_j$, we parametrize the action using some arbitrary parameter $\lambda$ and evaluate:

$$
\left.\frac{dS}{d\lambda}\right|_{\lambda=0} = \left.\frac{d}{d\lambda}\right|_{\lambda=0}\left[{}^{(3)}R - K^2 + K^{ij}K_{ij}\right]N\sqrt{\gamma}. \tag{F.9}
$$

We realize that ${}^{(3)}R$ is independent of $\lambda$ and only depends on the 3−metric. Thus we get:

$$
\Rightarrow \left.\frac{dS}{d\lambda}\right|_{\lambda=0} = 2N\sqrt{\gamma}\left(-K\gamma^{ij} + K^{ij}\right)\left.\frac{dK_{ij}}{d\lambda}\right|_{\lambda=0}. \tag{F.10}
$$

But from eq. (85) we identify $\sqrt{\gamma}\left(K\gamma^{ij} - K^{ij}\right) = \pi^{ij}$ to get:

$$
\Rightarrow \left.\frac{dS}{d\lambda}\right|_{\lambda=0} = -2N\pi^{ij}\left.\frac{dK_{ij}}{d\lambda}\right|_{\lambda=0}. \tag{F.11}
$$

Next we use the relation in eq. (63), namely $K_{ij} = \frac{1}{2N}\left[D_iN_j + D_iN_j - \dot{\gamma}_{ij}\right]$ and use the symmetric properties of $K_{ij} = +K_{ji}$ to get:

$$
\left.\frac{dK_{ij}}{d\lambda}\right|_{\lambda=0} = \frac{1}{2N}2D_i\left(\left.\frac{dN_j}{d\lambda}\right|_{\lambda=0}\right). \tag{F.12}
$$

Plugging back, we have:

$$
\Rightarrow \left.\frac{dS}{d\lambda}\right|_{\lambda=0} = -2\pi^{ij}D_i\left(\left.\frac{dN_j}{d\lambda}\right|_{\lambda=0}\right). \tag{F.13}
$$

But using integration by parts, we have:

$$
\pi^{ij}D_i\left(\left.\frac{dN_j}{d\lambda}\right|_{\lambda=0}\right) = D_i\left(\pi^{ij}\left.\frac{dN_j}{d\lambda}\right|_{\lambda=0}\right) - D_i\left(\pi^{ij}\right)\left.\frac{dN_j}{d\lambda}\right|_{\lambda=0}, \tag{F.14}
$$

where the first term on the RHS is a pure divergence and we ignore it. Thus we are left with:

$$
\Rightarrow \left.\frac{dS}{d\lambda}\right|_{\lambda=0} = +2D_i\left(\pi^{ij}\right)\delta N_j, \tag{F.15}
$$

where we defined $\delta N_j \equiv \frac{dN_j}{d\lambda}\Big|_{\lambda=0}$. We have imposed $\delta N_j|_{\partial \Sigma_t} = 0$.

Thus we get:

$$\frac{\delta S}{\delta N_j} = 2D_i \pi^{ij} \overset{!}{=} 0. \tag{F.16}$$

Comparing with eq. (98), we get:

$$\mathbb{D}_j = 0. \tag{F.17}$$

We now show that $\mathbb{D}_j = 0 \Leftrightarrow p_\alpha = 0$ where $p_\alpha$ is defined in eq. (69). Consider the projection of Einstein field equation in eq. (73) reproduced here for convenience:

$$D_i K_j^i - D_j K = 8\pi p_j. \tag{F.18}$$

Then we make use of eq. (85) to get (recall $D_i \gamma_{ab} = 0$)

$$D_i \pi^{ij} = D_i \left( \sqrt{\gamma} \left( K \gamma^{ij} - K^{ij} \right) \right) = \sqrt{\gamma} \left( D^j K - D_i K^{ij} \right). \tag{F.19}$$

Thus $D_i \pi^{ij} = 0$ implies the LHS of eq. (F.18) being zero, which in turn implies $p_j = 0$. Hence we have established the *three constraint relations*:

$$\boxed{\frac{\delta S_{ADM}}{\delta N_j} \overset{!}{=} 0 \quad \Leftrightarrow \quad p_j = 0 \quad \Leftrightarrow \quad \mathbb{D}_j = 0.} \tag{F.20}$$

**Variation with respect to conjugate momenta $\pi^{ij}$**

We start with the ADM action eq. (81) and use eq. (94) to get:

$$\begin{aligned}
S_{\text{ADM}} &= \int_{t_1}^{t_2} dt \int_{\Sigma_t} d^3x \left( \pi^{ij} \dot{\gamma}_{ij} - \mathcal{H}_{\text{ADM}} \right) \\
&= \int_{t_1}^{t_2} dt \int_{\Sigma_t} d^3x \left[ \pi^{ij} \dot{\gamma}_{ij} - \left( 2\pi^{ij} D_i N_j - N \sqrt{\gamma}^{(3)}R + \frac{N}{\sqrt{\gamma}} \left( \pi_{ij} \pi^{ij} - \frac{\pi^2}{2} \right) \right) \right].
\end{aligned} \tag{F.21}$$

Here $\mathbb{S} \equiv \int_{\Sigma_t} d^3x \left( \pi^{ij} \dot{\gamma}_{ij} \right)$ is known as the *symplectic potential*.

We recall that $\{\gamma_{ij}, N, N_j, \pi^{ij}\}$ are all independent from one another. Also $^{(3)}R$ just depends on the $3-$metric. Moreover:

$$\begin{aligned}
\frac{\partial \pi^2}{\partial \pi^{ij}} &= 2\pi \frac{\partial \pi}{\partial \pi^{ij}} \\
&= 2\pi \frac{\partial \pi^{ab} \gamma_{ab}}{\partial \pi^{ij}} \\
&= 2\pi \gamma_{ab} \frac{\partial \pi^{ab}}{\partial \pi^{ij}} \\
&= 2\pi \gamma_{ab} \delta_i^a \delta_j^b \\
&= 2\pi \gamma_{ij}.
\end{aligned} \tag{F.22}$$

Again if we define for our convenience

$$\mathcal{S} \equiv \pi^{ij} \dot{\gamma}_{ij} - \left( 2\pi^{ij} D_i N_j - N \sqrt{\gamma}^{(3)}R + \frac{N}{\sqrt{\gamma}} \left( \pi_{ij} \pi^{ij} - \frac{\pi^2}{2} \right) \right),$$

then we have:

$$\frac{\delta \mathcal{S}}{\delta \pi^{ij}} = \delta \pi^{ij} \left[ \dot{\gamma}_{ij} - 2D_i N_j - \frac{N}{\sqrt{\gamma}} \left( 2\pi_{ij} - \pi \gamma_{ij} \right) \right] \overset{!}{=} 0. \tag{F.23}$$

This finally gives us the *equation of motion* for the 3−metric:

$$\dot{\gamma}_{ij} = \frac{\delta H}{\delta \pi^{ij}} = D_i N_j + D_j N_i - \frac{N}{\sqrt{\gamma}}\left(2\pi_{ij} - \pi\gamma_{ij}\right).$$

(F.24)

As a redundant check, from eq. (88), we identify $\left(2\pi_{ij} - \pi\gamma_{ij}\right) = -\frac{2}{\sqrt{\gamma}}K_{ij}$ and we get back the result from eq. (63), namely $K_{ij} = \frac{1}{2N}\left[D_i N_j + D_i N_j - \dot{\gamma}_{ij}\right]$.

**Variation with respect to 3-metric $\gamma_{ij}$**

We start with:

$$S_{\text{ADM}} = \int_{t_1}^{t_2} dt \int_{\Sigma_t} d^3x \left(\pi^{ij}\dot{\gamma}_{ij} - \mathcal{H}_H\right).$$

(F.25)

Then using integration by parts on the first term on the RHS and ignoring boundary terms, we get:

$$\frac{\delta S_{\text{ADM}}}{\delta \gamma_{ij}} = -\int_{t_1}^{t_2} dt \left[\int_{\Sigma_t} \delta\gamma_{ij}\left(\dot{\pi}^{ij} + \frac{\delta\mathcal{H}_H}{\delta\gamma_{ij}}\right)d^3x\right] \overset{!}{=} 0.$$

(F.26)

Here we will use eq. (94) for evaluating $\delta_{\gamma_{ij}}\mathcal{H}_H$.

We recall that $\frac{\pi^{ab}}{\gamma_{ij}} = 0$ but $\frac{\delta\pi_{ab}}{\delta\gamma_{ij}} \neq 0$ as $\pi_{ab} = \pi^{rs}\gamma_{ra}\gamma_{sb}$, thus, for example, $\frac{\delta\pi_{ab}\pi^{ab}}{\delta\gamma_{ij}} = 2\pi_a^i\pi^{aj}$.

Here we will make use of relations obtained in Chapter 2.2 but in the context of 3−D on a hypersurface $\Sigma_t$. We will use, for example, the Jacobi's formula $\delta\gamma = \gamma\gamma^{ab}\delta\gamma_{ab}$.

So now we focus on $\frac{\delta\mathcal{H}_H}{\delta\gamma_{ij}}$ in eq. (F.26), which is explicitly written as (using eq. (94)):

$$\delta_{\gamma_{ij}}\mathcal{H}_H = \delta_{\gamma_{ij}}\left[\underbrace{2\pi^{ij}D_i N_j}_{\text{Term A}}\underbrace{-N\sqrt{\gamma}^{(3)}R}_{\text{Term B}} + \underbrace{\frac{N}{\sqrt{\gamma}}\left(\pi_{ij}\pi^{ij} - \frac{\pi^2}{2}\right)}_{\text{Term C}}\right].$$

(F.27)

Term C is the simplest to evaluate where we make use of the aforementioned Jacobi's formula to get:

$$\delta_{\gamma_{ij}}\left[\frac{N}{\sqrt{\gamma}}\left(\pi_{ij}\pi^{ij} - \frac{\pi^2}{2}\right)\right] = \delta_{\gamma_{ij}}\left[\frac{N}{\sqrt{\gamma}}\right]\left(\pi_{ij}\pi^{ij} - \frac{\pi^2}{2}\right) + \frac{N}{\sqrt{\gamma}}\delta_{\gamma_{ij}}\left[\left(\pi_{ij}\pi^{ij} - \frac{\pi^2}{2}\right)\right]$$
$$= \frac{N}{\sqrt{\gamma}}\left[-\frac{1}{2}\left(\pi_{cd}\pi^{cd} - \frac{\pi^2}{2}\right)\gamma^{ab} + 2\pi^a_{\ c}\pi^{bc} - \pi\pi^{ab}\right].$$

(F.28)

Term B is relatively straightforward as well if we call from Chapter 2.2 the variations of 4−Ricci scalar and apply the same formula for 3−D case here along with using the Jacobi's identity again (recall that $N$ and $\gamma_{ij}$ are independent variables):

$$\delta_{\gamma_{ij}}\left[-N\sqrt{\gamma}^{(3)}R\right] = -N\delta_{\gamma_{ij}}(\sqrt{\gamma})^{(3)}R - N\sqrt{\gamma}\delta_{\gamma_{ij}}\left(^{(3)}R\right)$$
$$= N\sqrt{\gamma}\left(^{(3)}R^{ab} - \frac{1}{2}\gamma^{ab(3)}R\right) - N\sqrt{\gamma}\gamma^{ab}\delta^{(3)}R_{ab}.$$

(F.29)

But using the variation of 4−Ricci tensor for 3−D case from Chapter 2.2 (eq. (25)), we see that $\sqrt{\gamma}\gamma^{ab}\delta^{(3)}R_{ab} = \partial_a\left[\sqrt{\gamma}\left(\gamma^{ij}\delta^{(3)}\Gamma^a_{ij} - g^{aj}\delta^{(3)}\Gamma^j_{ij}\right)\right] = \partial_a\left[\sqrt{\gamma}\delta Z^a\right]$. But using the 3−divergence

formula similar to 4−divergence from Appendix A, we get $D_a\delta Z^a = \frac{1}{\sqrt{\gamma}}\partial_a(\sqrt{\gamma}\delta Z^a)$. So we have for term B:

$$\delta_{\gamma_{ij}}\left[-N\sqrt{\gamma}^{(3)}R\right] = N\sqrt{\gamma}\left(^{(3)}R^{ab} - \frac{1}{2}\gamma^{ab\,(3)}R\right) - N\sqrt{\gamma}D_a(\delta Z^a). \tag{F.30}$$

We apply integration by parts on the last term on the RHS and ignore the boundary terms to get (recall that $D_a\gamma = 0$):

$$\delta_{\gamma_{ij}}\left[-N\sqrt{\gamma}^{(3)}R\right] = N\sqrt{\gamma}\left(^{(3)}R^{ab} - \frac{1}{2}\gamma^{ab\,(3)}R\right) + \sqrt{\gamma}\delta Z^a D_a(N). \tag{F.31}$$

Now we simplify the expression for $\delta Z^\rho$ using the expression for the variation of Christoffel symbol from Chapter 2.2 for our 3−D case:

$$\begin{aligned}
\delta Z^\rho &= \gamma^{\mu\nu}\delta^{(3)}\Gamma^\rho_{\mu\nu} - \gamma^{\rho\nu}\delta^{(3)}\Gamma^\mu_{\mu\nu} \\
&= \left(\gamma^{\mu\nu}\delta\gamma^{\rho\lambda} - \gamma^{\rho\nu}\delta\gamma^{\mu\lambda}\right)^{(3)}\Gamma_{\lambda\mu\nu} + \left(\gamma^{\mu\nu}\gamma^{\rho\lambda} - \gamma^{\rho\nu}\gamma^{\mu\lambda}\right)\delta^{(3)}\Gamma_{\lambda\mu\nu}.
\end{aligned} \tag{F.32}$$

But using the variation for $\delta^{(3)}\Gamma_{\lambda\mu\nu}$ from Chapter 2.2:

$$\begin{aligned}
\left(\gamma^{\mu\nu}\gamma^{\rho\lambda} - \gamma^{\rho\nu}\gamma^{\mu\lambda}\right)\delta^{(3)}\Gamma_{\lambda\mu\nu} &= \frac{1}{2}\left[\left(\gamma^{\mu\nu}\gamma^{\rho\lambda} - \gamma^{\rho\nu}\gamma^{\mu\lambda}\right)\left(\delta\gamma_{\lambda\nu,\mu} + \delta\gamma_{\mu\lambda,\nu} - \delta\gamma_{\mu\nu,\lambda}\right)\right] \\
&= \gamma^{\mu\nu}\gamma^{\rho\lambda}\delta\gamma_{\lambda\mu,\nu} - \gamma^{\mu\nu}\gamma^{\rho\lambda}\delta\gamma_{\mu\nu,\lambda}.
\end{aligned}$$

Thus on plugging this back and simplifying, we have the following result after switching to 3−covariant derivatives:

$$\delta Z^a = \gamma^{\mu\nu}\gamma^{\rho\lambda}\left(\nabla_\mu\delta\gamma_{\lambda\nu} - \nabla_\lambda\delta\gamma_{\mu\nu}\right). \tag{F.33}$$

Now we use this result to simplify the second term on the RHS of term B in eq. (F.31) as follows:

$$\begin{aligned}
\delta V^a D_a N &= \gamma^{ab}\gamma^{cd}\left(D_a\delta\gamma_{bc} - D_c\delta\gamma_{ab}\right)D_d N \\
&= D_a\left[\left(\gamma^{ab}D^c N - \gamma^{bc}D^a N\right)\delta\gamma_{bc}\right] - \left(D^a D^b N - \gamma^{ab}D_c D^c N\right)\delta\gamma_{ab},
\end{aligned} \tag{F.34}$$

where we again integrate by parts and use the boundary condition $\delta\gamma_{ab}|_{\partial\Sigma_t} = 0$ to finally get for term B:

$$\delta_{\gamma_{ij}}\left[-N\sqrt{\gamma}^{(3)}R\right] = N\sqrt{\gamma}\left(^{(3)}R^{ab} - \frac{1}{2}\gamma^{ab\,(3)}R\right) - \sqrt{\gamma}\left(D^a D^b N - \gamma^{ab}D_c D^c N\right). \tag{F.35}$$

Finally for term A, we need to expand $D_i N_j$ by using the formula for 4−covariant derivative in Appendix A for 3−D case to get:

$$D_i N_j = \frac{\partial N_j}{\partial x^i} - {}^{(3)}\Gamma^a_{ij}N_a, \tag{F.36}$$

and then use the variation of 3−Christoffel symbols with respect to the 3−metric from Chapter 2.2. After ignoring the boundary terms, we get for term A:

$$\delta_{\gamma_{ij}}\left[2\pi^{ij}D_i N_j\right] = \left(2\pi^{ab}D_a N^c - \pi^{bc}D_a N^a\right). \tag{F.37}$$

Now we make use of the constraint relation eq. (F.20) which implies $D_i\pi^{ij} = 0$ as well as the symmetrization on indices $i$ and $j$ (because we are calculating $\delta_{\gamma_{ij}}\mathcal{H}_H$ which is symmetric in its indices) to finally get for term A:

$$\delta_{\gamma_{ij}}\left[2\pi^{ij}D_i N_j\right] = D_c\left(\pi^{ac}N^b + \pi^{bc}N^a - \pi^{ab}N^c\right). \tag{F.38}$$

Thus we plug eqs. (F.28, F.35, F.38) in eq. (F.27) and then plug this back into eq. (F.26) leads to the following *equation of motion* for the 3−conjugate momenta:

$$\dot{\pi}^{ij} = -\frac{\delta \mathcal{H}_H}{\delta \gamma_{ij}} = -N\sqrt{\gamma}\left(R^{ij} - \frac{1}{2}\gamma^{ij}R\right) + \frac{N}{2\sqrt{\gamma}}\left(\pi_{cd}\pi^{cd} - \frac{\pi^2}{2}\right)\gamma^{ij}$$
$$-\frac{2N}{\sqrt{\gamma}}\left(\pi^{ic}\pi^j_c - \frac{1}{2}\pi\pi^{ij}\right) + \sqrt{\gamma}\left(D^iD^jN - \gamma^{ij}D_cD^cN\right)$$
$$+ D_c\left(\pi^{ij}N^c\right) - \pi^{ic}D_cN^j - \pi^{jc}D_cN^i. \tag{F.39}$$

# G  Imposing homogeneous ansatz on electromagnetic Hamiltonian

We have the electromagnetic Hamiltonian derived in eq. (326) where the constraint relations are given in eqs. (325, 327). The homogeneous ansatz that we need to impose is provided in eq. (334).

## Hamiltonian constraint

Let's start with the Hamiltonian constraint which we reproduce here from eq. (325) for convenience:

$$\mathbb{H}_{\text{EM}}[N] = N\mathbb{H}_{\text{EM}} = \int_{\Sigma_t} d^3x \left[N\left(\underbrace{\frac{1}{2\sqrt{\gamma}}\Pi^i\Pi_i}_{\text{Term A}} + \underbrace{\frac{1}{4}\sqrt{\gamma}F_{ij}F^{ij}}_{\text{Term B}}\right)\right]. \tag{G.1}$$

Term A becomes after using eq. (334) and orthogonality relations of triads (eq. (142)):

$$\text{Term A} = \frac{1}{2\sqrt{\gamma}}\Pi^i\Pi_i = \frac{1}{2\sqrt{h}\sin\theta}\Pi^\alpha e^i_\alpha \sin\theta\, \Pi_\beta e^\beta_i \sin\theta$$
$$= \frac{1}{2\sqrt{h}}\sin\theta\, \Pi^\alpha \Pi_\beta \delta^\beta_\alpha \tag{G.2}$$
$$= \frac{1}{2\sqrt{h}}\sin\theta\, \Pi^\alpha \Pi_\alpha.$$

Term B simplifies to:

$$\text{Term B} = \frac{1}{4}\sqrt{\gamma}F_{ij}F^{ij} = \frac{1}{2}\sqrt{h}\sin\theta\left[\left(D^iA^j\right)\left(D_iA_j\right) - \left(D^iA^j\right)\left(D_jA_i\right)\right]$$
$$= \frac{1}{2}\sqrt{h}\sin\theta\left[\underbrace{\gamma^{im}\gamma^{jn}\left(D_mA_n\right)\left(D_iA_j\right)}_{\text{Term B.1}} - \underbrace{\gamma^{im}\gamma^{jn}\left(D_mA_n\right)\left(D_jA_i\right)}_{\text{Term B.2}}\right], \tag{G.3}$$

where we can simplify term B.1 by using eqs. (146, 147) as well as relations in invariant basis such as $D_i = e^\delta_i D_\delta$:

$$\text{Term B.1} = h^{\mu\alpha}e^i_\mu e^m_\alpha h^{\nu\sigma}e^j_\nu e^n_\sigma \left(e^\delta_m D_\delta\left\{A_\beta(t)e^\beta_n\right\}\right)\left(e^\gamma_i D_\gamma\left\{A_\tau(t)e^\tau_j\right\}\right)$$
$$= h^{\mu\alpha}\delta^\gamma_\mu \delta^\delta_\alpha h^{\nu\sigma}A_\beta(t)A_\tau(t)e^j_\nu e^n_\sigma \underbrace{\left(D_\delta e^\beta_n\right)}_{=^{(3)}\Gamma^\beta_{\alpha\delta}e^\delta_n}\underbrace{\left(D_\gamma e^\tau_j\right)}_{=^{(3)}\Gamma^\tau_{\mu\gamma}e^\gamma_j} \tag{G.4}$$
$$= h^{\mu\alpha}h^{\nu\sigma}A_\beta A_\tau e^j_\nu e^n_\sigma {}^{(3)}\Gamma^\beta_{\alpha\delta}e^\delta_n {}^{(3)}\Gamma^\tau_{\mu\gamma}e^\gamma_j$$
$$= h^{\mu\alpha}h^{\nu\sigma}A_\beta A_\tau {}^{(3)}\Gamma^\beta_{\alpha\sigma}{}^{(3)}\Gamma^\tau_{\mu\nu},$$

where we took out $A_\beta(t)$ and $A_\tau(t)$ out of spatial derivatives because they only depend on time in the invariant basis. Also we suppressed the explicit notation $A_\beta(t)$ to $A_\beta$ where the time dependence is understood.

We observe from eq. (G.3) that term B.2 is the same as term B.1 with indices $i$ and $j$ swapped. Thus we have similarly for term B.2:

$$
\begin{aligned}
\text{Term B.2} &= h^{\mu\alpha}e^i_\mu e^m_\alpha h^{\nu\sigma}e^j_\nu e^n_\sigma \left(e^\delta_m D_\delta\left\{A_\beta(t)e^\beta_n\right\}\right)\left(e^\gamma_j D_\gamma\left\{A_\tau(t)e^\tau_i\right\}\right) \\
&= h^{\mu\alpha}\delta^\gamma_\nu\delta^\delta_\alpha h^{\nu\sigma}A_\beta(t)A_\tau(t)e^i_\mu e^n_\sigma \underbrace{\left(D_\delta e^\beta_n\right)}_{=^{(3)}\Gamma^\beta_{\alpha\delta}e^\delta_n}\underbrace{\left(D_\gamma e^\tau_i\right)}_{=^{(3)}\Gamma^\tau_{\nu\gamma}e^\gamma_i} \\
&= h^{\mu\alpha}h^{\nu\sigma}A_\beta A_\tau e^i_\mu e^n_\sigma{}^{(3)}\Gamma^\beta_{\alpha\delta}e^{\delta(3)}\Gamma^\tau_{\nu\gamma}e^\gamma_i \\
&= h^{\mu\alpha}h^{\nu\sigma}A_\beta A_\tau{}^{(3)}\Gamma^\beta_{\alpha\sigma}{}^{(3)}\Gamma^\tau_{\nu\mu},
\end{aligned}
\tag{G.5}
$$

We plug eqs. (G.4, G.5) into eq. (G.3) to get for term B:

$$
\Rightarrow \text{Term B} = \frac{1}{2}\sqrt{h}\sin\theta\left[h^{\mu\alpha}h^{\nu\sigma}A_\beta A_\tau{}^{(3)}\Gamma^\beta_{\alpha\sigma}\left({}^{(3)}\Gamma^\tau_{\mu\nu} - {}^{(3)}\Gamma^\tau_{\nu\mu}\right)\right].
\tag{G.6}
$$

We already noted in eq. (148) that in invariant basis for Bianchi IX universe, the connection is not symmetric and for a torsion-free case which we have been considering throughout, we have:

$$
{}^{(3)}\Gamma^\tau_{\mu\nu} - {}^{(3)}\Gamma^\tau_{\nu\mu} = D^\tau_{\mu\nu} = C^\tau_{\mu\nu},
\tag{G.7}
$$

where $C^\tau_{\mu\nu} = \epsilon^\tau_{\mu\nu}$ for Bianchi IX universe (eq. (135)). Also from eq. (135), we have $C^\tau_{\mu\nu} = \epsilon_{\mu\nu\sigma}C^{\sigma\tau}$ where $C^{\sigma\tau} = \text{diag}(1,1,1) = \delta^{\sigma\tau}$. *Therefore in the invariant basis, the Levi-Civita tensor ($\epsilon_{\mu\nu\sigma}$) which acts as a structure constant ($C^\tau_{\mu\nu}$) for Bianchi IX universe can be raised/lowered using a Kronecker delta function ($\delta^{\sigma\tau}$) and not the 3$-$metric $h_{\alpha\beta}$:*

$$
\delta^{\sigma\tau}\epsilon_{\mu\nu\sigma} = \epsilon^\tau_{\mu\nu}, \quad \delta_{\tau\sigma}\epsilon^\tau_{\mu\nu} = \epsilon_{\sigma\mu\nu} \qquad \text{(Bianchi IX).}
\tag{G.8}
$$

Then using eq. (G.7) in eq. (G.6) to get:

$$
\begin{aligned}
\Rightarrow \text{Term B} &= \frac{1}{2}\sqrt{h}\sin\theta\left[h^{\mu\alpha}h^{\nu\sigma}A_\beta A_\tau{}^{(3)}\Gamma^\beta_{\alpha\sigma}\epsilon^\tau_{\mu\nu}\right] \\
&= \frac{1}{2}\sqrt{h}\sin\theta\left[h^{\mu\alpha}h^{\nu\sigma}A_\beta A_\tau{}^{(3)}\Gamma^\beta_{\alpha\sigma}\epsilon_{\gamma\mu\nu}\delta^{\gamma\tau}\right].
\end{aligned}
\tag{G.9}
$$

Next we can express the only affine connection remaining in term B as:

$$
{}^{(3)}\Gamma^\beta_{\alpha\sigma} = \frac{1}{2}\left[\Gamma^\beta_{\alpha\sigma} + \Gamma^\beta_{\sigma\alpha}\right] + \frac{1}{2}\left[{}^{(3)}\Gamma^\beta_{\alpha\sigma} - {}^{(3)}\Gamma^\beta_{\sigma\alpha}\right],
\tag{G.10}
$$

where the RHS is "symmetric+anti-symmetric" parts respectively of the LHS. We also note that $h^{\mu\alpha}h^{\nu\sigma(3)}\Gamma^\beta_{\alpha\sigma} = {}^{(3)}\Gamma^{\beta\mu\nu}$ which is then contracted with the (completely anti-symmetric) Levi-Civita tensor in eq. (G.9). Thus the symmetric part on the RHS of eq. (G.10) vanishes and we are left with the anti-symmetric part. Thus we replace ${}^{(3)}\Gamma^\beta_{\alpha\sigma}$ by its anti-symmetric part in eq. (G.10) into eq. (G.9) to get:

$$
\Rightarrow \text{Term B} = \frac{1}{2}\sqrt{h}\sin\theta\left[h^{\mu\alpha}h^{\nu\sigma}A_\beta A_\tau\left(\frac{1}{2}\left({}^{(3)}\Gamma^\beta_{\alpha\sigma} - {}^{(3)}\Gamma^\beta_{\sigma\alpha}\right)\right)\epsilon_{\gamma\mu\nu}\delta^{\gamma\tau}\right].
\tag{G.11}
$$

But ${}^{(3)}\Gamma^\beta_{\alpha\sigma} - {}^{(3)}\Gamma^\beta_{\sigma\alpha} = \epsilon^\beta_{\alpha\sigma} = \epsilon_{\lambda\alpha\sigma}\delta^{\lambda\beta}$ to get:

$$
\begin{aligned}
\Rightarrow \text{Term B} &= \frac{1}{4}\sqrt{h}\sin\theta\left[h^{\mu\alpha}h^{\nu\sigma}A_\beta A_\tau\epsilon_{\lambda\alpha\sigma}\delta^{\lambda\beta}\epsilon_{\gamma\mu\nu}\delta^{\gamma\tau}\right] \\
&= \frac{1}{4}\sqrt{h}\sin\theta\left[h^{\mu\alpha}h^{\nu\sigma}A_\beta A_\tau\epsilon^\beta_{\alpha\sigma}\epsilon^\tau_{\mu\nu}\right].
\end{aligned}
\tag{G.12}
$$

Finally we plug in eqs. (G.2, G.12) into eq. (G.1) and collect the spatially dependent term as $n \equiv \left( \int_{\Sigma_t} d^3 x N \sin \theta \right)$ to get for the Hamiltonian constraint of the electromagnetic field:

$$\Rightarrow \boxed{\mathbb{H}_{\text{EM}}[N] = N \mathbb{H}_{\text{EM}} = \frac{n}{\sqrt{h}} \left[ \frac{1}{2} \Pi^\alpha \Pi_\alpha + \frac{h}{4} h^{\mu\alpha} h^{\nu\sigma} A_\beta A_\tau \epsilon^\beta_{\alpha\sigma} \epsilon^\tau_{\mu\nu} \right]}, \tag{G.13}$$

where we keep in mind eq. (G.8) and the text above. Also we can always choose $n$ to scale as $\sqrt{h}$ so that we can put $n' = \frac{n}{\sqrt{h}} = 1$ without loss of generality.

## Diffeomorphism constraints

We reproduce the diffeomorphism constraints from eq. (325) for convenience:

$$\mathbb{D}_{\text{EM}}[N^i] = N^i \mathbb{D}_{(\text{EM})i} = \int_{\Sigma_t} d^3 x \left[ N^i \underbrace{\left( \Pi^j F_{ij} \right)}_{\text{Term C}} \right]. \tag{G.14}$$

We focus on term C where we use $F_{ij} = D_i A_j - D_j A_i$ and impose the homogeneous ansatz in eq. (334) to get:

$$\begin{aligned} \text{Term C} &= \Pi^j F_{ij} = \sin(\theta) \Pi^\alpha e_\alpha^j \left[ e_i^\kappa D_\kappa \left( A_\beta(t) e_j^\beta \right) - e_j^\kappa D_\kappa \left( A_\beta(t) e_i^\beta \right) \right] \\ &= \sin(\theta) \Pi^\alpha e_\alpha^j \left[ e_i^\kappa A_\beta(t) D_\kappa \left( e_j^\beta \right) - e_j^\kappa A_\beta(t) D_\kappa \left( e_i^\beta \right) \right], \end{aligned} \tag{G.15}$$

where we use eqs. (146, 147), namely $D_\kappa \left( e_j^\beta \right) = {}^{(3)}\Gamma^\beta_{\kappa\mu} e_j^\mu$ and $D_\kappa \left( e_i^\beta \right) = {}^{(3)}\Gamma^\beta_{\kappa\nu} e_i^\nu$ to get:

$$\begin{aligned} \Rightarrow \text{Term C} &= \Pi^j F_{ij} = \sin(\theta) \Pi^\alpha e_\alpha^j \left[ e_i^\kappa A_\beta(t) {}^{(3)}\Gamma^\beta_{\kappa\mu} e_j^\mu - e_j^\kappa A_\beta(t) {}^{(3)}\Gamma^\beta_{\kappa\nu} e_i^\nu \right] \\ &= \sin(\theta) \Pi^\alpha A_\beta \left[ \Gamma^\beta_{\delta\mu} \delta^\mu_\alpha - \Gamma^\beta_{\kappa\delta} \delta^\kappa_\alpha \right] \\ &= \sin(\theta) \Pi^\alpha A_\beta \underbrace{\left[ \Gamma^\beta_{\delta\alpha} - \Gamma^\beta_{\alpha\delta} \right]}_{=\epsilon^\beta_{\delta\alpha}} \\ &= \sin(\theta) \underbrace{\Pi^\alpha A_\beta \epsilon_{\lambda\delta\alpha} \delta^{\lambda\beta}}_{\text{depends only on time}}, \end{aligned} \tag{G.16}$$

where we used in the second line the orthogonality relations of triads (eq. (142)), eq. (G.7) & the texts below it in the third line as well as eq. (G.8) in the fourth line.

We plug eq. (G.16) into eq. (G.14) to get for the diffeomorphism constraints for the electromagnetic field:

$$\Rightarrow \boxed{\mathbb{D}_{\text{EM}}[N^i] = N^i \mathbb{D}_{(\text{EM})i} = n^\delta \Pi^\alpha A_\beta \epsilon^\beta_{\delta\alpha}}, \tag{G.17}$$

where we again keep in mind the relation eq. (G.8) and the corresponding text below it.

## Gauss constraint

We reproduce the Gauss constraint from eq. (325) for convenience:

$$\mathbb{G}[A_0] = A_0 \mathbb{G} = \int_{\Sigma_t} d^3 x \left[ -A_0 \underbrace{\left( D_i \Pi^i \right)}_{\text{Term D}} \right]. \tag{G.18}$$

Now we impose the homogeneous ansatz (eq. (334)) here to get:

$$
\begin{aligned}
\text{Term D} &= A_0 \left( D_i \Pi^i \right) = A_0 D_i \left( \Pi^\alpha(t) \sin(\theta) e^i_\alpha \right) \\
&= A_0 \Pi^\alpha(t) D_i \left( \sin(\theta) e^i_\alpha \right).
\end{aligned}
\tag{G.19}
$$

Now we use the relation corresponding to the explicit forms of triads for Bianchi IX universe provided in Chapter 4.2 at the end of the subsection "Geometry of Hypersurfaces" (see footnote 7 therein for the proof of this identity), namely $D_i \left( \sin(\theta) e^i_\alpha \right) = 0$, to realize that the Gauss constraint is identically zero for electromagnetic field in a Bianchi IX universe:

$$
\Rightarrow \boxed{ \mathbb{G}[A_0] = A_0 \mathbb{G} = 0 } \qquad \text{(identically).}
\tag{G.20}
$$

# H  Derivation of the hypersurface deformation algebra

We prove here the following constraint algebra, known as the Hypersurface Deformation Algebra (HDA), or the Dirac algebra:

$$
\begin{aligned}
&\text{(a)} \ \left\{ \mathbb{D}[N^i], \mathbb{D}[M^j] \right\} \big|_{(\gamma,\pi)} = \mathbb{D}\left[ \mathcal{L}_{N^i} M^j \right] = -\mathbb{D}\left[ \mathcal{L}_{M^j} N^i \right] = \mathbb{D}\left[ [\vec{N}, \vec{M}] \right], \\
&\text{(b)} \ \left\{ \mathbb{D}[N^j], \mathbb{H}[N] \right\} \big|_{(\gamma,\pi)} = \mathbb{H}\left[ \mathcal{L}_{N^j} N \right] = \mathbb{H}\left[ N^j \partial_j N \right], \\
&\text{(c)} \ \left\{ \mathbb{H}[N], \mathbb{H}[M] \right\} \big|_{(\gamma,\pi)} = \mathbb{D}\left[ \gamma^{jk}(N \partial_j M - M \partial_j N) \right],
\end{aligned}
\tag{H.1}
$$

where we use the following definition of the Poisson brackets:

$$
\{ f(x), h(y) \} \big|_{(\gamma,\pi)} = \int d^3 z \left[ \frac{\delta f(x)}{\delta \gamma_{ij}(z)} \frac{\delta h(y)}{\delta \pi^{ij}(z)} - \frac{\delta f(x)}{\delta \pi^{ij}(z)} \frac{\delta h(y)}{\delta \gamma_{ij}(z)} \right].
\tag{H.2}
$$

The definitions of the Hamiltonian contraint (sometimes also known as the *super-Hamiltonian*) and diffeomorphism constraints (sometimes also known as the *super-momentum*) are taken from eqs. (97, 98) reproduced here for convenience:

$$
\begin{aligned}
\mathbb{H}[N] &\equiv \int_{\Sigma_t} d^3 x N \left[ -\sqrt{\gamma}\,^{(3)}R - \frac{1}{\sqrt{\gamma}} \left( \frac{\pi^2}{2} - \pi^{ij} \pi_{ij} \right) \right], \\
\mathbb{D}[\vec{N}] &\equiv \int_{\Sigma_t} d^3 x N^i \left[ -2 D^j \pi_{ij} \right].
\end{aligned}
\tag{H.3}
$$

Just like in classical mechanics where momentum generates translations, the diffeomorphism contraints (or the super-momentum) generates spatial deformations on the spatial hypersurface $\Sigma_t$ which are tangential to the hypersurface and described by the spatial vector fields $\vec{N}$. Similarly the Hamiltonian constraint (or the super-Hamiltonian) generates normal deformations of the spatial hypersurface, moving it forward as described by the lapse function $N$. This is how $N$ and $\vec{N}$ describe the evolution of any physical object living on the spatial hypersurface. This can be quantified as follows: for an infinitesimal deformation of the spatial hypersurface by $\delta N$ and $\delta \vec{N}$, any function $F$ that depends on the phase space variables changes by an amount $\delta F$ (thus $F \to F + \delta F$) given by:

$$
\delta F = \{ F, \mathbb{H}[\delta N] \} \big|_{(\gamma,\pi)} + \{ F, \mathbb{D}[\delta \vec{N}] \} \big|_{(\gamma,\pi)}.
\tag{H.4}
$$

This is shown in Fig. (4) where it is also illustrated that the HDA (eq. (H.1)) implies a closed constraint algebra. We first prove (c) and then proceed to prove (a) & (b) together.

**Proof of** 

We use the definition of Poisson brackets in eq. (H.2) to evaluate the LHS of (c):

$$\{\mathbb{H}[N], \mathbb{H}[M]\}|_{(\gamma,\pi)} = \int d^3x \left( \frac{\delta \mathbb{H}[N]}{\delta \gamma_{ab}(x)} \frac{\delta \mathbb{H}[M]}{\delta \pi^{ab}(x)} - (N \leftrightarrow M) \right). \tag{H.5}$$

Thus we need to evaluate two variations, namely $\delta_\gamma \mathbb{H}[N]$ and $\delta_\pi \mathbb{H}[N]$ (where we can anytime swtich $N \to M$) where it is understood that $\delta_\gamma = \frac{\delta}{\delta \gamma_{ab}}$ and $\delta_\pi = \frac{\delta}{\delta \pi^{ab}}$. Also the $\gamma_{ij}$ is taken independent from the $\pi^{ab}$.

**Variation with respect to** $3-$**conjugate momenta**

Using the definition of the Hamiltonian constraint from eq. (H.3), we get:

$$\begin{aligned}
\delta_\pi \mathbb{H}[N] &= \delta_\pi \left[ \int d^3x \frac{N}{\sqrt{\gamma}} \left( \pi^{ab} \pi^{cd} \gamma_{ac} \gamma_{bd} - \frac{\pi^2}{2} \right) \right] \\
&= \int d^3x \frac{N}{\sqrt{\gamma}} \left( 2\pi^{cd} \gamma_{ac} \gamma_{bd} \delta \pi^{ab} - \pi \delta \left( \pi^{ab} \gamma_{ab} \right) \right) \\
&= \int d^3x \frac{2N}{\sqrt{\gamma}} \left( \pi_{ab} - \frac{1}{2} \pi \gamma_{ab} \right) \delta \pi^{ab}.
\end{aligned} \tag{H.6}$$

**Variation with respect to** $3-$**metric**

Next we evaluate variation with respect to $\gamma_{ab}$ where we make use of variations derived in Chapter 2.2.

$$\begin{aligned}
\delta_\gamma \mathbb{H}[N] = \int d^3x N &\left[ -(\delta_\gamma \sqrt{\gamma})^{(3)}R - \sqrt{\gamma} \delta_\gamma{}^{(3)}R + \delta_\gamma \left( \frac{1}{\sqrt{\gamma}} \right) \left( \pi^{ab} \pi_{ab} - \frac{\pi^2}{2} \right) \right. \\
&\left. + \frac{1}{\sqrt{\gamma}} \left( \pi^{ab} \delta_\gamma \pi_{ab} - \pi \delta_\gamma \pi \right) \right].
\end{aligned} \tag{H.7}$$

We make use of the following results:

(i) $\delta \sqrt{\gamma} = \frac{1}{2\sqrt{\gamma}} \delta \gamma = \frac{\gamma \gamma^{ab} \delta \gamma_{ab}}{2\sqrt{\gamma}} = \frac{\sqrt{\gamma}}{2} \gamma^{ab} \delta \gamma_{ab}$,

(ii) $\delta \left( \frac{1}{\sqrt{\gamma}} \right) = \frac{-1}{2\gamma^{3/2}} \delta \gamma = \frac{-1}{2\gamma^{3/2}} \gamma \gamma^{ab} \delta \gamma_{ab} = \frac{-1}{2\sqrt{\gamma}} \gamma^{ab} \delta \gamma_{ab}$,

(iii) $\pi_{ab} = \pi^{cd} \gamma_{ac} \gamma_{bd} \Rightarrow \delta_\gamma \pi_{ab} = \gamma_{ac} \gamma_{bd} \underbrace{\delta_\gamma \pi^{cd}}_{=0} + 2\pi^{cd} \gamma_{bd} \delta \gamma_{ac}$,

(iv) $\pi = \pi^{ab} \gamma_{ab} \Rightarrow \delta_\gamma \pi = \gamma_{ab} \underbrace{\delta_\gamma \pi^{ab}}_{=0} + \pi^{ab} \delta \gamma_{ab}$.

Then eq. (H.7) becomes:

$$\begin{aligned}
\delta_\gamma \mathbb{H}[N] = \int d^3x N &\left[ -\frac{\sqrt{\gamma}}{2} \gamma^{ab} \delta \gamma_{ab}{}^{(3)}R \underbrace{-\sqrt{\gamma} \delta_\gamma{}^{(3)}R}_{\text{Term A}} - \frac{1}{2\sqrt{\gamma}} \gamma^{ab} \delta \gamma_{ab} \left( \pi^{cd} \pi_{cd} - \frac{\pi^2}{2} \right) \right. \\
&\left. + \frac{1}{\sqrt{\gamma}} (\underbrace{2\pi^{ab} \pi^{cd} \gamma_{bd} \delta \gamma_{ac}}_{(b \leftrightarrow c)=2\pi^{ac} \pi_c^b \delta \gamma_{ab}} - \pi \pi^{ab} \delta \gamma_{ab}) \right].
\end{aligned} \tag{H.8}$$

We focus on term A:

$$\text{Term A} = -\int d^3x N \sqrt{\gamma} \delta_\gamma \left( {}^{(3)}R_{ab}\gamma^{ab}\right) = -\int d^3x N \sqrt{\gamma} \left(\gamma^{ab}\delta^{(3)}R_{ab} + {}^{(3)}R_{ab}\delta\gamma^{ab}\right). \quad \text{(H.9)}$$

We can write the second term as: ${}^{(3)}R_{ab}\delta\gamma^{ab} = -{}^{(3)}R_{ab}\gamma^{ac}\gamma^{bd}\delta\gamma_{cd}$. We have already derived for the first term appearing here in eq. (25) to get (in 3−D):

$$\gamma^{ab}\delta^{(3)}R_{ab} = -D_a\left[D^a\left(\gamma^{bc}\delta\gamma_{bc}\right) - D^b\left(\gamma^{ac}\delta\gamma_{bc}\right)\right]. \quad \text{(H.10)}$$

Thus we have for term A (recalling that $D_a\gamma_{ij} = 0$):

$$\Rightarrow \text{Term A} = \int d^3x N \sqrt{\gamma}\left[D_c\left\{D^c(\gamma^{ab}\delta\gamma_{ab}) - D^a(\gamma^{cb}\delta\gamma_{ab})\right\} + {}^{(3)}R_{cd}\gamma^{ca}\gamma^{db}\delta\gamma_{ab}\right]$$

$$= \int d^3x \sqrt{\gamma}\left[(D^cD_cN)\gamma^{ab}\delta\gamma_{ab} - (D^aD_cN)\gamma^{cb}\delta\gamma_{ab} + N\sqrt{\gamma}{}^{(3)}R_{cd}\gamma^{ca}\gamma^{db}\delta\gamma_{ab}\right]$$

$$= \int d^3x \left[\sqrt{\gamma}(D^cD_cN)\gamma^{ab}\delta\gamma_{ab} - \sqrt{\gamma}\left(D^aD^bN\right)\delta\gamma_{ab} + N\sqrt{\gamma}{}^{(3)}R^{ab}\delta\gamma_{ab}\right], \quad \text{(H.11)}$$

where we applied integration by parts when going from the first line to the second on the first two terms on the RHS and ignored the boundary terms. Therefore using eq. (H.11) in eq. (H.8), we get:

$$\Rightarrow \delta_\gamma\mathbb{H}[N] = \int d^3x 7\left[-\frac{N\sqrt{\gamma}}{2}\gamma^{ab}\delta\gamma_{ab}{}^{(3)}R + N\sqrt{\gamma}{}^{(3)}R^{ab}\delta\gamma_{ab} + \sqrt{\gamma}(D^cD_cN)\gamma^{ab}\delta\gamma_{ab}\right.$$

$$-\sqrt{\gamma}\left(D^aD^bN\right)\delta\gamma_{ab} - \frac{N}{2\sqrt{\gamma}}\gamma^{ab}\left(\pi^{cd}\pi_{cd} - \frac{\pi^2}{2}\right)\delta\gamma_{ab} \quad \text{(H.12)}$$

$$\left.+\frac{N}{\sqrt{\gamma}}\left(2\pi^{ac}\pi_c^b\delta\gamma_{ab} - \pi\pi^{ab}\delta\gamma_{ab}\right)\right].$$

Rearranging finally gives us:

$$\Rightarrow \delta_\gamma\mathbb{H}[N] = \int d^3x \left[N\sqrt{\gamma}\left({}^{(3)}R^{ab} - \frac{1}{2}\gamma^{ab(3)}R\right) - \frac{N}{2\sqrt{\gamma}}\gamma^{ab}\left(\pi^{cd}\pi_{cd} - \frac{\pi^2}{2}\right)\right.$$

$$\left.+\frac{2N}{\sqrt{\gamma}}\left(\pi^{ac}\pi_c^b - \frac{\pi\pi^{ab}}{2}\right) + \sqrt{\gamma}\left\{(D_cD^cN)\gamma^{ab} - \left(D^aD^bN\right)\right\}\right]\delta\gamma_{ab}. \quad \text{(H.13)}$$

**Putting together**

We have successfully calculated the variations of the Hamiltonian constraint with respect to the 3−metric in eq. (H.13) and its conjugate momenta in eq. (H.6). We use these two equations in eq. (H.5) and simplify. One helpful observation is that there is a symmetry between $N \leftrightarrow M$ and thus any term that does not contain derivatives of $N$ &/or $M$ will cancel out in eq. (H.5). Hence we need to only keep terms in eqs. (H.6, H.13) where derivatives of lapse function

occurs while plugging in eq. (H.5). We have:

$$
\begin{aligned}
\{\mathbb{H}[N], \mathbb{H}[M]\}|_{(\gamma,\pi)} &= \int d^3x \left( \frac{\delta \mathbb{H}[N]}{\delta \gamma_{ab}} \frac{\delta \mathbb{H}[M]}{\delta \pi^{ab}} - (N \leftrightarrow M) \right) \\
&= \int d^3x \left\{ \left[ \left( \sqrt{\gamma}(D_c D^c N)\gamma^{ab} - \sqrt{\gamma}(D^a D^b N) \right) \frac{2M}{\sqrt{\gamma}} \left( \pi_{ab} - \frac{1}{2}\pi\gamma_{ab} \right) \right] - (N \leftrightarrow M) \right\} \\
&= 2 \int d^3x \left\{ \left[ \underbrace{M\gamma^{ab}(D_c D^c N)\left( \pi_{ab} - \frac{1}{2}\pi\gamma_{ab} \right)}_{\text{Term B}} \underbrace{- M\left( \pi_{ab} - \frac{1}{2}\pi\gamma_{ab} \right)(D^a D^b N)}_{\text{Term C}} \right] - (N \leftrightarrow M) \right\}.
\end{aligned}
\tag{H.14}
$$

We apply once the integration by parts on terms B & C and ignore the boundary terms:

$$
\begin{aligned}
\text{Term B} &= M\gamma^{ab}(D_c D^c N)\left( \pi_{ab} - \frac{1}{2}\pi\gamma_{ab} \right) \\
&= M(D_c D^c N)\left( \pi - \frac{3\pi}{2} \right) = -\frac{M\pi}{2}(D^c D_c N) \\
&= +(D_c N)D^c\left( \frac{M\pi}{2} \right) \\
&= \underbrace{(D_c N)(D^c M)\frac{\pi}{2}}_{\text{symmetric in N and M, thus cancels out}} + M(D_c N)D^c\left( \frac{\pi}{2} \right),
\end{aligned}
\tag{H.15}
$$

and

$$
\begin{aligned}
\text{Term C} &= -M\left( \pi_{ab} - \frac{1}{2}\pi\gamma_{ab} \right)(D^a D^b N) \\
&= (D^b N)D^a\left[ M\left( \pi_{a_b} - \frac{1}{2}\pi\gamma_{ab} \right) \right] \\
&= \underbrace{(D^b N)(D^a M)\left( \pi_{ab} - \frac{1}{2}\pi\gamma_{ab} \right)}_{\text{symmetric in 'a' and 'b', thus in N and M}} + M(D^b N)D^a\left( \pi_{ab} - \frac{1}{2}\pi\gamma_{ab} \right).
\end{aligned}
\tag{H.16}
$$

Plugging eqs. (H.15, H.16) in eq. (H.14), we get:

$$
\begin{aligned}
\Rightarrow \{\mathbb{H}[N], \mathbb{H}[M]\}|_{(\gamma,\pi)} &= 2 \int d^3x \left\{ \left[ M(D_c N)D^c\left( \frac{\pi}{2} \right) + M(D^b N)D^a\left( \pi_{ab} - \frac{1}{2}\pi\gamma_{ab} \right) \right] \right. \\
&\qquad\qquad\qquad\qquad \left. - (N \leftrightarrow M) \right\} \\
&= 2 \int d^3x \left\{ M(D^b N)\gamma_{bd}(D_c \pi^{cd}) - (N \leftrightarrow M) \right\} \\
&= 2 \int d^3x \left[ (D^b N)M - N(D^b M) \right]\gamma_{bd}D_c \pi^{cd} \\
&= 2 \int d^3x \left[ (D_b N)M - N(D_b M) \right]\gamma^{bd}D_c \pi^c_d \\
&= 2 \int d^3x \left[ M\partial_b N - N\partial_b M \right]\gamma^{bd}D_c \pi^c_d.
\end{aligned}
\tag{H.17}
$$

Then we use the definition of the diffeomorphism constraints in eq. (H.3) to get:

$$
\Rightarrow \{\mathbb{H}[N], \mathbb{H}[M]\}|_{(\gamma,\pi)} = -\mathbb{D}\left[ \gamma^{bd}(M\partial_b N - N\partial_b M) \right].
\tag{H.18}
$$

But $-\mathbb{D}\left[\gamma^{bd}\left(M\partial_b N - N\partial_b M\right)\right] = +\mathbb{D}\left[\gamma^{bd}\left(N\partial_b M - M\partial_b N\right)\right]$, therefore we have proved $\text{c}$:

$$\Rightarrow \boxed{\{\mathbb{H}[N], \mathbb{H}[M]\}|_{(\gamma,\pi)} = \mathbb{D}\left[\gamma^{jk}\left(N\partial_j M - M\partial_j N\right)\right].} \qquad (\text{H.19})$$

**Proofs of $\text{a}$ & $\text{b}$**

We now prove $\text{a}$ & $\text{b}$ together. First we prove a general relation whose special cases directly lead us to relations $\text{a}$ & $\text{b}$.

**General result**

We define:

$$f[\mathbf{M}] \equiv \int d^3 x\, M^{a_1...a_n}{}_{b_1...b_m}\tilde{f}_{a_1...a_n}{}^{b_1...b_m}(\gamma_{ij}, \pi^{ij}), \qquad (\text{H.20})$$

where $\tilde{f}$ is a function of phase space variables and $\mathbf{M}$ is independent of them. Then by definition, we have:

$$\Rightarrow f[\mathcal{L}_{\vec{N}}\mathbf{M}] = \int d^3 x\left(\mathcal{L}_{\vec{N}} M^{a_1...a_n}{}_{b_1...b_m}\right)\tilde{f}_{a_1...a_n}{}^{b_1...b_m}(\gamma_{ij}, \pi^{ij}). \qquad (\text{H.21})$$

We calculate the Poisson bracket $\{\mathbb{D}[N^i], f[M]\}|_{(\gamma,\pi)}$ for a general $f$ and as always, keep ignoring the boundary terms whenever we apply integration by parts. We start using the definition in eq. (H.2) to get:

$$\{\mathbb{D}[N^i], f[\mathbf{M}]\}|_{(\gamma,\pi)} = \int d^3 x\left[\frac{\delta\mathbb{D}}{\delta\gamma_{ij}(x)}\frac{\delta f}{\delta\pi^{ij}(x)} - \frac{\delta\mathbb{D}}{\delta\pi^{ij}(x)}\frac{\delta f}{\delta\gamma_{ij}(x)}\right]. \qquad (\text{H.22})$$

Using the variation of the diffeomorphism constraints with respect to the 3−metric $\gamma_{ij}$ and its conjugate momenta $\pi^{ij}$ from Appendix F, we have:

$$\frac{\delta\mathbb{D}[\vec{N}]}{\delta\gamma_{ij}(x)} = -\mathcal{L}_{\vec{N}}\pi^{ij}(x), \qquad \frac{\delta\mathbb{D}[\vec{N}]}{\delta\pi^{ij}(x)} = +\mathcal{L}_{\vec{N}}\gamma_{ij}(x). \qquad (\text{H.23})$$

Therefore plugging eq. (H.23) in eq. (H.22) gives:

$$\Rightarrow \{\mathbb{D}[N^i], f[\mathbf{M}]\}|_{(\gamma,\pi)} = \int d^3 x\left[-\left(\mathcal{L}_{\vec{N}}\pi^{ij}\right)\left(\frac{\delta f(\mathbf{M})}{\delta\pi^{ij}}\right) - \left(\mathcal{L}_{\vec{N}}\right)\gamma_{ij}\left(\frac{\delta f(\mathbf{M})}{\delta\gamma_{ij}}\right)\right]$$
$$= \int d^3 x\, M^{a_1...a_n}{}_{b_1...b_m}\left[-\left(\mathcal{L}_{\vec{N}}\pi^{ij}\right)\left(\frac{\delta\tilde{f}_{a_1...a_n}{}^{b_1...b_m}}{\delta\pi^{ij}}\right) - \left(\mathcal{L}_{\vec{N}}\gamma_{ij}\right)\left(\frac{\delta\tilde{f}_{a_1...a_n}{}^{b_1...b_m}}{\delta\gamma_{ij}}\right)\right]. \qquad (\text{H.24})$$

But $\tilde{f}_{a_1...a_n}{}^{b_1...b_m}$ is a function of phase space variables, hence using chain rule, we have:

$$\mathcal{L}_{\vec{N}}\tilde{f}_{a_1...a_n}{}^{b_1...b_m}(\gamma_{ij}, \pi^{ij}) = \left(\mathcal{L}_{\vec{N}}\pi^{ij}\right)\left(\frac{\delta\tilde{f}_{a_1...a_n}{}^{b_1...b_m}}{\delta\pi^{ij}}\right) + \left(\mathcal{L}_{\vec{N}}\gamma_{ij}\right)\left(\frac{\delta\tilde{f}_{a_1...a_n}{}^{b_1...b_m}}{\delta\gamma_{ij}}\right). \qquad (\text{H.25})$$

Thus we get:

$$\Rightarrow \{\mathbb{D}[N^i], f[\mathbf{M}]\}|_{(\gamma,\pi)} = -\int d^3 x\, M^{a_1...a_n}{}_{b_1...b_m}\left(\mathcal{L}_{\vec{N}}\tilde{f}_{a_1...a_n}{}^{b_1...b_m}(\gamma_{ij}, \pi^{ij})\right)$$
$$= \int d^3 x\left(\mathcal{L}_{\vec{N}} M^{a_1...a_n}{}_{b_1...b_m}\right)\tilde{f}_{a_1...a_n}{}^{b_1...b_m}(\gamma_{ij}, \pi^{ij}), \qquad (\text{H.26})$$

where we applied integration by parts in the second line and ignored the boundary terms. But from eq. (H.21), we identify the RHS to be $f[\mathcal{L}_{\vec{N}}\mathbf{M}]$. Thus we get the general result to be:

$$\Rightarrow \boxed{\{\mathbb{D}[N^i], f[\mathbf{M}]\}|_{(\gamma,\pi)} = f\left[\mathcal{L}_{\vec{N}}\mathbf{M}\right],} \tag{H.27}$$

for any general $f$ defined in eq. (H.20).

**Proof of (a)**

We substitute $f[\mathbf{M}] = \mathbb{D}[\vec{M}]$ in eq. (H.27) to get:

$$\boxed{\{\mathbb{D}[N^i], \mathbb{D}[\vec{M}]\}|_{(\gamma,\pi)} = \mathbb{D}\left[\mathcal{L}_{\vec{N}}\vec{M}\right] = -\mathbb{D}\left[\mathcal{L}_{\vec{M}}\vec{N}\right] = \mathbb{D}\left[[\vec{N},\vec{M}]\right].} \tag{H.28}$$

**Proof of (b)**

We substitute $f[\mathbf{M}] = \mathbb{H}[N]$ in eq. (H.27) to get:

$$\boxed{\{\mathbb{D}[N^i], \mathbb{H}[N]\}|_{(\gamma,\pi)} = \mathbb{H}[\mathcal{L}_{\vec{N}}N] = \mathbb{H}[N^j \partial_j N].} \tag{H.29}$$

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
