# Peer review of "Introduction to Hamiltonian Formulation of General Relativity and Homogeneous Cosmologies"

_SciPost Physics Lecture Notes, doi:SciPost Phys. Lect. Notes 73 (2023)_

## Round 1 · Referee Report · Anonymous · 2023-6-29

Strengths

1-Systematic presentation of the Hamiltonian formulation of general relativity with applications to anisotropic cosmological models.

2-Contains several detailed pedagogical derivations of well-known but often difficult results.

3-Emphasis on topics of current interest, such as the approach to cosmological singularities and the inclusion of various kinds of matter fields.

Weaknesses

1-Just a few imprecise statements.

Report

This review meets the acceptance criteria for Lecture Notes. It presents a detailed discussion of the Hamiltonian formulation of general relativity, including several new pedagogical derivations that cannot be found in this form in the existing literature. A considerable part of these notes is devoted to questions of current interest mainly in cosmology, such as quiescence and the inclusion of electromagnetic fields.

There are a few statements in this paper, mainly of mathematical nature, that do not take into account recent developments. However, they are not crucial for the main part of the discussion and can easily be corrected by rewording the corresponding sentences. In particular, in arXiv:2201.02883, Blohmann, Schiavina and Weinstein showed that the appropriate algebraic structure for hypersurface deformations is not a Lie algebroid but rather an L-infinity algebroid in which the Jacobi identity does not hold (in contrast to the Poisson bracket used in the physical derivation). Physical implications of this result have not been worked out yet and therefore cannot be included in these notes. However, the mathematical results indicate that the term "Lie algebroid" should be avoided in this context.

Requested changes

1-page 3, second paragraph: See comments above on "Lie algebroid"

2-page 12: What is the meaning of "direction" assigned to \hat{t}? This object is not a vector field, which is the usual mathematical structure that describes a direction.

3-page 27, footnote 4: The statement "which are spacetime functions, not structure constants, arising as one of the consequences of the nonlinearity of general relativity" is not clear because non-Abelian Yang-Mills theories are non-linear but have structure constants. The appearance of structure functions in Hamiltonian general relativity rather seems to be a property of hypersurface deformations, which is geometrical rather than dynamical. Notice also that the Lagrangian formulation of general relativity has structure constants in its symmetry algebra (space-time diffeomorphisms) even though its dynamics is as non-linear as it is in the Hamiltonian formulation. At the beginning of the footnote, the term "Lie algebroid" should be avoided or corrected.

4-page 56: Missing section number in bottom paragraph.

5-page 63: Missing citation reference in top paragraph and first line of section 6.1.

  • validity: high
  • significance: high
  • originality: good
  • clarity: high
  • formatting: good
  • grammar: good

Author:  Rishabh Jha  on 2023-06-30  [id 3773]

(in reply to Report 1 on 2023-06-29)

I would like to thank the referee for providing this useful report by taking their time out. I have incorporated the following changes in the manuscript based on the referee's suggestions:

  1. Page 3: All references to Lie algebroids are completely removed.

  2. Page 12: Indeed, the notation was confusing with $\hat{t}$, so I have replaced that sentence with the following clarification in the manuscript: "We have defined our hypersurface $\Sigma_t$ as that of a surface with constant $t$ where $t$ is a scalar field on $\mathcal{M}$. So the $1$-form $\mathrm{d}t$ is normal to $\Sigma_t$ in the sense that every vector on $\Sigma_t$ has a vanishing inner product with $\mathrm{d}t$. Accordingly the metric dual of $\mathrm{d}t$, namely $\partial_{\mu}t$, is also normal to the hypersurface $\Sigma_t$ where $\partial_{\mu}t$ is timelike if $\Sigma_t$ is spacelike. Thus we see a resemblance between the structure of $\partial_{\mu}t$ and $n_{\mu}$. Indeed upto a normalization constant, we can write $n_{\mu} = \Omega \partial_{\mu} t$ where $\Omega = \Omega(x^{\mu})$ is a normalization constant which is fixed by the condition $n^{\mu}n_{\mu} = -1$."

  3. Page 27: Footnote 4 has been rephrased as: "$\gamma^{jk}(x)$ appearing in (c) are spacetime functions, not just constants, that emerge from the geometrical deformations of hypersurface."

  4. Page 27: In accordance with point 3 above, I have made explicit the spacetime dependence $x$ of $\gamma^{jk}$ in eq. 105.(c).

  5. Page 56: The missing section number has been corrected.

  6. Page 63: Two missing citations have been corrected.

Attachment:

scipost_lecture_notes_resubmission_Rishabh_Jha.pdf

---

## Round 2 · Referee Report · Anonymous · 2023-6-30

Report

All the previous comments have been addressed.

---

## Editorial Decision

published